# A function-based typology for Earth's ecosystems

David A. Keith[1,2,3✉], José R. Ferrer-Paris[1,3], Emily Nicholson[3,4], Melanie J. Bishop[5], Beth A. Polidoro[6], Eva Ramirez-Llodra[7,8], Mark G. Tozer[1,2], Jeanne L. Nel[9,10], Ralph Mac Nally[11], Edward J. Gregr[12,13], Kate E. Watermeyer[4], Franz Essl[14,15], Don Faber-Langendoen[16], Janet Franklin[17], Caroline E. R. Lehmann[18,19], Andrés Etter[20], Dirk J. Roux[9,21], Jonathan S. Stark[22], Jessica A. Rowland[3,4], Neil A. Brummitt[23], Ulla C. Fernandez-Arcaya[24], Iain M. Suthers[1], Susan K. Wiser[25], Ian Donohue[26], Leland J. Jackson[27], R. Toby Pennington[18,28], Thomas M. Iliffe[29], Vasilis Gerovasileiou[30,31], Paul Giller[32,33], Belinda J. Robson[34], Nathalie Pettorelli[35], Angela Andrade[3,36], Arild Lindgaard[37], Teemu Tahvanainen[38], Aleks Terauds[22], Michael A. Chadwick[39], Nicholas J. Murray[1,3,40], Justin Moat[41], Patricio Pliscoff[42,43], Irene Zager[44] & Richard T. Kingsford[1]

As the United Nations develops a post-2020 global biodiversity framework for the Convention on Biological Diversity, attention is focusing on how new goals and targets for ecosystem conservation might serve its vision of 'living in harmony with nature'[1,2]. Advancing dual imperatives to conserve biodiversity and sustain ecosystem services requires reliable and resilient generalizations and predictions about ecosystem responses to environmental change and management[3]. Ecosystems vary in their biota[4], service provision[5] and relative exposure to risks[6], yet there is no globally consistent classification of ecosystems that reflects functional responses to change and management. This hampers progress on developing conservation targets and sustainability goals. Here we present the International Union for Conservation of Nature (IUCN) Global Ecosystem Typology, a conceptually robust, scalable, spatially explicit approach for generalizations and predictions about functions, biota, risks and management remedies across the entire biosphere. The outcome of a major cross-disciplinary collaboration, this novel framework places all of Earth's ecosystems into a unifying theoretical context to guide the transformation of ecosystem policy and management from global to local scales. This new information infrastructure will support knowledge transfer for ecosystem-specific management and restoration, globally standardized ecosystem risk assessments, natural capital accounting and progress on the post-2020 global biodiversity framework.

Sustaining ecosystem functions and services requires an understanding of ecological processes and mechanisms that drive ecosystem change[6]. Ecosystem functioning not only underpins biomass production, but also depends on and regulates the stocks and fluxes of resources, energy and biota[7]. These functions, together with ecological processes and species traits—collectively referred to as 'ecosystem properties' (see Supplementary Information, Glossary)—define and sustain ecosystem identity and shape ecosystem responses to environmental change, including anthropogenic changes[8]. Ecosystems with different species compositions may show functional convergence if their biota share similar traits and contribute to similar ecological processes (for example, in ref. [9]). Together with ecosystem function, the identity of constituent biota is central to biodiversity concepts, conservation goals and human values[10]. Although ecosystem functions and ecological processes support both the diversity of biota and human well-being, global assessments of ecosystems[11,12] continue to rely heavily on species metrics or simplistic land-cover proxies that convey limited information about ecosystems themselves. This limits our ability to diagnose trends and to design and resource on-ground management and policy solutions for slowing and reversing current declines in biodiversity and ecosystem services.

To serve the dual needs of sustaining ecosystem services and conserving biodiversity, ecosystem assessments require a global typology to frame comparisons and standardize data aggregation for analysing ecosystem trends and diagnosing their causes. To support applications throughout Earth's diverse ecosystems, users and scales of analysis, this typology should encapsulate: (1) ecosystem functions and ecological processes; (2) their characteristic biota; (3) conceptual consistency throughout the whole biosphere; (4) a scalable structure; (5) spatially explicit units; and (6) descriptive detail and minimal complexity (see Supplementary Information, Appendix 1 and Supplementary Table 1.1 for rationale).

We used these 6 design criteria to review a sample of 23 global-scale ecological typologies, finding none that explicitly represented both

ecosystem functions and biota (Supplementary Table 1.2). This limits the ability of ecosystem managers to learn from related ecosystems with similar operating mechanisms and drivers of change. Only three typologies encompassed the whole biosphere, but these lacked a clear theoretical basis, limiting their ability to generalize about properties of ecosystems grouped together. Ecological classifications based on tested and established theory are more likely to be robust to new information than classifications based only on observed patterns and correlations, which may prove unstable when new information emerges. Many typologies that we examined either did not describe their units in sufficient detail for reliable identification, or required diagnostic features that are difficult to observe. Others were based on biophysical attributes or biogeography, but approaches differed across terrestrial, freshwater and marine domains, precluding a truly global approach. In this study, we developed a Global Ecosystem Typology that meets all six design principles, thereby providing a stronger foundation for systematic ecosystem assessments, sustainable management and biodiversity conservation.

## Conceptual foundations

We developed a conceptual model to inform the construction of the Global Ecosystem Typology, consistent with the six design principles, and to serve as a template for describing the units of classification. The model (Fig. 1) frames working hypotheses about the processes (or 'drivers') that shape ecosystem properties and the interactions among drivers and properties. Ecosystem properties are attributes of ecosystems and their component biota that result from assembly processes[13]. They include aggregate ecosystem functions (productivity, stocks and fluxes), ecological processes (for example, trophic networks), structural features (for example, 3D spatial structure and diversity) and species-level traits of characteristic organisms (for example, ecophysiology, life histories and morphology).

Our model postulates five groups of ecological drivers that may shape ecosystems by acting both as assembly filters and evolutionary pressures (Fig. 1 and Supplementary Information, Appendix 2, for details). Filters are biotic and abiotic processes that determine community assembly from a species pool, given initial occupancy or dispersal (based on community assembly theory[13,14]). Evolutionary pressures are agents of selection that influence ecosystem function and constituent species traits, typically over longer time scales, through evolution and extinction within a dynamic species pool[13,15].

'Resource drivers' (Supplementary Information, Appendix 2, page 2) supply water, oxygen, nutrients, carbon and energy, the resources essential for life. The 'ambient environment' (Supplementary Information, Appendix 2, page 2) includes environmental features (for example, temperature, pH, salinity) that continually influence the availability of resources or the ability of organisms to acquire them. The model distinguishes these continuous factors from 'disturbance regimes' (Supplementary Information, Appendix 2, page 2), which are sequences of discrete events with different intensities and patterns of occurrence (for example, fires, floods, storms and earth mass movement) that destroy living biomass, liberate and redistribute resources, and regulate life-history processes. 'Biotic interactions' (Supplementary Information, Appendix 2, page 3) include competition, predation, pathogenicity, mutualisms and facilitation, which operate at local scales but may shape ecosystem properties at landscape and seascape scales (for example, reef-building symbioses). 'Human activities' (Supplementary Information, Appendix 2, page 3) are a special class of biotic interaction that influence ecosystem disassembly and reassembly through resource appropriation, physical restructuring, movement of biota, and climate change[16]. These anthropic processes operate largely, but not exclusively, through effects on other drivers. Although our model portrays humans as integral drivers of ecosystem assembly, we separated human activity from other biotic interactions

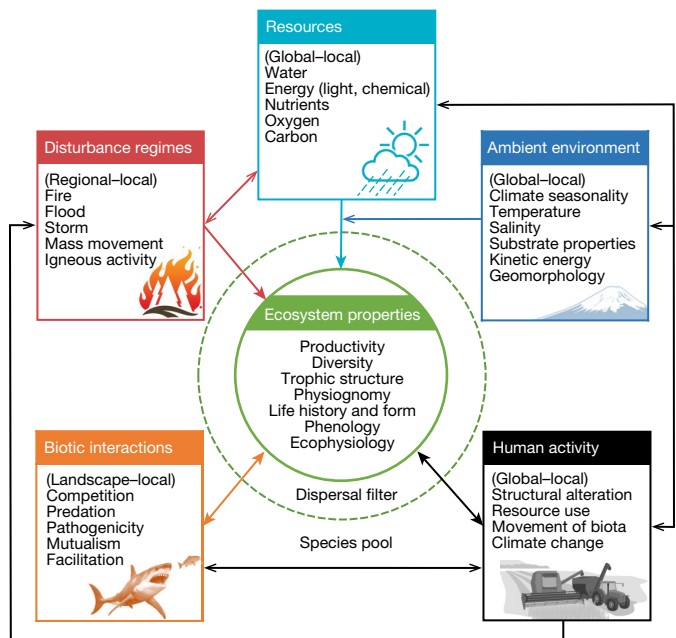

**Fig. 1 | The generic model of ecosystem assembly underlying the Global Ecosystem Typology.** Boxes represent abiotic (resources, the ambient environment and disturbance regimes) and biotic (biotic interactions and human activity) drivers that filter assemblages and form evolutionary pressures that in turn, shape ecosystem-level properties (inner green circle). The range of major organizational scales at which drivers operate are shown in parentheses, followed by a list of the major expressions of the drivers. The species pool is the set of 'available' traits on which the assembly filters and evolutionary pressures operate over short and longer time frames, respectively. Species pools are dynamic products of vicariance, dispersal and evolution that depend on biogeographic context and history. The outer green circle (dashed line) represents the contemporary dispersal filter that mediates the biota currently subjected to local selection by the abiotic and biotic filters and pressures. The inner green circle represents the properties (aggregate ecosystem functions and species-level traits) that characterize the ecosystem. Closed arrows show the influence of filtering processes on ecosystem properties. Feedbacks can occur whereby ecosystem properties modulate filtering processes (examples are indicated by bidirectional arrows). Interactions among drivers include indirect effects of human activity on assembly through other drivers (black open arrows) and the indirect effects of ambient environmental conditions on assembly by modulating resource availability or uptake (dark blue open arrow). Interactions among other drivers (omitted here for simplicity) are shown in ecosystem-specific adaptations of this generic model for each ecosystem functional group (level 3 of the typology) in Supplementary Information, Appendix 4. See Supplementary Information, Glossary, for explanation of terms. Details are in Supplementary Information, Appendix 2. Illustrations (wildfire icon; Japan mt. fuji; shark) DigitalVision Vectors via Getty Images.

to highlight connections between ecosystems and socio-economic systems that drive anthropogenic change[17], and the need to assess and mitigate the human impacts on biodiversity and ecosystem functioning.

Interactions may exist among drivers, modulating their effects on ecosystem properties (Fig. 1 and Supplementary Information, Appendix 2, page 4). For example, resource levels may influence ecosystem assembly directly through niche partitioning or indirectly through alteration of biotic interactions[18]. Similarly, feedbacks exist between ecosystem properties and drivers. For example, human land-use intensification initiates changes in ecosystems that, in turn, influence human social structure, markets and consumption patterns, driving changes in resource appropriation and further change in ecosystem properties[17]. Variations on the model template applied to different groups of

ecosystems in our typology (Supplementary Information, Appendices 3 and 4, pages 52–186) reflect our hypotheses about how drivers influence ecosystem properties directly, or indirectly through interactions with other drivers. The model posits that ecosystems share convergent ecological processes and functional properties if they are shaped by similar drivers—and conversely, major changes to these drivers (or their interactions) cause disassembly, transformation and ultimately ecosystem collapse, with consequent losses of biodiversity, ecosystem function and services[8].

Convergences in ecosystem properties are axiomatic to a functionally based ecosystem typology because they underpin robust generalizations and predictions about ecosystem responses to environmental change and management. Convergences in species traits may arise from common evolutionary origins and niche conservatism[19,20], but similarities in ecological drivers (selection pressures and assembly filters) may also produce functional convergences in independent lineages. These convergences are enablers of a functional classification framework represented in the upper three levels of our typology. Functional constraints may be imposed by the species pool, which is a dynamic outcome of vicariance, dispersal and evolution, depending on ecosystem location and biogeographic history[21].

Only a few ecological drivers are likely to be important in shaping the key properties of any particular ecosystem[13], despite the vast array of potential drivers on Earth and the complex interactions among them. This principle was critical to design of assembly models of each ecosystem functional group and for developing a parsimonious global typology (Supplementary Table 1, principle 6).

## Typology structure

Our ecosystem typology, adopted by the IUCN at the 2020 World Conservation Congress[22,23] has six hierarchical levels, enabling applications at different thematic scales (Methods and Supplementary Fig. 3.1). Three upper levels (Supplementary Table 3.1) differentiate functional groupings and three lower levels (Methods and Supplementary Information, Appendix 3, pages 19 and 20) accommodate differences in biotic composition among functionally convergent ecosystems. The scalable hierarchical structure (Supplementary Table 1.1, principle 4) and the explicit description of properties and drivers enables units at any thematic level to be mapped at different spatial scales. These units may be tracked through different temporal scales according to needs of specific applications and constraints arising from the resolution of available data.

Level 3 units of the typology (ecosystem functional groups, described in Supplementary Information, Appendix 4, pages 52–186 and summarized in Extended Data Tables 1–4) are fundamental to generalizations and predictions about ecosystems with similar functional properties, and therefore have key roles in global synthesis and knowledge transfer for ecosystems. Their distribution across landscapes and seascapes (Fig. 2) is governed by the expression of ecological drivers along temporally variable multidimensional gradients[24,25] (Fig. 3). Interactions between the drivers that operate at different spatial scales in this multidimensional space determine the dominant filters and evolutionary pressures that shape ecosystem properties in different parts of the biosphere (see Methods, 'Hierarchical levels' and Supplementary Information, Appendix 3 for key drivers that differentiate ecosystem functional groups along landscape and seascape gradients visualized in Figs. 2 and 3).

## Applications for ecosystem management

Decisions about effective action to conserve biodiversity and sustain ecosystem services require evidence of which ecosystems are most exposed to risks of collapse[6] and which ecosystems contribute most to particular human benefits[5]. These analyses are conspicuously lacking in global ecosystem assessments[11,12,26], but the IUCN Global Ecosystem Typology and a rapidly growing body of spatial data[27] have established an ecologically robust and powerful capability, and signal a growing readiness for such syntheses.

The IUCN Global Ecosystem Typology facilitates integrated assessment of Earth's ecosystems, enabling a more powerful and complete evaluation of progress towards biodiversity targets and sustainable development goals than previously possible. This fills a significant gap, exemplified by the limited range of ecosystems assessed in the Convention on Biological Diversity (CBD) Global Biodiversity Outlook 5[26] and the IPBES Global Assessment[12]. It will also strengthen the evidence base for setting science- and knowledge-based specific, measurable, ambitious, realistic and time-bound (SMART) biodiversity targets in the forthcoming post-2020 CBD global biodiversity framework and for reviewing progress towards them[2]. The United Nations Statistical Commission recently adopted the IUCN typology as a reference classification for extending the System of Environmental Economic Accounting (SEEA) framework to Ecosystem Accounts[28], meeting a long-recognized need for a spatially explicit, functionally based ecosystem typology to underpin natural capital accounting[29].

Integrating both functions and biota into the hierarchical structure of the typology confers versatility for diverse applications in ecosystem management and conservation (Fig. 4 and Supplementary Information, Appendix 6). Our typology and developing archive of maps (see caveats in Supplementary Information, Appendix 4) provide a globally consistent framework for advancing the IUCN Red List of Ecosystems[6,30] and Key Biodiversity Areas[31], as well as broadly based nature education[32].

Diagnostic models of ecosystem dynamics, as developed in Red List assessments[30], with improved ecosystem and threat distribution data, will strengthen capacity to forecast state changes that result in loss of ecosystem function, services and biota. Ecosystem groupings based on convergent drivers, properties and environmental relationships will reveal similarities in threats and mechanisms of degradation, and therefore inform the development of ecosystem-specific management strategies for recovery. Embracing the dynamic nature of ecosystems and its dependency on ecological processes is a key feature that differentiates the IUCN Global Ecosystem Typology from other ecological typologies (Supplementary Table 1.2). This will enable policy and management actions to be targeted towards causes of ecosystem degradation, with knowledge transfer and adaptive learning[33] about local ecosystems from functionally similar ecosystems elsewhere (Supplementary Fig. 6.1).

## Limitations and the way forward

We expect progressive improvements in future versions of the IUCN Global Ecosystem Typology as knowledge increases. Several aspects of the typology warrant further development to address uncertainties. In particular, models of assembly for each ecosystem functional group represent working hypotheses, for which available empirical evidence varies greatly (Methods, 'Limitations'). Redressing research biases across different ecosystem types and among different assembly filters will help improve not only the assembly models, but also the distinctions between ecosystem functional groups and units within other levels of the typology.

By highlighting poorly known systems in the atmosphere, deep sea floors, subterranean freshwaters, lithosphere and beneath ice, and by prompting researchers and other users to ask where particular ecosystems belong in the scheme, we foresee the typology promoting research to fill significant knowledge gaps that will improve outcomes of its application and inform future amendments of its structure, as well as descriptions of its units.

Ecosystem mapping is another component of the information base that urgently requires further development, as the currently available indicative global maps for ecosystem functional groups vary

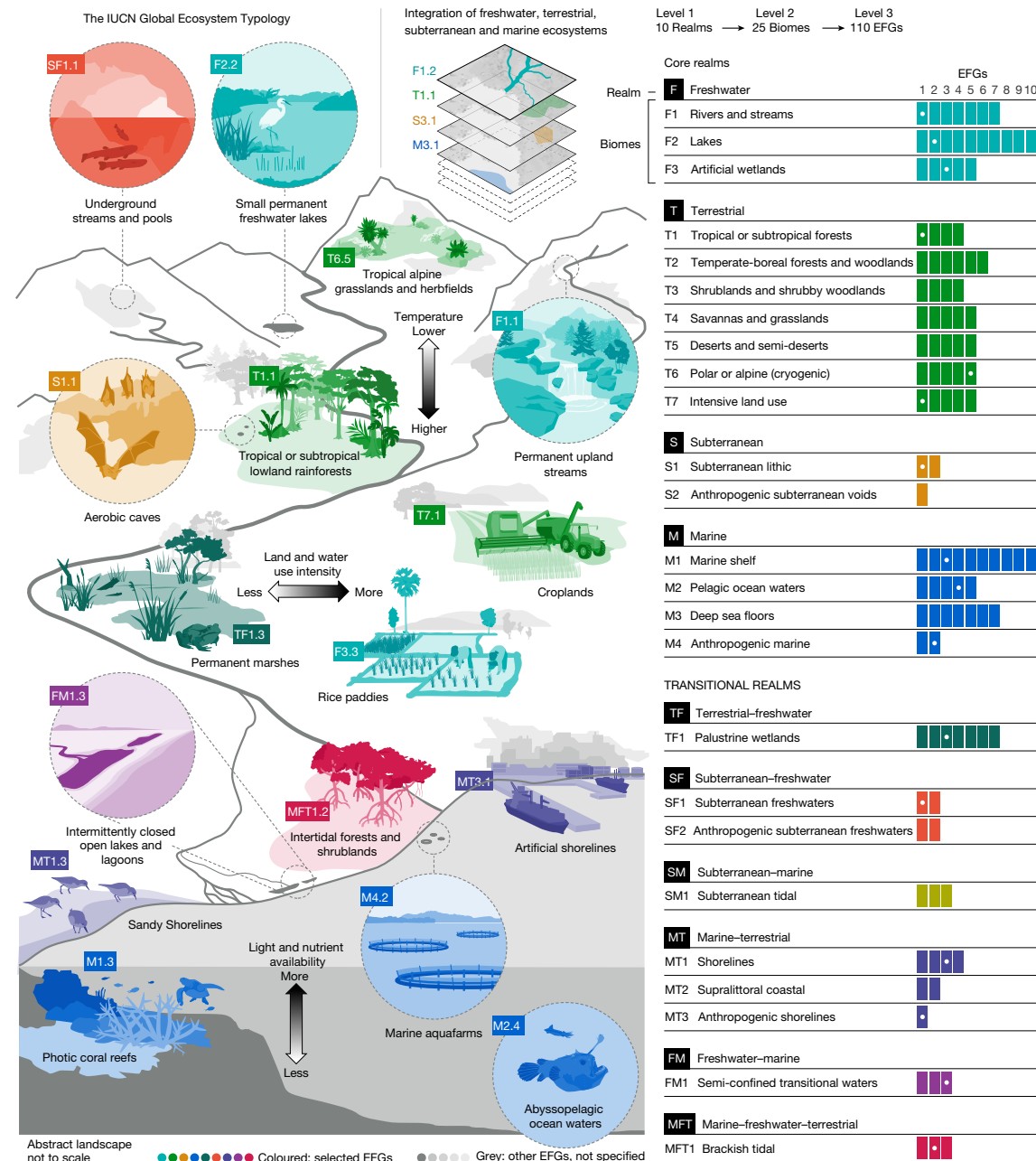

**Fig. 2 | Landscape and seascape relationships of ecosystem functional groups.** Left, a sample of ecosystem functional groups (EFGs) from the Global Ecosystem Typology distributed across a hypothetical tropical landscape and seascape. Right, the total number of ecosystem functional groups (coloured boxes) within each realm and functional biome listed (the ecosystem functional groups illustrated on the left are represented by white dots). Multidimensional environmental gradients—three examples are shown: temperature, intensity of human use and light and nutrient availability—influence the strength and spatial expression of ecological drivers (resources, ambient environment, disturbance regimes, biotic interactions and human activity) across landscapes and seascapes, and therefore the spatial relationships of ecosystem types.

substantially in accuracy and precision (Methods, 'Limitations'). Many uses of the typology (Fig. 3) do not require a full set of comprehensive and globally consistent maps because they are non-spatial (that is, knowledge transfer and framing generalizations), national in scope, or specific to particular ecosystem groups (for example, forests, coral reefs and mangroves). Reliable global maps of suitable resolution, however, are pivotal to the global synthesis of ecosystems, as required for systematic reporting on CBD targets and some other applications[2].

By decoupling the mapping process from prior development of the classification, our approach liberates the definition of ecosystem units from constraints imposed by the current availability of spatial data and allows for progressive improvement in maps (Supplementary Information, Appendix 4, page 13). New technologies in cloud computing and artificial intelligence, improved global environmental data and deepening time archives of satellite images are paving the way[34,35]. High-resolution maps, some with extended time series, that match the concepts of ecosystem functional groups have been produced for contrasting ecosystem groups such as tidal mudflats[36] (TM1.2), glacial lakes[37] (F2.4) and tropical cloud forests[38] (T1.3) (Supplementary Table 4.1); whereas generic data cubes for forest cover[39] and surface water[40] suggest that global high-resolution time-series mapping should be possible for most ecosystem functional groups within the next decade. Future versions of the typology will progressively improve map standards to support applications that depend on spatial analysis.

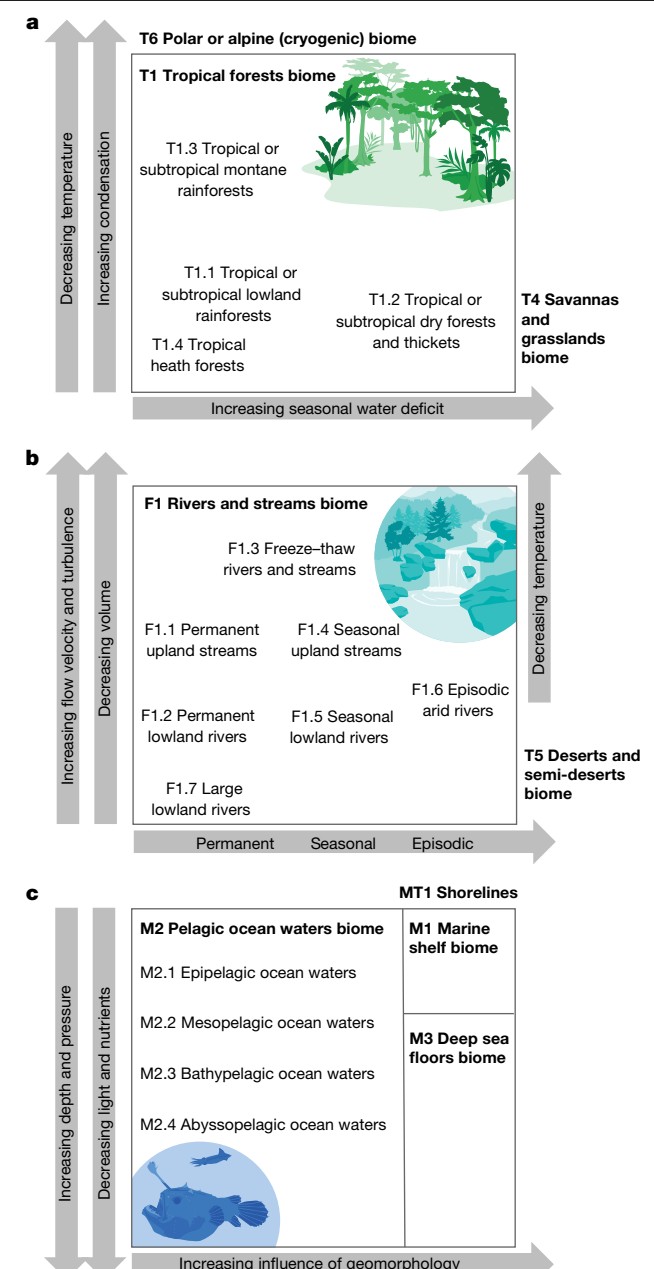

**Fig. 3 | Hypothesized relationships of functional groups differentiated along gradients of selected assembly filters. a**, The Tropical forests biome (T1), with temperature, elevation and water availability gradients. **b**, The Rivers and streams biome (F1), with stream gradient and temporal flow pattern. **c**, The Marine pelagic biome (M2), with depth and current gradients. In **a**, a third filter related to an edaphic environmental gradient differentiates group T1.4 from T1.1, but is not shown here (see Supplementary Information, Appendix 4, for details on the respective functional groups).

Improved mapping of threats and degradation is similarly required to support ecosystem assessments[41], particularly in marine environments.

We acknowledge the limitations associated with discrete representation of continuous ecological patterns in nature (Supplementary Information, Appendix 3, page 23). Even though our descriptive framework recognizes core and transitional units, its discrete structure generates boundary and other uncertainties among ecosystems that are ultimately unavoidable, even with extensive description or splitting of classes[42]. However, this fallibility is outweighed by a

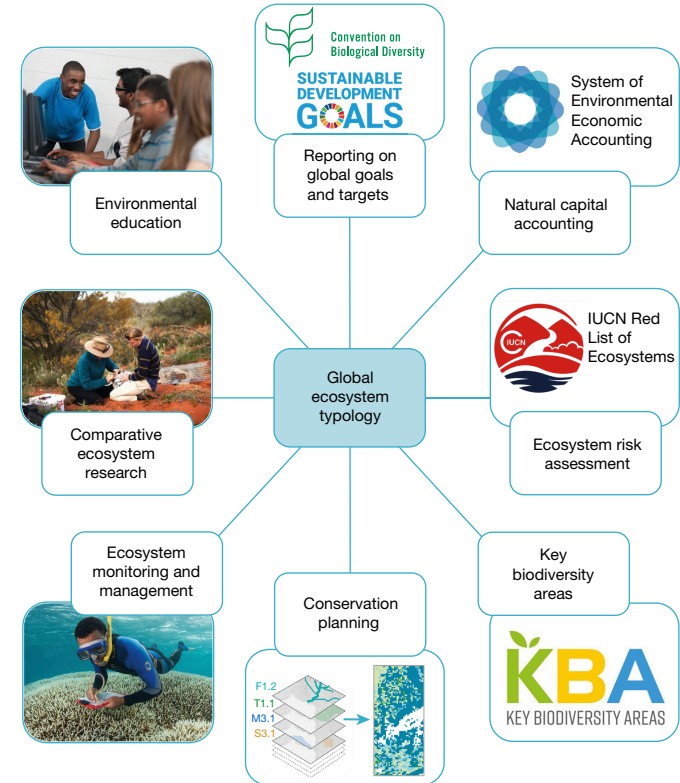

**Fig. 4 | Current and potential applications of the Global Ecosystem Typology to conserve biodiversity and sustain ecosystem services.** The typology provides a common ecosystem vocabulary and supports consistent treatment of ecosystems across applications where policy links exist between multiple initiatives. Details are presented in Supplementary Information, Appendix 4. Photo credit: Keith Ellenbogen (Ecosystem monitoring and management); Getty Images (Environmental education); KBA World Database of Key Biodiversity Areas at www.keybiodiversityareas.org; United Nations Sustainable Development Goals at: www.un.org/sustainabledevelopment.

classificatory approach founded in deep-seated cognitive processes that govern how humans understand and manage environmental, social, economic and cultural dimensions of their conscious universe by dividing it into parts[43]. This will facilitate the widespread uptake of the IUCN typology for effective storage, retrieval and transfer of ecosystem information.

The hierarchical structure of our typology should enable global imperatives to be linked directly with on-ground, nature-based solutions[44], supporting international mandates for sustainable development and biodiversity conservation. Viewing Earth's ecosystems through a dynamic functional lens, rather than through largely biogeographic or biophysical ones, will enable a more powerful and direct basis to address the dual goals of conserving biodiversity and sustaining ecosystem services.

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

¹Centre for Ecosystem Science, University of New South Wales, Sydney, New South Wales, Australia. ²New South Wales Department of Planning, Industry and Environment, Hurstville, New South Wales, Australia. ³IUCN Commission on Ecosystem Management, Gland, Switzerland. ⁴Centre for Integrative Ecology, Deakin University, Burwood, Victoria, Australia. ⁵Department of Biological Sciences, Macquarie University, Sydney, New South Wales, Australia. ⁶School of Mathematics and Natural Sciences, Arizona State University, Glendale, AZ, USA. ⁷Norwegian Institute for Water Research, Oslo, Norway. ⁸REV Ocean, Lysaker, Norway. ⁹Sustainability Research Unit, Nelson Mandela University, Port Elizabeth, South Africa. ¹⁰Wageningen Environmental Research, Wageningen University, Wageningen, The Netherlands. ¹¹School of BioSciences, The University of Melbourne, Melbourne, Victoria, Australia. ¹²Institute for Resources, Environment and Sustainability, University of British Columbia, Vancouver, British Columbia, Canada. ¹³SciTech Environmental Consulting, Vancouver, British Columbia, Canada. ¹⁴BioInvasions, Global Change, Macroecology-Group, Department of Botany and Biodiversity Research, University of Vienna, Vienna, Austria. ¹⁵Centre for Invasion Biology, Stellenbosch University, Stellenbosch, South Africa. ¹⁶NatureServe, Arlington, VA, USA. ¹⁷University of California, Riverside, CA, USA. ¹⁸Royal Botanic Garden Edinburgh, Edinburgh, UK. ¹⁹School of GeoSciences, University of Edinburgh, Edinburgh, UK. ²⁰Departamento de Ecología y Territorio, Pontificia Universidad Javeriana, Bogotá, Colombia. ²¹Scientific Services, South African National Parks, George, South Africa. ²²Australian Antarctic Division, Department of Climate Change, Energy, the Environment and Water, Hobart, Tasmania, Australia. ²³Department of Life Sciences, Natural History Museum, London, UK. ²⁴Instituto Español de Oceanografía, Centro Oceanográfico de Baleares, Palma, Spain. ²⁵Manaaki Whenua—Landcare Research, Lincoln, New Zealand. ²⁶Department of Zoology, School of Natural Sciences, Trinity College Dublin, Dublin, Ireland. ²⁷University of Calgary, Calgary, Alberta, Canada. ²⁸College of Life and Environmental Sciences Geography, University of Exeter, Exeter, UK. ²⁹Department of Marine Biology, Texas A&M University, Galveston, TX, USA. ³⁰Hellenic Centre for Marine Research (HCMR), Institute of Marine Biology, Biotechnology and Aquaculture (IMBBC), Heraklion, Greece. ³¹Department of Environment, Faculty of Environment, Ionian University, Zakynthos, Greece. ³²School of Biological Earth and Environmental Sciences, University College Cork, Cork, Ireland. ³³School of Life Sciences, South China Normal University, Guangzhou, China. ³⁴Centre for Sustainable Aquatic Ecosystems, Harry Butler Institute, Murdoch University, Perth, Western Australia, Australia. ³⁵Institute of Zoology, Zoological Society of London, London, UK. ³⁶Conservation International Colombia, Bogota, Colombia. ³⁷Norwegian Biodiversity Information Centre, Trondheim, Norway. ³⁸Department of Environmental and Biological Sciences, University of Eastern Finland, Joensuu, Finland. ³⁹Department of Geography, King's College London, London, UK. ⁴⁰College of Science and Engineering, James Cook University, Townsville, Queensland, Australia. ⁴¹Royal Botanic Gardens Kew, Richmond, UK. ⁴²Institute of Geography, Department of Ecology, Center of Applied Ecology and Sustainability (CAPES), Universidad Católica de Chile, Santiago, Chile. ⁴³Instituto de Ecología y Biodiversidad, Santiago, Chile. ⁴⁴Provita, Caracas, Venezuela. ✉e-mail: david.keith@unsw.edu.au

## Methods

We developed the IUCN Global Ecosystem Typology in the following sequence of steps: design criteria; hierarchical structure and definition of levels; generic ecosystem assembly model; top-down classification of the upper hierarchical levels; iterative circumscription of the units and ecosystem-specific adaptations of the assembly model; full description of the units; and map compilation. Some iteration proved necessary, as the description and review process sometimes revealed a need for circumscribing additional units.

### Design criteria and other typologies

Under the auspices of the IUCN Commission on Ecosystem Management, we developed six design principles to guide the development of a typology that would meet the needs for global ecosystem reporting, risk assessment, natural capital accounting and ecosystem management: (1) representation of ecological processes and ecosystem functions; (2) representation of biota; (3) conceptual consistency throughout the biosphere; (4) scalable structure; (5) spatially explicit units; and (6) parsimony and utility (see Supplementary Table 1.1 and Supplementary Information, Appendix 1 for definitions and rationale).

We assessed 23 existing ecological classifications with global coverage of terrestrial, freshwater, and/or marine environments against these principles to determine their fitness for IUCN's purpose (Supplementary Information, Appendix 1). These include general classifications of land, water or bioclimate, as well as classifications of units that conform with the definition of ecosystems adopted in the United Nations Convention on Biological Diversity[45] or an equivalent definition in the IUCN Red List of Ecosystems[30]. We reviewed documentation on methods of derivation, descriptions of classification units and maps to assess each classification against the six design principles (Supplementary Table 1.2 for details).

### Typology structure and ecosystem assembly

We developed the structure of the Global Ecosystem Typology and the generic ecosystem assembly model at a workshop attended by 48 terrestrial, freshwater and marine ecosystem experts at Kings College London, UK, in May 2017. Participants agreed that a hierarchical structure would provide an effective framework for integrating ecological processes and functional properties (Supplementary Table 1.1, design principle 1), and biotic composition (principle 2) into the typology, while also meeting the requirement for scalability (principle 4). Although neither function nor composition were intended to take primacy within the typology, we reasoned that a hierarchy representing functional features in the upper levels is likely to support generalizations and predictions by leveraging evolutionary convergence[13]. By contrast, a typology reflecting compositional similarities in its upper levels is less likely to be stable owing to dynamism of species assemblages and evolving knowledge on species taxonomy and distributions. Furthermore, representation of compositional relationships at a global scale would require many more units in upper levels, and possibly more hierarchical levels. Therefore, we concluded that a hierarchical structure recognizing compositional variants at lower levels within broad functionally based groupings at upper levels would be more parsimonious and robust (principle 6) than one representing composition at upper levels and functions at lower levels.

Workshop participants initially agreed that three hierarchical levels for ecosystem function and three levels for biotic composition could be sufficient to represent global variation across the whole biosphere. Participants developed the concepts of these levels into formal definitions (Supplementary Table 3.1), which were reviewed and refined during the development process.

To ensure conceptual consistency of the typology and its units throughout the biosphere (principle 3), we drew from community assembly theory to develop a generic model of ecosystem assembly.

The traditional community assembly model incorporates three types of filters (dispersal, the abiotic environment and biotic interactions) that determine which biota from a larger pool of potential colonists can occupy and persist in an area[13]. We extended this model to ecosystems by: (1) defining three groups of abiotic filters (resources, ambient environment and disturbance regimes) and two groups of biotic filters (biotic interactions and human activity); (2) incorporating evolutionary processes that shape characteristic biotic properties of ecosystems over time; (3) defining the outcomes of filtering and evolution in terms of all ecosystem properties including both ecosystem-level functions and species-level traits, rather than only in terms of species traits and composition; and (4) incorporating interactions and feedbacks among filters and selection agents and ecosystem properties to elucidate hypotheses about processes that influence temporal and spatial variability in the properties of ecosystems and their component biota. In community assembly, only a small number of filters are likely to be important in any given habitat[13]. In keeping with this proposition, we used the generic model to identify biological and physical features that distinguish functionally different groups of ecosystems from one another by focusing on different ecological drivers that come to the fore in structuring their assembly and shaping their properties.

### Hierarchical levels

The top level of classification (Fig. 2 and Extended Data Tables 1–4) defines five core realms of the biosphere based on contrasting media that reflect ecological processes and functional properties: terrestrial; freshwaters and inland saline waters (hereafter freshwater); marine; subterranean; and atmospheric. Biome gradient concepts[25] highlight continuous variation in ecosystem properties, which is represented in the typology by transitional realms that mark the interfaces between the five core realms (for example, floodplains (terrestrial–freshwater), estuaries (freshwater–marine), and so on). In Supplementary Information, Appendix 3 (pages 3–16) and Supplementary Table 3.1, we describe the five core realms and review the hypothesized assembly filters and ecosystem properties that distinguish different groups within them. The atmospheric realm is included for comprehensive coverage, but we deferred resolution of its lower levels because its biota is poorly understood, sparse, itinerant and represented mainly by dispersive life stages[46].

Functional biomes (level 2) are components of the biosphere united by one or more major assembly processes that shape key ecosystem functions and ecological processes, irrespective of taxonomic identity (Supplementary Information, Appendix 3, page 17). Our interpretation aligns broadly with 'functional biomes' described elsewhere[24,25,47], extended here to reflect dominant assembly filters and processes across all realms, rather than the more restricted basis of climate-vegetation relationships that traditionally underpin biome definition on land. Hence, the 25 functional biomes (Supplementary Information, Appendix 4, pages 52–186 and https://global-ecosystems.org/) include some 'traditional' terrestrial biomes[47], as well as lentic and lotic freshwater systems, pelagic and benthic marine systems, and anthropogenic functional biomes assembled and usually maintained by human activity[48].

Level 3 of the typology defines 110 ecosystem functional groups described with illustrated profiles in Supplementary Information, Appendix 4 (pages 52–186) and at https://global-ecosystems.org/. These are key units for generalization and prediction, because they include ecosystem types with convergent ecosystem properties shaped by the dominance of a common set of drivers (Supplementary Information, Appendix 3, pages 17–19). Ecosystem functional groups are differentiated along environmental gradients that define spatial and temporal variation in ecological drivers (Figs. 2 and 3 and Supplementary Figs. 3.2 and 3.4). For example, depth gradients of light and nutrients differentiate functional groups in pelagic ocean waters (Fig. 3c and Extended Data Table 4), influencing assembly directly and indirectly through predation. Resource gradients defined by flow regimes

(influenced by catchment precipitation and evapotranspiration) and water chemistry, modulated by environmental gradients in temperature and geomorphology, differentiate functional groups of freshwater ecosystems[25] (Fig. 3b and Extended Data Table 3). Terrestrial functional groups are distinguished primarily by gradients in water and nutrient availability and by temperature and seasonality (Fig. 3a and Extended Data Table 1), which mediate uptake of those resources and regulate competitive dominance and productivity of autotrophs. Disturbance regimes, notably fire, are important global drivers in assembly of some terrestrial ecosystem functional groups[49].

Three lower levels of the typology distinguish functionally similar ecosystems based on biotic composition. Our focus in this paper is on global functional relationships of ecosystems represented in the upper three levels of the typology, but the lower levels (Supplementary Information, Appendix 3, pages 19 and 20) are crucial for representing the biota in the typology, and facilitate the scaling up of information from established local-scale typologies that support decisions where most conservation action takes place. These lower levels are being developed progressively through two contrasting approaches with different trade-offs, strengths and weaknesses. First, level 4 units (regional ecosystem subgroups) are ecoregional expressions of ecosystem functional groups developed from the top-down by subdivisions based on biogeographic boundaries (for example, in ref. [50]) that serve as simple and accessible proxies for biodiversity patterns[51]. Second, level 5 units (global ecosystem types) are also regional expressions of ecosystem functional groups, but unlike level 4 units they are explicitly linked to local information sources by bottom-up aggregation[52] and rationalization of level 6 units from established subglobal ecological classifications. Subglobal classifications, such as those for different countries (see examples for Chile and Myanmar in Supplementary Tables 3.3 and 3.4), are often developed independently of one another, and thus may involve inconsistencies in methods and thematic resolution of units (that is, broadly defined or finely split). Aggregation of level 6 units to broader units at level 5 based on compositional resemblance is necessary to address inconsistencies among different subglobal classifications and produce compositionally distinctive units suitable for global or regional synthesis.

Integrating local classifications into the global typology, rather than replacing them, exploits considerable efforts and investments to produce existing classifications, already developed with local expertise, accuracy and precision. By placing national and regional ecosystems into a global context, this integration also promotes local ownership of information to support local action and decisions, which are critical to ecosystem conservation and management outcomes (Supplementary Information, Appendix 3, page 20). These benefits of bottom-up approaches come at the cost of inevitable inconsistencies among independently developed classifications from different regions, a limitation avoided in the top-down approach applied to level 4.

## Circumscribing upper-level units

We formed specialist working groups (terrestrial/subterranean, freshwater and marine) to develop descriptions of the units within the upper levels of the hierarchy, subdividing realms into functional biomes, and biomes into ecosystem functional groups. We used definitions of the hierarchical levels (Supplementary Table 3.1) and the conceptual model of ecosystem assembly (Fig. 1) to maintain consistency in defining the units at each level during iterative discussions within and between the working groups.

Working groups agreed on preliminary lists of functional biomes and ecosystem functional groups by considering variation in major drivers along ecological gradients (Figs. 2 and 3 and Supplementary Figs. 3.2 and 3.4) based on published literature, direct experience and expertise of working group members, and consultation with colleagues in their respective research networks. After the workshop, working groups sought recent global reviews of the candidate units and recent case studies of exemplars to shape descriptions of the major groups of ecosystem drivers and properties for each unit. Circumscriptions and descriptions of the units were reviewed and revised iteratively to ensure clear distinctions among units, with a total of 206 reviews of descriptive profiles undertaken by 60 specialists, a mean of 2.4 reviews per profile (Supplementary Table 5.1). The working groups concurrently adapted the generic model of ecosystem assembly (Fig. 1) to represent working hypotheses on salient drivers and ecosystem properties for each ecosystem functional group.

## Incorporating human influence

Very few of the ecological typologies reviewed in Supplementary Information, Appendix 1 integrate anthropogenic ecosystems in their classificatory frameworks. Anthropogenic influences create challenges for ecosystem classification, as they may modify defining features of ecosystems to a degree that varies from negligible to major transformation across different locations and times. We addressed this problem by distinguishing transformative outcomes of human activity at levels 2 and 3 of the typology from lesser human influences that may be represented either at levels 5 and 6, or through measurements of ecosystem integrity or condition that reflect divergence from reference states arising from human activity.

Anthropogenic ecosystems grouped within levels 2 and 3 were thus defined as those created and sustained by intensive human activities, or arising from extensive modification of natural ecosystems such that they function very differently. These activities are ultimately driven by socio-economic and cultural-spiritual processes that operate across local to global scales of human organization. In many agricultural and aquacultural systems and some others, cessation of those activities may lead to transformation into ecosystem types with qualitatively different properties and organizational processes (see refs. [53,54] for cropland and urban examples, respectively). Indices such as human appropriation of net primary productivity[55], combined with land-use maps[56], offer useful insights into the distribution of some anthropogenic ecosystems, but further development of indices is needed to adequately represent others, particularly in marine, and freshwater environments. Beyond land-use classification and mapping approaches (Supplementary Information, Appendix 1, page 6), a more comprehensive elaboration of the intensity of human influence underpinning the diverse range of anthropogenic ecosystems requires a multidimensional framework incorporating land-use inputs, outputs, their interactions, legacies of earlier activity and changes in system properties[17].

Where less intense human activities occur within non-anthropogenic ecosystem types, we focused descriptions on low-impact reference states. Therefore, human activities are not shown as drivers in the assembly models for non-anthropogenic ecosystem groups, even though they may have important influences on the contemporary ecosystem distribution. This approach enables the degree and nature of human influence to be described and measured against these reference states using assessment methods such as the Red List of Ecosystems protocol[30], with appropriate data on ecosystem change.

## Indicative distribution maps

Finally, to produce spatially explicit representations of the units at level 3 of the typology (principle 5), we sought published global maps (sources in Supplementary Table 4.1) that were congruent with the concepts of respective ecosystem functional groups. Where several candidate maps were available, we selected maps with the closest conceptual alignment, finest spatial resolution, global coverage, most recent data and longest time series. The purpose of maps for our study was to visualize global distributions. Prior to applications of map data to spatial analysis, we recommend critical review of methods and validation outcomes reported in each data source to ensure fitness for purpose (Supplementary Information, Appendix 4).

Extensive searches of published literature and data archives identified high-quality datasets for some ecosystem functional groups (for example, T1.3 Tropical–subtropical montane rainforests; MT1.4 Muddy shorelines; M1.5 Sea ice) and datasets that met some of these requirements for a number of other ecosystem functional groups (see Supplementary Table 4.1 for details). Where evaluations by authors or reviewers identified limitations in available maps, we used global environmental data layers and biogeographic regionalizations as masks to adjust source maps and improve their congruence to the concept of the relevant functional group (for example, F1.2 Permanent lowland rivers). For ecosystem functional groups with no specific global mapping, we used ecoregions[50,57,58] as biogeographic templates to identify broad areas of occurrence. We consulted ecoregion descriptions, global and regional reviews, national and regional ecosystem maps, and applied in situ knowledge of participating experts to identify ecoregions that contain occurrences of the relevant ecosystem functional group (for example, T4.4 Temperate woodlands) (see Supplementary Table 4.1 for details). We mapped ecosystem functional groups as major occurrences where they dominated a landscape or seascape matrix and minor occurrences where they were present, but not dominant in landscape–seascape mosaics, or where dominance was uncertain. Although these two categories in combination communicate more information about ecosystem distribution than binary maps, simple spatial overlays using minor occurrences are likely to inflate spatial statistics. The maps are progressively upgraded in new versions of the typology as explicit spatial models are developed and new data sources become available (see ref. [27] for a current archive of spatial data).

The classification and descriptive profiles, including maps, for each functional biome and ecosystem functional group underwent extensive consultation, and targeted peer review and revision through a series of four phases described in Supplementary Information, Appendix 5 (pages 2–4). The reviewer comments and revisions from targeted peer review are documented in Supplementary Table 5.1. In all, more than 100 ecosystem specialists have contributed to the development of v2.1 of the typology.

## Limitations

Uneven knowledge of Earth's biosphere has constrained the delimitation and description of units within the typology. There is a considerable research bias across the full range of Earth's ecosystems, with few formal research studies evaluating the relative influence of different ecosystem drivers in many of the functional groups, and abiotic assembly filters generally receiving more attention than biotic and dispersal filters. This poses challenges for developing standardized models of assembly for each ecosystem functional group. The models therefore represent working hypotheses, for which available evidence varies from large bodies of published empirical evidence to informal knowledge of ecosystem experts and their extensive research networks. Large numbers of empirical studies exist for some forest functional groups, savannas, temperate heathlands in Mediterranean-type climates, coral reefs, rocky shores, kelp forests, trophic webs in pelagic waters, small permanent freshwater lakes, and others (see references in the respective profiles (Supplementary Information, Appendix 4)). For example, Bond[49] reviewed empirical and modelling evidence on the assembly and function of tropical savannas that make up three ecosystem functional groups, showing that they have a large global biophysical envelope that overlaps with tropical dry forests, and that their distribution and dynamics within that envelope is strongly influenced by top-down regulation via biotic filters (large herbivores and their predators) and recurrent disturbance regimes (fires). Despite the development of this critical knowledge base, savannas suffer from an awareness disparity that hinders effective conservation and management[59]. In other ecosystems, our assembly models rely more heavily on inferences and generalizations of experts drawn from related ecosystems, are more sensitive to interpretations of participating experts, and await

empirical testing and adjustment as understanding improves. Empirical tests could examine hypothesized variation in ecosystem properties along gradients within and between ecosystem functional groups and should return incremental improvements on group delineation and description of assembly processes.

High-quality maps at suitable resolution are not yet available for the full set of ecosystem functional groups, which limits current readiness for global analysis. The maps most fit for global synthesis are based on remote sensing and environmental predictors that align closely to the concept of their ecosystem functional group, incorporate spatially explicit ground observations and have low rates of omission and commission errors, 'high' spatial resolution (that is, rasters of 1 km$^2$ (30 arcsec) or better), and time series of changes. Sixty of the maps currently in our archive[27] aligned directly or mostly with the concept of their corresponding ecosystem functional group, while the remainder were based on indirect spatial proxies, and most were derived from polygon data or rasters of 30 arcsec or finer (Supplementary Table 4.1). Maps for 81 functional groups were based either on known records, or on spatial data validated by quantitative assessments of accuracy or efficacy. Therefore, we suggest that maps currently available for 60–80 of the 110 functional groups are potentially suitable for global spatial analysis of ecosystem distributions. Although, a significant advance on broad proxies such as ecoregions, the maps currently available for ecosystem functional groups would benefit from expanded application of recent advances in remote sensing, environmental datasets, spatial modelling and cloud computing to redress inequalities in reliability and resolution. The most urgent priorities for this work are those identified in Supplementary Table 4.1 as relying on indirect proxies for alignment to concept, qualitative evaluation by experts and coarse resolution (>1 km$^2$) spatial data.

## Reporting summary

Further information on research design is available in the Nature Research Reporting Summary linked to this article.

## Data availability

Descriptions, images and interactive maps for the typology are updated periodically at https://global-ecosystems.org/. The spatial data for this study are available at Zenodo (https://doi.org/10.5281/zenodo.3546513).

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

**Acknowledgements** The PLuS Alliance supported a workshop in London to initiate development. D.A.K., E.N., R.T.K., J.R.F.-P., J.A.R. and N.J.M. were supported by ARC Linkage Grants LP170101143 and LP180100159 and the MAVA Foundation. E.N. was supported by ARC Future Fellowship FT190100234. The IUCN Commission on Ecosystem Management supported travel for D.A.K. to present aspects of the research to peers and stakeholders at International Congresses on Conservation Biology in 2017 and 2019, and at meetings in Africa, the Middle East and Europe. E. Woischin drafted Fig. 2. We thank the many specialist contributors to the typology listed in Supplementary Information, Appendices 4 and 5.

**Author contributions** D.A.K. conceived the project and led development of the conceptual model and typology structure with input from E.N., J.R.F.-P., M.J.B., B.A.P., M.G.T., J.L.N., E.J.G., F.E., D.F.-L., D.J.R., N.A.B., I.D., L.J.J., R.T.P., N.P., A.L., M.A.C., J.A.R., N.J.M., J.M., P.P., I.Z. and R.T.K. Descriptive profiles of functional groups were contributed by D.A.K., R.T.K., M.J.B., E.R.-L., M.G.T., B.A.P., K.E.W., R.M.N., F.E., D.F.-L., A.E., J.L.N., C.E.R.L., R.T.P., J.F., U.C.F.-A., T.M.I., V.G., P.G., B.J.R., D.J.R., J.S.S., N.J.M., N.A.B., I.M.S., S.K.W., L.J.J., A.L., T.T., A.T., J.M. and P.P., and others as listed in Supplementary Information, Appendix 4. E.N. managed reviews of profiles. Policy implications were contributed by D.A.K., E.N., A.A. and N.P. All authors contributed to manuscript preparation, led by D.A.K.

**Competing interests** The authors declare no competing interests.

**Additional information**
**Correspondence and requests for materials** should be addressed to David A. Keith.

# Extended Data Table 1 | Key features of Ecosystem Functional Groups In The Terrestrial Realm And The Terrestrial-freshwater transitional realm of the IUCN Global Ecosystem Typology v2.1

| Ecosystem Functional Group | Typical Key features * | Distribution |
|---|---|---|
| **T1. Tropical-subtropical forests biome** | | |
| T1.1 Tropical/Subtropical lowland rainforests | Tall closed-canopy evergreen forests in warm wet climates, phylogenetically & functionally highly diverse life forms | Global wet tropics & subtropics |
| T1.2 Tropical/Subtropical dry forests and thickets | Closed-canopy deciduous and semi-deciduous forests in warm seasonally wet/dry climates, diverse life forms | Global wet/dry tropics & subtropics |
| T1.3 Tropical/Subtropical montane rainforests | Closed-canopy evergreen forests with abundant non-vascular epiphytes in warm/cool wet cloudy climates, diverse life forms | Global tropical & subtropical mountains |
| T1.4 Tropical heath forests | Low closed-canopy evergreen forests in warm wet climates on low-nutrient substrates, structurally simple cf T1 | Amazon basin, southeast Asia, possibly Congo basin |
| **T2. Temperate-boreal forests and woodlands biome** | | |
| T2.1 Boreal and temperate high montane forests and woodlands | Closed to open, evergreen (conifers) or deciduous forests in cold climates with short growth periods, low vascular plant species diversity, but abundant cryptogams | Cool regions (boreal zone or mountains in temperate or mediterranean regions) of the Northern Hemisphere, limited occurrences in southern South America |
| T2.2 Deciduous temperate forests | Closed canopy broadleaved forests in seasonally warm and cold humid climates, with low to moderate woody species diversity | Temperate regions of the Northern Hemisphere, limited occurrences in southern South America |
| T2.3 Oceanic cool temperate rainforests | Closed canopy evergreen or semi-deciduous forests in cool wet climates, high endemism with low tree diversity and abundant epiphytes | Cool temperate coasts of Chile, Patagonia, New Zealand, Tasmania and Pacific Northwest |
| T2.4 Warm temperate laurophyll forests | Simple, closed-canopy mostly evergreen forests in warm environments with modest summer rainfall deficits; moderate diversity and endemism | Patchy warm temperate-subtropical distribution at 26–43° latitude, north or south of the Equator |
| T2.5 Temperate pyric humid forests | Tall, moist and complex multi-layered forests in wet-temperate climates; characterised by sclerophyll dominant trees and diverse mesophyll understorey; population processes driven by fire regimes | Subtropical - temperate southeast and temperate southwest Australia |
| T2.6 Temperate pyric sclerophyll forests and woodlands | Sclerophyll forests and woodlands in warm climates with winter precipitation and a canopy-fire regime | Temperate regions of Australia, the Mediterranean, and California |
| **T3. Shrublands and shrubby woodlands biome** | | |
| T3.1 Seasonally dry tropical shrublands | Mostly evergreen, sclerophyll shrublands on nutrient-poor soils, C4 grasses can be important | Global seasonally-dry tropics : South America, Australia, oceanic high islands |
| T3.2 Seasonally dry temperate heath and shrublands | Sclerophyll evergreen shrublands of humid and subhumid mid-latitudes with a canopy-fire regime | Temperate regions adjacent to cold ocean currents with summer dry season |
| T3.3 Cool temperate heathlands | Low-diversity, low productivity mixed graminoid ericoid shrublands of maratime environments, supporting mammalian browsers | Boreal and cool temperate coasts, North America, Europe, Magellenic South America |
| T3.4 Young rocky pavements, lava flows and screes | Low-diversity cryptogam-dominated systems with scattered herbs and shrubs on skeletal substrates with limited nutrients and moisture | Around the Pacific Rim, African Rift Valley, Mediterranean and north Atlantic |
| **T4. Savannas and grasslands biome** | | |
| T4.1 Trophic savannas | Grassy woodlands and grasslands dominated by C4 grasses in seasonal climates with lower rainfall and higher soil fertility. | African and Asian wet/dry tropics & subtropics |
| T4.2 Pyric tussock savannas | Grasslands and grassy woodlands dominated by C4 tussock grasses. Strong seasonal (winter) drought, low fertility, and fires major consumer of biomass. | Global wet/dry tropics & subtropics |
| T4.3 Hummock savannas | Sparse to open low-productivity woodlands in nutrient poor often rocky landscapes with C4 hummock grasses, rich reptile fauna, abundant termites, moderate herbivore densities and irregular fires. | Restricted to northern Australia in the wet-dry and semi-arid tropics. |
| T4.4 Temperate woodlands | Open-canopy woodlands, trees microphyll and evergreen, with herbaceous understory including C3 and/or C4 grasses | Temperate regions worldwide with summer water deficit, some with winter precipitation |
| T4.5 Temperate subhumid grasslands | Tussock grasslands with mixtures of C3 and C4 grasses and interstitial forbs, high productivity and complex trophic networks | Temperate regions worldwide with summer water deficit, aseasonal precipitation that along with temperatures are lower than T4.4 |
| **T5. Deserts and semi-deserts biome** | | |
| T5.1 Semi-desert steppe | Low-productivity and low-stature shrublands, tussock-grass and mixed, with episodic trophic pulses driven by variable rainfall | Global temperate-arid regions with high temperatures and low and variable precipitation |
| T5.2 Succulent or Thorny deserts and semi-deserts | Characterized by tall succulent plants, diverse annuals and geophytes, supporting diverse mammals, reptiles and invertebrates | Subtropical latitudes of the Americas, southern Africa and southern Asia |
| T5.3 Sclerophyll hot deserts and semi-deserts | Perennial sclerophyll shrubs and Hummock C4 grasses on nutrient-poor soils; highly variable rainfall, high diversity and endemism | Central Australia on sandy substrates; extremely arid with hot summers and cool winters. |
| T5.4 Cool deserts and semi-deserts | Xeromorphic suffrutescent or non-sclerophyll shrublands or grasslands; freezing temperatures in winter low, rainfall offset by reduced evapotranspiration burdon; low diversity and endemism | Cool temperate plains and plateaus from sea level to 4,000 m elevation in central Eurasia, western North America, and Patagonia. Extreme cold deserts are placed in the polar/alpine biome |
| T5.5 Hyper-arid deserts | Very sparsely vegetated ecosystems in areas with very low or no precipitation; very low productivity and simple trophic structures; low diversitybut high endemism | Driest parts of the Sahara-Arabian, Atacama, and Namib deserts in subtropical latitudes |
| **T6. Polar/alpine (cryogenic) biome** | | |
| T6.1 Ice sheets, glaciers and perennial snowfields | Permanent, dynamic ice cover where extreme cold limits productivity and diversity, biota dominated by microorganisms, migratory/overwintering birds may occur | Polar regions and high mountains in the western Americas, central Asia, Europe, and New Zealand |
| T6.2 Polar/alpine cliffs, screes, outcrops and lava flows | Environments free of permanent ice where extreme cold, winds, skeletal substrates and periodic mass movement limit biota to cryptogams, invertebrates and microorganisms, nesting birds may occur. | Permanently ice-free areas of Antarctica, Greenland, the Arctic Circle, and high mountains in the western Americas, central Asia, Europe, Africa and New Zealand. |
| T6.3 Polar tundra and deserts | Open and low vegetation of herbaceous plants (e.g. tussocks, cushions, rosette plants) and abundant kryptogams in very cold climates with permafrost | Locally in northern Europe (Scandinavia, Russia), northern Siberia and North America |
| T6.4 Temperate alpine grasslands and shrublands | Mountain systems above the physiological limits of trees, with sparse to continuous cover of herbaceous plants, cryptogams and dwarf shrubs that may be morphologically adapted to extreme cold. | Ttemperate and boreal zones of the Americas, Europe, central Eurasia, west and north Asia, Australia, and New Zealand |
| T6.5 Tropical alpine grasslands and herbfields | Dense perennial C3 cold tolerant tussock grasslands, with distinctive arborescent rosette and cushion growth forms, treeless except for sheltered gullies. | High mountain tops of tropics |
| **T7. Intensive land-use biome** | | |
| T7.1 Annual croplands | Structurally simple, very low- diversity, high-productivity annual croplands are maintained by the intensive anthropogenic supplementation of nutrients, water and artificial disturbance regimes | Tropical to temperate humid climatic zones or river flats in dry climates across south sub-Saharan and North Africa, Europe, Asia, southern Australia, Oceania, and the Americas. |
| T7.2 Sown pastures and fields | Structurally simple, very low- diversity, high-productivity grasslands dominated by one or few species of perennial grasses (Poaceae) maintained by intensive addition of nutrients, water and artificial disturbance regimes (mowing or grazing) | Abundant in humid or sub-humid, boreal to tropical climates worldwide |
| T7.3 Plantations | Structurally simple, low-diversity forests of one (rarely, a few) planted tree species of mostly same age, lack of structural elements of old-growth forests such as deadwood or cavities | Abundant in humid or sub-humid, boreal to tropical climates worldwide |
| T7.4 Urban and industrial ecosystems | Ecosystems dominated by anthropogenic structures (e.g. buildings, roads, wastelands) associated with human infrastructures, intensive anthropogenic disturbance regimes, and severely altered biogeochemical site conditions | Abundant worldwide in all regions settled by humans |
| T7.5 Derived semi-natural pastures and old fields | Extensively used, low-input grasslands (no or moderate fertilizer application, no sowing), rich in vascular plant species | In humid or sub-humid, boreal to tropical climates worldwide, mostly in regions with long agricultural tradition (e.g. Europe, western Asia) |
| **TF1. Palustrine wetlands biome** | | |
| TF1.1 Tropical flooded forests and peat forests | Evergreen closed-canopy forests in tropical swamps and riparian zones, differing between high and low nutrients waters, and supporting complex trophic networks | Equatorial lowlands of Southeast Asia, South America and Central and West Africa |
| TF1.2 Subtropical/temperate forested wetlands | Permently to seasonally wet (or flooded), nutrient poor, to nutrient rich, open to closed canopy forests, often on organic soils (peat); poor in woody species, high abundance of mosses and sedges and no to open woody species cover | Subtropical to temperate regions of both hemispheres, mostly in humid climates |
| TF1.3 Permanent marshes | Shallow permanently inundated freshwater wetlands, dominated by herbaceous macrophytes, supporting high primary productivity and complex trophic networks with abundant insects, birds and amphibians | Mainly on floodplains in catchments with humid tropical or temperate climates |
| TF1.4 Seasonal floodplain marshes | High productivity wetlands with strongly seasonal water regimes, supporting functionally diverse mosaics of aquatic plants and seasonally variable trophic networks of invertebrates, amphibians, crocodilians and birds | Seasonal tropics and subhumid temperate regions |
| TF1.5 Episodic arid floodplains | Highly productive floodplains when flooded, supporting highly diverse and complex trophic networks, followed by long periods of low productivity when dry | Semi-ard and arid regions |
| TF1.6 Boreal, temperate and montane peat bogs | Permanently ground water-logged (by rainwater-fed ground water,) nutrient poor, acidic sites on organic soils (peat); species poor, but high abundance of mosses, sedges and no to open woody species cover | Boreal and temperate humid zones of the northern hemisphere, limited occurrences in the southern hemisphere (southern South America, southern Australasia) |
| TF1.7 Boreal and temperate fens | Permanently groundwater-logged, nutrient poor to (moderately) nutrient-rich sites, often organic soils; high abundance of mosses, sedges and no to open woody species cover | Boreal and temperate zones of the northern hemisphere, limited occurrences in the southern hemisphere (southern South America, southern Australasia, possibly South Africa) |

See Appendix S3 for further details of typology structure and Appendix S4 for descriptions of functional biomes (Level 2) and Ecosystem Functional Groups (Level 3).

**Extended Data Table 2 | Key features of Ecosystem Functional Groups in the Subterranean realm, Subterranean-Freshwater transitional realm and Subterranean-Marine transitional realm of the IUCN Global Ecosystem Typology v2.1**

| Ecosystem Functional Group | Typical Key features * | Distribution |
|---|---|---|
| **S1. Subterranean lithic biome** | | |
| S1.1 Aerobic caves | Dark dry or humid geological cavities with microbial chemoautotrophs, detrivores, decomposers, endemic invertebrates & no photoautotrophs | Scattered globally throughout land masses |
| S1.2 Endolithic systems | Microbial systems within lithic matrices and interstitial spaces with truncated trophic networks founded on lithautotrophs and lacking photoautotrophs (except near surface) and high-order predators. | Throughout the Earth's crust to depths of 4-7 km |
| **S2. Anthropogenic subterranean voids biome** | | |
| S2.1 Anthropogenic subterranean voids | Dry or humid subterranean voids created by mining or infrastructure development and colonised by opportunistic microbes, invertebrates and sometimes vertebrates. | Associated with urban and industrial infrastructure worldwide |
| **SF1. Subterranean freshwaters biome** | | |
| SF1.1 Underground streams and pools | Water-filled subterranean voids with low diversity of light-limited bacteria, fungi, detrivores and predators. | Scattered lobally in limestone or more rarely basalt or other lithic substrates |
| SF1.2 Groundwater ecosystems | Saturated ecosystems at or below the watertable with low diversity communities of heterotrophic microbes and invertebrates | Scattered globally throughout land masses |
| **SF2. Anthropogenic subterranean freshwaters biome** | | |
| SF2.1 Water pipes and subterranean canals | Artificial flowing waterbodies that carry water with variable flow regime, limited light, sometimes with high carbon and nutrients supporting opportunities aquatic detritivores and predators | Ubiquitous in developed regions of the world, most commonly in urban landscapes and irrigation areas |
| SF2.2 Flooded mines and other voids | Underground largely static low-productivity waterbodies often with large of warm groundwater or seepage, colonised by opportunistic microbes and invertebrates | Common in mineral rich regions of the world |
| **SM1. Subterranean tidal biome** | | |
| SM1.1 Anchialine caves | Cave-bound waterbodies connected to the sea with a gradient of tidal influence and salinity. Filter feeders, scavengers and predators limited by light and nutrients | Limestone, basalt and more rarely lithic substrates coastal regions globally |
| SM1.2 Anchialine pools | Open pools with subterranean connections to the sea and groundwater, and dynamic, diverse trophic networks | Limestone, basalt and more rarely lithic substrates coastal regions globally |
| SM1.3 Sea caves | Wave-exposed caves provide dim light and shelter to cave-exclusive, resident and transient/ migratory invertebrates and fish. | Coastal headlands, rocky and coral reefs globally |

See Appendix S3 for further details of typology structure and Appendix S4 for descriptions of functional biomes (Level 2) and Ecosystem Functional Groups (Level 3).

**Extended Data Table 3 | Key features of Ecosystem Functional Groups in the Freshwater realm and Freshwater-Marine transitional realm of the IUCN Global Ecosystem Typology v2.1**

| Ecosystem Functional Group | Typical Key features * | Distribution |
|---|---|---|
| **F1. Rivers and streams biome** | | |
| F1.1 Permanent upland streams | High-medium velocity, low-medium volume perennial flows with abundant benthic filter feeders, algal biofilms & small fish | Global uplands with wet climates |
| F1.2 Permanent lowland rivers | Low-medium velocity, high volume, perennial flows with abundant zooplankton, fish, macrophytes, macroinvertebrates & piscivores | Global lowlands fed by wet uplands |
| F1.3 Freeze-thaw rivers and streams | Cold-climate streams with seasonally frozen surface water and variable melt flows and aquatic biota with cold-resistance and/or seasonal dormancy | High latitudes and/or high mountains, especially boreal regions |
| F1.4 Seasonal upland streams | High-medium velocity, low-medium volume, highly seasonal flows with abundant benthic filter feeders, algal biofilms & small fish | Extensive in wet-dry tropics and temperate zones |
| F1.5 Seasonal lowland rivers | Highly productive large rivers with seasonal hydrology large floodplain subsidies. Short food chains support large mobile predaors | Tropical, subtropical and temperate lowlands |
| F1.6 Episodic arid rivers | Rivers with high temporal flow variability which determines periods of high and low productivity, supporting high levels of biodiversity and complex trophic networks during floods and simple trophic networks during dry periods | Arid and semi-arid landscapes in mid-latitudes mostly in lowlands |
| F1.7 Large lowland rivers | Large highly productive rivers with megaflow rates and complex food webs, reflecting the extent of habitat, connections with floodplains and available niches for plants, invertebrates and large vertebrates including aquatic mammals. | Tropical and subtropical lowlands, with some in temperate regions with large catchments topped by wet mountain ranges |
| **F2. Lakes biome** | | |
| F2.1 Large permanent freshwater lakes | Large (usually >100km2) permanent freshwater lakes connected to rivers, with high spatial and bathymetric niche diversity supporting complex trophic networks supported by planktonic algae, high diversity and endemism | Humid temperate and tropical regions |
| F2.2 Small permanent freshwater lakes | Small permanent freshwater lakes or ponds with niche diversity strongly related to size and depth, and resource subsidies from catchments. Littoral zones and benthic macrophytes are important contributors to productivity | Predominantly in humid temperate and tropical regions |
| F2.3 Seasonal freshwater lakes | Mostly small and shallow well mixed freshwater lakes with seasonal patterns of filling and seasonally variable abundance and composition of aquatic biota, including species with dormant life phases and some that retreat to refuges in dry seasons | Mainly subhumid temperate (including Mediterranean-type climate zones) and wet-dry tropical regions |
| F2.4 Freeze-thaw freshwater lakes | Waterbodies with frozen surfaces for at least one month of the year, with spring thaw initiating trophic successional dynamics beginning with a flush of diatom productivity. Deeper lakes may be cold stratified and fish tolerate oxygen depletion in winter | Boreal regions, cool temperate continental Eurasia and North America and high altitudes of South America regions |
| F2.5 Ephemeral freshwater lakes | Shallow temporary lakes, depressions or pans with long dry periods of low productivity, punctuated by episodes of inflow that bring large resource subsidies from catchments, resulting in high productivity, population turnover and trophic connectivity | Semi-arid and arid regions at mid latitudes of Africa, southern Australia, Eurasia, Europe and western parts of North and South America |
| F2.6 Permanent salt and soda lakes | Permanent waterbodies with high inorganic solute concentrations (particularly sodium), supporting simple trophic networks, including cyanobacteria and algae, invertebrates and specialist birds | Mostly in semi-arid regions of Africa, southern Australia, Eurasia, Europe and western North and South America |
| F2.7 Ephemeral salt lakes | Salt lakes with salt crusts in long dry phases and short productive wet phases. Trophic networks are simple but high productivity is driven by bacteria and phytoplankton, supporting specialist birds | Mostly in arid and semi-arid Africa, Eurasia, Australia and North and South America |
| F2.8 Artesian springs and oases | Groundwater dependent ecosystems from artesian waters discharged to the surface, maintaining relatively stable water levels. Often insular systems with high endemism | Mostly in arid regions in Africa, the Middle East, central Eurasia, southwest of North America and Australia's Great Artesian basin |
| F2.9 Geothermal pools and wetlands | Hot springs, geysers and mud pots dependent on groundwater interactions with magma and hot rocks, supporting highly specialised low diversity biota tolerate of high temperatures and high concentrations of inorganic salts | Tectonically or active volcanic areas from the tropical to subpolar latitudes |
| F2.10 Subglacial lakes | Lakes beneath permanent ice sheets with a truncated microbial food web, including chemoautotrophic and heterotrophic of bacteria and archaea | Antarctica, Greenland, Iceland and Canada |
| **F3. Artificial wetlands biome** | | |
| F3.1 Large reservoirs | Large, usually deep stratifed waterbodies impounded by walls across outflow channels. Productivity and biotic diversity are lower than unregulated lakes of simila rsize and complexity. Trophic networks are simple | Scattered across all continents with high concentrations in Asia, Europe and North America |
| F3.2 Constructed lacustrine wetlands | Small, shallow open waterbodies with high or low productivity depending on nutrient subsidies and complexity of littoral zones and benthos Relatively simple trophic networks with algae, macrophytes, zooplankton, aquatic invertebrates and amphibians | Scattered across all regions of the world |
| F3.3 Rice paddies | Artificial wetlands with limited horizontal and vertical heterogeneity, filled seasonally with water from rivers or rainfall and frequently disturbed by planting and harvest of rice. Simple trophic networks with colonists from rivers and wetlands that may also include managed fish populations | Mostly in tropical and subtropical southeastand south Asia, also in Africa, Europe, South America, North America and southeast Australia |
| F3.4 Freshwater aquafarms | Artificial mostly permanent waterbodies managed for production of fish or crustaceans with managed inputs of nutrients and energy Simple trophic networks of opportunistic colonists supported mainly be algal productivity | Mostly in Asia but also in northern and western Europe, North and West Africa, the Americas, and southeast Australia and New Zealand |
| F3.5 Canals, ditches and drains | Artificial streams often with low horizontal and vertical heterogeneity, but with productivity, diversity and trophic structure highly dependent on fringing vegetation and subsidies of nutrients and carbon from catchments | In urban and irrigation landscapes mostly in temperate and subtropical latitudes |
| **FM1. Semi-confined transitional waters biome** | | |
| FM1.1 Deepwater coastal inlets | Strong gradients between adjacent terrestrial and freshwater systems,e.g. fjords. Seasonaly abundant plankton, jellies, fish and mammals. | Glaciated coastlines (current or historical) in polar or cool-temperate regions |
| FM1.2 Permanently open riverine estuaries and bays | Productive mosaic systems with variable salinity, often nuseries for fish and supporting abundant seabirds and mammals. | Coastlines globally |
| FM1.3 Intermittently closed and open lakes and lagoons | Shallow water systems, highly variability depending on opening or closing of lagoonal entrance. Detritus-based foodwebs with plankton, invertebrates and small fish. | Wave-dominated coastlines globally |

See Appendix S3 for further details of typology structure and Appendix S4 for descriptions of functional biomes (Level 2) and Ecosystem Functional Groups (Level 3).

**Extended Data Table 4 | Key features of Ecosystem Functional Groups in the Marine realm, Marine-Terrestrial transitional realm and Marine-Freshwater-Terrestrial transitional realm of the IUCN Global Ecosystem Typology v2.1**

| Ecosystem Functional Group | Typical Key features * | Distribution |
|---|---|---|
| **M1. Marine shelf biome** | | |
| M1.1 Seagrass meadows | Soft, mostly subtidal substrates in low-energy waters with abundant vascular macrophytes, associated epibiota, infauna and fish | Shallow tropical- temperate nearshore waters |
| M1.2 Kelp forests | Hard subtidal substrates in cold, clear nutrient-rich waters with dominant brown algal macrophytes, associated epibiota, benthic macrofauna, fish & mammals | Cool temperate coastal waters or regions receiving cold currents |
| M1.3 Photic coral reefs | Biogenic reefs formed by hard coral-algal symbionts with phylogentically & functionally diverse biota in clear, warm subtidal waters | Warm tropical & subtropical coastal waters |
| M1.4 Shellfish beds and reefs | Intertidal or subtidal three-dimensional stuctures, formed primarily by oysters and mussels, and supporting algae, invertebrates and fishes. | Tropical to temperate estuarine and coastal waters |
| M1.5 Photo-limited marine animal forests | Largely heterotrophic systems dominated by megabenthic suspension feeders and associated diverse epifauna, microphytobenthos and fish | Low light tropical to polar coastal waters |
| M1.6 Subtidal rocky reefs | Productive systems with functionally diverse sessile and mobile biota, and a strong depth gradient | Continental and island shelves |
| M1.7 Subtidal sand beds | Medium to coarse-grained soft sediment with burrowing invertebrate detrivores and suspension-feeders mostly relying on allochthonous energy. | Continental and island shelves |
| M1.8 Subtidal mud plains | Soft sediment with limited primary production, abundant micro- and macro-detritivores and associated foraging predators | Low energy waters of continental and island shelves |
| M1.9 Upwelling zones | Cool, wind-driven systems with high productivity and variability, supporting abundant plankton, fish, mammals and seabirds | Coastal eastern-boundary current systems and some localised areas in open oceans |
| M1.10 Rhodolith/Maërl beds | Biogenic beds formed by non-geniculate (non-jointed), free-living coralline algae on soft substrates supporting diverse benthic and demersal fauna and bacterial biofilms | Continental and island shelves at depths up to 270 m from the subtropics to subpolar waters |
| **M2. Pelagic ocean waters biome** | | |
| M2.1 Epipelagic ocean waters | Uppermost euphotic ocean, where phytoplankton production supports abundant mobile zooplankton, fish, cephalopods, mammals and seabirds | Surface layer of the open ocean |
| M2.2 Mesopelagic ocean water | Dimly lit 'twilight' zone below the epipelagic with a high biomass of diverse detrivores and predators and where bioluminescence is common | Oceans between ~200m depth/where <1% of light penetrates, down to 1000m. |
| M2.3 Bathypelagic ocean waters | Lightless, high pressure depths where adapted zooplankton, crustaceans, jellies, cephalopods and fish rely on nutrients falling from above | Deep oceans between 1000 - 3000m |
| M2.4 Abyssopelagic ocean waters | Lightless, high pressure depths with limited nutrients and low biodiversity of adapted detrivores, jellies, scavengers and predatory fish | Deep oceans between 3000 - 6000m |
| M2.5 Sea ice | Highly dynamic, seasonally frozen surface waters support diverse ice-associated organisms from plankton to seabirds and whales | Polar oceans |
| **M3. Deep sea floors biome** | | |
| M3.1 Continental and island slopes | Large sedimentary, aphotic, and heterotrophic slopes where depth gradients result in a bathymetric faunal zonation of high taxonomic diverstiy. | Continental slopes from shelf break (~250 m) to abyssal basins (4000 m) |
| M3.2 Submarine canyons | Dinamics and heterogenous geomorphic features, supporting highly diverse heterotrophic communities through enhaced transport of energy from the continents to the deep sea. | Submarine canyons incising continental margins globally |
| M3.3 Abyssal plains | Largest benthic heterotrophic system, mostly of fine sediment, supporing high biodiversity of small organisms (microbes, meio- and macro-fauna) | Seafloor between 3000 and 6000 m depth |
| M3.4 Seamounts, ridges and plateaus | Elevated geomorhic features with modified hydrography and heterogeneous habitat supporting high bnethic and pelagic productivity | Elevated rocky topographic features rising from deep seafloor |
| M3.5 Deepwater biogenic beds | Benthic sessile suspension feeders that crate structurally complex 3D habitat, supporting high biodiversity | Aphotic biogenic structures from benthic fauna |
| M3.6 Hadal trenches and troughs | Deepest ocean systems, poorly explored, mostly of fine nutrient-poor sediment dominated by scavangers and detritivors | Seafloor between 6000 and 11 000 m |
| M3.7 Chemosynthetic-based-ecosystems (CBE) | Systems supported by microbial chemoautotrophy with high biomass of relatively low diversity, highly speciliased, fauna | Hydrothermal vents, cold seeps, large organic falls on the deep seafloor |
| **M4. Anthropogenic marine biome** | | |
| M4.1 Submerged artificial structures | Hard surfaces of oil and gas infrastructure, artificial reefs and wrecks form habitat for sessile filter feeders, invertebrates and some reef fish. | Coastal waters globally |
| M4.2 Marine aquafarms | High density, productive, enclosed systems with variable permeability, for breeding and harvesting marine species. Allochthonous nutrients from human sources is common. | Largely coastal or shore-based, some open-ocean facilities |
| **MFT1. Brackish tidal biome** | | |
| MFT1.1 Coastal river deltas | Depositional, mosaic systems with strong gradients between terrestrial, freshwater and marine elements. Productive with diverse plankton, fish, birds and mammals. | Continental margins of high rainfall catchments globally |
| MFT1.2 Intertidal forests and shrublands | Intertidal mangrove-dominated systems, producing high amounts of organic matter that is both buried in situ and exported; sediments dominated by detritivores and leaf shredders, with birds , mammals, reptiles and terrestrial invertebrates occupying the canopy | Tropical and warm temperate coastlines with good sediment supply |
| MFT1.3 Coastal saltmarshes and reedbeds | Variable salinity tidal system dominated by salt-tolerant plants, with invertebrates, small/juvenile fish and birds. | Mostly low energy coasts from tropical to arctic and subantarctic latitudes |
| **MT1. Shorelines biome** | | |
| MT1.1 Rocky Shorelines | Hard intertidal substrate, dominated by sessile and mobile invertebrates, and macroalgae | High-energy shorelines globally |
| MT1.2 Muddy Shorelines | Intertidal soft-sediment, of fine particle-size, dependent on allochtonous production and dominated by deposit feeding and detritivorous invertebrates that provide a prey resource for shore birds and fishes | Low-energy shorelines globally |
| MT1.3 Sandy Shorelines | Intertidal soft-sediment, of large particle-size, lacking conspicuous macrophytes, and dominated by suspension-feeding invertebrates that provide a prey resource for shore birds and fishes | Medium-high energy shorelines, particularly at temperate latitudes |
| MT1.4 Boulder and cobble shores | Unstable intertidal hard substrate, that supports encrusting and fouling species at low elevations and in some instances vegetation, though largely dependent on allochtonous production | High-latitude shorelines receiving cobbles from rivers, glaciers or erosion of cliffs |
| **MT2. Supralittoral coastal biome** | | |
| MT2.1 Coastal shrublands and grasslands | Coastal scrub limited by salinity, water deficit and disturbances (e.g. cliff collapse). Strong gradients from sea to land and highly mobile fauna. | Coastal dunes and cliffs in tropical, temperate and boreal latitudes |
| MT2.2 Large seabird and pinniped colonies | Localised areas of bare or vegetated ground with diverse microbial communities at the ocean interface receiving massive nutrient subsidies and disturbance from large concentrations of roosting or nesting seabirds and pinnipeds that function as mobile links between land and sea | Scattered globally on islands and coastlines, but most common in polar and subpolar regions |
| **MT3. Anthropogenic shorelines biome** | | |
| MT3.1 Artificial shorelines | Coastal infrastructure, such as seawalls, breakwaters, pilings and piers, extending from the intertidal to subtidal, supporting cosmopolitan sessile and mobile invertebrates and macroalgae on their hard surfaces, and in some instances serving as artificial reefs for fish | Globally, along urbanised coastlines |

See Appendix S3 for further details of typology structure and Appendix S4 for descriptions of functional biomes (Level 2) and Ecosystem Functional Groups (Level 3).

# Reporting Summary

## Statistics

For all statistical analyses, confirm that the following items are present in the figure legend, table legend, main text, or Methods section.

| n/a | Confirmed | |
|---|---|---|
| ☒ | ☐ | The exact sample size (*n*) for each experimental group/condition, given as a discrete number and unit of measurement |
| ☒ | ☐ | A statement on whether measurements were taken from distinct samples or whether the same sample was measured repeatedly |
| ☒ | ☐ | The statistical test(s) used AND whether they are one- or two-sided<br>*Only common tests should be described solely by name; describe more complex techniques in the Methods section.* |
| ☒ | ☐ | A description of all covariates tested |
| ☒ | ☐ | A description of any assumptions or corrections, such as tests of normality and adjustment for multiple comparisons |
| ☒ | ☐ | A full description of the statistical parameters including central tendency (e.g. means) or other basic estimates (e.g. regression coefficient) AND variation (e.g. standard deviation) or associated estimates of uncertainty (e.g. confidence intervals) |
| ☒ | ☐ | For null hypothesis testing, the test statistic (e.g. *F*, *t*, *r*) with confidence intervals, effect sizes, degrees of freedom and *P* value noted<br>*Give P values as exact values whenever suitable.* |
| ☒ | ☐ | For Bayesian analysis, information on the choice of priors and Markov chain Monte Carlo settings |
| ☒ | ☐ | For hierarchical and complex designs, identification of the appropriate level for tests and full reporting of outcomes |
| ☒ | ☐ | Estimates of effect sizes (e.g. Cohen's *d*, Pearson's *r*), indicating how they were calculated |

*Our web collection on statistics for biologists contains articles on many of the points above.*

## Software and code

Policy information about availability of computer code

| | |
|---|---|
| Data collection | All data used in this study came from published sources as cited. Arc Map v 10.2.2, GRASS GIS v 7.4.0, PostGIS v 2.4.3 and Google Earth Engine were used to import, edit and curate input spatial data. |
| Data analysis | Table S4.1 details the assembly methods for thumbnail maps presented in descriptive profiles of Ecosystem Functional Groups in Appendix S4. GRASS GIS v 7.4.0, R statistical package v 3.6.1 and Python v 3.7.3 were used for data analysis. Detailed descriptions and code will also be available at: https://doi.org/10.5281/zenodo.6459843. Code for visualisation of the data in Earth Engine is available at: https://zenodo.org/record/6459698, users can also add the following repository to the Earth Engine Code Editor: https://code.earthengine.google.com/?accept_repo=users/jrferrerparis/IUCN-GET |

For manuscripts utilizing custom algorithms or software that are central to the research but not yet described in published literature, software must be made available to editors and reviewers. We strongly encourage code deposition in a community repository (e.g. GitHub). See the Nature Portfolio guidelines for submitting code & software for further information.

## Data

Policy information about availability of data

All manuscripts must include a data availability statement. This statement should provide the following information, where applicable:
- Accession codes, unique identifiers, or web links for publicly available datasets
- A description of any restrictions on data availability
- For clinical datasets or third party data, please ensure that the statement adheres to our policy

Profiles, diagrammatic assembly models and interactive maps are available at https://global-ecosystems.org. Permanent record of the current version of the profiles is available at  https://doi.org/10.5281/zenodo.6459844 (All versions available at https://doi.org/10.5281/zenodo.6459843). Permanent record of the current

# Field-specific reporting

Please select the one below that is the best fit for your research. If you are not sure, read the appropriate sections before making your selection.

☐ Life sciences  ☐ Behavioural & social sciences  ☒ Ecological, evolutionary & environmental sciences

For a reference copy of the document with all sections, see nature.com/documents/nr-reporting-summary-flat.pdf

# Ecological, evolutionary & environmental sciences study design

All studies must disclose on these points even when the disclosure is negative.

| | |
|---|---|
| Study description | Our study presents a new typology for Earth's ecosystems and reviews its strengths, weakness and recent and potential applications to conservation and sustainability from global to local scales. |
| Research sample | The typology was developed by consensus among the 41 authors and 55 reviewers, selected based on published expertise encompassing terrestrial, freshwater and marine ecosystems, as well as global synthesis. |
| Sampling strategy | Not applicable |
| Data collection | Spatial data were compiled from published sources (documented in Appendix S4) primarily by DAK, JRFP and NJM. |
| Timing and spatial scale | Most spatial data sets were published between years 2000 and 2021 (see Table S4.1 for full details of sources), spatial resolution was as published in original sources or else reclassified to 30 arc seconds to ensure clear representation in the thumbnail maps in Appendix S4. |
| Data exclusions | Not applicable |
| Reproducibility | Development, revision and update history for the typology and its units are fully documented. No experiments were undertaken. |
| Randomization | Not applicable |
| Blinding | Not applicable |

Did the study involve field work?  ☐ Yes  ☒ No

# Reporting for specific materials, systems and methods

We require information from authors about some types of materials, experimental systems and methods used in many studies. Here, indicate whether each material, system or method listed is relevant to your study. If you are not sure if a list item applies to your research, read the appropriate section before selecting a response.

### Materials & experimental systems

| n/a | Involved in the study |
|---|---|
| ☒ | ☐ Antibodies |
| ☒ | ☐ Eukaryotic cell lines |
| ☒ | ☐ Palaeontology and archaeology |
| ☒ | ☐ Animals and other organisms |
| ☒ | ☐ Human research participants |
| ☒ | ☐ Clinical data |
| ☒ | ☐ Dual use research of concern |

### Methods

| n/a | Involved in the study |
|---|---|
| ☒ | ☐ ChIP-seq |
| ☒ | ☐ Flow cytometry |
| ☒ | ☐ MRI-based neuroimaging |

