## [Peer Review File · Nature]

Manuscript Title: A function-based typology for Earth's ecosystems

Reviewer Comments & Author Rebuttals

Reviewer Reports on the Initial Version:

Referees' comments:

Referee #1 (Remarks to the Author):

First, I need to state that this is the most unusual paper I have ever been asked to review. That is neither good nor bad, but it is an important piece of context. This is really a book with many hundreds of pages. Ideally all of this material needs to be peer reviewed, and reviewed by a panel of perhaps a dozen or more experts, which is not going to happen in the typical Nature review turnaround window. It's not going to be read nor used like a typical Nature paper either. Most of its uses will dive into the extended appendices. And to be truly useful, GIS files will need to be published, which I did not see (but easily could have missed). This type of effort is more typical of an NGO report. For me it brings to mind the Millennium Ecosystem Assessment, the WWF Global 200 ecoregions, and the IUCN 105 habitats. The point is all of those were not really published as a peer reviewed paper in a journal. They went through massive peer review processes with dozens (hundreds in some cases) of reviewers via public comment periods. Then they were published as a report by an NGO. And then their ultimate uptake was very heavily dependent on the backing and resources as well as the public process of the NGO behind the report. As such, not knowing the provenance of this effort (at least I have not yet found it) is a serious shortcoming. Is there an NGO behind this effort? Who? What mass peer review has been done? What kind of buy-in to this proposed typology can we expect? It is my opinion that if Nature decides it wants to publish this, it will need to: a) find out what has already been done in peer review (and make all of this more apparent in the manuscript), and b) use a modified review process that uses more reviewers and longer time frames, c) publish this as a Review to allow more space. But one could very easily argue this should be handled in a more traditional fashion like other NGO reports, or like a book.

Alternatively it could be published as half a dozen separate papers published in long-form venues like QRB, AREES, Ecological Monographs, etc (although towards this last option I applaud the authors for not salami slicing and think the user community also benefits from everything in one place so I'm not really advocating for this approach)

Turning now to a more traditional review ...

This paper represents a massive effort to develop a rigorous, first-principles list of ecosystems to be used in global conservation management efforts. They then use these classifications to divide the globe up into a map of ~100 ecosystems. They then produce two main results (Fig 3 & 4) - 1) a current assessment of % protected vs human pressures and 2) the temporal trend in human pressure for each separate ecosystem. These maps directly to assessment of how well the world is meeting Aichi/CBD targets for differing ecosystem types/locations on the globe.

As already noted, the typology has peers/competitors so the concept itself is not novel. The

assessments could have been done (and at least partially have been done) using other typologies that are more familiar and have more buy-in and vetting. So the first question that arises is do we need another typology? and if so is this particular form the best? This is not a simple yes/no question. But some of my thoughts:

Pros

- a) this came out of a careful process that identified 6 major metrics of quality for a typology and evaluated 21 prior typologies (and found them all wanting against the totality of criteria)
- b) this typology attempts to be first principles looking at major ecosystem processes (Figure 1, but this figure is then customized to each of the 81 ecosystems identified).
- c) this typology is relatively rare in targeting both the ecosystem services view (exemplified by functional group classifications in DGVM models) and the biodiversity view (exemplified by the WWF focus on distinct zootas)

Cons

- d) mostly just relating to whether we need yet another typology. Introducing a new typology immediately outdates all existing results using other typologies and prevents long-term comparisons. On some level the burden of proof is on the authors to demonstrate we have better/different results than we could have gotten with the IUCN or WWF typologies. I think it is unlikely we get a fundamentally different story/outcome
- e) Appendix 3 that lists each of the 81 ecosystems over 145 pages is a gold mine of information but needs serious peer review by experts in each area. I found myself fascinated by the ecosystem-specific versions of Figure 1 given for each of the 81 ecosystems. This is a bold and useful statement. But of course I then immediately started questioning whether they were right and how much scientific evidence we had for each of these figures.
- f) adoption of this new typology is mostly going to depend on social factors (who it comes from, what kind of money is behind it) rather than scientific merit and this is not currently obvious to me. I do see a brief sentence in the Methods this came out of a working group, but more information is needed. Specifically, any ties to NGOs? Any NGOs lined up to adopt this? or is this purely a research effort?
- g) adoption of this typology will depend heavily on providing GIS (shapefile, raster) implementations of this for others to use and I did not see where this is done

Or to summarize, scientifically superior but not unprecedented, but socially connected and adept is probably more important and unclear to me.

Some other thoughts if this goes forward at Nature:

- 1) The WWF ecosystems was not assessed/compared that I could see. I know this is in some ways targeted at animals which makes it more narrow, but the authors included many plant-based functional grouping systems. WWF needs to be included
- 2) Providing GIS files needs to be done
- 3) The main paper needs to be reprioritized on allocation of space. Much less detail on levels of classification should be given (or pushed to Methods). More details need to go into human impacts

(not just pushed into Methods). More interpretation on management implications. Explain more why combining the ecosystem function and biodiversity perspective is important

4) There has to be a simple summary table that makes explicit what has been accomplished.

Something that looks like the table of contents of Appendix 3 - i.e. revealing the top 3 levels of the hierarchy breaking into 81 ecosystems, then providing columns for total (and/or percent) area, key processes, % protected, human impacts, change in human impacts. I know this might seem like a dry table. But it is the only way to make the main output of the project (the ecosystems) tangible. And no such summary exists anywhere right now (the table of contents of Appendix 3 is the closest). Space spent on this table is far more concrete and useful than Figure 2 and a page of describing hierarchical structure abstractly.

5) Combining functional and biodiversity perspectives is a claim of novelty and importance for this classification. To my eye the biodiversity perspective mostly shows up in the lower levels of the hierarchy (i.e. levels 4-6) which implies it is less important. Please address this. Also please give an example or two of ecosystems further broken out in to these biodiversity classifications. Can biodiversity be pulled back up into the table in #4 - e.g. estimated # of birds, mammals and perhaps plants in each of the 81 ecosystems? I think this last task, while some work, would greatly help with the claim of merging perspectives and with the use and uptake of this typology.

Referee #2 (Remarks to the Author):

This paper includes a substantial volume of work related to a new system for a global ecosystem classification. The authors introduce a typology for classifying global ecosystems, describe the process that was used to refine and develop a comprehensive classification across 6 hierarchical levels, and analyse the third level (Ecosystem Functional Groups) against the CBD Aichi biodiversity targets. My review covers these three steps.

A. The typology.

There is no question that the authors are correct in asserting that there is no global ecosystem typology of general applicability. There are good reasons for this – in particular ecosystems are not fixed units in space and time, but rather multi-dimensional and multi-scale dynamic units. A general ecosystem classification is therefore lacking for good reasons, and any ecosystem classification will have to be designed with a specific purpose in mind. One classification will not meet all needs. The review in Appendix S1 includes some typologies that are ideal for the purpose for which they were designed but clearly not for what the authors here are seeking. So the key issue in the paper must be to define the purpose here. The authors state the identified requirement is for an ecosystem classification that groups ecosystems with functionally similar responses relevant to evaluating progress with conservation assessments (Aichi targets) and sustainability goals (SDGs). They are seeking a classification that will allow predictions and metrics across all areas of the Earth in particular for these conservation and sustainability assessments. In fact the paper addresses the Aichi targets mainly, but includes a much broader requirement for the system to underpin predictions and generalisations about ecosystem responses to environmental change ... etc. (lines 65-68). In any case, it is not surprising then that there is no existing typology that meets the authors design requirement (line 100).

The authors lay out their design criteria in Figure 1 and Appendix 2. The system that they develop is described but not tested or defended. I cannot judge how well it meets its purpose. There seem to be some relatively arbitrary decisions made about the drivers (in the boxes).

One question is why the system includes human pressures as a driver defining the ecosystem type if the system is to be used to predict ecosystem responses to environmental change and management. If human pressures are already defining the ecosystem type then how can pressures be analysed independently?

What are the design criteria for a degraded ecosystem? How does the system reflect loss of function or services as a result of additional human pressures? Doesn't this require some process or dynamic model rather than a static hierarchical classification?

At the end of the paper, the authors list a series of other potential uses for the ecosystem typology including for environmental accounts and natural capital accounting, and for conservation planning. Surely the design criteria here would be different? For natural capital and environmental accounting there needs to be a human demand and ideally economic valuation of ecosystem service benefit flows. Conservation planning is usually based on biotic units of conservation importance (species, ecological communities, habitats). So this section seems to be somewhat at odds with the design criteria.

Overall, I find the objective of developing a typology for ecosystem mapping and prediction under changing drivers to be an important task and clearly a lot of thought and work has gone into it here. Some more justification of design decisions seems important given the very elaborate structure that emerges, especially considering the elements of ecosystem science and community ecology that are relevant. Appendix S2 and S3 describe the five driver boxes but does not explain why these were selected or what was rejected. What process was used to decide that this was the right classification?

B. The system

The system is described and developed in detail in Appendix 4, including the detailed description of all the functional groups. I cannot review this section, and in fact its provenance is unclear as there are other citations. Is this being published elsewhere in whole or in part?

C. The analysis

Having developed and described the system, the authors use it to examine progress with two Aichi targets, numbers 5 and 11. This analysis presented in Figure 3 and 4 are the main research findings from the paper. According to the CBD, Target 5 is "By 2020, the rate of loss of all natural habitats, including forests, is at least halved and where feasible brought close to zero, and degradation and fragmentation is significantly reduced", and Target 11 is "By 2020, at least 17 per cent of terrestrial and inland water, and 10 per cent of coastal and marine areas, especially areas of particular importance for biodiversity and ecosystem services, are conserved through effectively and equitably managed, ecologically representative and well connected systems of protected areas and other

effective area-based conservation measures, and integrated into the wider landscapes and seascapes.”

I think there needs to be a clearer explanation of how these analyses in Figure 3 and 4 relate to the target text. Figure 3 is a mapping of pressures from two composite indicators, the Human Footprint and the Marine Cumulative Impact index. As the authors describe (lines 503 onwards) these composite indicators have several problems. But most importantly for this analysis, what is the evidence that ecosystem degradation in the EFG classes will be a consequence of increasing ecosystem area exposed to pressures above the median exposure level? Similarly, what is the evidence that having a greater proportion of the ecosystem area under protection reduces the degradation and loss of the EFGs? There is a very high level of abstraction here and no testing that the system is actually working for what it was designed for. So the findings in Figure 3 are difficult to interpret in the context of the Aichi targets. Figures 3 and 4 provide rather little new and robust evidence about trends in ecosystems relevant to the CBD

Overall, the paper is interesting and obviously includes a huge amount of detailed and careful work. If the key findings are related to the Aichi targets, then this needs more justification and development. If the paper is really a presentation of an ecosystem classification, then I think more needs to be done to show that this is robust and useful and providing added value in some way.

Author Rebuttals to Initial Comments:

Referees' comments:

Referee #1 (Remarks to the Author):

Reviewer #1: First, I need to state that is is the most unusual paper I have ever been asked to review. That is neither good nor bad, but it is an important piece of context. This is really a book with many hundreds of pages. Ideally all of this material needs to be peer reviewed, and reviewed by a panel of perhaps a dozen or more experts, which is not going to happen in the typical Nature review turnaround window. Its not going to be read nor used like a typical Nature paper either. Most of its uses will dive into the extended appendices. And to be truly useful, GIS files will need to be published, which I did not see (but easily could have missed).

Author response 7: We acknowledge the unusual character of our paper – we believe it reflects its ground-breaking qualities, and in submitting it to *Nature*, we expect all content will be duely exposed to the journal’s peer review process, additional that already undertaken through IUCN. As noted above (Author response 3), the data are available for review on Zenodo.

Reviewer #1: This type of effort is more typical of an NGO report. For me it brings to mind the Millenium Ecosystem Assessment, the WWF Global 200 ecoregions, and the IUCN 105 habitats. The point is all of those were not really published as a peer reviewed paper in a journal. They went through massive peer review processes with dozens (hundreds in some cases) of reviewers via public comment periods. Then they were published as a report by an NGO. And then their ultimate uptake was very heavily dependent on the backing and resources as well as the public process of the NGO behind the report. As such, not knowing the provenance of this effort (at least I have not yet found it) is a serious shortcoming. Is there an NGO behind this effort? Who? What mass peer review has been done? What kind of buy-in to this proposed typology can we expect?

Author response 8: Our paper has a developmental research component coupled to a significant global case study that distinguish it from the NGO reports mentioned by the

reviewer. The research component of our paper establishes critical information infrastructure to address a serious shortfall in capacity – the lack of a globally comprehensive typology explicitly for ecosystems – which has been a hindrance to progress on global ecosystem conservation efforts. The solution that we present is novel, scientifically robust, backed by IUCN, demonstrably fit-for-purpose and has enduring utility for ecosystem assessments in the future. In contrast, the NGO reports aim to review existing information on the status of biodiversity, rather than develop new frameworks that strengthen capacity for synthesis into the future. It is noteworthy that the WWF Global 200 ecoregions report drew its foundation from three publications in the peer-reviewed literature that established ecoregion classifications for terrestrial, freshwater and marine environments (Olson et al. 2001; Spalding 2007; Abell et al. 2008). The Millenium Ecosystem Assessment similarly drew from foundations in peer-reviewed literature.

The global case study in our paper shows how the typology can be applied to identify global conservation problems and to monitor impact of ecosystem management solutions as they are implemented. The significance of this advance is highlighted by a comparison of our structured analysis based on the IUCN Global Ecosystem Typology (Figs 3 & 4, Appendix S6) with the recently released CBD Global Biodiversity Outlook 5. For the first time, our Global Ecosystem Typology enables a comprehensive global assessment of progress on CBD targets each major type of ecosystem on earth. In contrast, Outlook 5 assesses progress on Aichi target 5 only generically for forests and wetlands (including mangroves), and ecological representation of Protected Areas (a component of target 11) only through a broad comparison of terrestrial and marine environments. The new information infrastructure developed in our paper therefore enables significantly more powerful global insights on ecosystem status and trends as a basis for problem diagnosis and design of locally relevant solutions because the units are broadly recognisable and functionally relevant on the ground.

In our original ms, we did not make it sufficiently clear that the development of the Global Ecosystem Typology was initiated, co-ordinated and backed by IUCN under the auspices of its Commission on Ecosystem Management. The IUCN (International Union for Conservation of Nature) is composed of both government and civil society organisations with more than 1,400 Member organisations and input from more than 17,000 experts. The diversity, expertise, experience, resources and reach of IUCN make it the global authority on the status of the natural world and the measures needed to safeguard it. IUCN and its members have invested significant resources to ensure rigorous scientific review and guarantee wide uptake of our new Global Ecosystem Typology. In response 6 above, we detail early evidence of uptake and buy-in by global and national institutions. We also detail the IUCN’s role and the mass review process in Appendix S5.

To emphasise the shortfall in capacity for ecosystem analysis and the need for a typology to redress the disparity between assessments of ecosystems and species, we amended text on Lines 79-80 as follows,

“Analogous to species taxonomy, this new typology establishes vital information infrastructure to support globally standardised ecosystem risk assessments...”

To make IUCN's backing more explicit, we now introduce the typology in the Summary paragraph (lines 69-70) as follows:

"Here we present IUCN's new conceptually robust, scalable, spatially explicit ecosystem typology..."

We also refer to the IUCN Global Ecosystem Typology in the main text, e.g. on Line 95-96, *"To serve the dual needs for conservation and sustainability, the IUCN Global Ecosystem Typology was designed to..."*

Reviewer #1: It is my opinion that if Nature decides it wants to publish this, it will need to: a) find out what has already been done in peer review (and make all of this more apparent in the manuscript), and b) use a modified review process that uses more reviewers and longer time frames, c) publish this as a Review to allow more space. But one could very easily argue this should be handled in a more traditional fashion like other NGO reports, or like a book.

Author response 9: a) As noted above (Author responses 1, 2 & 6), the extensive peer-review process of the granular detail of typology and its units of classification is now detailed in Appendix S5, including the comments, responses and revisions to more than 250 reviews by more than 50 ecosystem specialists. b) We will engage in whatever further review process deemed necessary by the journal. c) Again, we are open to suggestions on the type of article, but we emphasise the importance of the original content. IUCN will publish descriptive content from appendices S3 and S4 in a book. However that publication is essentially a technical manual that will not deal with content in the main text of our ms or the results presented in Figs 3 and 4, and Appendix S6 of our submission to *Nature*.

Reviewer #1: Alternatively it could be published as half a dozen separate papers published in long-form venues like QRB, AREES, Ecological Monographs, etc (although towards this last option I applaud the authors for not salami slicing and think the user community also benefits from everything in one place so I'm not really advocating for this approach)

Author response 10: We acknowledge the large volume of content in our submission, which reflects a large global effort and the breadth of issues that need to be addressed to present and justify the novel rationale and approach to development, explain the structure and interpretation of the typology, and the methods and results of the case study. We fully concur with the reviewer's opinion that the cohesion of having all these inter-dependent components published in one paper is essential for valid and effective application in the user community, and thus we do not favour fragmenting our contribution into half a dozen separate papers. Publication of all components in *Nature*, the world's most visible journal, is crucial to the timely launch and uptake of a much-needed innovation to advance the forthcoming post-2020 CBD agenda.

Reviewer #1: Turning now to a more traditional review ...

This paper represents a massive effort to develop a rigorous, first-principles list of ecosystems to be used in global conservation management efforts. They then use these classifications to divide the globe up into a map of ~100 ecosystems. They then produce two main results (Fig 3 & 4) - 1) a current assessment of % protected vs human pressures and 2) the temporal trend in human pressure for each separate ecosystem. These map directly to

assessment of how well the world is meeting Aichi/CBD targets for differing ecosystem types/locations on the globe.

As already noted, the typology has peers/competitors so the concept itself is not novel. The assessments could have been done (and at least partially have been done) using other typologies that are more familiar and have more buy-in and vetting. So the first question that arises is do we need another typology? and if so is this particular form the best? This is not a simple yes/no question. But some of my thoughts:

Author response 11: Thank you for the positive comments. We contend that a new typology is imperative to the CBD and related applications because existing ecological typologies do not explicitly focus on ecosystems (as defined in the CBD) and lack comprehensive coverage of the biosphere. This deficiency was made very clear in the recent Global Biodiversity Outlook 5 report, which mentioned ecosystems no less than 519 times, but offered no breakdown by ecosystem on progress toward biodiversity targets. Only very general summaries of progress were possible for Aichi targets relevant to ecosystems (5, 10, 11, 14 and 15). Our case study demonstrates how this critical reporting gap can be filled with a globally comprehensive typology explicitly designed to represent ecosystems. The IUCN Global Ecosystem Typology was designed uniquely fit-for-purpose to assess the dual contributions of ecosystems to biodiversity conservation and human well-being.

While the notion of an ecological typology (in the broad sense) is not new, there substantial novelty in our new approach to address contemporary global demands for ecosystem targets and governance. Our typology is the only one with a foundation in ecosystem theory that covers the entire biosphere. In lines 97-100 and Appendix S1 (lines 52-74), we draw a distinction between a broad range of ‘ecological’ typologies (defined in Methods lines 444445 and S1 as “*a classification of land, water or bioclimate intended to represent variation in the expression of ecological features.*”) and the IUCN Global Ecosystem Typology, which is focussed explicitly on ecosystems, defined in the CBD as, “a dynamic complex of plant, animal and micro-organism communities and their non-living environment interacting as a functional unit” (a definition essentially mirrored in our previous work on ecosystem conservation; Keith et al. 2013, 2015). Therefore, ecosystem typologies are a subset of all ecological typologies. In that sense, many other ‘ecological’ typologies that we reviewed in S1 are not peers/competitors because they are not explicitly ecosystem typologies, even though some are misapplied as ecosystem proxies or for purposes that they were not designed to serve (see Author response 4 for further commentary on distinctions between our Global Ecosystem Typology and other ecological classifications). Our assessments could have been done (or partly done) with some of those other typologies, but the inferences drawn about ecosystem status would hold limited validity, because those units of assessment are not ecosystems per se.

We challenge the notion that other typologies have more vetting than the IUCN Global Ecosystem Typology - refer to Author responses 1 & 2, and the extensive peer review process documented in Appendix S5, which so far as we are aware is unmatched by any other classification. The familiarity and buy-in of those other classifications reflects: i) a longer period since their release; ii) use for purposes other than ecosystem assessment; or iii) the lack of an information base that is directly fit for this purpose. The lack of CBD

reporting by ecosystem types in the Outlook report and early evidence of uptake (Response 6) suggest that the IUCN Global Ecosystem Typology will rapidly meet these important needs.

Reviewer #1: Pros

- a) this came out of a careful process that identified 6 major metrics of quality for a typology and evaluated 21 prior typologies (and found them all wanting against the totality of criteria)
- b) this typology attempts to be first principles looking at major ecosystem processes (Figure 1, but this figure is then customized to each of the 81 ecosystems identified).
- c) this typology is relatively rare in targeting both the ecosystem services view (exemplified by functional group classifications in DGVM models) and the biodiversity view (exemplified by the WWF focus on distinct zootas)

Author response 12: Thank you, we agree that these are some of the strengths of our typology.

Reviewer #1: Cons

- d) mostly just relating to whether we need yet another typology. Introducing a new typology immediately outdates all existing results using other typologies and prevents longterm comparisons. On some level the burden of proof is on the authors to demonstrate we have better/different results than we could have gotten with the IUCN or WWF typologies. I think it is unlikely we get a fundamentally different story/outcome

Author response 13: In Author response 11 we explain why the IUCN Global Ecosystem Typology the IUCN Global Ecosystem Typology is more fit-for-purpose to assess the status of the world's ecosystems and monitor progress towards global CBD targets than other ecological typologies. We also suggest that it serves other purposes related to ecosystem management more directly than alternative. This is because the IUCN ecosystem typology offers much greater thematic and spatial granularity than is possible from IUCN Species Habitats v3.1 or WWF ecoregions. The IUCN Species Habitats were entirely unmapped until Jung et al. (2020) modelled the distributions of some of the terrestrial types. We have examined their maps in detail and found that only a few of their maps matched the concept of major ecosystems or Ecosystem Functional Groups (see evaluation of Jung et al. (2020) against design principles in Table S1.2 and relatively Table S4.1 of EFG maps in which only three EFGs were mapped using data from Jung et al. 2020). WWF ecoregions are also suboptimally fit for ecosystem assessment. We added text to Appendix S1 (Results and Discussion section, lines 74-104) to explain why ecoregion classifications this is so. See Author response 4 for details of this additional explanation.

Reviewer #1: e) Appendix 3 that lists each of the 81 ecosystems over 145 pages is a gold mine of information but needs serious peer review by experts in each area. I found myself fascinated by the ecosystem-specific versions of Figure 1 given for each of the 81 ecosystems. This is a bold and useful statement. But of course I then immediately started

questioning whether they were right and how much scientific evidence we had for each of these figures.

Author response 14: Thanks you for the positive comments about this component of our ms. We believe this highlights the outstanding novelty and enduring utility of the typology. Regarding questions about the correctness and evidence for each of the figures, please see Author responses 1, 2 and 6 on the extensive and rigorous peer review process documented in Appendix S5 of the ms.

Reviewer #1: f) adoption of this new typology is mostly going to depend on social factors (who it comes from, what kind of money is behind it) rather than scientific merit and this is not currently obvious to me. I do see a brief sentence in the Methods this came out of a working group, but more information is needed. Specifically, any ties to NGOs? Any NGOs lined up to adopt this? or is this purely a research effort?

Author response 15: Please refer to Author response 6, in which we list evidence of early uptake, and Author response 8, in which we describe amendments to make IUCN backing of the typology much clearer. We think adoption of the IUCN Global Ecosystem Typology will depend on both social factors and scientific excellence. An extensive social outreach and governance effort (e.g. <https://global-ecosystems.org/>, <https://www.iucncongress2020.org/motion/074>) accompanies the research effort to ensure uptake and implementation. However, we believe the impact and visibility of the typology would be significantly enhanced by publication in the world's leading scientific journal.

Reviewer #1: g) adoption of this typology will depend heavily on providing GIS (shapefile, raster) implementations of this for others to use and I did not see where this is done

Author response 16: The spatial data are already available to users and reviewers in a public repository (see Author response 3 for details). In addition, we have developed a dedicated website application including extensive descriptive content (text, diagrams, images) with search and browse capability and advanced mapping and spatial analysis functions (<https://global-ecosystems.org/>), facilitating easy access for a much broader public audience. If our ms is accepted for publication in *Nature*, this web application could be linked to the published article. A similar coupling of a *Nature* Letter with a web-app is illustrated by Pekel et al. (2016) [<https://doi.org/10.1038/nature2058>] and the Global Surface Water Explorer [<https://global-surface-water.appspot.com/>].

Reviewer #1: Or to summarize, scientifically superior but not unprecedented, but socially connected and adept is probably more important and unclear to me. Some other thoughts if this goes forward at Nature:

1) The WWF ecosystems was not assessed/compared that I could see. I know this is in someways targeted at animals which makes it more narrow, but the authors included many plant-based functional grouping systems. WWF needs to be included

Author response 17: See Author responses 4 and 13 and Appendix S1. In Author response 4 we draw attention to the evaluation of other ecological typologies (including WWF terrestrial ecoregions, Olson 2001, updated by Dinerstein 2017) against six design principles that guided the development of the IUCN Global Ecosystem Typology. Table S1.2 (3rd row) contains the evaluation of WWF ecoregions, which can be compared with the IUCN ecosystem typology (1st row). There are substantial differences between these two typologies in relation to principles 1, 3 and 4, and additional differences in relation to the other principles. In Author response 4, we quote new text added to the Results and Discussion section of Appendix S1 that addresses some key distinctions between ecoregions and our Global Ecosystem Typology, *“Three ecoregional classifications of terrestrial (Olson et al. 2001; Dinerstein et al. 2017); coastal marine (Spalding et al. 2007) and freshwater (Abell et al. 2008) environments are among the most widely used of global ecological classifications that we reviewed...”* [see Author response 4 or Appendix S1 for full text]. In Author response 13, we provide further commentary on the distinction between the IUCN Global Ecosystem Typology and ecoregions.

Reviewer #1: 2) Providing GIS files needs to be done

Author response 18: Please see Author responses 3 for details of the public repository where the spatial data are lodged and available for review and Author response 16 for details of the dedicated website application with maps.

Reviewer #1: 3) The main paper needs to be reprioritized on allocation of space. Much less detail on levels of classification should be given (or pushed to Methods). More details need to go into human impacts (not just pushed into Methods). More interpretation on management implications. Explain more why combining the ecosystem function and biodiversity perspective is important

Author response 19: We undertook considerable restructuring of our manuscript to address this comment as detailed in the following list:

- We moved detail on the levels of the typology to Methods [lines 483-517] as suggested.
- We added the following text on human impacts:
“In the tropics, ecosystem protection is low, except for heath forests (T1.4), but opportunities remain for more protection of lowland and montane rainforests (T1.1, T1.3). Dry forests (T1.2) are the most pressured and least protected tropical group, approaching the status of some highly transformed temperate forests, with options for protection rapidly diminishing.” [Lines 163-166].
“Pressures are rapidly increasing on the world’s largest rivers (F1.7) and seasonal lowland rivers (F1.5) (Fig. 4).” [Lines 174-175]
“When catchment processes are more fully incorporated into our limited analysis, more freshwater ecosystems will likely be identified as under high pressures, and the vital role of catchment management in restoration of freshwater ecosystems will be more apparent.” [Lines 180-182]
“High pressures were identified in seamounts (M3.4), deep biogenic beds (M3.5) and trenches (M3.6) mostly in unprotected international waters.” [Lines 186-187]

“e.g. MFT1.3 coastal saltmarshes (Figs. 4c, S5.1), or those well-known in shelf systems such as coral reefs [22].” [Lines 189-190]

- We elaborate further on human impacts in Appendix S6 lines 155-314.
- We added the following text on management implications [Lines 212-236]:
*“The effectiveness of PAs depends on the nature of threats and on management. For example, marine PAs mitigate overharvest and related threats [23], but additional measures are required to address severe climate stressors [22]. Opportunities to expand PA networks to abate conversion of natural and semi-natural terrestrial ecosystems to anthropogenic systems [17] are mediated by contrasting signatures of degradation (Figs. 4, S6.5). Tropical dry forests (T1.2), for example, are undergoing accelerating transformation to semi-natural mosaics and pasture, whereas extensive historical replacement of temperate grasslands (T4.5) by pastures (T7.2) and crops (T7.1) has abated. This indicates needs for different protection and restoration strategies tailored to land use context (Fig. S6.2).
Expansion of PAs is most urgent for ecosystems in which PAs offer effective preventative measures and there are rapidly diminishing options for protection of remnants. Less than 15% of unprotected, rapidly diminishing tropical dry forests have so far avoided high pressures. For unprotected ecosystems less exposed to high pressures (Fig. S6.2; e.g F2.6 permanent salt/soda lakes), substantial intact areas (>30%) offer more opportunities for preventative action by expanding the PA network before major loss and degradation occurs.
Active management is needed to maintain pressures at low levels and sustain ecosystem function and biodiversity both inside and outside PAs. For example, ecosystems threatened by degradation resulting from biological invasions are likely to require ongoing biosecurity and control measures throughout landscapes and seascapes where the sources of invasive biota are pervasive and PA boundaries are permeable. Moreover, active restoration is likely to be needed in ecosystems, such as temperate grasslands (T4.5), with legacies of prolonged exposure to high pressures, especially those with significant local or regional contributions to ecosystem services or supporting endemic biota. For EFGs well-represented in PAs, such as seagrass meadows (M1.1), management strategies should seek to mitigate threats within PAs, and maintain ecosystem functions in other parts of the landscape or seascape.”*
- We elaborate further on management implications in Appendices S6 [lines 256-296] and S7 [lines 129-145].
- To more clearly explain the importance of combining functional and biodiversity properties in a single typology, we honed the Summary paragraph [Lines 67-73]:
“Different types of ecosystems vary in their contributions to biodiversity [4], service provision [5], and their relative exposure to risks [3], yet there is no globally consistent framework for grouping ecosystems with functionally similar responses, hampering progress on conservation targets and sustainability goals. Here we present IUCN’s new conceptually robust, scalable, spatially explicit ecosystem typology for generalisations and predictions about functions, biodiversity, risks and management remedies across the entire biosphere.”
- We added the following text on the importance of combining the ecosystem function and biodiversity perspectives: *“Although ecosystem functions underpin both biodiversity and human benefits, global assessments of ecosystems [7] [11] continue to rely heavily on species metrics or simplistic landcover proxies that convey limited*

information about them. This limits capacity to diagnose trends and to design and resource on-ground solutions.” [Lines 89-93]

- We note in lines 132-136,
“...biodiversity, ecosystem functions and services [9]. Salient differences in these features are axiomatic to a functionally-based ecosystem typology because they underpin robust generalisations and predictions about ecosystem responses to environmental change and management.”
- We also modified the following sentence to more directly address the need for a functional approach:
“A robust function-based typology conceptual framework (Appendix S2) that groups ecosystems with based on convergent drivers, traits and environmental relationships should reveal similarities in threats, mechanisms of degradation and management strategies for recovery (Appendix S7).” [Lines 254-256]
- We elaborate further on the need for a typology that address both ecosystem function and biodiversity in Appendix S1 [Table S1.1 – rationale for principles 1 and 2; and lines 52-136].

Reviewer #1: 4) There has to be a simple summary table that makes explicit what has been accomplished. Something that looks like the table of contents of Appendix 3 - i.e. revealing the top 3 levels of the hierarchy breaking into 81 ecosystems, then providing columns for total (and/or percent) area, key processes, % protected, human impacts, change in human impacts. I know this might seem like a dry table. But it is the only way to make the main output of the project (the ecosystems) tangible. And no such summary exists anywhere right now (the table of contents of Appendix 3 is the closest). Space spent on this table is far more concrete and useful than Figure 2 and a page of describing hierarchical structure abstractly.

Author response 20: We thank the reviewer for this suggestion. We have compiled the suggested table and added it to Appendix S6 (as Table S6.1) for now, pending advice from the editor. We could move the existing Fig. 2 to Methods, to support the text already shifted there (see Response 19), however, we cannot see how a table with c. 100 rows (including all 3 levels as suggested) could fit within the 5-page limit. We also point to Fig. 4, which has all 85 Ecosystem Functional Groups (all those assessed) listed in a compact results format, including the codes, which enable users to identify the respective realms (Level 1) and biomes (Level 2) for each of the Level 3 units. In addition, Appendix S6 has a further three figures reporting on results of analyses for all 85 EFGs. Regarding Fig. 2, its content is comparatively modest, but we think it is a useful graphic overview of the typology structure, terms and the combined top-down bottom-up approach to development, which would otherwise be hard for readers to gain from the main text and methods, so we would like to see it retained at least in Methods.

Reviewer #1: 5) Combining functional and biodiversity perspectives is a claim of novelty and importance for this classification. To my eye the biodiversity perspective mostly shows up in the lower levels of the hierarchy (i.e. levels 4-6) which implies it is less important. Please address this. Also please give an example or two of ecosystems further broken out in to these biodiversity classifications. Can biodiversity be pulled back up into the table in #4 - e.g. estimated # of birds, mammals and perhaps plants in each of the 81 ecosystems? I think this

last task, while some work, would greatly help with the claim of merging perspectives and with the use and uptake of this typology.

Author response 21: We decided to adopt a hierarchical arrangement for the typology that represents functional resemblance in the upper levels and biodiversity resemblance in the lower levels because this has greater parsimony and robustness to new knowledge than alternative structures. In essence, we judged that a hierarchy representative biodiversity in the top levels and functions in the lower levels would not be workable. We have been careful to avoid any implication that one feature is more important than the other (i.e. resemblance based on function or biodiversity) – in the text we always refer to these as ‘dual’ objectives. Nonetheless, we acknowledge that some readers may misinterpret the hierarchy in terms of the importance of the features represented in the typology. We therefore added text to the Methods [lines 472-479] to address this explicitly:
“Neither function, nor biotic composition are intended to take primacy within the typology. Functional units are represented in the upper levels of the hierarchy because representation of compositional relationships at global scales requires many more units and is more likely to change with developing knowledge than representation of functional relationships. Therefore, a structure that recognises compositional variants within broad functional groupings is more parsimonious and robust than one that attempts to represent compositional resemblance at the upper levels and functions at lower levels.”

The focus of this paper is to examine global patterns in pressures and protection through a functional lens. Regarding exposition of biodiversity relationships in Levels 4-6 of the typology, we have a study in progress linking the IUCN Global Ecosystem typology to a sample of national and regional ecological classifications from around the world. This is incomplete, but we have included two examples in Appendix S3 [Tables S3.3, S3.4] to demonstrate the linkages that can be made between established national classifications at Level 6 and the upper three levels of the Global Ecosystem Typology. These examples represent ecosystems from tropical, temperate, montane and desert environments in Myanmar and Chile, and were developed by members of our author team.

Referee #2 (Remarks to the Author):

This paper includes a substantial volume of work related to a new system for a global ecosystem classification. The authors introduce a typology for classifying global ecosystems, describe the process that was used to refine and develop a comprehensive classification across 6 hierarchical levels, and analyse the third level (Ecosystem Functional Groups) against the CBD Aichi biodiversity targets. My review covers these three steps.

Reviewer #2: A. The typology.

There is no question that the authors are correct in asserting that there is no global ecosystem typology of general applicability. There are good reasons for this – in particular ecosystems are not fixed units in space and time, but rather multi-dimensional and multi-scale dynamic units. A general ecosystem classification is therefore lacking for good reasons, and any ecosystem classification will have to be designed with a specific purpose in mind. One classification will not meet all needs. The review in Appendix S1 includes some

typologies that are ideal for the purpose for which they were designed but clearly not for what the authors here are seeking. So the key issue in the paper must be to define the purpose here. The authors state the identified requirement is for an ecosystem classification that groups ecosystems with functionally similar responses relevant to evaluating progress with conservation assessments (Aichi targets) and sustainability goals (SDGs). They are seeking a classification that will allow predictions and metrics across all areas of the Earth in particular for these conservation and sustainability assessments. In fact the paper addresses the Aichi targets mainly, but includes a much broader requirement for the system to underpin predictions and generalisations about ecosystem responses to environmental change ... etc. (lines 65-68). In any case, it is not surprising then that there is no existing typology that meets the authors design requirement (line 100).

Author response 22: We agree with the reviewer's assessment, which reaffirms the outcome of our review in Appendix S1 and highlights the importance of the gap, which the IUCN Global Typology is designed to fill. As we note in Appendix S1, the majority of classifications we reviewed have a biodiversity focus (typically founded on biogeographic patterns or biophysical variables intended as proxies for biota), with comparatively little attention to functional and dynamic features of ecosystems. Previous classifications that explicitly address ecosystem function have a narrow scope.

Reviewer #2: The authors lay out their design criteria in Figure 1 and Appendix 2. The system that they develop is described but not tested or defended. I cannot judge how well it meets its purpose. There seem to be some relatively arbitrary decisions made about the drivers (in the boxes).

Author response 23: Please refer to Author response 2. The decisions made about salient ecological processes identified for each EFG, and all other aspects of the typology, were the outcome of a structured deductive process based on explicit design principles and application of a conceptual model of ecosystem assembly consistently across all EFGs. This guided development, review and revision process that is now described in an additional appendix (S5). The design principles are described in Appendix S1, the underlying conceptual model is described in Appendix S2, and the structure of the typology is described in Appendix S3. We believe the strong foundation in theory, attention to detail and extensive specialist input (see Author response 2) provide a basis for judging the efficacy of the typology. We believe this level of critical review far exceeds that undertaken during the development of any previous typology, including all those reviewed in Appendix S1. However, typologies are not strictly testable in an empirical sense and whether the IUCN Global Ecosystem Typology meets its purpose will ultimately be determined by its use. In Author response 6, we list early evidence of uptake.

We acknowledge that alternative classifications and alternative interpretations of ecosystem drivers might be similarly consistent with the available evidence (see caveats added to the text and quoted in Response 2). However, a strong consensus among an extensive group of specialists supports our contention that v2.0 of the IUCN typology will be workable and interpretable by a large group of future users. There is ofcourse, provision to make further refinements to the IUCN Global Ecosystem Typology in future versions as new

data emerge and experience builds from its application around the world.

Reviewer #2: One question is why the system includes human pressures as a driver defining the ecosystem type if the system is to be used to predict ecosystem responses to environmental change and management. If human pressures are already defining the ecosystem type then how can pressures be analysed independently?

Author response 24: Anthropogenic ecosystems within biomes T7, S2, SF2, F3, M4 and MT3, (see Appendix S4), including croplands, aquafarms, urban systems, etc. are those that are shaped and maintained by human activities which, if ceased, would result in major transformation of ecosystem characteristics. The pressures analysis was applied only to the 85 non-anthropogenic EFGs (those not primarily defined by human activity), while anthropogenic ecosystems were excluded (see Methods text, lines 529-530). Therefore the pressure analysis reported in Figs. 4, 5 and Appendix S6 was not compromised by a lack of independence of human pressures.

We added that sentence to the Methods text (lines 533-537) to make this clear. We also state in Appendix S4 in the section on 'Diagrammatic assembly models' (lines 68-70) that, *"Only the major features are shown in the diagrammatic models and anthropogenic processes are only shown for anthropogenic functional groups (encompassing ecosystems that are shaped and maintained by humans)."*

Reviewer #2: What are the design criteria for a degraded ecosystem? How does the system reflect loss of function or services as a result of additional human pressures? Doesn't this require some process or dynamic model rather than a static hierarchical classification?

Author response 25: Firstly, we would like to clarify that analyses of the loss of ecosystem functions rely on indices and/or models of degradation, whereas the purpose of the typology is to structure those analyses and comparisons of degradation between different types of ecosystems. Therefore, the acknowledged limitations of our pressures analysis relate to the indices of impact that were available, not the hierarchical classification.

In our analysis of pressures (see Methods lines 528-556 and Appendix S6), we assumed that two composite indices (Human Footprint and Marine Cumulative Human Impact) represented relevant pressures and that values above the raw median threshold indicated high pressures, which we assumed to be associated with ecosystem degradation. We explored the sensitivity of the analyses to uncertainty in ecosystem distribution and found it had little effect on results. We also evaluated the limitations of the composite indices (acknowledged in Methods text lines 612-624) and concluded, *"We therefore considered the data sufficient for our demonstration purposes and for inferences about very general patterns in the global status of ecosystems, except where stated. Future refinements to improve data quality will enable more detailed inferences to be drawn about the global status of Ecosystem Functional Groups."*

In agreement with the reviewer, we acknowledge that dynamic process models would give a higher grain and more reliable assessment of human impact and risks than application of these static composite indices, and have applied such models to more detailed risk

assessments of selected ecosystems where available data permit (e.g. Bland et al. 2017. Using multiple lines of evidence to assess the risk of ecosystem collapse. Proc. R. Soc. B 284: 20170660. <http://dx.doi.org/10.1098/rspb.2017.0660>). However, the available data are well short of what is required to support the development of suitable process models for assessments of the 85 global ecosystem groups examined in this paper. Even so, our function-based typology provides guidance on the types of ecosystems that require models, and the parameters and variables that should be considered in their construction.

We suggest that ecosystem-specific application of risk assessment protocols, such as the IUCN Red List of Ecosystems criteria, provide an intermediate means of assessment that avoid some significant limitations of generic composite indices of pressures, yet are less data-demanding than stochastic process models. Again, our typology provides guidance for defining the units for such assessments because of its representation of both functions and biodiversity at different levels. We plan to pursue this avenue in future work and comment in lines 248-252 of the main text, as follows,

“The limitations of our analysis, noted above for freshwater and marine ecosystems, point to the need for more ecosystem-specific analysis of threats guided by conceptual models of ecosystem dynamics developed in Red List assessments (Keith et al. 2013). With improved threat and ecosystem distribution data, these will strengthen capacity to forecast state changes that result in loss of ecosystem function, services and biodiversity [28].” and in Appendix S7 [lines 77-98].

Reviewer #2: At the end of the paper, the authors list a series of other potential uses for the ecosystem typology including for environmental accounts and natural capital accounting, and for conservation planning. Surely the design criteria here would be different? For natural capital and environmental accounting there needs to be a human demand and ideally economic valuation of ecosystem service benefit flows. Conservation planning is usually based on biotic units of conservation importance (species, ecological communities, habitats). So this section seems to be somewhat at odds with the design criteria.

Author response 26: We had to reduce this part of the text to make space for other edits, but we now explain the versatility of the typology more directly in the main text and Appendix S7. The key point here is that the typology has built-in flexibility for a selection of different applications, all of which rely on representation of both ecosystem functions and biodiversity.

In lines 247-248, we now note,

“Integration of both functions and biodiversity into the hierarchical structure of the typology confers versatility for a diversity of applications that require those features”

We elaborate in the following text to Appendix S7 [lines 15-33] to explain:

“Our ecosystem assembly model and global typology will support multi-disciplinary action on sustainable development, ecosystem governance, ecosystem management, communication and education around the world. The typology has built-in versatility for a range of different applications related to ecosystem functions and biodiversity, because:

- i) the scalable hierarchy enables “representation of different features at particular hierarchical levels and facilitates applications across a range of spatial and organisational scales” (see Rationale column in Table S1.1); and*

ii) the typology represents both ecosystem functions (in the upper levels) and biodiversity (in the lower levels).

Here we identify eight applications that would benefit from a classificatory framework provided by the IUCN Global Ecosystem Typology (Fig. S7.1). All of these applications require either a function-based framework for ecosystem assessment (global analysis of SDGs, ecosystem accounts), assemblages of biodiversity above the species level (e.g. landscape/seascape conservation planning, representation analyses of protected areas) or both (e.g. Red List of Ecosystems risk assessments).

The diverse themes and sales of these applications suggest that the IUCN Global Ecosystem Typology will provide a much-needed information infrastructure to address dual overarching goals to conserve biodiversity and sustain ecosystem services.”

We address the two specific examples raised by the reviewer below.

Natural capital accounting is focussed on natural assets in so far as they produce benefits for human well-being. These benefits, or ‘ecosystem services’, stem from ecosystem functions. Level 3 units (Ecosystem Functional Groups) of the typology are therefore an appropriate framework for defining ecosystem assets and for upscaling and comparing national accounts internationally. These Level 3 units provide a more direct representation of ecosystem functions than land use and land cover units which have served as a reporting framework since the United Nations first adopted its System for Environmental Economic Accounting in 2014 (UN SEEA). In recognition of this advantage, Level 3 of our IUCN Global Ecosystem Typology was recently adopted as the reference classification in the revision of the UN SEEA-Experimental Ecosystem Accounts. We have included a reference to this in the main text (lines 242-245), as we think it is strong evidence of an emerging use of the typology (not simply a potential use):

“A recent UN Forum of Experts on the System of Environmental-Economic Accounting (SEEA) has adopted EFGs as a reference classification for extending the SEEA framework to Experimental Ecosystem Accounts [31], meeting a long-recognised need for a spatially explicit, functionally-based ecosystem typology to underpin natural capital accounting [30].”

In Appendix S7 (p2-3, in section on ‘Natural Capital Accounting’), we further elaborate on how our typology will be used for global synthesis of national ecosystem accounts:

“A recent United Nations Forum of Experts adopted our Global Ecosystem Typology as the reference classification for implementing SEEA-EEA (UNSD 2019). The core of the reference classification is Level 3, which will be used to summarise globally across national ecosystem accounts. Users are expected to use the highest quality high-resolution classification available for their jurisdiction when developing their national accounts (i.e. Level 6 units), and assign the units of that classification to Ecosystem Functional Groups (Level 3) to enable consistent international reporting. This flexible use of the reference classification will enable jurisdictions to report additional detail nationally, provided that the accounts are also summarised to units of the international reference classification (UNSD 2019).

For countries that currently lack a national classification for ecosystem accounting, the GET may be used to develop one. Murray et al. (2020) provide a recent example of using the GET to scale down to locally-derived, locally-relevant ecosystem types in Myanmar.”

Level 6 of the IUCN Global Ecosystem Typology provides a suitable framework for conservation planning, as these finer-scale units represent different assemblages of species and thus landscape/seascape scale variation in biodiversity. We note in the main text (lines 509-513) that the lower levels of classification

“are crucial for representing biodiversity in the typology, but also have an important role in scaling up information from established local-scale typologies. We aimed to facilitate integration of these local classifications, rather than replace them, because they have considerable investment, information richness, accuracy and especially local ownership (Appendix S3).”

In Appendix S3 (section on ‘Top-down and bottom-up construction’, lines 328-332), we elaborate on this rationale as follows:

“This flexibility to define compositional relationships from the bottom-up is critical to utility, local ownership and wide use of the typology (Principle 6) because: i) expertise and data on compositional relationships reside primarily at national and subnational levels; and ii) ecosystem management and biodiversity conservation is implemented through locally-based on-ground action.”

We note further that, *“Incorporating these classifications into a global framework acknowledges the value of substantial investments in data acquisition and development, as well as the integration of these classifications into policy instruments and management plans.”* (Appendix S3, section on ‘Lower levels of classification’, lines 281-283)

As for natural capital accounting, there is evidence for early uptake of our IUCN Global Ecosystem Typology in conservation planning applications at national levels. Murray et al. (2020) (*Threatened Ecosystems of Myanmar. An IUCN Red List of Ecosystems Assessment*. Version 1.0. Wildlife Conservation Society. ISBN: 978-0-9903852-5-7 DOI 10.19121/2019.Report.37457) used our framework to develop a national classification and map of ecosystems for conservation planning in Myanmar to assess risks to different ecosystem types, which is now being used to identify Key Biodiversity Areas and priorities for expanding the network of protected areas in that country. We added text to the Conservation planning section of Appendix S7 [lines 132-138] to note this recent progress, *“Evidence for early uptake of our IUCN Global Ecosystem Typology in conservation planning applications at national levels. Murray et al. (2020) used our typological framework to develop a national classification and map of ecosystems for conservation planning in Myanmar to assess risks to different ecosystem types, which is now being used to identify Key Biodiversity Areas and priorities for expanding the network of protected areas in that country. It will also be used to structure Myanmar’s next National Biodiversity Strategy and Action Plan for reporting to Convention Biological Diversity.”*

Reviewer #2: Overall, I find the objective of developing a typology for ecosystem mapping and prediction under changing drivers to be an important task and clearly a lot of thought and work has gone into it here. Some more justification of design decisions seems important given the very elaborate structure that emerges, especially considering the elements of ecosystem science and community ecology that are relevant. Appendix S2 and S3 describe the five driver boxes but does not explain why these were selected or what was rejected. What process was used to decide that this was the right classification?

Author response 27: Thank you. We hope improved content mentioned in Responses 2 and 23 clarifies this point.

Reviewer #2: B. The system

The system is described and developed in detail in Appendix 4, including the detailed description of all the functional groups. I cannot review this section, and in fact its provenance is unclear as there are other citations. Is this being published elsewhere in whole or in part?

Author response 28: We had no plans to publish the content of Appendix S4 elsewhere at the time of original submission. However, after advice from *Nature's* editor, we were encouraged to publish the detail of the descriptions in an IUCN report. This process is in train. We have also developed a website application with mapping and spatial query functionality that will allow users to explore and use the typology in relevant applications (<https://global-ecosystems.org/>).

Reviewer #2: C. The analysis

Having developed and described the system, the authors use it to examine progress with two Aichi targets, numbers 5 and 11. This analysis presented in Figure 3 and 4 are the main research findings from the paper. According to the CBD, Target 5 is “By 2020, the rate of loss of all natural habitats, including forests, is at least halved and where feasible brought close to zero, and degradation and fragmentation is significantly reduced”, and Target 11 is “By 2020, at least 17 per cent of terrestrial and inland water, and 10 per cent of coastal and marine areas, especially areas of particular importance for biodiversity and ecosystem services, are conserved through effectively and equitably managed, ecologically representative and well connected systems of protected areas and other effective area-based conservation measures, and integrated into the wider landscapes and seascapes.”

I think there needs to be a clearer explanation of how these analyses in Figure 3 and 4 relate to the target text. Figure 3 is a mapping of pressures from two composite indicators, the Human Footprint and the Marine Cumulative Impact index. As the authors describe (lines 503 onwards) these composite indicators have several problems. But most importantly for this analysis, what is the evidence that ecosystem degradation in the EFG classes will be a consequence of increasing ecosystem area exposed to pressures above the median exposure level?

Author response 29: The median values of the two indices were not used to assess either of the Aichi targets. The purpose of Fig. 3 is to identify EFGs with high exposure (>70% of distribution) to high pressures (defined as above-median values of respective indices) and low protection (defined as less than 17% of the distribution represented in Protected Areas, or less than 10% for marine ecosystems). We also used factorial combinations of exposure to high pressures and protection shown in Fig. 3 as a means of structuring discussion about ecosystem status and management strategies across the three major realms and their transitions.

To clarify our interpretation of Aichi target 5, and the limitations of our analysis, we added the following text to the Methods (lines 560-562),

“In a separate analysis, we evaluated progress against Aichi targets to illustrate how the typology could be used to structure such evaluations for these targets and those that will soon succeed them in the post-2020 CBD framework.”

...and the following text to Methods (lines 568-581),

“This analysis was constrained by three limitations on the available pressure data. First, we assumed that changes in pressure translated into loss and hence that there were negligible lags in ecosystem responses and that there was a linear relationship between increase in pressures and loss of habitat. This is likely to be true for some types of pressures represented in the HFP (e.g. land use change to built environments, crop land or pasture land), but not necessarily for others (e.g. human population density, transport corridors) (Ventor et al. 2016). Second, for each index, two comparable estimates of pressure were available for different times, 2000 and 2013 for HFP and 2008 and 2013 for MCHI. These intervals only intersect the early portion of the 2010-2020 reporting period for Aichi targets. Third, the availability of only two temporal estimates of the indices meant that change in pressures could only be estimated over a single interval, thus precluding any assessment of whether “rates of change had at least halved” or whether “degradation and fragmentation were significantly reduced”, as stated in parts of Aichi target 5. Instead, we used the data to assess whether the change in pressures over the interval for which data were available was “close to zero”, as also stated in Aichi target 5.”

This calculation was based on the raw difference between the two temporal pressure estimates for a random sample of 1000 cells, with the respective distributions of values given in Fig 4. It did not use the median values. We have clarified this interpretation in the Methods text [lines 562-568].

Reviewer #2: Similarly, what is the evidence that having a greater proportion of the ecosystem area under protection reduces the degradation and loss of the EFGs? There is a very high level of abstraction here and no testing that the system is actually working for what it was designed for. So the findings in Figure 3 are difficult to interpret in the context of the Aichi targets. Figures 3 and 4 provide rather little new and robust evidence about trends in ecosystems relevant to the CBD

Author response 30: This comment is relevant to evaluation of Protected Areas for Aichi target 11. In particular, we used the IUCN Global Ecosystem Typology to assess whether Protected Areas were ‘ecologically representative’ as specified in the target, and the progress of different ecosystem types toward the target during the reporting period. We clarified the purpose and scope of our analysis in relation to Aichi target 11 with the addition of the following text to Methods (lines 583-592):

To demonstrate how the typology could support evaluation of Aichi target 11, we focussed our analysis on whether “at least 17 per cent of terrestrial and inland water, and 10 per cent of coastal and marine areas... [were] conserved through ecologically representative systems ... of protected areas...” as specified in Aichi target 11. We used the IUCN Global Ecosystem Typology to assess whether Protected Areas were ‘ecologically representative’, and the progress of different ecosystem types toward the area target during the reporting period 2010-2020. We did not assess whether protected areas were, “effectively and equitably managed”, “especially areas of particular importance for biodiversity and ecosystem

services”, or “well connected”, and we did not assess “other effective area-based conservation measures”, as specified in Aichi target 11.

Reviewer #2: Overall, the paper is interesting and obviously includes a huge amount of detailed and careful work. If the key findings are related to the Aichi targets, then this needs more justification and development. If the paper is really a presentation of an ecosystem classification, then I think more needs to be done to show that this is robust and useful and providing added value in some way.

Author response 31: Thank you for this positive comment and for identifying the need to clarify the purpose of the analysis of Aichi targets, which was to demonstrate how the typology could support the evaluation of CBD targets. To clarify this, we modified text in lines 193-195 as follows:

“The data are sufficient to demonstrate the potential utility of the typology for assessing progress against CBD targets, such as Aichi targets 5 (loss and degradation of natural habitats) and 11 (representation in protected areas).”

We also make this clear in Appendix S6 (Introduction lines 20-26):

“The recent Global Biodiversity Outlook 5 (CBD 2020) reported on the outcomes for the 17 Aichi targets for their full reporting period 2010-2020, mentioning ecosystems no less than 519 times. However, Outlook 5 gave only limited summaries of progress for Aichi targets relevant to ecosystems (5, 10, 11, 14 and 15), and offered no breakdown by ecosystem types. Our case study demonstrates how this critical reporting gap can be filled with a globally comprehensive typology explicitly designed to represent ecosystems.”

References

Abell R, Thieme ML, Revenga C, Bryer M, Kottelat M, Bogutskaya N, Coad B, Mandrak N, Contreras Balderas S, Bussing W, Stiassny MLJ, Skelton P, Allen GR, Unmack P, Naseka A, Ng R, Sindorf N, Robertson J, Armijo E, Higgins JV, Heibel TJ, Wikramanayake E, Olson D, López HL, Reis RE, Lundberg JG, Sabaj Pérez MH, Petry P (2008) Freshwater ecoregions of the world: A new map of biogeographic units for freshwater biodiversity conservation. *BioScience* 58: 403–414. [<https://doi.org/10.1641/B580507>]

CBD (2020) Global Biodiversity Outlook 5. Secretariat of the Convention on Biological Diversity Montreal. [<https://www.cbd.int/gbo/gbo5/publication/gbo-5-en.pdf>]

Keith, D. A., Rodríguez, J. P., Brooks, T. M., Burgman, M. A., Barrow, E. G., Bland, L., . . . Spalding, M. D. (2015). The IUCN red list of ecosystems: Motivations, challenges, and applications. *Conservation Letters*, 8(3), 214-226. doi:10.1111/conl.12167

IUCN (2016). A Global Standard for the Identification of Key Biodiversity Areas, Version 1.0. First edition. Gland, Switzerland: IUCN.

Olson, DM, Dinerstein E, Wikramanayake ED, Burgess ND, Powell GVN, et al. (2001) Terrestrial Ecoregions of the World: A New Map of Life on Earth. *Bioscience* 51: 933-938.

Spalding MD, Fox HE, Allen GR, Davidson N, Ferdaña ZA, Finlayson M, Halpern BS, Jorge MA, Lombana A, Lourie SA, Martin KD, McManus E, Molnar J, Recchia CA, Robertson J (2007) Marine ecoregions of the world: A bioregionalization of coastal and shelf areas., *BioScience* 57: 573–583. [<https://doi.org/10.1641/B570707>]

Watson, J. E. M., Keith, D. A., Strassburg, B. B. N., Venter, O., Williams, B., & Nicholson, E. (2020). Set a global target for ecosystems. *Nature*, 578(7795), 360-362. doi:[10.1038/d41586-020-00446-1](https://doi.org/10.1038/d41586-020-00446-1)

Reviewer Reports on the First Revision:

Referees' comments:

Referee #1 (Remarks to the Author):

This manuscript does two main things:

- A) It presents a new ecosystem classification dividing the earth's surface up into 5 realms hierarchically further divided into 25 biomes and 108 ecosystem functional groups.
- B) It presents an analysis of the conservation status of each of the 108 EFG vs. Aichi targets in terms of preservation and human pressures.

On(A) I feel I have a much deeper insight into the background and depth of the work that has gone into this. I feel like this work is now more clearly presented across the methods in the main text, the comparison with existing classifications, and an appendix showing review comments. I also understand how this is situated viz existing conservation agencies (notably IUCN) and likely uptake of the system. In short I largely feel much more comfortable with the rigor of the process (not that I doubted it before but it just wasn't as fully documented). It remains a bit of a novelty to present something like this as a Nature paper, but no doubt it would be highly cited, so that remains an editorial decision. Overall I continue to like this particular effort, most notably because of its comprehensiveness (whole globe), the analysis of both ecosystem and biodiversity, the systematic and thoughtful approach to developing this and its focus on processes.

(B) is an obvious capstone or pinnacle outcome of this process, and is of high value to the conservation and policy communities. I do worry a bit that this output gets lost in the whole picture (swamped in the details) and much of the methods and interesting results get pushed into Appendix 6. I increasingly think it might be better to split (A) and (B) apart so that B can get its full exposition for people who are more interested in (A) than (B) and vice versa but that obviously is a big change at this point in the editorial process. I found many of the figures in S6 important and deeply interesting (e.g. Figure S6.2 would be in the main text in any normal presentation of Task B). Figure S6.3 is also information dense and could profitably be in the main text. I loved the Sankey diagrams but understand there are too many to go in the main text, but could a couple go in the main text so that readers know they are there? You can see what it is problematic to treat this topic fairly and thoroughly on top of (A).

Overall I am appreciative of the changes made. At this point I have reviewed the main text and each of the supplements although S4 and S5 (which cover each of the 100+ EFG) I have sampled from half a dozen EFGs across all 3 biomes that I have some professional exposure to. Some minor issues (other than the fact that (B) is getting buried) are:

On Task (A) establishment of an EFG classification:

- 1) I continue to think a 3-level table containing the actual realms/biomes/EFG is more useful and concrete than Figure 2 in the main paper (even though it takes a bit more space). This is essentially the primary output of task A. And yet it is buried in supplemental materials.
- 2) Of the half dozen EFG's in S4 that I looked at (which systems I know reasonably well). I was overall

impressed with their accuracy and the contribution of such succinct, careful summaries. Overall my concerns about the canonical figure on each page were allayed with one exception.

3) I found the abiotic environment and resources pieces and the ecological traits pieces fairly convincing. I was unable to come up with convincing arguments about the biotic interactions however - what e.g. would be the evidence that boreal and temperate broadleaved forests are dominated by herbivory and predation while maritime temperate rainforests (T2.3) are dominated by competition? Predation, competition and herbivory have all been deeply studied as important processes in all 3 ecosystems - but I am not aware of any good evidence the relative importance of these forces. I think this is at risk of capturing biases of researchers past more than true relative importance of processes. There are very active debates right now with Nature-level papers on latitudinal gradients in importance of different species interactions. I just don't think we know the answers yet. There is also a nod towards incorporating dispersal processes. Personally I do not think the biotic interactions and dispersal process components are on the same level of rigor as the resource and abiotic context boxes. Personally I think these two pieces should be dropped.

4) I found S5 helpful in appreciating the degree of review each EFG received and the issues raised. On the one hand it suggests a thorough process. On the other hand it left me with the impression there was only one reviewer per EFG? Elsewhere I thought I saw there were more reviewers. It would be beneficial to highlight number of reviewers as well as nature of issues raised.

5) Thank you for including geospatial files of the EFG. It would be highly desirable to allow them all to be downloaded as a single zip/tar archive file rather than having to click 108 separate links. Also I was a bit surprised to see them as raster rather than shape files. I can see arguments for raster. But if they are raster would it not make sense to have a unified file where the raster cells can an index to the dominant EFG type?

6) Although the paper continues to refer to sublevels 4-6 in the hierarchy that reference biodiversity, there is essentially zero information or examples included about them. It is probably more appropriate to delete their mention in this paper.

On Task B (in addition to feeling like it is in some ways presented as the central result and in other ways is a bit of toss-off only briefly introduced)

7) I think the biggest concern I have is methodological. What is the basis for determining human pressure by calculating % of area that is above a binary threshold of human pressure (% of area under high vs low pressure)? Binary thresholds can often be rather misleading. Given that human pressure was a continuous variable could not some other alternative like an area-weighted average be more informative? In general, I may have missed it among the text and supplements, but I did not find a strong rationale for this particular metric.

7) Figure 3 (main text) is by some reads of the paper the main result. This figure needs more information density

9) Figure 4 (main text) this figure is fairly informative about relative change in pressure, but since "change in pressure" index is an abstract concept it is hard to know how to interpret the scale

10) If you present a table of all 108 EFG you can include the scores for % protected, whatever measure of impact intensity (per #6 I'm not convinced you have the best one) and the change in pressure index. At a minimum such a table could I believe at Nature go as an Extended Table so it is downloaded with the main PDF. In general I missed Extended Figures that I expect to see in Nature - just half a dozen very long supplemental materials.

11) The text on pages 5 and 6 (Main text) is fairly awkward and long. The table mentioned in #8 plus

improved versions of figures 3 & 4 would be much preferable as a means of communication.

Referee #2 (Remarks to the Author):

This is quite an unusual paper, the tip of the iceberg for a very substantial amount of work. It is would be way beyond my expertise to review all of it. As a conservation biologist working at large spatial scales, I have reviewed specific facets of it:

- The need for, and usefulness of, a global ecosystem typology: I agree with the authors that this typology is much needed.
- The conceptual framework: I am well impressed, but have some reserves and recommendations for improvement;
- The assessment of pressures and protection: I am not fully convinced it is very robust.

*** 1) NEED FOR A GLOBAL TYPOLOGY OF ECOSYSTEMS ***

I am fully persuaded by the need for a global classification of ecosystems that is conceptually robust, scalable and spatially explicit. The authors make a very good job at detailing the main immediate applications of this classification in Appendix 7. This work fills a major current data gap and I foresee that its outputs (the classification, the maps) will be plugged immediately into conservation policy and management.

I also foresee it will become a key layer in macro-ecological analyses, even in theoretical studies (in the same way the species distribution data derived from Red List assessments have been underpinning an endless stream of studies).

Given the institutional background of the development of this framework and resulting typology (i.e. IUCN-led), I am persuaded it will gain immediate traction. I am also cautiously optimistic that it will be seen as so valuable to policy/monitoring that resources will materialise to allow it to continue to be refined and updated over time.

*** 2) CONCEPTUAL FRAMEWORK ***

I am impressed by the conceptual depth of the work presented here, and in particular by the ability to bring together terrestrial, marine, freshwater, subterranean and atmospheric into a common typology (Fig 2), underpinned by a common conceptual framework (Fig 1). I agree with the authors that the resulting typology represents a marked development in relation to existing classifications, in particular because of its conceptual underpinning, its comprehensiveness, and its more direct alignment with the concept of ecosystem as a level of biodiversity organisation (as defined by the Convention on Biological Diversity).

This said, I have some specific concerns, questions and recommendations.

* 2.1. Evolutionary processes

Disclaimer: I am not an evolutionary ecologist. I step in because this point does not seem to have been addressed at all in the previous rounds of reviews (either the one by Nature or that previously organised by the IUCN). Ideally, it would be good to have this reviewed by an evolutionary ecologist.

The conceptual framework disregards evolutionary processes. This reflects its grounding in community ecology. For example, the article the authors cite as the main basis for their conceptual framework (HilleRisLambers et al. 2012), clarifies that they do not take into account long-term evolutionary processes (which makes sense, in their case, as their focus is on much smaller geographical scales/shorter time frames). But this is something that cannot be ignored at the large geographic scales covered by the present ecosystem classification (and correspondingly deep evolutionary times), where community composition and function is as much the result of evolution in situ (adaptation, speciation, co-evolution, extinction...) as the result of dispersal/filtering from a pre-existing species' pool.

The word "evolution" is never even mentioned in the main text. In Figure 1, it does not appear at all, with Ecological Traits presented as the end result after a broad species pool goes through a set of filters. The very use of the term "community assembly" reflects an emphasis on a process of combination of emergence from pre-existing pieces (although this term can also be used in evolutionary terms).

Evolution is discussed in Appendix S2 (page 3), but only from the perspective of "evolutionary legacies": the authors discuss how "local evolutionary legacies whose progenitors had long histories of prior occupancy" can (through niche conservatism and limitations on dispersal) result in functional differences between ecosystems, for example in Savannas. This is of course true, but conversely convergent evolution under similar abiotic conditions (or functionally convergent biotic conditions) can also result in functional similarities between ecosystems dominated by distinct evolutionary lineages, for example: cushion vegetation in polar/alpine ecosystems; blind predators in underground ecosystems; swimming predator birds in polar waters (Alcidae/Spheniscidae).

The role of evolution is also currently lacking in the discussion of ecological traits in Appendix S2 (page 4) which are presented as solely the outcomes of the assembly process, even when referring to the traits of species ("life-histories, life-forms, morphology, phenology, behavioural and ecophysiological features"). Surely many of these traits evolved in situ through adaptation, speciation (as well as the disappearance of other species with other traits, via extinction).

The authors point out that biodiversity appears more strongly in the framework when distinguishing between categories at lower levels (4-5-6), whereas the upper levels are more focused on function. My point here is fully related with function: the role of evolutionary processes in determining the traits that determine ecosystem function. Of course evolution also means that we get different species compositions (i.e., different biodiversity) in functionally similar ecosystems distributed

across the world (to be reflected in levels 4-5-6).

As it stands, the lack of explicit integration of evolutionary processes into the framework undermines the assertion that this is a conceptually robust typology, based on a solid theoretical basis.

In my view (but, again, I am not an evolutionary ecologist) this current limitation of the framework does not necessarily invalidate the results (i.e., the classification obtained) because all the drivers currently presented as simple filters are also major evolutionary pressures. Accordingly, I think (but I may be too naïve here) that addressing this point does not require a substantial change in the conceptual framework, but to generalise it to also include evolutionary processes. Specifically, I would recommend:

- Acknowledging (main text, Appendix S2) that all “drivers” (abiotic and biotic, and including human activity) act both as “filters” (determining community assembly from a pre-existing species pool, given dispersal) as well as “evolutionary pressures” (determining community structure and function through in situ speciation and extinction). Also being careful to formalise that the term “community assembly” reflects processes at a diversity of temporal and spatial scales, including through speciation and extinction. And clarifying that current “ecological traits” are the end products both of long-term evolutionary processes and of shorter-term filtering from the current species pool.

- This broader perspective then needs to be reflected in Fig 1. Maybe it would suffice to broaden “Dispersal filter” to “Dispersal filter/speciation/extinction”. The legend would also need to mention evolutionary processes among the feedbacks.

A related reference:

Mittelbach, G.G. & Schemske, D.W. (2015). Ecological and evolutionary perspectives on community assembly. *Trends in Ecology & Evolution*, 30, 241–247.

* 2.2. Anthropogenic effects

Another weak aspect of the conceptual framework is how human activities have been integrated. Currently they are defined as “a special class of biotic interaction” and appear separately in Figure 1 and in Table S3.2 (at the same level as the other four groups of ecological drivers). Yet it seems to me that anthropogenic effects act by modifying particular aspects of each one of the other types of drivers, for example: of resource processes by adding nutrients in agricultural lands or removing water through climate change (leading to desertification); of ambient environmental processes by modifying temperature through climate change, or modifying substrate properties in urban areas; of disturbance regimes by yearly soil perturbation in agricultural fields, or by suppressing/adding fire in some ecosystems; of biotic interactions by adding/removing species, eliminating competitors (in crops); etc. This idea that human actions act through biotic and abiotic drivers does appear in the figures associated with the description of the highly modified EFGs, through arrows going from human activities to ecological traits via other drivers. But some direct arrows remain, which in my view are not justifiable. For example, for T7.1 Annual croplands (page 72 in Appendix 4), “annual

substrate manipulation” is a type of disturbance and should be represented accordingly.

Seeing all human activities as modifiers of various types of drivers would also contribute to addressing something else that bothers me in the current typology, which is a dichotomy between EFGs driven by human effects (croplands, urban, waterpipes, artificial shorelines) and those without any human effects (forests, shrublands, savannas, deserts, rivers...). In practice, most existing ecosystems have already suffered some degree of human modification (e.g. the extinction of terrestrial mega herbivores/mega predators in the Americas and Europe; addition of domesticated herbivores to grasslands in Africa; modification of fire regime in Australia; reduction of top predators in marine ecosystems; introduction of exotic fish in freshwater systems; melting of glaciers through climate change etc). This needs to be acknowledged – should be formalised in some cases in the future descriptions of lower-level units (e.g., African Lakes with their non-native species).

More broadly, it is important to make it clear that most drivers vary in time – through both natural processes and human actions – and that hence ecosystems are temporally and spatially dynamic. This is in line with the authors’ ambition that the proposed typology will “underpin robust generalisations and predictions about ecosystem responses to environmental change and management” but it needs to be better formalised. Also makes sense to mention that ecosystems are not necessarily at equilibrium, they may be responding to past extinctions for example (thinking of work by William Bond on the effects of Moa extinctions in New Zealand vegetation), or invasions, or still be undergoing a process of colonisation as more species find them/adapt to them (e.g. urban ecosystems).

As a related point: the main text (lines 128-130) states that “The model posits that ecosystems share convergent functional traits if they are shaped by similar drivers, and conversely, major changes to these drivers (or their dependencies) cause disassembly, transformation, and ultimately ecosystem collapse” – arguably there is no such thing as “ecosystem collapse”, simply ecosystem transformation (more or less abrupt).

* 2.3 Soil ecosystems

Soil is presented as a substrate (e.g. in Table S3.2), but it can also be seen as a (subterranean) ecosystem on its own right, with distinctive communities and structured by very specific biotic and abiotic drivers. I certainly cannot see the rationale for having a class for endolithic systems (S1.2) but not for soil systems.

* 2.4 The 6 levels

I realise this paper focuses on levels 1-3, and that is fine. But regarding the other three levels, I think a couple of extra sentences are needed in the main text to explain the logic – currently all that the reader has is the legend of Figure 2, which is very cryptic. Personally, I find the logic brilliant in its pragmatism: by allowing both a top-down subdivision of level 3 (e.g. using ecoregions) and a bottom-up aggregation of existing ecosystem classifications (e.g. at the national level) into level 3; these are complementary approaches making the best use of available data, with the second key to ensure local appropriation/uptake. But I had to dive deep into Appendix 3 to understand Figure 2.

Please make it easier to the reader in the main text.

I still don't understand Level 5. I can see how national classifications (Level 6) can be aggregated to EFGs (Level 3). If I am understanding it well, that is what is exemplified in Tables S3.3 and S3.4. But I don't understand what a Level 5 classification (Global Ecosystem Types) corresponds to, and how it would nest underneath the (also global) Ecosystem Functional Groups. As described, it gives the impression Level 5 is simply not needed. Please provide an example.

* 2.5 Terminology

I suspect the authors have already spent considerable time thinking about the nomenclature for their typology, but I will add my two cents.

- I find the term "Ecosystem Functional Group" really unappealing. It is not intuitive (a "group" of what?), the adjective "functional" is ambiguous ("functional group" sounds like a group that functions; whereas the point here is a group of ecosystems aggregated by function). Also, it is way too long and cumbersome, so it will default to another dry acronym in the conservation literature (EFG). Which is a shame, given that this is likely to be a widely used concept in policy and communication. I would recommend something simpler, like "ecosystem class" or "ecosystem category". I realise the authors want to emphasise the "functional" aspect, but it seems like a heavy trade-off with communication. Or they could go for "Functional Ecosystem Class/category" for equal levels of cumbersome but (in my view) more clarity in meaning.

- It is confusing to have "ecotype" (level 4) and "ecosystem type" (levels 5 and 6). It is not obvious if these are different things, or if "ecotype" is a contraction of "ecosystem type"? Why not "Biogeographic ecosystem type"? (note that in evolutionary ecology, an "ecotype" has a very different meaning – a population adapted to particular environmental conditions).

- Note that in Table S3.1. the term used is "Functional Biomes" but elsewhere it is simply "Biomes"

- Given that level 5 is not nested under 4 (they are instead at the same level), it could arguable be clearer to change the numbering to make this more explicit, perhaps as

1 -> 2 -> 3 -> 4A

1 -> 2 -> 3 -> 4B -> 5B

*** 3) ASSESSMENT OF PRESSURES/PROTECTION ***

For each EFG, the authors overlay the indicative distribution map (combining minor and major occurrences) with maps of pressure (human footprint, HF; or marine cumulative human impact,

MCHI) to evaluate relative pressure (EFGs with values of pressure lower than the median are classified as “low pressure” those above as “high pressure”); and with maps of protected areas to evaluate protection levels.

The results of this analysis are only as good as the underlying spatial data. The authors include plenty of caveats on the limitations of the HF/MCHI maps (and yes, they are a problem, particularly the point that these layers do not map well the relevant types of pressures for all ecosystems). But I am equally worried about the quality of the EFG maps. Indeed, many EFGs are mapped very coarsely, with multiple EFGs overlapping over some regions. For the purposes of the analyses done here, this is particularly problematic in regions with high human presence, where EFG boundaries correspond more to broad historical distributions (i.e., the boundaries of biomes/ecoregions) than to maps of the current occurrence of the specific ecosystem types detailed in Appendix 4. For example, EFG T2.2. “Deciduous temperate forests” is mapped as covering most of Europe, North America and China, but these same regions are simultaneously densely covered by (functionally very distinct) EFGs of the Intensive Landuse Biome (T7.1. “Annual croplands”, T7.4. “Urban and industrial ecosystems”, T7.5. “Derived semi-natural pastures and old fields”). So the extremely high pressure (76% degraded) the authors find when overlapping the map of EFG T2.2 with Human Footprint corresponds mostly to historical degradation, rather than reflecting ongoing pressure. The analysis of changes in pressure (by analysing change in HF between 2000 and 2013) is also flawed, because it is done over an area much larger than that actually covered by deciduous temperate forest. This may overestimate change, by treating intensification in EFGs that are not forest (e.g. conversion from pasture to urban) as if it corresponded to forest loss; but it may also underestimate change, by diluting rates of ongoing forest loss across a very wide, non-forest area. Finally, the overlap with protected areas tells us little about how the existing deciduous temperate forests are de facto protected, because the mapped EFGs include large areas that are currently not forest.

The authors state they have also explored weighing more strongly (by a factor of two) areas of major occurrence (“where an ecosystem functional group is very likely to occur”) than areas of minor occurrence (“where an ecosystem functional group is scattered in patches within matrices of other ecosystem functional groups or where they occur in substantial areas but only within a segment of a larger region”), having found “little effect on overall relationships in the degradation status of EFGs”. I wonder if focusing solely on areas of major occurrence would have made more sense. Although I note that for some EFGs, there are also highly transformed regions overlapping areas of major occurrence (e.g.: major urban areas within the area mapped as being of major occurrence for T3.2: Athens, Marseille, Barcelona, Tunis...).

Not quite sure what the solution here is – other than better refining the EFG maps to ensure that they correspond more to “area of occupancy” than to “extent of occurrence” of each ecosystem. As currently done, this analysis is not a good illustration of the advantages of “viewing the world’s ecosystems through a functional lens, rather than through largely biogeographic or biophysical ones” as the authors state, because the boundaries of at least some EFG maps mainly reflect biogeographic units (e.g. the boundaries of ecoregions) rather than functionally-defined ecosystems where they currently occur.

I realise that a previous reviewer (Reviewer #1, point 3) had recommended giving more emphasis to

these results and less to the typology itself, but I would actually recommend the opposite. I find the typology itself is the most useful/robust contribution to the literature, with the analysis of pressure/protection still very preliminary.

Referee #3 (Remarks to the Author):

I read with much interest this new typology of Earth's ecosystems. I did not review the original submission and, as requested by the editor, I have focused on the review of the biome classification provided in Appendix S4, and of the Savannas and grasslands and Deserts/semideserts biomes in particular. I have also carefully revised the response letter provided by the authors to the previous round of review and the main manuscript, so I will also provide general comments on these. But please note that I have not reviewed appendices other than S4, so I assume that other reviewers will have specifically focused on them.

Comments on the main text

I will start with my general impression about the paper itself (main text). This is not certainly the typical article/review one may expect in Nature and, as already highlighted by Reviewer 1, one may argue whether an article-like format is the best way to publish this new classification. I can fully appreciate the motivation of the authors to publish it in a journal like Nature rather than as a book or a report, and I praise them for attempting to synthesize the complex process leading to this new classification in this article. However, the article format makes necessarily that key information to understand and assess the classification proposed is "hidden" in the hundreds of pages of supplementary material, and thus may be missed/can't be found easily. To facilitate that important methodological details are fully understood by readers I would also point to specific lines/sections/figures/tables within the Appendices when citing them in the main text. Said so, it is the decision of the Editor, not that of a reviewer like myself, to decide whether this article is suitable for Nature. As a reader of the journal I certainly would find this classification both interesting and timely; its potential to influence policy and management is also very clear.

I think the authors have done a careful and effective revision of the manuscript and have addressed most of the main criticisms raised in the main round of review. Said so, I have some additional comments on the main text, focused mainly on the structure/clarity of the text and on key methodological issues that I would advice the authors to consider:

I found the structure of the first paragraphs of the manuscript, which are critical in a manuscript for Nature, not very engaging. The first paragraph is OK but I would not start right away in the second presenting the IUCN Global Ecosystem Typology as it comes here out of the blue. Before doing this talk about the existing typologies and set the stage/justify the need for a new typology encapsulating both biodiversity and ecosystem functioning. Then you can talk about the IUCN Global Ecosystem Typology and the novel aspects it includes to continue with the specific objectives of the study (are not clearly stated in the current version of the manuscript) and the main results/discussion implications for guiding both science and policy/management.

Overall, I think the authors should make an extra effort to be as less cryptic and clear as possible, and to better “guide” readers through the contents of this paper. As currently written, and with so much specific terminology and references to Appendices and methods, is easy to get lost, and thus to disconnect (before finishing the second page of the manuscript!).

The methods section should also be improved and expanded (Nature has no word limits for this section), so it includes all the relevant information to facilitate the understanding of how the ecosystem classification/analyses were made. With so many references to Appendices (and so much relevant content included there), so many key details missing from this section and the structure used by the authors is really difficult to get a proper understanding of what has been done, and thus to judge its suitability (believe me, I had to go back and forth several times through the Methods and even so I am not sure if I fully understood what the authors did; and I feel this will happen to most readers). Adding sections to the methodology would also help to navigate through this section without getting lost. Linking the different analyses done to specific objectives of the paper (e.g. evaluation of particular Aichi targets) would also help to better understand why the author are doing them, and thus to judge their suitability.

I have also some general concerns about the analyses conducted. The authors acknowledge the limitations of the pressure indices used, which failed to detect high levels of pressures in ecosystems known to suffer them (e.g., L186-L191). This raises questions about the overall exercise presented in this study, as if the pressure indices used cannot properly account for such pressures in the real world, then the whole analyses conducted here is of limited value to guide management and policy actions. Related to this comment, the temporal analyses of pressure impacts were limited by the availability of data to the periods 2000-2013 and 2008-2013 for HFP and MCHI, respectively (L447). Thus, they are missing the change being experienced by natural ecosystems in the recent 7 years, which as the authors known have been years with an intensification of climate change drivers (e.g. droughts, warming) and human impacts (e.g. wildfires, expansion of cropping areas, intensification of fisheries) across the world. Thus, I wonder whether the conclusions obtained with data from these periods are valid nowadays. Because of this, the suitability of the approach used must be properly justified to convince critical readers (like me!). The justification provided in lines 500-511 does not sound fully convincing to me.

Other specific comments are below

L 72-75: long sentence difficult to read

L 75: I know that the abstract is not the place to provide much details, but I think it would be useful if some of the main pressures are mentioned in brackets here

L 76-78: not clear which ecosystems you refer to, those degraded and least protected?

L 78-79: perhaps this sentence could be better framed as “The classification introduced here can guide policy transformation for ecosystem-specific action, including ... (list some of the key actions this classification may be particularly useful for)”

L 86-87: I may be missing something here, but could you define the “identity” of an ecosystem? I would say that stocks and fluxes of resources, energy, biomass and biodiversity already can accurately define a particular ecosystem.

L104: Mention/reference the typologies you are referring to. This is an example of what I said above about pointing to specific parts/content of the Appendices to facilitate readers finding important content there.

L115: Drivers of what? Specify. The term “ecological drivers” is somewhat vague.

L145-6: which pressures? Which indices? Be more explicit and cite them here.

L152-153: I can't follow the logic of this sentence, please rewrite for clarity.

L179-181: Not clear if this a direction for future work or not, please rewrite for clarity.

L450: unclear where these degrees of freedom come from, please provide more details on the analyses conducted. Mention also the statistical software/packages used to run the different analyses presented in the text.

Comments on Appendix S4 (general comments and comments on biomes T4 and T5)

I revised the general methodology presented in this Appendix and have some reservations about the mapping exercise conducted by the authors, which otherwise is needed in a project like this one. I truly appreciate the complexities involved in delivering reliable and sound-based maps when using so many different inputs and with so many data gaps. However, the authors should think about the use that can be done of the maps provided, particularly if they are validated by the publication of this classification in the peer-reviewed literature. There are some key general issues about the mapping I would like to comment on:

I found the mapping of major and minor occurrences a little bit misleading. I will exemplify this with the mapping of biome T4.5 (Temperate subhumid grasslands) in Spain. This biome occupies most of the territory of this country with minor occurrence. This is so despite most of this territory does not have a subhumid climate (it is rather semiarid and dry-subhumid) and temperate grasslands as described in this biome are quite rare throughout it. Since the biome classification being proposed here has a clearly defined goal to support policies and management actions related to the conservation of biodiversity and ecosystems, I think maps such as those provided for T4.5, which show a clear mismatch between the ecosystem mapped and that found in the field, will not be very helpful for managers and indeed can prompt not effective or even damaging actions when trying to protect them. Of course, the authors indicate that the habitat has a minor occurrence throughout this territory, but how a manager can make a useful/good use of this information? What's the point of mapping a given biome across large portions of territory where it is quite unlikely to find it on the ground? I may be missing something but to me this is a major drawback of the biome mapping presented in this article.

The authors already note that their “maps were designed to be indicative of global distribution patterns and are not intended to represent fine-scale patterns”. This is expected in a global exercise like this one from the scientific point of view given the data sources (and resolution) available. However, and getting back to what I mentioned before, many potential users, particularly land managers and other stakeholders working will use the information contained in the maps (widely accessible via the web [BTW a very nice webpage and map server!]) to guide management actions at local or regional scales. And as noted above these maps will not be very useful for them. Thus, a better justification of the approach used, or even better, a detailed guided about how to use (and not use) the maps and information provided to support biodiversity and ecosystem conservation

policies and actions is warranted.

Also, it would be certainly good if a box explaining how to interpret the diagram with the links between resources/biotic interactions/traits... is included. It is not convenient to go to the main text or to another appendix to understand this figure.

Overall, I found the different sub-biomes included within the T5 biome classification sound and correct to the best of my knowledge. However, it was surprising to me not finding a specific biome category for dry tussock steppes such as those that occupy vast spaces across the SE of Spain and the North of Africa (Morocco, Algeria, Tunisia, Le Hourérou 2001). Much of the area covered by these steppes is incorrectly (based on what one can observe in the field) included as temperate woodlands (T4.4). A similar problem to that discussed for biome T4.5 above is also found for biome T4.4 (temperate woodlands), as it is drawn as a dominant biome across many dryland areas of the Mediterranean Basin that have rainfall levels < 350 mm and that have a shrub- and grass-, rather than tree-dominated vegetation.

Other minor comments on the text of biomes T4 (Grasslands and savannas) and T5 (Deserts and semi-deserts) are the following:

- “Herbivory is the primary driver in highly fertile and productive systems, whereas fire is the primary driver in less fertile and lower productivity systems.” Driver of what? unclear.
- “Nutrient gradients are exacerbated volatilisation during fire and the loss of nutrients in smoke” I can’t understand this sentence
- I would not say that biotic interactions are weak in desert biomes, as it is well known that facilitative and competitive plant-plant interactions, as well as plant-herbivore interactions, can be intense in these ecosystems (e.g. Fowler 1986, Graff et al. 2007, Graff & Aguiar 2016).

I hope that despite my criticisms, which I have made with the idea to be constructive, you will find the comments provided useful to further revise and improve this work.

References

Fowler, N. The Role of Competition in Plant Communities in Arid and Semiarid Regions. *Annual Review of Ecology and Systematics* 17, 89–110 (1986).

Graff, P. & Aguiar, M. R. Do species’ strategies and type of stress predict net positive effects in an arid ecosystem? *Ecology* 98, 794–806 (2017).

Graff, P., Aguiar, M. R. & Chaneton, E. J. Shifts in Positive and Negative Plant Interactions Along a Grazing Intensity Gradient. *Ecology* 88, 188–199 (2007).

Le Houérou, H. N. Biogeography of the arid steppeland north of the Sahara. *Journal of Arid Environments* 48, 103–128 (2001).

Referee #4 (Remarks to the Author):

The manuscript by Keith et al. describes the IUCN global typology of ecosystems, based on the functional traits of the species and ecosystems. In this era of global change and human impacts there is a dire need to make global comparisons of ecosystems, the impacts they receive, and their threats. The rationale of the typology is explained, and the utility is illustrated by showing how different terrestrial, freshwater and marine ecosystems are protected and potentially threatened by global change.

The strength of the manuscript are:

- 1) a unifying global typology that allows to classify bewildering different ecosystems that are often assessed separately (e.g., terrestrial, freshwater, marine),
- 2) a consistent classification based on community assembly and the functioning of organisms and ecosystems, which allows for a better mechanistic understanding, assessment and prediction of the consequences of environmental change,
- 3) a separate inclusion of humans as an environmental driver, which allows to assess the consequences of human activity for the biodiversity and functioning of the planet, and allows to design policies to change human activities or mitigate their effects,
- 4) the application of the typology by describing and mapping the 100 ecosystem types. The description in Appendix S1 with one page factsheets are a pleasure to read, as they are nice and concrete, succinct, well written, conceptually consistent by showing the same conceptual diagram with different drivers, and nicely illustrated with a clear beautiful photo conveying the message, and reference for further reading

The weak points of the manuscript are

- 1) A very vague, poorly written main text article which was for large parts not understandable (even no for me as a functional ecologist!) and below publication standards
- 2) The conceptual diagram should be improved as the terminology is (in my opinion) not consistent, it should made more clear that humans affect ecosystems by affecting the other drivers
- 3) Conceptually, distinguish better between traits of individuals/species, and traits of ecosystems. It is unclear what you mean with 'ecosystem traits', and whether you suggest that ecosystems are filtered out by the environment (which I think is conceptually wrong).
- 4) Better explain and show how you used in practice these functional traits to classify the ecosystems. It seemed to me that you used a-priori defined ecosystems and described them afterwards with your functional typology, whereas it should of course have been done the other way around!

I was specifically asked to look at the descriptions of the tropical systems, which I do further below. I have reviewed 7 tropical systems. Given the fact that the nature reader thinks that everything what is published in Nature has been scrutinized and is true, I echo the concerns of the previous reviewer that it is imperative that ALL 100 ecosystem descriptions are checked by specialists. So you still have to look for reviewers for the other 93 ...

Overall, this has been an admirable and Herculean task, for which I congratulate the authors. Please improve the main text article, as it does not do justice to the rest of the work you have done. Please find my major and minor comments below, which I hope are of help to improve the manuscript.

MAJOR COMMENTS

1. IMPROVE THE INTRODUCTION TEXT. I am a functional ecologist and biologist, do fieldwork in a variety of ecosystems across the world, and have affinity with management and conservation. So I thought that I should be able to understand the main text, and get inspired by it. I must admit that I found the main text very vague, unclear, full of undefined jargon, and below publication standards. So if the authors want to reach and convince a wider audience, then please invest time to make it attractive, understandable and accessible.

2. IMPROVE YOUR CONCEPTUAL DIAGRAM. Appendix S2 (conceptual foundations) is well written, interesting, and a pleasure to read. Figure 1 presents your conceptual model, which is the cornerstone of your whole typology, and returns in each ecosystem description. The wording, should therefore be crisp and clear, and above, all, correct. In my opinion, several terms and classifications are now incorrect and should be improved:

- Human activity. The real conceptual problem I have is that human activity does not DIRECTLY affect the ecosystem traits and species traits (as your arrow suggests), but human activity INDIRECTLY affects the ecological traits by changing environmental conditions, resources, disturbance regimes and biotic interaction. This applies for all the four items you give as example in your human activity box. For example, structural alteration works through disturbance. Resource use is either a disturbance (you remove biomass), or a biotic interaction (it equals predation). Movement of biota is a biotic interaction (dispersal). Climate change works through changing temperature (an environmental condition), and carbon and water availability (resources). So I really think it is conceptually wrong to frame human activity as having an independent effect on the traits, You should have arrows from human activity to disturbance regime, biotic interactions, resources and environmental conditions. In the same way as you explicitly draw an arrow from environmental conditions to resources. It therefore does not suffice to say that those human effects are included through the circle with the broken line.

- Human activity. I understand that in the Anthropocene and as preservationists you want to frame humans not as being part of nature, but as being outside nature and affecting nature. Many people may disagree with you, so justify somewhere why you put human activity as a separate box.

- The term "Resource processes" is incorrect. What you show in the box are resources (water nutrients, etc), so no processes of resource uptake or loss. So rename this box "Resources"

- The term "Ambient environmental processes" is incorrect. What you show in the box are environmental conditions, not processes. The word "ambient" is also confusing. If these conditions are ambient, then you should also label your "resources" as "Ambient resources". I think the correct name of this box is "Environmental conditions". And please define in the legend "Kinetic energy". I did not have a clue what you were referring to

- The term "Biotic interactions" does not match the names inside the box. "Competitors" are organisms that are interacting, it is not a biotic interaction. So replace the names in such way that they reflect real interactions; "Competitors" should be "Competition", "Predators" should be "Predation", Mutualists should be mutualism. And think yourself how to rename "pathogens" and

“engineers”.

- “Engineer” is in my opinion not a biotic interaction. Facilitation is. Is that what you mean? If so, use facilitation. That is closer to your “community assembly theory”. Or do you mean that “ecosystem engineers” modify the landscape? Then it is in my opinion not a biotic interaction anymore.

- Disturbance regime. If you define a disturbance as a sudden event that destroys biomass then flooding is NOT a disturbance regime (it is not sudden, and generally does not remove biomass), but a stress. I think it conceptually belongs to the environmental box, because flooding modifies resource availability by reducing oxygen and light, and increasing water availability and nutrient availability (through deposition). And please define in the legend “Mass movement” and “Igneous activity”. I did not have a clue what you were referring to.

- Ecological traits. In the central circle make a distinction between what are your ecosystem traits and what are your species traits. For me as a functional trait ecologists it is VERY confusing that they are all listed as ecological traits, as the ecosystem traits are an emergent property of the species traits.

- Species pool. Move the word to a different place in your diagram. Now it erroneously suggest that it refers to the broken circle

3. CLARIFY CONCEPTUALLY ECOLOGICAL TRAITS AND ECOSYSTEM TRAITS. Ecological traits are the basis of your typology. Maybe I have missed it, but you nowhere define what a trait is. In traditional functional ecology a trait is a property of an individual, affecting its performance (growth, survival, reproduction, fitness) (Violle et al. 2014 PNAS). I can imagine that you assign a trait to a species. But then the question is: what is an “ecosystem level trait”? In my humble opinion, assembly theory is about the assembly of individuals and species into a community. A trait is therefore a property of an individual or a species. In my opinion, your ecosystem level traits (productivity, diversity, trophic structure and physiognomy) are the emergent consequences of the traits of the individuals in combination with the environmental conditions. For example, primary productivity of a forest is determined by the number of individuals, and by their size, total leaf area and photosynthetic capacity (all properties of an individual), in combination with the environmental conditions (light, temperature) that determine photosynthetic rates). So using this analogy, I do not see why productivity would be an ecological “trait” of the ecosystem. It is simply the emergent consequence of species traits and environmental conditions. To circumvent this problem, you could define a “trait” simply as an attribute of an individual, species, or ecosystem. I am fine with that, but in your conceptual framework you suggest that ecosystems are filtered out by the environment based on their ecosystem traits, and that is of course not true. I think the environment does not filter out ecosystem A or B. It filters the individuals and species belonging to ecosystem A or B.

4. CLARIFY HOW YOU CLASSIFIED YOUR ECOSYSTEMS BASED ON FUNCTIONAL TRAITS. Better explain and show how you used in practice these functional traits to classify the ecosystems. It seemed to me that you used a-priori defined ecosystems and described them afterwards with your functional typology, whereas it should of course have been done the other way around!

CONTENT-WISE COMMENTS MAIN TEXT

L139. You seem to have a very static view on ecosystems. Most are dynamic and resilient (to a

certain extent)

L140. ASSESSING THE RISK OF PRESSURES. Nice that you have a human pressure map. But who says that these are relevant to the functioning of ecosystems. So you assume a threat, but as you did not measure ecosystem response it is just a supposed threat. So how relevant is this exercise?

FIG 1. See my major comment 2 at the beginning of my review

FIG 2. I did not understand the figure, nor the legend

FIG 3. WHAT IS THE MEANING OF ABOVE OR BELOW AVERAGE. Everything is relative, so how important it is that an ecosystem faces above- or below pressures (your dotted lines). The question is how much a certain pressure results in an absolute response of the system (some systems are resistant or resilient, others are sensitive)

FIG3. WHAT IS THE MEANING THAT PROTECTION DECLINES WITH HIGH PRESSURE? It could be that people selected parks in remote areas as we do not need those (= probably areas with low pressure), it could be that conservation results in less pressure. We simply can not tell. So what can we learn from this? In that case you should have done a temporal analysis (once it is a park, does the pressure decline).

FIG 4., Why a one-tailed test? I think it is more robust and appropriate if you would do two-tailed test. As an ecologists my prediction often was the opposite of what I found, so a one-tailed test makes little sense. And what is in the end the added value of having once conceptual framework if you still analyze terrestrial, freshwater, and marine systems separately?

MINOR COMMENTS ON INTRO MAIN TEXT:

DEFINE TERMS. If this approach is to be a vehicle for multidisciplinary or interdisciplinary collaboration, then all terms should be defined very clearly. It would be great if the authors would add in the main text a box with a glossary. Please define what you mean with “functionally similar responses (l68), “functions” (l71), “Ecosystem Functional Groups” (l74), “resources” (l85), “ecological functions” (l102), “biota” (l102), “filters” (l115), “community assembly theory” (l116), “ecosystem traits” (l116) “disassembly and reassembly” (l125) “semi-deterministic resource appropriation” (l125), “physical restructuring” (l124), “movement of biota” (l126), “convergent functional traits” (do you mean converging from an evolutionary perspective? (l128), “functionally based ecosystem typology” (l132), “generic indices” (l148), “cryogenic ecosystems” (l167).

CLARIFY TEXT. Large parts of the introductory text are very vague or totally unclear.

l66. What do you mean that “ecosystems contribute to biodiversity”?? I would say that ecosystems vary in their biodiversity

L74-75. This sentence does not flow and it is unclear what <17 or <10 refers to,

L75-678 This is a mixed bag of many things. Provide some conceptual structure. Now it feels like a random list of things.

78-79: What do you mean with “globally comprehensive typology”, “policy transformation” and “ecosystem specific action.”? The links that you suggest are totally vague. L

80: a typology is not an infrastructure, it is a tool.

L83-84: This is a complicated sentence.

L86: what is the difference between energy and biomass?

L87-88. What do you mean with “The identity of biota is also central to biodiversity concepts, conservation goals, and human values”?

- L92. What do you mean with “design and resource on-ground solutions”?
- L83. Why is there a dual need for sustainability and conservation. Why do you need both? And do you suggest that there is a trade-off between the two?
- L95-97. Why are especially these design principles crucial? Please justify
- L99. How different are these “biophysical attributes” from your stocks and fluxes?
- L99-100. Very good point!
- L102. What is the difference between an ecological function and an ecosystem function? Please use your terms consistently throughout. It seems that sometimes you use synonyms, which is confusing
- L105 Why would that “limit the ability to generalise about properties of ecosystems grouped together”?
- L109. What do you mean with “to serve as a template”? What are your “units of classification”?
- L111. What do you mean with “Interactions and dependencies amongst drivers and traits”. This is very vague
- L127. What is the difference between “dependency” and “interactions”?
- L131-134. I am totally lost
- L134. To be adopted in 2020 or has it been adopted?
- L136 what you say about the upper and lower groupings is not understandable

MINOR COMMENTS ON APPENDIX S2

- P4 fifth par. Define assembly processes. This is crucial, as it is the cornerstone of your theory. Geomorphology and turbulence are not assembly processes. The assembly process is in my opinion dispersal, colonization, establishment, environmental filtering, etc.
- P2 last paragraph. Flooding is not a disturbance
- P4 first par. Define “Ecological traits” (I intuitively consider it a species attribute, rather than an ecosystem attribute).
- P4 third paragraph. Subterranean systems do have herbivores; nematodes and many other macrofauna browse on plant roots
- P4 third par. Define “aphotic sensory mechanisms”. This is really jargon

COMMENTS ON TROPICAL ECOSYSTEM DESCRIPTIONS (S4)

T1.1 TROPICAL -SUBTROPICAL LOWLAND RAINFORESTS

Ecological traits: What do you mean with “Bottom-up regulatory processes are fuelled by large autochthonous energy sources”?

- It seems that you like to mention SLA but that you have not captured the concept, as every time you say that the SLA is high (or low) and it should be the other way around. SLA is the product of (1/leaf thickness) and (1 leaf density). Rainforest trees have thick and dense leaves and therefore a LOW SLA instead of high SLA, as they are adapted to low light or nutrient-poor soils. Low SLA comes along with long-lived persistent leaves. So they can retain the scarce carbon and nutrients (and rain can leach out nutrients less easily from the leaves) for a longer time.
- Rainforest trees do not have rapid growth but slow growth (because they have low SLA and dense wood). A high growth potential does not make sense in a resource poor environment (low light, low nutrients). They grow on average 1mm in diameter per year. Only pioneer trees have fast growth.
- Add palms to the life forms

- Say “by tropical storms SUCH AS near coastal forests”. Large storms also occur in central amazonia where km² of trees might be blown down
- PLEASE remove “Many trees exhibit leaf form plasticity on a single individual” ALL plants over the world show a strong plastic response to light, this is not specific for tropical trees and it does not explain their success
- Please remove “some species germinate on tree trunks” . This is a detail which is not typical for most rainforests and not an important functional response
- Key ecological drivers. Add that soils range from very fertile, such as volcanic soils, to very infertile, such as on old weathered acidic soils. Because of high rainfall and strong weathering P often limits productivity
- Conceptual diagram: what do you mean with “limited” competitive release? I guess you mean “strong winds” instead of “high winds”. Why do biotic interactions lead to “bottom-up regulation”. Add to resources “generally nutrient-limited”

T1.2 TROPICAL SUBTROPICAL DRY FORESTS AND THICKETS

- Do you mean that tree and vertebrate diversity higher than most other TEMPERATE forest systems? If so, add temperate.
- I was highly surprised that you said that trees have typically thin bark and low fire tolerance. Compared to what? Compared to savanna yes, compared to rainforest no. My experience is that trees can have thick barks (e.g., several Bombacaceae, Apocynaceae, Baobabs, especially the larger drought deciduous ones that store water in the stem. Next to that there are many thin barked ones, especially the shrubby plants that can resprout easily. So I would say that bark variation, and hence, fire tolerance varies a lot
- I am highly surprised that you talk about gap-phase dynamics. Because dry forest have a seasonally deciduous and relatively open canopy and small leaflets, there is a lot of light in the understory. Many plants can recruit and hang on in the understory. Gap phase dynamics are generally NOT important, unless you talk about extremely light demanding species that may need large disturbances such as fires or hurricanes.
- What do you mean with “These forests may be involved in fire-regulated stable state dynamics with savannas”? That is a totally vague and random remark
- Maybe mention that many dry forests have a high abundance of nitrogen fixing species (e.g., M. Gei et al. 2018 Nature Ecology & Evolution)
- In what areas are tropical storms important? In the Mexican Yucatan and Caribbean isles?? I can not think of any other dry forest area where this would be important, so I wonder whether this is the exception rather than the rule
- Conceptual diagram. I would not say that light is limiting. Maybe if you compare it to a savanna yes, but in general not. Say for warm temperatures that they have RELATIVELY low diurnal and seasonal variability. The variability is higher than in tropical rainforests, I think, due to less clouds, and cooling due to open night skies. Why do you say that there are canopy herbivores. In all forest systems there are canopy herbivores. Why do you say the canopy is dense? The canopy is in my opinion relatively open compared to rainforests, and temperate forests. The tree density is higher. I would remove shade tolerance and gap dynamics
- Citations. Maybe cite Sanchez-Azofeifa et al. 2013 book: Tropical dry forests in the Americas: ecology, conservation, and management. I guess there is more ecosystem functioning there than in Toby Penningtons book (apologies!;)

T1.3 TROPICAL-SUBTROPICAL MONTANE RAINFOREST

- It is wrong that montane rainforest have a high specific leaf area (SLA). SLA is leaf area divided by leaf mass. Montane species have thick, dense, leathery or coriaceous leaves to retain scarce nutrients and reduce damage by UV radiation. I am flabbergasted that the expert authors have overseen this
- Why is elfin woodland not a separate functional ecosystem?? They have a totally different structure and functioning compared to cloudforests!!! I would make it a separate category
- What do you mean with “productivity is fueled by autochthonous energy??” That plants are autotrophic and photosynthesizing? If so say so, but this is an odd comment. An elevational transect through the Andes found that productivity is mainly limited by radiation (Fyllas et al. 2017 EcolLett Solar radiation and functional traits explain the decline of forest primary productivity along a tropical elevation gradient)
- Say that taxonomic TREE diversity is low, and add that there is a very high diversity of epiphytes (orchids, bromeliads, lichens, mosses, ferns)
- Add that gap-phase dynamics are also driven by landslides (often driven by geologically young substrates, steep terrain, high rainfall and waterlogged soils)
- Key ecological drivers. Say something that in general montane systems are especially limited by soil N? (Grubb 1977 Control of forest growth and distribution on wet tropical mountains: with special reference to mineral nutrition). Or has this proven to be wrong?
- Conceptual diagram. Why do you say that landslides lead to competitive release?? All trees are gone, so nobody is released. I can imagine that storms lead to competitive release. Plant competition is in nearly all closed vegetation cover systems of the world, so why do you mention it here? Because it is relatively shaded? Why do you say “bottom up regulation”? No clue what you are referring to. I disagree that diversity is moderate to low. It is moderate to low tree diversity but high epiphyte diversity. Add to “Abundant bryophytes” also “epiphytes and lichens”. Add also nutrients to your ambient environment, as it co-limits productivity?

T1.4 TROPICAL HEATH FOREST

- For a global classification I would NOT use the word “heath forest”. This comes traditionally from Asia (Kerangas forest) and is only used there. The term “Heidewald” was introduced by Winkler (1914). It gives to me associations with Ericaceae shrubland, which it is not. It is a forest on siliceous, acid, nutrient poor soil, so I think “white sand forest” is more clear and appropriate in the spirit of your FUNCTIONAL classification, and it is also known as such in the Neotropics.
- Ecological traits. I would add to the first sentence that it has a high density of thin stems. Also add that there are many plants and animals with special adaptations to low nutrients (pitcher plants), and that there are special mutualisms (abundant myrmecophytes). Richards mentions as feature a tendency towards dominance
- Drivers: Proctor suggests that aluminium toxicity plays a role. I do not know whether that has been accepted or rejected. Please check and add it to drivers and diagram if needed
- Conceptual diagram: sandy soils and shallow rooting lead not only to flooding but also to a water shortage in the dry season. So I would put an arrow from sandy substrate to resources, and not only put under resources water surplus but also shortage. In ecological traits you say “low diversity”. Low compared to what? Temperate forests? Tropical rainforest? I believe there are many species there. I would put high dominance as an ecological trait. Rather than microphyll I would say that leaves are

leathery (that better reflects the nutrient limitation and functioning)

- Distribution: P.W. Richards classifies Caatinga as heath forest. But you did not include it here on your map? Why not? You say that heath forest is not known from Africa, but Peace & MacDonald (Biotropica 1981) mention that it also occurs in Gabon (I do not know from where they have that wisdom)
- References: they feel a bit detailed and random. Do you expect the readers to get an overview of the system based on these? Maybe better cite PW Richards 1996. And I must imagine that there is a better book or review article on these forests

T4.1 Trophic savannas.

- I am not a savanna ecologist, but have travelled through the systems.
- Ecological traits: I would add that the mammals can represent a large biomass, and that mammals show rotational grazing. And that in East Africa (and South Africa) mammals show strong migratory patterns aligned with the rainfall and nutrient needs when lactating. I am not sure whether this applies to West Africa. What do you mean with “sustaining the system through positive feedbacks and limiting fire fuels”? Why do you add “Nitrogen fixation, recycling, and deposition by animals exceeds volatilisation”? Why is nitrogen volatilisation a problem when there is little fire?
- Key ecological drivers. You say that low intensity fires have return intervals of 5-50 years. I guess you refer here to natural fires? I think that in West Africa fire return interval is each year as people burn it regularly for hunting or to get rid of snakes etc., and to renew the grasses for their cattle
- Distribution: In the map you classify the west African Sahel zone as trophic savanna. Maybe originally, but I guess now most animals are hunted and little is left. The same applies to the Indian subcontinent. At some places you can find still intact communities. Maybe the West African savannas are now de-facto pyrogenic savannas? In the description be explicit that “Asia” is the Indian subcontinent
- Conceptual diagram. Ecological traits: mention high mammal abundance. Biotic interactions: It is not clear what you mean with “engineers (+ve feedbacks)” and “strong and weak top down processes” Maybe use “Strong or weak herbivore control”?

T4.1 PYRIC TUSSOCK SAVANNAS

- Ecological traits: what do you mean with “grasses cure in winter” and “local endemism is low across all taxa”?
- You highlight deciduousness, but some savannas are dominated by evergreen species (Australia) and others by deciduous species (South America).
- Shouldn't you mention the strong variation in tree cover (in the Brazilian cerrado you have at least campo sujo, cerrado, and cerradao) and the importance of gallery forests?
- You say that plant defences against herbivores such as spinescence are less prominent. Then be explicit that you mention about MAMALLIAN herbivores, as insect herbivory can be high, and at least in the cerrado many plants defend themselves with high silicate concentrations.
- When you talk about detritivores, do you refer about termites? If so, say so.
- In the cerrado the resprouters and reseederers are important; there is a high diversity because of the fire regime and the reseeding plants. Maybe worth mentioning? I do not know whether that applies to other savannas.
- Why do you say that plants have a high SLA? High for the deciduous species, but low for the

evergreen species!

- Maybe mention that belowground carbohydrate storage is important for resprouting success after fire, and many species have massive belowground storage organs?

Referee #5 (Remarks to the Author):

This paper represents a major step forward in a unified understanding and classification of the natural world in a way that can guide management and protection at very large scales. It directly addresses a common criticism of environmental protection: regulation and management is mostly based upon conservation of particular species or specific human concerns such as pollutants entering water or air. The authors' approach provides an important step toward conservation or restoration of ecosystems using a more holistic approach. Global classification of biotic (biodiversity as well as ecosystem function) patterns is the most important issue with respect to the biodiversity crisis and maintaining ecosystem services vital to humanity.

A criticism was in the prior reviews that the classification was not perfect for several specific reasons. The fact is that humans like to classify things as a way to deal with them conceptually, and it is necessary to do things like write laws and treaties. This creates problems similar to those seen in ecology with biome classification. In the real world biomes grade into each other and there are not clean lines. For example, if you go to the Cerrado in Brazil, the uplands are savannah and if you go to the riparian zones, the vegetation is more like the Atlantic Rainforest to the east. As long as the authors explicitly note this reality in the caveats to the general framework, I do not think that criticisms of classification are too much of a worry. It is just a tool for humans to deal with the world, and we only need to be aware of the limitations of that tool.

The manuscript and methodology behind it have gone through extensive external review, and the authors have made a strong effort to respond to those reviews. Appendix s5 on development, review, and revision was a pillar of this work. I think that this is reflective of a developing system that will be continuously revised and refined in the future (sort of like the IUCN red list when new data, taxonomic information, or human caused changes become available or occur). Thus, I do not view this paper as a static result as much as a milestone report on an ongoing journey. I think a parallel is the publication of the human genome. It was not complete, and continues to be refined, but it was a large step forward.

The functional diagram figure 1 should include seasonality? The methods do mention seasonality, but the key conceptual diagram only mentions it indirectly under climate change. Temporal factors are a missing major axis in classification that is ultimately included in the fine details, but not indicated at the top conceptual level.

The inclusion of subterranean habitats particularly impressed me. This highly underappreciated area needs protection, as well as representing an ecosystem that influences global biogeochemistry, linkages between land and aquatic habitats, and can be a hotspot of unique biodiversity. I am not aware of any efforts to make such classifications and determinations and applaud the authors for

including this habitat.

Some previous efforts have justified this manuscript. Whittaker (1970) stated biomes could be developed for aquatic systems based on physiognomy but noted that aquatic communities intergrade with each other in different ways and are less dependent on global climatic gradients than are terrestrial habitats (presumably because water is less limiting). My own efforts related to this paper might assist in the revision (Dodds et al. 2015, 2019). In these papers, we discuss the ideas of translating the terrestrial concept of biome to freshwaters. The Dodds et al. (2019) paper specifically discusses the fact that ecosystem characteristics, in addition to phylogenetic information, informs biomes. The author's manuscript acknowledges this view. Many of our predictions or reporting of patterns hinges upon the observation that the line between actual and potential evapotranspiration is a key factor in determination of fundamental characteristics of intermittency of freshwater habitats. This is a key factor in all freshwater habitats (in addition to terrestrial biomes).

While Figures 1 and 2 help understand the classification approach, a conceptual figure on the approach for assigning pressures would be useful for the naive reader (myself included), as they appear to be a combination of two independent indices of pressure. Figure 2 is somewhat nebulous, but it is trying to encapsulate a very complex process, and I am not sure how to better do that. The earlier suggestion of a detailed supplementary table goes a good way toward that.

Some minor comments:

This statement in the abstract

"Applying this typology, we find that the most degraded and least protected ecosystems on Earth include 18 of 85 terrestrial, freshwater and marine Ecosystem Functional Groups with >70% of extent exposed to high pressures and <17%, or <10% for marine, represented in protected areas."

Is a bit convoluted and could be made more strongly.

How about one fifth of Earths ecosystems are severely degraded and not protected with less than 70% of freshwater and marine areas protected

If there is room maybe talk about some of the rarest or most endangered habitats that clearly need the most protection? Sort of like biodiversity hot-spots in need of conservation?

Getting in the weeds a little, river regulation is a major impact, and I could not see any mention of dams or reservoirs.

Line 70 define IUCN

There is a link to the additional information on zenodo, and it needs updating. I got to the most recent version from there, but does not directly link.

Dodds, W. K., L. Bruckerhoff, D. Batzer, A. Schechner, C. Pennock, E. Renner, F. Tromboni, K. Bigham, and S. Grieger. 2019. The freshwater biome gradient framework: predicting macroscale properties based on latitude, altitude, and precipitation. *Ecosphere* 10:e02786.

Dodds, W. K., K. Gido, M. R. Whiles, M. D. Daniels, and B. P. Grudzinski. 2015. The Stream Biome Gradient Concept: factors controlling lotic systems across broad biogeographic scales. *Freshwater Science* 34:1-19.

Whittaker, R. H. 1970. *Communities and ecosystems*.

Referee #6 (Remarks to the Author):

Please see attached PDF.

Referee #7 (Remarks to the Author):

In this ambitious paper, the authors have attempted to distill a truly global classification scheme for ecosystems that can be used to guide international management, such as maintenance of biological diversity under the Aichi targets, by emphasizing and drawing on established ecological theory. The authors include a litany of supplemental material, lasting many of hundreds of pages, supporting their expert classifications that emerged from an international working group that initially convened in 2017 and considered two dozen existing typologies. They go on to show how their new classifications relate to protected status and human impacts, suggesting that fewer of these ecosystems are protected and more are vulnerable to human impacts than is needed to meet current sustainability goals.

In full disclosure, I did not review the first iteration of this manuscript but was brought on as additional referee with marine expertise. What the authors have produced is noteworthy based on sheer volume alone. But I honestly cannot say having read the paper and some of the appendices, I know exactly what the novel contribution is here, who is the intended audience, or how it is useful or improves on existing classifications. Some of this content is more clear in the response to the reviewers.

A primary goal appears to be to produce a globally consistent and integrative classification scheme for ecosystems (Appendix S4), which is rooted entirely in ecological theory and therefore unbiased by existing perceptions or treatments of these biomes. They adopt an environmental filtering-limiting similarity approach, working from large earth systems (marine, terrestrial, atmospheric) down to individual biomes whose constituents—and therefore biodiversity—are determined by similar abiotic and biotic drivers. Putting aside for the moment that the notion of environmental filtering and competitive exclusion can lead to both the same and wildly different assemblages (see transformative work by Jonathan Levine and Margaret Mayfield published in 2010 in *Ecology*), I am still struggling to understand how this new approach leads to categories that are fundamentally different than what we have always been managing for, e.g., coral reefs, temperate forests, etc. and therefore why this new typology is warranted. Or, rather, how including a comprehensive appendix crossing realms is any different than copy/pasting together 3 separate documents pertaining to terrestrial, marine, and freshwater systems. In other words, there is nothing synergistic about

bringing these realms together under this new framework; it is merely the sum of its parts. In fact, I can see one instance where the resulting categories have clearly suffered from pre-existing biases and do not necessarily follow the proposed scheme: seagrasses (M1.1) and kelps (M1.2). Both are marine primary producers inhabiting shallow coastal shelf regions, provide vertical structure that support diverse food webs, are limited by light, nutrients and oxygen, have similar dispersal strategies (forgiving reproductive differences, both use spores/seeds to disperse over large distances), and are subject to the same environmental (e.g., temperature, storms) and biological pressures (e.g., grazing), including human impacts (e.g., nutrient run-off, urbanization). They only differ in their geographic distribution (with seagrasses abundant in both temperate and tropical zones whereas kelps are largely temperate and restricted to a few areas in the tropics), but this is not a criterion in their Table S1.1. In fact, both kelp and seagrass ecosystems were recently grouped under “marine macrophytes” in a review for the Global Ocean Observing System (see Duffy et al. 2019 in *Frontiers in Marine Science*). I am wondering, given the description in Appendix S2, why these systems were kept separate? It seems that they would logically cluster together, as has been done by GOOS.

Nevertheless, such an overarching typology ignores the fact that most if not all of the management occurs not at the global (planetary) scale, but at the regional or even local scale, with rare exceptions (e.g., European Water Directive). I am not aware of management plans that consider globally broad categories and goals, but generally set targets based on regional historical baselines. Seagrasses in Florida are not managed the same as in Vancouver, or Norway, or Australia, even when attempting to meet international targets (such as Nationally Determined Contributions under the Paris Climate Agreement). As another example, pelagic fisheries stocks in the Atlantic are managed very differently than in the Pacific due to the much longer period of exploitation, even by the same country. I am not convinced by the current presentation of the benefits of plugging into this hierarchical topology (which again, leads to the same management entities, or EFGs, as previous schemes). Perhaps this is useful at the ultimate level, i.e., for international governance (IUCN), but the authors did not state who their audience or userbase is, so this is merely speculation on my part. (note: they elaborate more in the response to the reviewers, but this detail is not included in the manuscript, presumably due to space constraints).

Indeed, the new scheme fails to yield new insights into the value of protection or response to human stressors. We already know that marine protected areas are generally concentrated on the coast, are therefore biased towards particular foundational species (e.g., corals), and are nominally effective depending on the size, isolation, enforcement, and age of the reserve (see review by Edgar et al. 2013 in *Nature*). We also know the unregulated areas that are under heavy exploitation (e.g., deep-sea mining) are also under greatest threat. There is not a conclusion here that is not echoed in other recent syntheses, or present novel statistics that integrate across realms in a way that is not possible by simply combining the results of these individual studies (again: this typology is just the sum of its parts).

I think the major issue is that, while 20 previous typologies did not consider ecological constraints EXPLICITLY, they do so implicitly. Terrestrial forests are different than kelp forests: we can see that based on where they are found and what they look like, captured by their evolutionary history. Clearly, an explicit consideration of ecological notions of filtering and competition has not led us to any different classifications or, if it has, the authors have not made that contrast clear. The paper would benefit immensely if the authors could contrast the novelty of their EFGs with those in other schema. I also find the suggestion that because this typology is already used by some entities and

others have found it useful it should be published is rather weak: there are undoubtedly thousands of national and international schema that do not find their way into the pages of Nature because they are not stimulating groundbreaking science worth of the highest impact journal in the world, even if they are getting the job done.

In sum, I wish I could be more positive about this paper. Much of the content is useful review: the original Reviewer 1 is correct, the massive appendices could be better served as a technical report (and I see will be published in a forthcoming IUCN report). But I also question the value of this new framework and whether it adds anything substantial to our understanding or our capacity to meet global sustainability goals based on the current presentation. The world is now facing a proliferation of schema, typologies, classifications, and all manner of guiding frameworks: while this manuscript is undoubtedly an achievement and represents considerable consensus, I was not convinced it is, as we are always searching for, a “better mousetrap.”

The manuscript by Keith et al. describes the IUCN global typology of ecosystems, based on the functional traits of the species and ecosystems. In this era of global change and human impacts there is a dire need to make global comparisons of ecosystems, the impacts they receive, and their threats. The rationale of the typology is explained, and the utility is illustrated by showing how different terrestrial, freshwater and marine ecosystems are protected and potentially threatened by global change.

The strength of the manuscript are:

- 1) a unifying global typology that allows to classify bewildering different ecosystems that are often assessed separately (e.g., terrestrial, freshwater, marine),
- 2) a consistent classification based on community assembly and the functioning of organisms and ecosystems, which allows for a better mechanistic understanding, assessment and prediction of the consequences of environmental change,
- 3) a separate inclusion of humans as an environmental driver, which allows to assess the consequences of human activity for the biodiversity and functioning of the planet, and allows to design policies to change human activities or mitigate their effects,
- 4) the application of the typology by describing and mapping the 100 ecosystem types. The description in Appendix S1 with one page factsheets are a pleasure to read, as they are nice and concrete, succinct, well written, conceptually consistent by showing the same conceptual diagram with different drivers, and nicely illustrated with a clear beautiful photo conveying the message, and reference for further reading

The weak points of the manuscript are

- 1) A very vague, poorly written main text article which was for large parts not understandable (even not for me as a functional ecologist!) and below publication standards
- 2) The conceptual diagram should be improved as the terminology is (in my opinion) not consistent, it should be made more clear that humans affect ecosystems by affecting the other drivers
- 3) Conceptually, distinguish better between traits of individuals/species, and traits of ecosystems. It is unclear what you mean with 'ecosystem traits', and whether you suggest that ecosystems are filtered out by the environment (which I think is conceptually wrong).
- 4) Better explain and show how you used in practice these functional traits to classify the ecosystems. It seemed to me that you used a-priori defined ecosystems and described them afterwards with your functional typology, whereas it should of course have been done the other way around!

I was specifically asked to look at the descriptions of the tropical systems, which I do further below. I have reviewed 7 tropical systems. Given the fact that the nature reader thinks that everything what is published in Nature has been scrutinized and is true, I echo the concerns of the previous reviewer that it is imperative that ALL 100 ecosystem descriptions are checked by specialists. So you still have to look for reviewers for the other 93 ...

Overall, this has been an admirable and Herculean task, for which I congratulate the authors. Please improve the main text article, as it does not do justice to the rest of the work you have done. Please find my major and minor comments below, which I hope are of help to improve the manuscript.

MAJOR COMMENTS

1. IMPROVE THE INTRODUCTION TEXT. I am a functional ecologist and biologist, do fieldwork in a variety of ecosystems across the world, and have affinity with management and conservation. So I thought that I should be able to understand the main text, and get inspired by it. I must admit that I found the main text very vague, unclear, full of undefined jargon, and below publication standards. So if the authors want to reach and convince a wider audience, then please invest time to make it attractive, understandable and accessible.

2. IMPROVE YOUR CONCEPTUAL DIAGRAM. Appendix S2 (conceptual foundations) is well written, interesting, and a pleasure to read. Figure 1 presents your conceptual model, which is the cornerstone of

your whole typology, and returns in each ecosystem description. The wording, should therefore be crisp and clear, and above, all, correct. In my opinion, several terms and classifications are now incorrect and should be improved:

- **Human activity.** The real conceptual problem I have is that human activity does not DIRECTLY affect the ecosystem traits and species traits (as your arrow suggests), but human activity INDIRECTLY affects the ecological traits by changing environmental conditions, resources, disturbance regimes and biotic interaction. This applies for all the four items you give as example in your human activity box. For example, structural alteration works through disturbance. Resource use is either a disturbance (you remove biomass), or a biotic interaction (it equals predation). Movement of biota is a biotic interaction (dispersal). Climate change works through changing temperature (an environmental condition), and carbon and water availability (resources). So I really think it is conceptually wrong to frame human activity as having an independent effect on the traits, You should have arrows from human activity to disturbance regime, biotic interactions, resources and environmental conditions. In the same way as you explicitly draw an arrow from environmental conditions to resources. It therefore does not suffice to say that those human effects are included through the circle with the broken line.
- **Human activity.** I understand that in the Anthropocene and as preservationists you want to frame humans not as being part of nature, but as being outside nature and affecting nature. Many people may disagree with you, so justify somewhere why you put human activity as a separate box.
- The term **"Resource processes"** is incorrect. What you show in the box are resources (water nutrients, etc), so no processes of resource uptake or loss. So rename this box "Resources"
- The term **"Ambient environmental processes"** is incorrect. What you show in the box are environmental conditions, not processes. The word "ambient" is also confusing. If these conditions are ambient, then you should also label your "resources" as "Ambient resources". I think the correct name of this box is "Environmental conditions". And please define in the legend "Kinetic energy". I did not have a clue what you were referring to
- The term **"Biotic interactions"** does not match the names inside the box. "Competitors" are organisms that are interacting, it is not a biotic interaction. So replace the names in such way that they reflect real interactions; "Competitors" should be "Competition", "Predators" should be "Predation", Mutualists should be mutualism. And think yourself how to rename "pathogens" and "engineers".
- **"Engineer"** is in my opinion not a biotic interaction. Facilitation is. Is that what you mean? If so, use facilitation. That is closer to your "community assembly theory". Or do you mean that "ecosystem engineers" modify the landscape? Then it is in my opinion not a biotic interaction anymore.
- **Disturbance regime.** If you define a disturbance as a sudden event that destroys biomass then flooding is NOT a disturbance regime (it is not sudden, and generally does not remove biomass), but a stress. I think it conceptually belongs to the environmental box, because flooding modifies resource availability by reducing oxygen and light, and increasing water availability and nutrient availability (through deposition). And please define in the legend "Mass movement" and "Igneous activity". I did not have a clue what you were referring to.
- **Ecological traits.** In the central circle make a distinction between what are your ecosystem traits and what are your species traits. For me as a functional trait ecologists it is VERY confusing that they are all listed as ecological traits, as the ecosystem traits are an emergent property of the species traits.
- Species pool. Move the word to a different place in your diagram. Now it erroneously suggest that it refers to the broken circle

3. CLARIFY CONCEPTUALLY ECOLOGICAL TRAITS AND ECOSYSTEM TRAITS. Ecological traits are the basis of your typology. Maybe I have missed it, but you nowhere define what a trait is. In traditional functional ecology a trait is a property of an individual, affecting its performance (growth, survival, reproduction, fitness) (Violle et al. 2014 PNAS). I can imagine that you assign a trait to a species. But then the question is: what is an "ecosystem level trait"? In my humble opinion, assembly theory is about the assembly of individuals and species into a community. A trait is therefore a property of an individual or a species. In my opinion, your ecosystem level traits (productivity, diversity, trophic structure and physiognomy) are

the emergent consequences of a the traits of the individuals in combination with the environmental conditions. For example, primary productivity of a forest is determined by the number of individuals, and by their size, total leaf area and photosynthetic capacity (all properties of an individual), in combination with the environmental conditions (light, temperature) that determine photosynthetic rates). So using this analogy, I do not see why productivity would be an ecological "trait" of the ecosystem. It is simply the emergent consequence of species traits and environmental conditions. To circumvent this problem, you could define a "trait" simply as an attribute of an individual, species, or ecosystem. I am fine with that, but in your conceptual framework you suggest that ecosystems are filtered out by the environment based on their ecosystem traits, and that is of course not true. I think the environment does not filter out ecosystem A or B. It filters the individuals and species belonging to ecosystem A or B.

4. CLARIFY HOW YOU CLASSIFIED YOUR ECOSYSTEMS BASED ON FUNCTIONAL TRAITS. Better explain and show how you used in practice these functional traits to classify the ecosystems. It seemed to me that you used a-priori defined ecosystems and described them afterwards with your functional typology, whereas it should of course have been done the other way around!

CONTENT-WISE COMMENTS MAIN TEXT

L139. You seem to have a very static view on ecosystems. Most are dynamic and resilient (to a certain extent)

L140. ASSESSING THE RISK OF PRESSURES. Nice that you have a human pressure map. But who says that these are relevant to the functioning of ecosystems. So you assume a threat, but as you did not measure ecosystem response it is just a supposed threat. So how relevant is this exercise?

FIG 1. See my major comment 2 at the beginning of my review

FIG 2. I did not understand the figure, nor the legend

FIG 3. WHAT IS THE MEANING OF ABOVE OR BELOW AVERAGE. Everything is relative, so how important it is that an ecosystem faces above- or below pressures (your dotted lines). The question is how much a certain pressure results in an absolute response of the system (some systems are resistant or resilient, others are sensitive)

FIG3. WHAT IS THE MEANING THAT PROTECTION DECLINES WITH HIGH PRESSURE? It could be that people selected parks in remote areas as we do not need those (= probably areas with low pressure), it could be that conservation results in less pressure. We simply can not tell. So what can we learn from this? In that case you should have done a temporal analysis (once it is a park, does the pressure decline).

FIG 4., Why a one-tailed test? I think it is more robust and appropriate if you would do two-tailed test. As an ecologists my prediction often was the opposite of what I found, so a one-tailed test makes little sense. And what is in the end the added value of having once conceptual framework if you still analyze terrestrial, freshwater, and marine systems separately?

MINOR COMMENTS ON INTRO MAIN TEXT:

DEFINE TERMS. If this approach is to be a vehicle for multidisciplinary or interdisciplinary collaboration, then all terms should be defined very clearly. It would be great if the authors would add in the main text a box with a glossary. Please define what you mean with "functionally similar responses (l68), "functions" (l71), "Ecosystem Functional Groups" (l74), "resources" (l85), "ecological functions" (l102), "biota" (l102), "filters" (l115), "community assembly theory" (l116), "ecosystem traits" (l116)"disassembly and reassembly" (l125)"semi-deterministic resource appropriation" (l 125), "physical restructuring" (l124), "movement of biota" (l126), "convergent functional traits" (do you mean converging from an evolutionary perspective? (l128), "functionally based ecosystem typology" (l132), "generic indices" (l 148), "cryogenic ecosystems" (l167).

CLARIFY TEXT. Large parts of the introductory text are very vague or totally unclear.

L66. What do you mean that “ecosystems contribute to biodiversity”? I would say that ecosystems vary in their biodiversity

L74-75. This sentence does not flow and it is unclear what <17 or <10 refers to,

L75-678 This is a mixed bag of many things. Provide some conceptual structure. Now it feels like a random list of things.

78-79: What do you mean with “globally comprehensive typology”, “policy transformation” and “ecosystem specific action.”? The links that you suggest are totally vague. L

80: a typology is not an infrastructure, it is a tool.

L83-84: This is a complicated sentence.

L86: what is the difference between energy and biomass?

L87-88. What do you mean with “The identity of biota is also central to biodiversity concepts, conservation goals, and human values”?

L92. What do you mean with “design and resource on-ground solutions”?

L83. Why is there a dual need for sustainability and conservation. Why do you need both? And do you suggest that there is a trade-off between the two?

L95-97. Why are especially these design principles crucial? Please justify

L99. How different are these “biophysical attributes” from your stocks and fluxes?

L99-100. Very good point!

L102. What is the difference between an ecological function and an ecosystem function? Please use your terms consistently throughout. It seems that sometimes you use synonyms, which is confusing

L105 Why would that “limit the ability to generalise about properties of ecosystems grouped together”?

L109. What do you mean with “to serve as a template”? What are your “units of classification”?

L111. What do you mean with “Interactions and dependencies amongst drivers and traits”. This is very vague

L127. What is the difference between “dependency” and “interactions”?

L131-134. I am totally lost

L134. To be adopted in 2020 or has it been adopted?

L136 what you say about the upper and lower groupings is not understandable

MINOR COMMENTS ON APPENDIX S2

- P4 fifth par. Define assembly processes. This is crucial, as it is the cornerstone of your theory. Geomorphology and turbulence are not assembly processes. The assembly process is in my opinion dispersal, colonization, establishment, environmental filtering, etc.
- P2 last paragraph. Flooding is not a disturbance
- P4 first par. Define “Ecological traits” (I intuitively consider it a species attribute, rather than an ecosystem attribute).
- P4 third paragraph. Subterranean systems do have herbivores; nematodes and many other macrofauna browse on plant roots
- P4 third par. Define “aphotic sensory mechanisms”. This is really jargon

COMMENTS ON TROPICAL ECOSYSTEM DESCRIPTIONS (S4)

T1.1 TROPICAL -SUBTROPICAL LOWLAND RAINFORESTS

Ecological traits: What do you mean with “Bottom-up regulatory processes are fuelled by large autochthonous energy sources”?

- It seems that you like to mention SLA but that you have not captured the concept, as every time you say that the SLA is high (or low) and it should be the other way around. SLA is the product of (1/leaf thickness) and (1 leaf density). Rainforest trees have thick and dense leaves and therefore a LOW SLA instead of high SLA, as they are adapted to low light or nutrient-poor soils. Low SLA comes along with long-lived persistent leaves. So they can retain the scarce carbon and nutrients (and rain can leach out nutrients less easily from the leaves) for a longer time.

- Rainforest trees do not have rapid growth but slow growth (because they have low SLA and dense wood). A high growth potential does not make sense in a resource poor environment (low light, low nutrients). They grow on average 1mm in diameter per year. Only pioneer trees have fast growth.
- Add palms to the life forms
- Say "by tropical storms SUCH AS near coastal forests". Large storms also occur in central amazonia where km² of trees might be blown down
- PLEASE remove "Many trees exhibit leaf form plasticity on a single individual" ALL plants over the world show a strong plastic response to light, this is not specific for tropical trees and it does not explain their success
- Please remove "some species germinate on tree trunks" . This is a detail which is not typical for most rainforests and not an important functional response
- Key ecological drivers. Add that soils range from very fertile, such as volcanic soils, to very infertile, such as on old weathered acidic soils. Because of high rainfall and strong weathering P often limits productivity
- Conceptual diagram: what do you mean with "limited" competitive release? I guess you mean "strong winds" instead of "high winds". Why do biotic interactions lead to "bottom-up regulation". Add to resources "generally nutrient-limited"

T1.2 TROPICAL SUBTROPICAL DRY FORESTS AND THICKETS

- Do you mean that tree and vertebrate diversity higher than most other TEMPERATE forest systems? If so, add temperate.
- I was highly surprised that you said that trees have typically thin bark and low fire tolerance. Compared to what? Compared to savanna yes, compared to rainforest no. My experience is that trees can have thick barks (e.g., several Bombacaceae, Apocynaceae, Baobabs, especially the larger drought deciduous ones that store water in the stem. Next to that there are many thin barked ones, especially the shrubby plants that can resprout easily. So I would say that bark variation, and hence, fire tolerance varies a lot
- I am highly surprised that you talk about gap-phase dynamics. Because dry forest have a seasonally deciduous and relatively open canopy and small leaflets, there is a lot of light in the understory. Many plants can recruit and hang on in the understory. Gap phase dynamics are generally NOT important, unless you talk about extremely light demanding species that may need large disturbances such as fires or hurricanes.
- What do you mean with "These forests may be involved in fire-regulated stable state dynamics with savannas"? That is a totally vague and random remark
- Maybe mention that many dry forests have a high abundance of nitrogen fixing species (e.g., M. Gei et al. 2018 Nature Ecology & Evolution)
- In what areas are tropical storms important? In the Mexican Yucatan and Caribbean isles?? I can not think of any other dry forest area where this would be important, so I wonder whether this is the exception rather than the rule
- Conceptual diagram. I would not say that light is limiting. Maybe if you compare it to a savanna yes, but in general not. Say for warm temperatures that they have RELATIVELY low diurnal and seasonal variability. The variability is higher than in tropical rainforests, I think, due to less clouds, and cooling due to open night skies. Why do you say that there are canopy herbivores. In all forest systems there are canopy herbivores. Why do you say the canopy is dense? The canopy is in my opinion relatively open compared to rainforests, and temperate forests. The tree density is higher. I would remove shade tolerance and gap dynamics
- Citations. Maybe cite Sanchez-Azofeifa et al. 2013 book: Tropical dry forests in the Americas: ecology, conservation, and management. I guess there is more ecosystem functioning there than in Toby Pennington's book (apologies!;)

T1.3 TROPICAL-SUBTROPICAL MONTANE RAINFOREST

- It is wrong that montane rainforest have a high specific leaf area (SLA). SLA is leaf area divided by leaf mass. Montane species have thick, dense, leathery or coriaceous leaves to retain scarce nutrients and reduce damage by UV radiation. I am flabbergasted that the expert authors have overseen this

- Why is elfin woodland not a separate functional ecosystem?? They have a totally different structure and functioning compared to cloudforests!!! I would make it a separate category
- What do you mean with "productivity is fueled by autochthonous energy??" That plants are autotrophic and photosynthesizing? If so say so, but this is an odd comment. An elevational transect through the Andes found that productivity is mainly limited by radiation (Fyllas et al. 2017 EcolLett Solar radiation and functional traits explain the decline of forest primary productivity along a tropical elevation gradient)
- Say that taxonomic TREE diversity is low, and add that there is a very high diversity of epiphytes (orchids, bromeliads, lichens, mosses, ferns)
- Add that gap-phase dynamics are also driven by landslides (often driven by geologically young substrates, steep terrain, high rainfall and waterlogged soils)
- Key ecological drivers. Say something that in general montane systems are especially limited by soil N? (Grubb 1977 Control of forest growth and distribution on wet tropical mountains: with special reference to mineral nutrition). Or has this proven to be wrong?
- Conceptual diagram. Why do you say that landslides lead to competitive release?? All trees are gone, so nobody is released. I can imagine that storms lead to competitive release. Plant competition is in nearly all closed vegetation cover systems of the world, so why do you mention it here? Because it is relatively shaded? Why do you say "bottom up regulation"? No clue what you are referring to. I disagree that diversity is moderate to low. It is moderate to low tree diversity but high epiphyte diversity. Add to "Abundant bryophytes" also "epiphytes and lichens". Add also nutrients to your ambient environment, as it co-limits productivity?

T1.4 TROPICAL HEATH FOREST

- For a global classification I would NOT use the word "heath forest". This comes traditionally from Asia (Kerangas forest) and is only used there. The term "Heidewald" was introduced by Winkler (1914). It gives to me associations with Ericaceae shrubland, which it is not. It is a forest on siliceous, acid, nutrient poor soil, so I think "white sand forest" is more clear and appropriate in the spirit of your FUNCTIONAL classification, and it is also known as such in the Neotropics.
- Ecological traits. I would add to the first sentence that it has a high density of thin stems. Also add that there are many plants and animals with special adaptations to low nutrients (pitcher plants), and that there are special mutualisms (abundant myrmecophytes). Richards mentions as feature a tendency towards dominance
- Drivers: Proctor suggests that aluminium toxicity plays a role. I do not know whether that has been accepted or rejected. Please check and add it to drivers and diagram if needed
- Conceptual diagram: sandy soils and shallow rooting lead not only to flooding but also to a water shortage in the dry season. So I would put an arrow from sandy substrate to resources, and not only put under resources water surplus but also shortage. In ecological traits you say "low diversity". Low compared to what? Temperate forests? Tropical rainforest? I believe there are many species there. I would put high dominance as an ecological trait. Rather than microphyll I would say that leaves are leathery (that better reflects the nutrient limitation and functioning)
- Distribution: P.W. Richards classifies Caatinga as heath forest. But you did not include it here on your map? Why not? You say that heath forest is not known from Africa, but Peace & MacDonald (Biotropica 1981) mention that it also occurs in Gabon (I do not know from where they have that wisdom)
- References: they feel a bit detailed and random. Do you expect the readers to get an overview of the system based on these? Maybe better cite PW Richards 1996. And I must imagine that there is a better book or review article on these forests

T4.1 Trophic savannas.

- I am not a savanna ecologist, but have travelled through the systems.
- Ecological traits: I would add that the mammals can represent a large biomass, and that mammals show rotational grazing. And that in East Africa (and South Africa) mammals show strong migratory patterns aligned with the rainfall and nutrient needs when lactating. I am not sure whether this applies to West Africa. What do you mean with "sustaining the system through positive feedbacks and

limiting fire fuels"? Why do you add "Nitrogen fixation, recycling, and deposition by animals exceeds volatilisation"? Why is nitrogen volatilisation a problem when there is little fire?

- Key ecological drivers. You say that low intensity fires have return intervals of 5-50 years. I guess you refer here to natural fires? I think that in West Africa fire return interval is each year as people burn it regularly for hunting or to get rid of snakes etc., and to renew the grasses for their cattle
- Distribution: In the map you classify the west African Sahel zone as trophic savanna. Maybe originally, but I guess now most animals are hunted and little is left. The same applies to the Indian subcontinent. At some places you can find still intact communities. Maybe the West African savannas are now de-facto pyrogenic savannas? In the description be explicit that "Asia" is the Indian subcontinent
- Conceptual diagram. Ecological traits: mention high mammal abundance. Biotic interactions: It is not clear what you mean with "engineers (+ve feedbacks)" and "strong and weak top down processes" Maybe use "Strong or weak herbivore control"?

T4.1 PYRIC TUSSOCK SAVANNAS

- Ecological traits: what do you mean with "grasses cure in winter" and "local endemism is low across all taxa"?
- You highlight deciduousness, but some savannas are dominated by evergreen species (Australia) and others by deciduous species (South America).
- Shouldn't you mention the strong variation in tree cover (in the Brazilian cerrado you have at least campo sujo, cerrado, and cerrado) and the importance of gallery forests?
- You say that plant defences against herbivores such as spinescence are less prominent. Then be explicit that you mention about MAMALLIAN herbivores, as insect herbivory can be high, and at least in the cerrado many plants defend themselves with high silicate concentrations.
- When you talk about detritivores, do you refer about termites? If so, say so.
- In the cerrado the resprouters and reseederers are important; there is a high diversity because of the fire regime and the reseeding plants. Maybe worth mentioning? I do not know whether that applies to other savannas.
- Why do you say that plants have a high SLA? High for the deciduous species, but low for the evergreen species!
- Maybe mention that belowground carbohydrate storage is important for resprouting success after fire, and many species have massive belowground storage organs?

Author Rebuttals to First Revision:

Referees' comments:

Referee #1 (Remarks to the Author):

This manuscript does two main things:

- A) It presents a new ecosystem classification dividing the earth's surface up into 5 realms hierarchically further divided into 25 biomes and 108 ecosystem functional groups.
- B) It presents an analysis of the conservation status of each of the 108 EFG vs. Aichi targets in terms of preservation and human pressures.

On(A) I feel I have a much deeper insight into the background and depth of the work that has gone into this. I feel like this work is now more clearly presented across the methods in the main text, the comparison with existing classifications, and an appendix showing review comments. I also understand how this is situated viz existing conservation agencies (notably IUCN) and likely uptake of the system. In short I largely feel much more comfortable with the rigor of the process (not that I doubted it before but it just wasn't as fully documented). It remains a bit of a novelty to present something like this as a Nature paper, but no doubt it would be highly cited, so that remains an editorial decision. Overall I continue to like this particular effort, most notably because of its

comprehensiveness (whole globe), the analysis of both ecosystem and biodiversity, the systematic and thoughtful approach to developing this and its focus on processes.

AU RESPONSE 5: Thank you for these positive remarks

(B) is an obvious capstone or pinnacle outcome of this process, and is of high value to the conservation and policy communities. I do worry a bit that this output gets lost in the whole picture (swamped in the details) and much of the methods and interesting results get pushed into Appendix 6. I increasingly think it might be better to split (A) and (B) apart so that B can get its full exposition for people who are more interested in (A) than (B) and vice versa but that obviously is a big change at this point in the editorial process. I found many of the figures in S6 important and deeply interesting (e.g. Figure S6.2 would be in the main text in any normal presentation of Task B). Figure S6.3 is also information dense and could profitably be in the main text. I loved the Sankey diagrams but understand there are too many to go in the main text, but could a couple go in the main text so that readers know they are there? You can see what it is problematic to treat this topic fairly and thoroughly on top of (A).

AU RESPONSE 6: We removed the spatial analyses of pressures from the manuscript and Appendix S6 as requested, and now focus the manuscript solely around the development of the new classification scheme. We will pursue spatial analyses in a separate publication in which we will address technical concerns about some of the maps (see Response 1).

Overall I am appreciative of the changes made. At this point I have reviewed the main text and each of the supplements although S4 and S5 (which cover each of the 100+ EFG) I have sampled from half a dozen EFGs across all 3 biomes that I have some professional exposure to. Some minor issues (other than the fact that (B) is getting buried) are:

On Task (A) establishment of an EFG classification:

1) I continue to think a 3-level table containing the actual realms/biomes/EFG is more useful and concrete than Figure 2 in the main paper (even though it takes a bit more space). This is essentially the primary output of task A. And yet it is buried in supplemental materials.

AU RESPONSE 7: Agree, we have replaced Fig 2 with a table listing the 3 top levels of the typology and salient characteristics of the units.

2) Of the half dozen EFG's in S4 that I looked at (which systems I know reasonably well). I was overall impressed with their accuracy and the contribution of such succinct, careful summaries. Overall my concerns about the canonical figure on each page were allayed with one exception.

AU RESPONSE 8: Thank you.

3) I found the abiotic environment and resources pieces and the ecological traits pieces fairly convincing. I was unable to come up with convincing arguments about the biotic interactions

however - what e.g. would be the evidence that boreal and temperate broadleaved forests are dominated by herbivory and predation while maritime temperate rainforests (T2.3) are dominated by competition? Predation, competition and herbivory have all been deeply studied as important processes in all 3 ecosystems - but I am not aware of any good evidence the relative importance of these forces. I think this is at risk of capturing biases of researchers past more than true relative importance of processes. There are very active debates right now with Nature-level papers on latitudinal gradients in importance of different species interactions. I just don't think we know the answers yet. There is also a nod towards incorporating dispersal processes. Personally I do not think the biotic interactions and dispersal process components are on the same level of rigor as the resource and abiotic context boxes. Personally I think these two pieces should be dropped.

AU RESPONSE 9: The intent of the assembly models is to recognise the most salient processes influential on ecosystem assembly. Pragmatically, this means that some drivers of assembly were omitted because they were not considered by expert contributors to be consistently influential, even though they may be influential in some places or circumstances. As the Referee #1 points out, there is no objective means to determine this due to reporting (and research) biases, and there is indeed greater uncertainty about the biotic drivers of assembly than the abiotic drivers, as the former are more challenging to research and fewer data exist on strength of effects. Consequently, we rely on the judgements of expert contributors, their collaborative networks and the subsequent reviewers of the profiles, with consistency checks by the lead author, as explained in Appendix S4. This is a limitation that we now acknowledge in the main text in an expanded discussion of strengths and weaknesses. We also now clearly present the assembly models as hypotheses based on the authors' appraisal of current knowledge (see lines 226-232 quoted in Response 4). Further, version 2.01 of the typology, including the assembly models is open to revisions and updates as knowledge improves and uncertainties are resolved.

Although Referee #1 proposes deleting biotic filters and dispersal filters from the models, we suggest that retaining them as hypotheses about salient assembly processes for each functional group is essential to the completeness of the descriptions. Although empirical studies are not abundant, those that are available suggest that biotic and dispersal filters are critical to ecosystem assembly in a range of contrasting ecosystem types (see for example Scheffer et al. 2001 Nature 413, 591-596; Thebault et al. 2007 Oikos 116, 163-173; Estes et al. 2009 Phil; Trans Roy Soc B 364, 1647- 1658; Fritz et al. 2011 Ecography 34, 196-202). Conversely, omitting these processes from the assembly models would imply that these processes are never important to assembly, contrary to available evidence. Therefore we think it is important to retain these elements of the assembly models, even though their roles are more uncertain than abiotic filters in many functional groups.

4) I found S5 helpful in appreciating the degree of review each EFG received and the issues raised. On the one hand it suggests a thorough process. On the other hand it left me with the impression there was only one reviewer per EFG? Elsewhere I thought I saw there were more reviewers. It would be beneficial to highlight number of reviewers as well as nature of issues raised.

AU RESPONSE 10: A total of 264 reviews were undertaken for the 108 Ecosystem Functional Groups, a mean of 2.44 reviews per EFG, similar to a standard journal review process. We now report these statistics in the Methods section (lines 458-460). In Table S5.1, we withheld the individual identifiers of reviewers to ensure anonymity. However, the caption states, "Each row represents a different review. Multiple rows for the same Ecosystem Functional Group or biome represent different

individual reviewers.” Thus, for example, there are two reviews for Biome T1, two reviews EFG T1.1, of which the second addresses two specific issues numbered 1 and 2 within the third column.

5) Thank you for including geospatial files of the EFG. It would be highly desirable to allow them all to be downloaded as a single zip/tar archive file rather than having to click 108 separate links. Also I was a bit surprised to see them as raster rather than shape files. I can see arguments for raster. But if they are raster would it not make sense to have a unified file where the raster cells can an index to the dominant EFG type?

AU RESPONSE 11: We omitted the spatial analysis- from the manuscript as requested by the editor (see Response 1), and hence no longer use the geospatial files except to display the thumbnail maps in Appendix S4. Many of the original data sources were rasters, so we chose to make all data available in that common format. A raster format also enabled us to ensure that map resolution reflects reliability for global applications, and to keep file sizes manageable. We are progressively updating our cited repository to make alternative download options and formats available for users. Given that no single global source exists for all ecosystem groups, and that insufficient evidence exists to determine dominance, we refrain, at present, from assigning a dominant EFG type per pixel. We expect to develop an approach enabling this improvement in the future, in collaboration with ecosystem experts and mappers to produce more accurate, precise and integrated distribution maps for EFGs.

6)Although the paper continues to refer to sublevels 4-6 in the hierarchy that reference biodiversity, there is essentially zero information or examples included about them. It is probably more appropriate to delete their mention in this paper.

AU RESPONSE 12: We think it is important to present the structure of the full typology, while emphasising the upper three levels. Several key points made in our main text hinge on a basic understanding of Levels 4-6 including: i) the duality of function and biodiversity in design of the typology (principles 1 and 2); ii) the importance of integrating, rather than replacing local classifications within the global framework; and iii) explicit combination of top-down and bottom-up approaches in the development of the typology (commended by Reviewer 2 as ‘brilliant logic’). In the revised ms, we have expanded the paragraph on Levels 4-6 to give more detail on their role in the typology and explicitly referencing examples of integrating Level 6 classifications from Chile and Myanmar in Appendix S3. The expanded main text is on lines 419-445:

“Three lower levels of the typology distinguish functionally similar ecosystems based on biotic composition. Our focus in this paper is on global functional relationships of ecosystems represented in the upper three levels of the typology, but the lower levels are crucial for representing the biota in the typology, and facilitate the scaling up of information from established local-scale typologies (Appendix S3, pp19-20). These lower levels are being developed progressively through two contrasting approaches with different trade-offs, strengths and weaknesses. Firstly, Level 4 units (Regional ecosystem subgroups) are ecoregional expressions of EFGs developed from the top-down by subdivisions based on biogeographic boundaries (e.g. [54]) that serve as simple and accessible proxies for biodiversity patterns [55]. Secondly, Level 5 units (Global ecosystem types) are also regional expressions of EFGs, but are derived from bottom-up aggregation and rationalisation of units from established subglobal ecological classifications, which define Level 6 (e.g. see hierarchical integration of national ecological classifications for Chile and Myanmar in Tables S3.3 and S3.4). Subglobal classifications, such as those for different

countries, are often developed independently of one another, and thus may involve inconsistencies in methods and thematic resolution of units (i.e. broadly defined or finely split). Aggregation of Level 6 units to broader units at Level 5 based on compositional resemblance is necessary to address inconsistencies among different subglobal classifications and produce compositionally distinctive units suitable for global or regional synthesis, but (unlike Level 4 units) explicitly linked to local information sources.

Integrating local classifications into the global typology, rather than replacing them, exploits considerable efforts at producing existing classifications, already developed with local expertise, accuracy and precision. By placing national and regional ecosystems into a global context, this integration also promotes local ownership of information to support local action and decisions, which are critical to ecosystem conservation and management outcomes (Appendix S3, p20). These benefits of bottom-up approaches come at the cost of inevitable inconsistencies among independently developed classifications from different regions, a limitation avoided in the top-down approach applied to Level 4."

On Task B (in addition to feeling like it is in some ways presented as the central result and in other ways is a bit of toss-off only briefly introduced)

7) I think the biggest concern I have is methodological. What is the basis for determining human pressure by calculating % of area that is above a binary threshold of human pressure (% of area under high vs low pressure)? Binary thresholds can often be rather misleading. Given that human pressure was a continuous variable could not some other alternative like an area-weighted average be more informative? In general, I may have missed it among the text and supplements, but I did not find a strong rationale for this particular metric.

AU RESPONSE 13: We removed the spatial analyses from the manuscript as requested (see Response 1).

7) Figure 3 (main text) is by some reads of the paper the main result. This figure needs more information density

AU RESPONSE 14: We removed the spatial analyses from the manuscript as requested (see Response 1).

9) Figure 4 (main text) this figure is fairly informative about relative change in pressure, but since "change in pressure" index is an abstract concept it is hard to know how to interpret the scale

AU RESPONSE 15: We removed the spatial analyses from the manuscript as requested (see Response 1).

10) If you present a table of all 108 EFG you can include the scores for % protected, whatever measure of impact intensity (per #6 I'm not convinced you have the best one) and the change in pressure index. At a minimum such a table could I believe at Nature go as an Extended Table so it is downloaded with the main PDF. In general I missed Extended Figures that I expect to see in Nature - just half a dozen very long supplemental materials.

AU RESPONSE 16: We replaced Fig. 2 with the new table suggested by Reviewer #1 (see Response 7). We also removed the spatial analyses from the manuscript (see Response 1) and do not think it is appropriate to report quantitative estimates of % protected and impact intensity due to acknowledged limitations on the spatial data for a number of EFGs.

11) The text on pages 5 and 6 (Main text) is fairly awkward and long. The table mentioned in #8 plus improved versions of figures 3 & 4 would be much preferable as a means of communication.

AU RESPONSE 17: The text on pp5-6 of the previous version has been deleted, as it was a commentary on the outcomes of spatial analyses which were removed from the ms (see Response 1).

Referee #2 (Remarks to the Author):

This is quite an unusual paper, the tip of the iceberg for a very substantial amount of work. It is would be way beyond my expertise to review all of it. As a conservation biologist working at large spatial scales, I have reviewed specific facets of it:

- The need for, and usefulness of, a global ecosystem typology: I agree with the authors that this typology is much needed.
- The conceptual framework: I am well impressed, but have some reserves and recommendations for improvement;
- The assessment of pressures and protection: I am not fully convinced it is very robust.

*** 1) NEED FOR A GLOBAL TYPOLOGY OF ECOSYSTEMS ***

I am fully persuaded by the need for a global classification of ecosystems that is conceptually robust, scalable and spatially explicit. The authors make a very good job at detailing the main immediate applications of this classification in Appendix 7. This work fills a major current data gap and I foresee that its outputs (the classification, the maps) will be plugged immediately into conservation policy and management.

I also foresee it will become a key layer in macro-ecological analyses, even in theoretical studies (in the same way the species distribution data derived from Red List assessments have been underpinning an endless stream of studies).

Given the institutional background of the development of this framework and resulting typology (i.e. IUCN-led), I am persuaded it will gain immediate traction. I am also cautiously optimistic that it will be seen as so valuable to policy/monitoring that resources will materialise to allow it to continue to be refined and updated over time.

AU RESPONSE 18: Thank you for these positive remarks

*** 2) CONCEPTUAL FRAMEWORK ***

I am impressed by the conceptual depth of the work presented here, and in particular by the ability to bring together terrestrial, marine, freshwater, subterranean and atmospheric into a common typology (Fig 2), underpinned by a common conceptual framework (Fig 1). I agree with the authors that the resulting typology represents a marked development in relation to existing classifications, in particular because of its conceptual underpinning, its comprehensiveness, and its more direct alignment with the concept of ecosystem as a level of biodiversity organisation (as defined by the Convention on Biological Diversity).

This said, I have some specific concerns, questions and recommendations.

* 2.1. Evolutionary processes

Disclaimer: I am not an evolutionary ecologist. I step in because this point does not seem to have been addressed at all in the previous rounds of reviews (either the one by Nature or that previously organised by the IUCN). Ideally, it would be good to have this reviewed by an evolutionary ecologist.

The conceptual framework disregards evolutionary processes. This reflects its grounding in community ecology. For example, the article the authors cite as the main basis for their conceptual framework (HilleRisLambers et al. 2012), clarifies that they do not take into account long-term evolutionary processes (which makes sense, in their case, as their focus is on much smaller geographical scales/shorter time frames). But this is something that cannot be ignored at the large geographic scales covered by the present ecosystem classification (and correspondingly deep evolutionary times), where community composition and function is as much the result of evolution in situ (adaptation, speciation, co-evolution, extinction...) as the result of dispersal/filtering from a pre-existing species' pool.

The word "evolution" is never even mentioned in the main text. In Figure 1, it does not appear at all, with Ecological Traits presented as the end result after a broad species pool goes through a set of filters. The very use of the term "community assembly" reflects an emphasis on a process of combination of emergence from pre-existing pieces (although this term can also be used in evolutionary terms).

Evolution is discussed in Appendix S2 (page 3), but only from the perspective of "evolutionary legacies": the authors discuss how "local evolutionary legacies whose progenitors had long histories of prior occupancy" can (through niche conservatism and limitations on dispersal) result in functional differences between ecosystems, for example in Savannas. This is of course true, but conversely convergent evolution under similar abiotic conditions (or functionally convergent biotic conditions) can also result in functional similarities between ecosystems dominated by distinct evolutionary lineages, for example: cushion vegetation in polar/alpine ecosystems; blind predators in underground ecosystems; swimming predator birds in polar waters (Alcidae/Spheniscidae).

The role of evolution is also currently lacking in the discussion of ecological traits in Appendix S2 (page 4) which are presented as solely the outcomes of the assembly process, even when referring to the traits of species ("life-histories, life-forms, morphology, phenology, behavioural and ecophysiological features"). Surely many of these traits evolved in situ through adaptation, speciation (as well as the disappearance of other species with other traits, via extinction).

The authors point out that biodiversity appears more strongly in the framework when distinguishing between categories at lower levels (4-5-6), whereas the upper levels are more focused on function. My point here is fully related with function: the role of evolutionary processes in determining the traits that determine ecosystem function. Of course evolution also means that we get different

species compositions (i.e., different biodiversity) in functionally similar ecosystems distributed across the world (to be reflected in levels 4-5-6).

As it stands, the lack of explicit integration of evolutionary processes into the framework undermines the assertion that this is a conceptually robust typology, based on a solid theoretical basis.

In my view (but, again, I am not an evolutionary ecologist) this current limitation of the framework does not necessarily invalidate the results (i.e., the classification obtained) because all the drivers currently presented as simple filters are also major evolutionary pressures. Accordingly, I think (but I may be too naïve here) that addressing this point does not require a substantial change in the conceptual framework, but to generalise it to also include evolutionary processes. Specifically, I would recommend:

- Acknowledging (main text, Appendix S2) that all “drivers” (abiotic and biotic, and including human activity) act both as “filters” (determining community assembly from a pre-existing species pool, given dispersal) as well as “evolutionary pressures” (determining community structure and function through in situ speciation and extinction). Also being careful to formalise that the term “community assembly” reflects processes at a diversity of temporal and spatial scales, including through speciation and extinction. And clarifying that current “ecological traits” are the end products both of long-term evolutionary processes and of shorter-term filtering from the current species pool.

- This broader perspective then needs to be reflected in Fig 1. Maybe it would suffice to broaden “Dispersal filter” to “Dispersal filter/speciation/extinction”. The legend would also need to mention evolutionary processes among the feedbacks.

A related reference:

Mittelbach, G.G. & Schemske, D.W. (2015). Ecological and evolutionary perspectives on community assembly. *Trends in Ecology & Evolution*, 30, 241–247.

AU RESPONSE 19: We thank the reviewer for these very constructive suggestions. Our submitted version of the ms made minimal reference to the role of evolutionary processes in shaping the structure and function of communities because of our primary focus on the contemporary applications of the typology. However, we do agree that assembly filters and evolutionary pressures may be viewed as two sides of the same coin, albeit with influences that play out over different time scales, and that reference to an evolutionary perspective better represents the broader theoretical basis of the typology. Removing the spatial analysis from the ms allowed use to address this aspect explicitly by adding two passages of text and relevant references as follows:

Lines 126-132:

“Our model postulates five groups of ecological drivers that may shape ecosystems by acting both as assembly filters and evolutionary pressures (Fig. 1; Appendix S2 for details). Filters are biotic and abiotic processes that determine community assembly from a species pool, given initial occupancy or dispersal (based on community assembly theory [15]). Evolutionary pressures are agents of selection that influence ecosystem function and constituent species traits, typically over longer time scales, through evolution and extinction within a dynamic species pool [16].”

and Lines 164-172:

“Convergences in ecosystem properties are axiomatic to a functionally-based ecosystem typology because they underpin robust generalisations and predictions about ecosystem responses to

environmental change and management. Convergences in species traits may arise from common evolutionary origins and niche conservatism [19] [20], but similarities in ecological drivers (selection pressures, assembly filters) may also produce functional convergences in independent lineages. These convergences are the enablers of a functional classification framework represented in the upper three levels of our typology. Functional constraints may be imposed by the species pool, which is a dynamic outcome of vicariance, dispersal and evolution, depending on ecosystem location and biogeographic history [21]."

The additional references are:

[13] J. HilleRisLambers, P. B. Adler, W. S. Harpole, J. M. Levine and M. M. Mayfield, "Rethinking Community Assembly through the Lens of Coexistence Theory," in ANNUAL REVIEW OF ECOLOGY, EVOLUTION, AND SYSTEMATICS, VOL 43, vol. 43, D. J. Futuyma, Ed., 4139 EL CAMINO WAY, PO BOX 10139, PALO ALTO, CA 94303-0897 USA, ANNUAL REVIEWS, 2012, pp. 227-248.

[14] G. G. Mittelbach and D. W. Schemske, "Ecological and evolutionary perspectives on community assembly," Trends in Ecology & Evolution, vol. 30, no. 5, pp. 241-247, 2015.

[16] M. D. Crisp, M. T. K. Arroyo, L. G. Cook, M. A. Gandolfo, G. J. Jordan, M. S. McGlone, P. H. Weston, M. Westoby, P. Wilf and H. P. Linder, "Phylogenetic biome conservatism on a global scale. Nature 458, 754–756 (2009).," Nature, vol. 458, p. 754–756, 2009.

[17] R. A. Segovia, R. T. Pennington, T. R. Baker, F. Coelho de Souza, D. M. Neves, C. C. Davis, J. J. Armesto, A. T. Olivera-Filho and K. G. Dexter, "Freezing and water availability structure the," Scientific Advances, vol. 6, p. eaaz5373, 2020.

[18] M. D. Crisp and L. G. Cook, "How was the Australian flora assembled over the last 65 million years? A molecular perspective.," Annual Review of Ecology, Evolution and Systematics, vol. 44, pp. 303-324, 2013.

We also amended the caption of Fig. 1 to incorporate evolutionary processes in the interpretation as follows:

"The generic model of ecosystem assembly underlying the global ecosystem typology (Appendix S2 for details). Boxes represent abiotic (resources, the ambient environment, disturbance regimes) and biotic (biotic interactions, human activity) drivers that filter assemblages and form evolutionary pressures, in turn, shaping ecosystem-level properties (filled green ellipse). The range of major organisational scales at which drivers operate are shown in italics in the boxes, followed by a list of the major expressions of the drivers. The species pool is the set of 'available' traits on which the assembly filters and evolutionary pressures operate over short and longer time frames, respectively. Species pools are dynamic products of vicariance, dispersal and evolution, that depend on biogeographic context and history. The outer green ellipse represents the contemporary dispersal filter that mediates the biota currently subjected to local selection by the abiotic and biotic filters/pressures..."

Other additions to Fig. 1 caption were made in response to comments by Reviewer #3 and 4 (see Responses 47, 52, 55).

*** 2.2. Anthropogenic effects**

Another weak aspect of the conceptual framework is how human activities have been integrated. Currently they are defined as “a special class of biotic interaction” and appear separately in Figure 1 and in Table S3.2 (at the same level as the other four groups of ecological drivers). Yet it seems to me that anthropogenic effects act by modifying particular aspects of each one of the other types of drivers, for example: of resource processes by adding nutrients in agricultural lands or removing water through climate change (leading to desertification); of ambient environmental processes by modifying temperature through climate change, or modifying substrate properties in urban areas; of disturbance regimes by yearly soil perturbation in agricultural fields, or by suppressing/adding fire in some ecosystems; of biotic interactions by adding/removing species, eliminating competitors (in crops); etc. This idea that human actions act through biotic and abiotic drivers does appear in the figures associated with the description of the highly modified EFGs, through arrows going from human activities to ecological traits via other drivers. But some direct arrows remain, which in my view are not justifiable. For example, for T7.1 Annual croplands (page 72 in Appendix 4), “annual substrate manipulation” is a type of disturbance and should be represented accordingly.

AU RESPONSE 20: We agree that human activity often (and perhaps most strongly) affects ecosystem assembly indirectly through influences on other drivers. The assembly model for each anthropogenic ecosystem functional group (adapted from the generic model, Fig. 1) in Appendix S4 show these salient indirect effects with black arrows from the Human activity box to other drivers. In the previous version of our ms, we represented all interactions among drivers with an elliptical broken line in the generic model (Fig. 1) to simplify the many potential pairwise combinations among drivers. We now see that presentation of ideas in that way understated the importance of indirect effects of drivers on ecosystem assembly, particularly human activity. Recognising that some additional complexity is warranted, we revised Fig. 1 to remove the dashed ellipse and show indirect effects of human activities on assembly through other drivers as examples of a large set of potential interactions among drivers.

A slightly different issue is whether all effects of human activity are indirect or whether some direct effects may also exist. In part, this depends on how the assembly model (a diagrammatic representation of our assembly hypotheses) is framed. Our assembly models for anthropogenic ecosystems reflect our hypotheses that human influences can sometimes operate directly on ecosystem traits. We posit that addition or removal of ecosystem components by humans may directly alter ecosystem traits such as diversity, productivity, dominance, characteristic life-histories, etc., independently of any indirect effects through modifications of other drivers. We therefore suggest that models providing for both direct and indirect effects of human activity offer more complete and parsimonious hypotheses about assembly processes than models that exclude all possibility of direct human effects (see further comment in Response 52). We made the following additions to the main text and Appendix S2 to acknowledge the reviewer’s concerns and to clarify these points:

Main text Lines 142-149 (Here we also elaborate on why human activities were treated as a separate driver):

“These anthropic processes operate largely, but not exclusively, through effects on other drivers. While our model portrays humans as integral drivers of ecosystem assembly, we separated human activity from other biotic interactions to highlight interactions and feedbacks between ecosystems and socio-economic systems [18], and the need to assess and mitigate the human impacts on biodiversity and ecosystem functioning.”

Main text Lines 150-156:

“Interactions may exist among drivers, modulating their effects on ecosystem properties (Fig. 1, Appendix S2 p4). For example, resource levels may influence ecosystem assembly directly through niche partitioning or indirectly through an interaction that alters biotic interactions. Similarly, feedbacks exist between ecosystem properties and drivers. For example, human land use intensification initiates changes in ecosystems that, in turn, influence human social structure, markets and consumption patterns, driving changes in resource appropriation and further change in ecosystem properties [18].”

Figure 1. We replaced the broken-line ellipse with arrows representing selected examples of effects of drivers (Human activity, Ambient environment) on other drivers and added the following text to the caption:

“...Interactions among drivers, include indirect effects of human activity on assembly through other drivers (black open arrows) and the indirect effects of ambient environmental conditions on assembly by modulating resource availability or uptake (dark blue open arrow). Interactions among other drivers (omitted here for simplicity) are shown in ecosystem-specific adaptations of this generic model for each ecosystem functional group (Level 3 of the typology) in Appendix S4...”

Appendix S2, p6 2nd para:

“Different assembly processes (Fig. 1) do not act independently in shaping the properties of ecosystems, but rather covary and interact through space and time (Cadotte & Tucker 2010). Resource levels, for example, may influence ecosystem assembly directly through niche partitioning or indirectly by altering biotic interactions. Variations on the model template applied to different groups of ecosystems (Fig. 1 & Appendix S4) reflect our hypotheses about drivers that operate on ecosystem traits directly or indirectly through effects on other drivers.”

Appendix S2, pp6-7:

“Terrestrial anthropogenic ecosystems (biome T7 in Appendix S4) provide instructive examples of interactions between human activity and other assembly filters. Assembly processes may involve complex interactions and feedbacks between ecosystems and socio-economic systems with varied settings for labour and capital inputs, technology, market dynamics, cultural beliefs, economic decision making and geopolitics (Meyfroidt et al. 2016).

We posited that addition or removal of ecosystem components by humans may directly or indirectly alter ecosystem attributes such as diversity, productivity, dominance, characteristic life-histories, etc. Direct effects may occur independently of any modifications of other drivers. For example in T7.1 Annual croplands, the introduction of new crop strains with faster growth rates is an anthropogenic assembly process that directly influences ecosystem productivity, without operating through other drivers, because the new strains are inherently more efficient in resource extraction even when supply levels are unchanged (e.g. Pan et al. 2011). These direct effects contrast with interactive assembly processes, which operate on ecosystem properties through indirect effects on other drivers. For example, introduction of disease-resistant crop strains or control agents act indirectly on ecosystem assembly by altering interactions with pathogens (Kohl et al. 2019), while addition of fertilisers operates indirectly by altering resource levels (Berzsenyi et al. 2000). Tillage can be viewed as having direct effects (e.g. by altering soil structure and function, and removing indigenous biota) and indirect interactive effects (e.g. by increasing aeration, water permeability and oxygen availability, promoting nutrient release, etc.) (Hamza & Anderson 2003).

Uncertainties exist in distinguishing direct and indirect effects of all drivers, and the key drivers and their interactions remain poorly understood in many ecosystems. However, assembly models that provide for both direct and indirect interactive effects of drivers offer more complete and

parsimonious hypotheses about assembly processes than models that exclude all possibility of interactions, dependencies and feedbacks. The conceptual relationships posited in the assembly models offer hypotheses to advance our understanding of these underlying mechanisms of ecosystem dynamics and assembly.”

Seeing all human activities as modifiers of various types of drivers would also contribute to addressing something else that bothers me in the current typology, which is a dichotomy between EFGs driven by human effects (croplands, urban, waterpipes, artificial shorelines) and those without any human effects (forests, shrublands, savannas, deserts, rivers...). In practice, most existing ecosystems have already suffered some degree of human modification (e.g. the extinction of terrestrial mega herbivores/mega predators in the Americas and Europe; addition of domesticated herbivores to grasslands in Africa; modification of fire regime in Australia; reduction of top predators in marine ecosystems; introduction of exotic fish in freshwater systems; melting of glaciers through climate change etc). This needs to be acknowledged – should be formalised in some cases in the future descriptions of lower-level units (e.g., African Lakes with their non-native species).

AU RESPONSE 21: We added text to Methods to more clearly explain how we addressed the influence of human activity with different approaches to description of anthropogenic and non-anthropogenic Ecosystem Functional Groups in the typology (lines 466-491).

“Anthropogenic ecosystems grouped within Levels 2 and 3 were thus defined as those created and sustained by intensive human activities, or arising from extensive modification of natural ecosystems such that they function very differently. In many agricultural and aquacultural systems and some others, cessation of those activities may lead to transformation into ecosystem types with qualitatively different properties and organisational processes (see [49] and [9] for cropland and urban examples, respectively). Indices such as human appropriation of net primary productivity [51], combined with land-use maps [52], offer useful insights into the distribution of some anthropogenic ecosystems, but further development of indices is needed to adequately represent others, particularly in marine, and freshwater environments. Beyond land-use classification and mapping approaches (Appendix S1, p6), a more comprehensive elaboration of the intensity of human influence underpinning the diverse range of anthropogenic ecosystems requires a multidimensional framework incorporating land-use inputs, outputs, their interactions, legacies of earlier activity and changes in system properties [15].

Where less intense human activities occur within non-anthropogenic ecosystem types, we focussed descriptions on low-impact reference states. Therefore, human activities are not shown as drivers in the assembly models for non-anthropogenic ecosystem groups, even though they may have important influences on the contemporary ecosystem distribution. This approach enables the degree and nature of human influence to be described and measured against these reference states using assessment methods such as the Red List of Ecosystems protocol [28], with appropriate data on ecosystem change.”

While anthropogenic drivers could be added to the assembly models of virtually all EGFs, we focussed the descriptions on non-human components and processes in all but the anthropogenic EFGs to avoid confounding the assessments of ecosystem modification with definition and description of units. As noted in Response 1, we will address assessments of ecosystem status and effects of anthropogenic disturbance in a separate publication.

More broadly, it is important to make it clear that most drivers vary in time – through both natural processes and human actions – and that hence ecosystems are temporally and spatially dynamic. This is in line with the authors’ ambition that the proposed typology will “underpin robust

generalisations and predictions about ecosystem responses to environmental change and management” but it needs to be better formalised. Also makes sense to mention that ecosystems are not necessarily at equilibrium, they may be responding to past extinctions for example (thinking of work by William Bond on the effects of Moa extinctions in New Zealand vegetation), or invasions, or still be undergoing a process of colonisation as more species find them/adapt to them (e.g. urban ecosystems).

AU RESPONSE 22: We agree with this point. We avoid any implication in our ms that ecosystems and their drivers are static or necessarily in equilibrium. In revised introductory text of our ms we make it clear that ecosystem dynamics (of both drivers and traits) is central to our conceptual framework (Lines 81-84):

“Sustaining ecosystem functions and services [10] requires an understanding of ecological processes and mechanisms that drive ecosystem change, irrespective of specific biota within the ecosystems [7]. Ecosystem functioning not only underpins biomass production, but also depends on, and regulates the stocks and fluxes of resources, energy, and biota [11].”

To highlight our emphasis on dynamic ecological processes as a key point of difference between our approach and other classifications, we added the following text (lines 219-221):

“Embracing the dynamic nature of ecosystems and its dependency on ecological processes is a key feature that differentiates the IUCN Global Ecosystem Typology from other ecological typologies (Table S1.2).”

As a related point: the main text (lines 128-130) states that “The model posits that ecosystems share convergent functional traits if they are shaped by similar drivers, and conversely, major changes to these drivers (or their dependencies) cause disassembly, transformation, and ultimately ecosystem collapse” – arguably there is no such thing as “ecosystem collapse”, simply ecosystem transformation (more or less abrupt).

AU RESPONSE 23: Our usage of ‘ecosystem collapse’ follows the definition stated in the IUCN Red List of Ecosystems (RLE) Categories and Criteria, an internationally agreed standard for ecosystem risk assessment. The guidelines for the RLE (Bland et al. 2017) state (p13), ‘Transitions to collapse may be gradual, sudden, linear, non-linear, deterministic or highly stochastic,’ consistent with the referee’s interpretation of ecosystem transformation (more or less abrupt). In essence, ecosystem collapse (as defined by the IUCN) is a type of transformation. We prefer to retain both terms in our text order to communicate the point clearly to the widest possible audience, and use terminology consistent with international standards.

* 2.3 Soil ecosystems

Soil is presented as a substrate (e.g. in Table S3.2), but it can also be seen as a (subterranean) ecosystem on its own right, with distinctive communities and structured by very specific biotic and abiotic drivers. I certainly cannot see the rationale for having a class for endolithic systems (S1.2) but not for soil systems.

AU RESPONSE 24: We use ‘substrate’ in our ms as a generic term to encompass soils, lake beds, stream beds, shorelines, sea floors and outcropping rock in a range of ecosystem types. Substrates are integral parts of the respective ecosystem functional groups for which they are included in

descriptions and models. This is because there is a strong flux of resources, energy, matter and organisms between the substrate and other parts of the ecosystem. In adopting the IUCN Global Ecosystem Typology as the reference classification for ecosystem accounting, the United Nations Statistic Commission specifically included soils as part of terrestrial ecosystems for accounting purposes. In contrast to the substrates of various ecosystem types, endolithic systems (S1.2) are largely insular ecosystems within rocks of the earth's crust (not regolith) that have no strong association with other systems. The fluxes of resources, energy, matter and organisms to and from other ecosystems is generally limited, they have a distinctive biota with specialised microbes, extremophiles, and truncated trophic webs. They are among the most poorly known ecosystems on earth. For these reasons, we think it is important to highlight them as an important and unique group of subterranean ecosystems that warrant more attention from the scientific and wider communities. Referee #5 commended this aspect of the typology.

* 2.4 The 6 levels

I realise this paper focuses on levels 1-3, and that is fine. But regarding the other three levels, I think a couple of extra sentences are needed in the main text to explain the logic – currently all that the reader has is the legend of Figure 2, which is very cryptic. Personally, I find the logic brilliant in its pragmatism: by allowing both a top-down subdivision of level 3 (e.g. using ecoregions) and a bottom-up aggregation of existing ecosystem classifications (e.g. at the national level) into level 3; these are complementary approaches making the best use of available data, with the second key to ensure local appropriation/uptake. But I had to dive deep into Appendix 3 to understand Figure 2. Please make it easier to the reader in the main text.

AU RESPONSE 25: Thank you for recognising the advantages of combining top-down and bottom-up approaches in construction of the typology. We have expanded the relevant paragraph in the main text to give further details of the three lower levels of the typology. See response 12 for details.

I still don't understand Level 5. I can see how national classifications (Level 6) can be aggregated to EFGs (Level 3). If I am understanding it well, that is what is exemplified in Tables S3.3 and S3.4. But I don't understand what a Level 5 classification (Global Ecosystem Types) corresponds to, and how it would nest underneath the (also global) Ecosystem Functional Groups. As described, it gives the impression Level 5 is simply not needed. Please provide an example.

AU RESPONSE 26: Level 5 units are intermediate aggregations of Level 6 units that are regional variants of EFGs that are suitable compositional units for global analysis. They are necessary to rationalise inconsistencies among different local classifications that sit in Level 6. For example, the national classification of Finland recognises more than 400 different ecosystem types in the boreal zone, whereas the national ecosystem classification for Myanmar, a larger country in the tropics, currently describes only 62 ecosystem types. In other words, Level 6 units of Finland are split more finely than those in Myanmar, largely as a legacy of the independent development of those classifications. Although Level 5 is yet to be developed for these areas, it should reduce methodologically based differences in thematic resolution at Level 6 by aggregating units within and across different source classifications on the basis of compositional resemblance, with more aggregation of Level 6 units likely for Finland than for Myanmar. To clarify the nature and purpose of Level 5 in the typology, we expanded relevant text in Methods as follows (lines 428-437):

“Secondly, Level 5 units (Global ecosystem types) are also regional expressions of EFGs, but unlike Level 4 units they are explicitly linked to local information sources by bottom-up aggregation and rationalisation of Level 6 units from established subglobal ecological classifications. Subglobal classifications, such as those for different countries (see examples for Chile and Myanmar in Tables S3.3 and S3.4), are often developed independently of one another, and thus may involve inconsistencies in methods and thematic resolution of units (i.e. broadly defined or finely split). Aggregation of Level 6 units to broader units at Level 5 based on compositional resemblance is necessary to address inconsistencies among different subglobal classifications and produce compositionally distinctive units suitable for global or regional synthesis.”

Subsequent text describes contrasting strengths and weaknesses of Level 4 (top-down) and Level 5 (bottom-up) units for global analysis.

* 2.5 Terminology

I suspect the authors have already spent considerable time thinking about the nomenclature for their typology, but I will add my two cents.

- I find the term “Ecosystem Functional Group” really unappealing. It is not intuitive (a “group” of what?), the adjective “functional” is ambiguous (“functional group” sounds like a group that functions; whereas the point here is a group of ecosystems aggregated by function). Also, it is way too long and cumbersome, so it will default to another dry acronym in the conservation literature (EFG). Which is a shame, given that this is likely to be a widely used concept in policy and communication. I would recommend something simpler, like “ecosystem class” or “ecosystem category”. I realise the authors want to emphasise the “functional” aspect, but it seems like a heavy trade-off with communication. Or they could go for “Functional Ecosystem Class/category” for equal levels of cumbersome but (in my view) more clarity in meaning.

- It is confusing to have “ecotype” (level 4) and “ecosystem type” (levels 5 and 6). It is not obvious if these are different things, or if “ecotype” is a contraction of “ecosystem type”? Why not “Biogeographic ecosystem type”? (note that in evolutionary ecology, an “ecotype” has a very different meaning – a population adapted to particular environmental conditions).

- Note that in Table S3.1. the term used is “Functional Biomes” but elsewhere it is simply “Biomes”

- Given that level 5 is not nested under 4 (they are instead at the same level), it could arguable be clearer to change the numbering to make this more explicit, perhaps as

1 -> 2 -> 3 -> 4A

1 -> 2 -> 3 -> 4B -> 5B

AU RESPONSE 27: Thank you for these suggestions. There are risks and trade-offs associated with all potential names, but we think it’s important to have names as well as numbers to convey some sense of what the hierarchical levels represent.

Firstly, the label for Level 3 (Ecosystem Functional Groups) was the outcome of workshop discussions among terrestrial, freshwater and marine specialists. We think it conveys the definition of the units well – in longhand, ‘groups of functionally similar ecosystems’, information that is not so clearly conveyed by terms such as ecosystem categories or classes. We think this outweighs risks of defaulting to the EFG acronym.

Secondly, we agree that the name for Level 4 in the previous version of our ms was problematic and confusing. Level 4 units are the different expressions of Level 3 groups across different biogeographic (ecoregional) zones. We reserve usage of “ecosystem type” for Levels 5 and 6, which are derived from bottom-up from direct observations. We think it is important to differentiate these two levels of the typology from Level 4 units, which are derived from the top-down using ecoregional proxies for biodiversity to subdivide functional groups. To avoid confusion with other usages of the term ‘ecotype’, and reflect the definition as transparently as possible, we revised the name of Level 4 to “*Regional Ecosystem Subgroups*”. We hope this conveys the relationship of Level 4 units as regional subgroups within Level 3 units (Ecosystem Functional Groups).

Third, we resolved the inconsistencies between Functional biomes and biomes except where it was essential to abbreviate the full label.

Finally, although we recognise that alternative numbering of levels suggested by Referee #2 has merit, we think the ordinal numbering is simpler and emphasises the independent derivation of Levels 4 and 5, avoiding implications that are equivalent.

*** 3) ASSESSMENT OF PRESSURES/PROTECTION ***

For each EFG, the authors overlay the indicative distribution map (combining minor and major occurrences) with maps of pressure (human footprint, HF; or marine cumulative human impact, MCHI) to evaluate relative pressure (EFGs with values of pressure lower than the median are classified as “low pressure” those above as “high pressure”); and with maps of protected areas to evaluate protection levels.

The results of this analysis are only as good as the underlying spatial data. The authors include plenty of caveats on the limitations of the HF/MCHI maps (and yes, they are a problem, particularly the point that these layers do not map well the relevant types of pressures for all ecosystems). But I am equally worried about the quality of the EFG maps. Indeed, many EFGs are mapped very coarsely, with multiple EFGs overlapping over some regions. For the purposes of the analyses done here, this is particularly problematic in regions with high human presence, where EFG boundaries correspond more to broad historical distributions (i.e., the boundaries of biomes/ecoregions) than to maps of the current occurrence of the specific ecosystem types detailed in Appendix 4. For example, EFG T2.2. “Deciduous temperate forests” is mapped as covering most of Europe, North America and China, but these same regions are simultaneously densely covered by (functionally very distinct) EFGs of the Intensive Landuse Biome (T7.1. “Annual croplands”, T7.4. “Urban and industrial ecosystems”, T7.5. “Derived semi-natural pastures and old fields”). So the extremely high pressure (76% degraded) the authors find when overlapping the map of EFG T2.2 with Human Footprint corresponds mostly to historical degradation, rather than reflecting ongoing pressure. The analysis of changes in pressure (by analysing change in HF between 2000 and 2013) is also flawed, because it is done over an area much larger than that actually covered by deciduous temperate forest. This may overestimate change, by treating intensification in EFGs that are not forest (e.g. conversion from pasture to urban) as if it corresponded to forest loss; but it may also underestimate change, by diluting rates of ongoing forest loss across a very wide, non-forest area. Finally, the overlap with protected areas tells us little about how the existing deciduous temperate forests are de facto protected, because the mapped EFGs include large areas that are currently not forest.

The authors state they have also explored weighing more strongly (by a factor of two) areas of major occurrence (“where an ecosystem functional group is very likely to occur”) than areas of minor

occurrence (“where an ecosystem functional group is scattered in patches within matrices of other ecosystem functional groups or where they occur in substantial areas but only within a segment of a larger region”), having found “little effect on overall relationships in the degradation status of EFGs”. I wonder if focusing solely on areas of major occurrence would have made more sense. Although I note that for some EFGs, there are also highly transformed regions overlapping areas of major occurrence (e.g.: major urban areas within the area mapped as being of major occurrence for T3.2: Athens, Marseille, Barcelona, Tunis...).

Not quite sure what the solution here is – other than better refining the EFG maps to ensure that they correspond more to “area of occupancy” than to “extent of occurrence” of each ecosystem. As currently done, this analysis is not a good illustration of the advantages of “viewing the world’s ecosystems through a functional lens, rather than through largely biogeographic or biophysical ones” as the authors state, because the boundaries of at least some EFG maps mainly reflect biogeographic units (e.g. the boundaries of ecoregions) rather than functionally-defined ecosystems where they currently occur.

I realise that a previous reviewer (Reviewer #1, point 3) had recommended giving more emphasis to these results and less to the typology itself, but I would actually recommend the opposite. I find the typology itself is the most useful/robust contribution to the literature, with the analysis of pressure/protection still very preliminary.

AU RESPONSE 28: We acknowledge the limitations in the spatial data and have deleted this analysis from the paper (as per Response 1)

Referee #3 (Remarks to the Author):

I read with much interest this new typology of Earth’s ecosystems. I did not review the original submission and, as requested by the editor, I have focused on the review of the biome classification provided in Appendix S4, and of the Savannas and grasslands and Deserts/semideserts biomes in particular. I have also carefully revised the response letter provided by the authors to the previous round of review and the main manuscript, so I will also provide general comments on these. But please note that I have not reviewed appendices other than S4, so I assume that other reviewers will have specifically focused on them.

Comments on the main text

I will start with my general impression about the paper itself (main text). This is not certainly the typical article/review one may expect in Nature and, as already highlighted by Reviewer 1, one may argue whether an article-like format is the best way to publish this new classification. I can fully appreciate the motivation of the authors to publish it in a journal like Nature rather than as a book or a report, and I praise them for attempting to synthesize the complex process leading to this new classification in this article. However, the article format makes necessarily that key information to understand and assess the classification proposed is “hidden” in the hundreds of pages of supplementary material, and thus may be missed/can’t be found easily. To facilitate that important methodological details are fully understood by readers I would also point to specific lines/sections/figures/tables within the Appendices when citing them in the main text. Said so, it is

the decision of the Editor, not that of a reviewer like myself, to decide whether this article is suitable for Nature. As a reader of the journal I certainly would find this classification both interesting and timely; its potential to influence policy and management is also very clear.

AU RESPONSE 29: Thank you for the positive remarks. We appreciate the challenge for readers to assimilate the large and complex volumes of material that we present in this ms, and welcome the referee's suggestions to improve and simplify the presentation. Deletion of the analysis from the ms (as per Response 1) freed space in the main text, enabling inclusion more of the important explanatory detail about the typology in the main text, rather than in appendices. We have also taken the referee's suggestion to provide more specific cross-referencing to the Appendices throughout the main text and appendices. Finally, we note that much of the volume (particularly in Appendix S4) is reference material for look-up and use, rather than for end-to-end reading, and have tried to identify the key content essential to interpretation of the reference material.

I think the authors have done a careful and effective revision of the manuscript and have addressed most of the main criticisms raised in the main round of review. Said so, I have some additional comments on the main text, focused mainly on the structure/clarity of the text and on key methodological issues that I would advise the authors to consider:

I found the structure of the first paragraphs of the manuscript, which are critical in a manuscript for Nature, not very engaging. The first paragraph is OK but I would not start right away in the second presenting the IUCN Global Ecosystem Typology as it comes here out of the blue. Before doing this talk about the existing typologies and set the stage/justify the need for a new typology encapsulating both biodiversity and ecosystem functioning. Then you can talk about the IUCN Global Ecosystem Typology and the novel aspects it includes to continue with the specific objectives of the study (are not clearly stated in the current version of the manuscript) and the main results/discussion implications for guiding both science and policy/management.

Overall, I think the authors should make an extra effort to be as less cryptic and clear as possible, and to better "guide" readers through the contents of this paper. As currently written, and with so much specific terminology and references to Appendices and methods, is easy to get lost, and thus to disconnect (before finishing the second page of the manuscript!).

AU RESPONSE 30: As suggested, we restructured the 2nd and 3rd paragraphs to set the need for a new typology in context with existing systems, culminating in our overall aim. We also refined the 1st paragraph in line with suggestion of other referees. The revised text is as follows (lines 81-116):

"Sustaining ecosystem functions and services requires an understanding of ecological processes and mechanisms that drive ecosystem change, irrespective of specific biota within the ecosystems [6]. Ecosystem functioning not only underpins biomass production, but also depends on, and regulates the stocks and fluxes of resources, energy, and biota [7]. These functions, together with ecological processes and species traits (collectively, 'properties', see Glossary), define and sustain ecosystem identity, and shape ecosystem responses to environmental change, including anthropogenic changes [8]. Together with ecosystem function, the identity of constituent biota (individual species) is central to biodiversity concepts, conservation goals, and human values [9]. Although ecosystem functions and ecological processes support both the diversity of biota and human wellbeing, global assessments of ecosystems [10] [11] continue to rely heavily on species metrics or simplistic land-cover proxies that convey limited information about ecosystems themselves. This limits our ability to diagnose trends and to design and resource on- ground

management and policy solutions for slowing and reversing current declines in biodiversity and ecosystem services.

To serve dual needs of sustaining ecosystem services and conserving biodiversity, ecosystem assessments require a global typology to frame comparisons and standardise data aggregation for analysing ecosystem trends and diagnosis. To support applications throughout Earth's diverse ecosystems, users and scales of analysis, this typology should encapsulate: 1) ecosystem functions and ecological processes; 2) their characteristic biota; 3) conceptual consistency throughout the whole biosphere; 4) a scalable structure; 5) spatially explicit units; and 6) descriptive detail and minimal complexity (see Table S1.1 and Appendix S1 for rationale).

We used these six design criteria to review a sample of 23 global-scale ecological typologies, finding none that explicitly represented both ecological functions and biota (Table S1.2). This limits the ability of ecosystem managers to learn from related ecosystems with similar operating mechanisms and drivers of change. Only three typologies encompassed the whole biosphere, but these lacked a clear theoretical basis, limiting their ability to generalise about properties of ecosystems grouped together. Ecological classifications based on tested and established theory are more likely to be robust to new information than classifications based only on observed patterns and correlations, which may prove unstable when new information emerges. Many typologies that we examined either failed to describe their units in sufficient detail for reliable identification, or required diagnostic features that are hard to observe. Others were based on biophysical attributes or biogeography, but approaches differed across terrestrial, freshwater and marine domains, precluding a truly global approach. In this study, we developed a global ecosystem typology that meets all six design criteria, thereby providing a stronger foundation for systematic ecosystem assessments, sustainable management, and biodiversity conservation."

The methods section should also be improved and expanded (Nature has no word limits for this section), so it includes all the relevant information to facilitate the understanding of how the ecosystem classification/analyses were made. With so many references to Appendices (and so much relevant content included there), so many key details missing from this section and the structure used by the authors is really difficult to get a proper understanding of what has been done, and thus to judge its suitability (believe me, I had to go back and forth several times through the Methods and even so I am not sure if I fully understood what the authors did; and I feel this will happen to most readers). Adding sections to the methodology would also help to navigate through this section without getting lost. Linking the different analyses done to specific objectives of the paper (e.g. evaluation of particular Aichi targets) would also help to better understand why the author are doing them, and thus to judge their suitability.

AU RESPONSE 31: We first simplified the Methods section by removing all details related to the spatial analyses now deleted from the manuscript (see Response 1). This allowed us to expand on the methods used to develop the typology, while keeping within the 3000 word limit specified in Nature's guidelines to authors. We also added more specific cross references (page numbers, tables, figures) to appendices in supplementary material (see Response 29).

I have also some general concerns about the analyses conducted. The authors acknowledge the limitations of the pressure indices used, which failed to detect high levels of pressures in ecosystems known to suffer them (e.g., L186-L191). This raises questions about the overall exercise presented in

this study, as if the pressure indices used cannot properly account for such pressures in the real world, then the whole analyses conducted here is of limited value to guide management and policy actions. Related to this comment, the temporal analyses of pressure impacts were limited by the availability of data to the periods 2000-2013 and 2008-2013 for HFP and MCHI, respectively (L447). Thus, they are missing the change being experienced by natural ecosystems in the recent 7 years, which as the authors known have been years with an intensification of climate change drivers (e.g. droughts, warming) and human impacts (e.g. wildfires, expansion of cropping areas, intensification of fisheries) across the world. Thus, I wonder whether the conclusions obtained with data from these periods are valid nowadays. Because of this, the suitability of the approach used must be properly justified to convince critical readers (like me!). The justification provided in lines 500-511 does not sound fully convincing to me.

AU RESPONSE 32: We removed the spatial analyses and research findings from the manuscript and Appendix S6 as requested by the editor (see Response 1).

Other specific comments are below

L 72-75: long sentence difficult to read

AU RESPONSE 33: Deleted - We removed the spatial analyses and research findings from the manuscript and Appendix S6 as requested by the editor (see Response 1).

L 75: I know that the abstract is not the place to provide much details, but I think it would be useful if some of the main pressures are mentioned in brackets here

AU RESPONSE 34: Deleted - We removed the spatial analyses and research findings from the manuscript and Appendix S6 as requested by the editor (see Response 1).

L 76-78: not clear which ecosystems you refer to, those degraded and least protected?

AU RESPONSE 35: Deleted - We removed the spatial analyses and research findings from the manuscript and Appendix S6 as requested by the editor (see Response 1).

L 78-79: perhaps this sentence could be better framed as “The classification introduced here can guide policy transformation for ecosystem-specific action, including ... (list some of the key actions this classification may be particularly useful for)”

AU RESPONSE 36: We revised sentence to incorporate some specific actions as follows (lines 74-77):

“This new information infrastructure will support knowledge transfer for ecosystem-specific management and restoration, globally standardised ecosystem risk assessments, natural capital accounting and progress on the post-2020 global biodiversity framework.”

L 86-87: I may be missing something here, but could you define the “identity” of an ecosystem? I would say that stocks and fluxes of resources, energy, biomass and biodiversity already can accurately define a particular ecosystem.

AU RESPONSE 37: This is a fair point. We intended to deal with misperceptions that ecosystem function equates with productivity (an oft-used summational proxy of function), but the original text did not do that. We deleted the reference to identity, which is superfluous to the point, and revised it as follows (lines 83-84):

“Ecosystem functioning not only underpins biomass production, but also depends on, and regulates the stocks and fluxes of resources, energy, and biota (11).”

In the following sentence we state that the properties of an ecosystem (comprising functions, ecological processes and traits of constituent species) define its identity (lines 84-87):

“These functions, together with ecological processes and species traits (collectively, ‘properties’, see Glossary), define and sustain ecosystem identity, and shape ecosystem responses to environmental change, including anthropogenic changes (12).”

L104: Mention/reference the typologies you are referring to. This is an example of what I said above about pointing to specific parts/content of the Appendices to facilitate readers finding important content there.

AU RESPONSE 38: We deleted that sentence, but now reference Table S1.2 (line 104), which lists and gives sources for the 23 typologies that we reviewed.

L115: Drivers of what? Specify. The term “ecological drivers” is somewhat vague.

AU RESPONSE 39: Lines 119-120 of the revised text define “drivers” as ecological processes that shape ecosystem traits. In lines 126-127, the text notes that drivers may act as assembly filters or evolutionary pressures over different time scales (see Response 19) and in lines 132-149 we discuss the five groups of drivers shown in the model of ecosystem assembly in Fig. 1: resources, ambient environmental factors, disturbance regimes, biotic interactions and human activity.

L145-6: which pressures? Which indices? Be more explicit and cite them here.

AU RESPONSE 40: Deleted - We removed the spatial analyses and research findings from the manuscript and Appendix S6 as requested by the editor (see Response 1).

L152-153: I can’t follow the logic of this sentence, please rewrite for clarity.

AU RESPONSE 41: Deleted - We removed the spatial analyses and research findings from the manuscript and Appendix S6 as requested by the editor (see Response 1).

L179-181: Not clear if this a direction for future work or not, please rewrite for clarity.

AU RESPONSE 42: Deleted - We removed the spatial analyses and research findings from the manuscript and Appendix S6 as requested by the editor (see Response 1).

L450: unclear where these degrees of freedom come from, please provide more details on the analyses conducted. Mention also the statistical software/packages used to run the different analyses presented in the text.

AU RESPONSE 43: Deleted - We removed the spatial analyses and research findings from the manuscript and Appendix S6 as requested by the editor (see Response 1).

Comments on Appendix S4 (general comments and comments on biomes T4 and T5)

I revised the general methodology presented in this Appendix and have some reservations about the mapping exercise conducted by the authors, which otherwise is needed in a project like this one. I truly appreciate the complexities involved in delivering reliable and sound-based maps when using so many different inputs and with so many data gaps. However, the authors should think about the use that can be done of the maps provided, particularly if they are validated by the publication of this classification in the peer-reviewed literature.

AU RESPONSE 44: As noted in Response 1, we have deleted the spatial analysis from the manuscript, hence inferences drawn in the paper no longer depend on the maps. We retained the indicative thumbnail maps in the descriptive profiles of Ecosystem Functional Groups (EFGs) presented in Appendix S4 because: i) these maps add important contextual information to the descriptions of EFGs not communicated in the text; ii) the quality of even the most limited maps is sufficient to give a global overview of the distributions as thumbnail maps occupying ~10% of an A4 page.

There are some key general issues about the mapping I would like to comment on:

I found the mapping of major and minor occurrences a little bit misleading.

AU RESPONSE 45: We think the use of two broad categories of occurrence (major and minor) is an important and effective means of communicating information about the global distributions of ecosystem functional groups. We define minor occurrences (Appendix S4, p15 2nd last paragraph) as:

“areas where an ecosystem functional group is scattered in patches within matrices of other ecosystem functional groups or where they occur in substantial areas but only within a segment of a larger region.”

Omitting minor occurrences from maps, or merging minor and major occurrences (in which the EFG comprises the majority of the landscape matrix) would involve significant loss of information, notwithstanding acknowledged limitations on accuracy and precision.

I will exemplify this with the mapping of biome T4.5 (Temperate subhumid grasslands) in Spain. This biome occupies most of the territory of this country with minor occurrence. This is so despite most of this territory does not have a subhumid climate (it is rather semiarid and dry-subhumid) and temperate grasslands as described in this biome are quite rare throughout it. Since the biome

classification being proposed here has a clearly defined goal to support policies and management actions related to the conservation of biodiversity and ecosystems, I think maps such as those provided for T4.5, which show a clear mismatch between the ecosystem mapped and that found in the field, will not be very helpful for managers and indeed can prompt not effective or even damaging actions when trying to protect them. Of course, the authors indicate that the habitat has a minor occurrence throughout this territory, but how a manager can make a useful/good use of this information? What's the point of mapping a given biome across large portions of territory where it is quite unlikely to find it on the ground? I may be missing something but to me this is a major drawback of the biome mapping presented in this article.

AU RESPONSE 46: Thank you for this specific observation. We believe the discrepancy is in the name of this EFG, rather than the map of its distribution. Descriptions of ecoregions indicate that T4.4 and T4.5 are both present in central Spain. The full text description (Appendix S4 p63 under Key Ecological Drivers) notes that “*Mean annual rainfall varies [globally] from 250 mm to 750 mm*”, consistent with a subhumid-semi-arid climate and with the reviewer's remarks. We endeavoured to keep EFG names as brief as possible (Occam's Razor), in this case aiming to communicate that these grasslands were found in climates that are drier than 'humid'. However, we see how this may cause misapprehension that T4.5 is not found in semiarid climates. We therefore adjusted the name of EFG T4.5 to remove reference to climatic variation and focus on a common structural feature of the ecosystems in this group: “*Temperate tussock grasslands*”.

The authors already note that their “maps were designed to be indicative of global distribution patterns and are not intended to represent fine-scale patterns”. This is expected in a global exercise like this one from the scientific point of view given the data sources (and resolution) available. However, and getting back to what I mentioned before, many potential users, particularly land managers and other stakeholders working will use the information contained in the maps (widely accessible via the web [BTW a very nice webpage and map server!]) to guide management actions at local or regional scales. And as noted above these maps will not be very useful for them. Thus, a better justification of the approach used, or even better, a detailed guided about how to use (and not use) the maps and information provided to support biodiversity and ecosystem conservation policies and actions is warranted.

AU RESPONSE 47: We fully agree that more reliable and consistent mapping is needed for to support ecosystem management and that clear guidance on map use is needed. We are careful not to recommend our indicative maps for ecosystem management. Instead, their purpose is to provide general global-scale descriptive information on EFG distributions that cannot be communicated efficiently in text. As noted in our revised ms (lines 250-252), maps for about two-thirds of the 108 EFGs are suitable for global analysis. Fewer would meet the standard required to inform ecosystem management on the ground, which is why locally developed maps should be used for that purpose. We identify the attributes of reliable ecosystem maps, in lines 246-249:

“Maps that are most fit for purpose would be based on remote sensing and environmental predictors that align closely to the concept of their ecosystem functional group, incorporate spatially explicit ground observations and have low rates of omission and commission errors, high spatial resolution and time series of changes.”

In lines 254-266 copied below, we discuss how our approach to typology development facilitates mapping updates, give examples of the most suitable global maps, discuss the need to harness

rapidly developing remote sensing and computing power to improve map quality, and describe IUCN's governance system to progressively update the typology and its maps with release of future versions via the web page and future publications. For example, we have incorporated new maps for six EFGs into our archive during the review of our ms, and updated Appendix S4 and the web page accordingly with v2.01 of the typology.

“By decoupling the mapping process from prior development of the classification, our approach liberates the definition of ecosystem units from constraints imposed by the current availability of spatial data and allows for progressive improvement (Appendix S4, p15). New technologies in cloud computing and artificial intelligence, improved global environmental data, and deepening time archives of satellite images are paving the way [10] [42]. High-resolution maps, some with extended time series, that match the concepts of EFGs have been produced for contrasting ecosystem groups such as tidal mudflats TM1.2 [43], glacial lakes F2.4 [44] and tropical cloud forests T1.3 [45] (Table S4.1); recently-developed data cubes for a diverse range of species habitats [46] suggest that global high-resolution time series mapping should be possible for most ecosystem functional groups within the next decade. Future versions of the typology will progressively strengthen map standards and improve applications that depend on spatial analysis. Improved mapping of threats and degradation is similarly required to support ecosystem assessments [47], particularly in marine environments.”

Also, it would be certainly good if a box explaining how to interpret the diagram with the links between resources/biotic interactions/traits... is included. It is not convenient to go to the main text or to another appendix to understand this figure.

AU RESPONSE 48: The generic form of the diagram is presented in Fig. 1, which is adapted for each Ecosystem Functional Group presented in Appendix S4. Fig. 1 has a detailed caption containing the information requested by the Referee #3. The figure and its caption are reproduced in Appendix S2, which has a detailed commentary on the development and interpretation of the assembly model. Appendix S4 has a section entitled “*Diagrammatic assembly models*” that gives basic guidance on the interpretation of these diagrams and cross-references Fig. 1 and Appendix S2. We think this should be helpful guidance for users.

Overall, I found the different sub-biomes included within the T5 biome classification sound and correct to the best of my knowledge. However, it was surprising to me not finding a specific biome category for dry tussock steppes such as those that occupy vast spaces across the SE of Spain and the North of Africa (Morocco, Algeria, Tunisia, Le Hourérou 2001). Much of the area covered by these steppes is incorrectly (based on what one can observe in the field) included as temperate woodlands (T4.4). A similar problem to that discussed for biome T4.5 above is also found for biome T4.4 (temperate woodlands), as it is drawn as a dominant biome across many dryland areas of the Mediterranean Basin that have rainfall levels < 350 mm and that have a shrub- and grass-, rather than tree-dominated vegetation.

AU RESPONSE 48: Dry tussock steppes are included within EFG T5.1 Semi-desert steppes, which is indeed mapped across the north of Africa in Morocco, Algeria and Tunisia (Appendix S4, p60). In the southeast of Spain, as well as Morocco, Algeria and Tunisia, we map minor occurrences of T4.5, which forms a continuous transition to T5.1 with increasing aridity. It is possible that T5.1 also occurs

in southeast Spain, but we defer to more definitive mapping to represent the distinctions between these inter-grading units in future versions of the typology.

Other minor comments on the text of biomes T4 (Grasslands and savannas) and T5 (Deserts and semi-deserts) are the following:

- “Herbivory is the primary driver in highly fertile and productive systems, whereas fire is the primary driver in less fertile and lower productivity systems.” Driver of what? unclear.

AU RESPONSE 49: We modified the sentence to clarify as follows (Appendix S4, p58):
“*Herbivory is the primary driver of ecosystem assembly...*”

- “Nutrient gradients are exacerbated volatilisation during fire and the loss of nutrients in smoke” I can’t understand this sentence

AU RESPONSE 50: We added missing words to clarify the sentence as follows (Appendix S4, p58): “*Nutrient gradients are exacerbated by volatilisation during fire and the consequent loss of nutrients in smoke.*”

- I would not say that biotic interactions are weak in desert biomes, as it is well known that facilitative and competitive plant-plant interactions, as well as plant-herbivore interactions, can be intense in these ecosystems (e.g. Fowler 1986, Graff et al. 2007, Graff & Aguiar 2016).

AU RESPONSE 51: We base this hypothesis on the stress-gradient theory (Maestre et al. 2009, J. Ecol.), which states that competition becomes less important relative to facilitation as resource availability declines and stress increases. As deserts and semi-deserts are resource-limited environments, we suggest that competitive interactions are weaker there than in other biomes with higher resource availability, such as forests, shrublands and grasslands. We acknowledge that competitive effects on grasses may be strong where they occur within neighbourhoods of other plants. To clarify the context of the hypothesised relationship with appropriate qualification, we made the following adjustment to text (Appendix S4, p64):

“*Competitive interactions are generally weak, relative to most other terrestrial biomes (T1-T4), although ...*”

I hope that despite my criticisms, which I have made with the idea to be constructive, you will find the comments provided useful to further revise and improve this work.

Thank you, we found them very useful.

References

Fowler, N. The Role of Competition in Plant Communities in Arid and Semiarid Regions. Annual Review of Ecology and Systematics 17, 89–110 (1986).

Graff, P. & Aguiar, M. R. Do species' strategies and type of stress predict net positive effects in an arid ecosystem? *Ecology* 98, 794–806 (2017).

Graff, P., Aguiar, M. R. & Chaneton, E. J. Shifts in Positive and Negative Plant Interactions Along a Grazing Intensity Gradient. *Ecology* 88, 188–199 (2007).

Le Houérou, H. N. Biogeography of the arid steppeland north of the Sahara. *Journal of Arid Environments* 48, 103–128 (2001).

Referee #4 (Remarks to the Author):

The manuscript by Keith et al. describes the IUCN global typology of ecosystems, based on the functional traits of the species and ecosystems. In this era of global change and human impacts there is a dire need to make global comparisons of ecosystems, the impacts they receive, and their threats. The rationale of the typology is explained, and the utility is illustrated by showing how different terrestrial, freshwater and marine ecosystems are protected and potentially threatened by global change.

The strength of the manuscript are:

- 1) a unifying global typology that allows to classify bewildering different ecosystems that are often assessed separately (e.g., terrestrial, freshwater, marine),
- 2) a consistent classification based on community assembly and the functioning of organisms and ecosystems, which allows for a better mechanistic understanding, assessment and prediction of the consequences of environmental change,
- 3) a separate inclusion of humans as an environmental driver, which allows to assess the consequences of human activity for the biodiversity and functioning of the planet, and allows to design policies to change human activities or mitigate their effects,
- 4) the application of the typology by describing and mapping the 100 ecosystem types. The description in Appendix S1 with one page factsheets are a pleasure to read, as they are nice and concrete, succinct, well written, conceptually consistent by showing the same conceptual diagram with different drivers, and nicely illustrated with a clear beautiful photo conveying the message, and reference for further reading

AU: Thank you for these positive remarks

The weak points of the manuscript are

- 1) A very vague, poorly written main text article which was for large parts not understandable (even no for me as a functional ecologist!) and below publication standards
- 2) The conceptual diagram should be improved as the terminology is (in my opinion) not consistent, it should made more clear that humans affect ecosystems by affecting the other drivers
- 3) Conceptually, distinguish better between traits of individuals/species, and traits of ecosystems. It is unclear what you mean with 'ecosystem traits', and whether you suggest that ecosystems are filtered out by the environment (which I think is conceptually wrong).
- 4) Better explain and show how you used in practice these functional traits to classify the ecosystems. It seemed to me that you used a-priori defined ecosystems and described them

afterwards with your functional typology, whereas it should of course have been done the other way around!

AU: Thank you for these critical insights. We address them below under Major comments

I was specifically asked to look at the descriptions of the tropical systems, which I do further below. I have reviewed 7 tropical systems. Given the fact that the nature reader thinks that everything what is published in Nature has been scrutinized and is true, I echo the concerns of the previous reviewer that it is imperative that ALL 100 ecosystem descriptions are checked by specialists. So you still have to look for reviewers for the other 93 ...

Overall, this has been an admirable and Herculean task, for which I congratulate the authors. Please improve the main text article, as it does not do justice to the rest of the work you have done. Please find my major and minor comments below, which I hope are of help to improve the manuscript.

MAJOR COMMENTS

1. IMPROVE THE INTRODUCTION TEXT. I am a functional ecologist and biologist, do fieldwork in a variety of ecosystems across the world, and have affinity with management and conservation. So I thought that I should be able to understand the main text, and get inspired by it. I must admit that I found the main text very vague, unclear, full of undefined jargon, and below publication standards. So if the authors want to reach and convince a wider audience, then please invest time to make it attractive, understandable and accessible.

AU RESPONSE 52: We have substantially revised the main text to improve clarity. Removing the analysis component of the study enabled us to simply the content and structure of the main text, and also provided an opportunity to expand and clarify explanations of the approach, rationale and application of the typology. We restructured the first paragraphs to lead readers through the need for a global typology, the inadequacy of the current information infrastructure to serve the need, and the barriers to ecosystem management solutions, culminating in the aim of our study (see Response 30). We also revised text on the assembly model, and its interpretation in relation to assembly and evolutionary theory, as well as the structure of the typology (see Response 19).

2. IMPROVE YOUR CONCEPTUAL DIAGRAM. Appendix S2 (conceptual foundations) is well written, interesting, and a pleasure to read. Figure 1 presents your conceptual model, which is the cornerstone of your whole typology, and returns in each ecosystem description. The wording, should therefore be crisp and clear, and above, all, correct. In my opinion, several terms and classifications are now incorrect and should be improved:

- Human activity. The real conceptual problem I have is that human activity does not DIRECTLY affect the ecosystem traits and species traits (as your arrow suggests), but human activity INDIRECTLY affects the ecological traits by changing environmental conditions, resources, disturbance regimes and biotic interaction. This applies for all the four items you give as example in your human activity box. For example, structural alteration works through disturbance. Resource use is either a disturbance (you remove biomass), or a biotic interaction (it equals predation). Movement of biota is a biotic interaction (dispersal). Climate change works through changing temperature (an environmental condition), and carbon and water availability (resources). So I really think it is conceptually wrong to frame human activity as having an independent effect on the traits, You

should have arrows from human activity to disturbance regime, biotic interactions, resources and environmental conditions.

In the same way as you explicitly draw an arrow from environmental conditions to resources. It therefore does not suffice to say that those human effects are included through the circle with the broken line.

AU RESPONSE 53: We address specific issues related to terminology below and have amended the text to clarify the context of human activity in the model and the typology (see Responses 20 & 21). We accept the referee's perspective about the indirect nature of human influence on ecosystem assembly through other drivers. We agree that most effects of humans operate this way, and that the generic model in Fig. 1 did not communicate this as effectively as the ecosystem-specific models in Appendix S4. We therefore modified Fig. 1 to represent indirect effects of human activity more explicitly (see Response 20). However, we think it is useful to frame the model in a way that allows for some direct effects as well as indirect effects (see Response 20). While we generally agree with the reviewer's interpretation of his/her examples above, we see difficulty in interpreting all interventions as indirect, particularly where humans are 'driving' the construction and persistence of anthropogenic ecosystems. For example, in Ecosystem Functional Group T7.1, the transformative removal of indigenous plants and animals, replacement by crops and instigation of new types of disturbance regimes (harvesting, ploughing) could be viewed as indirect human effects on biotic movement and disturbance regimes, but we think this understates the effect and the nature of intervention. In segregating human activity as a 'special kind' of biotic interaction we distinguish (for example) simply removing and introducing biota from manipulating species dominance in a way that shapes productivity, trophic structure, diversity and other ecosystem traits. Similarly, we propose a distinction between altering disturbance regimes that exist without human intervention (e.g. floods, fires) as an indirect effect, and imposing new types of disturbance regimes (e.g. harvesting, ploughing) as a direct effect that would not exist without human intervention. We think this distinction between direct and indirect effects is useful in other anthropogenic systems, such as M4.1 artificial reefs, SF2.2 flooded mines or F3.1 large reservoirs, where humans deterministically construct infrastructure of an ecosystem that generates secondary effects on biotic movement, resource supply, the ambient environment, etc.

- Human activity. I understand that in the Anthropocene and as preservationists you want to frame humans not as being part of nature, but as being outside nature and affecting nature. Many people may disagree with you, so justify somewhere why you put human activity as a separate box.

AU RESPONSE 54: It was not our intention to frame humans as not part of 'nature', rather to highlight their influence as integral drivers of ecosystem assembly. We added the following text to clarify (lines 146-149).

"While our model portrays humans as integral drivers of ecosystem assembly, we separated human activity from other biotic interactions to highlight interactions and feedbacks between ecosystems and socio-economic systems [18], and the need to assess and mitigate the human impacts on biodiversity and ecosystem functioning."

- The term "Resource processes" is incorrect. What you show in the box are resources (water nutrients, etc), so no processes of resource uptake or loss. So rename this box "Resources"

AU RESPONSE 55: We renamed the box as suggested. Our original intent was to represent resource supply processes, but we agree “Resources” is sufficient.

- The term “Ambient environmental processes” is incorrect. What you show in the box are environmental conditions, not processes. The word “ambient” is also confusing. If these conditions are ambient, then you should also label your “resources” as “Ambient resources”. I think the correct name of this box is “Environmental conditions”. And please define in the legend “Kinetic energy”. I did not have a clue what you were referring to

AU RESPONSE 56: We renamed this box, “Ambient environment” in Fig. 1 and explained our use of the term “ambient”, which was to help distinguish aspects of the environment surrounding (in a broad sense) an ecosystem that continually regulate availability of resources or ability to use them from “Disturbance regimes” – sequences of discrete events that operate as triggers for processes identified in the text. We revised the text to explain this rationale as follows (lines: 134-139)

“The ambient environment (Appendix S2, p2) includes surrounding environmental features (e.g. temperature, pH, salinity) that continually influence the availability of resources or the ability of organisms to acquire them. The model distinguishes these continuous factors from disturbance regimes (Appendix S2, p2), which are sequences of discrete events with different intensities and patterns of occurrence (e.g. fires, floods, storms, earth mass movement) that destroy living biomass, liberate and redistribute resources, and regulate life history processes.”

We added a definition of kinetic energy in the context of ecosystem assembly (see below) to the Glossary in Appendix S4 (Table S4.1), and a cross reference to the Glossary in the caption of Fig. 1:

“Kinetic energy – a property of aquatic ecosystems describing the motion of water in terms of velocity and mass. It influences the assembly of aquatic systems through movement and supply of resources (oxygen, nutrients, organic particles containing energy and carbon) and through influences on other components of the ambient environment (e.g. substrate stability), particularly for ecosystems in the marine shelf (M1), shorelines (MT1) and rivers and streams (F1) biomes. For example, streams with rapid flows of high volumes of water have high kinetic energy, influencing life histories and feeding traits of their biota and stability of their substrates.”

- The term “Biotic interactions” does not match the names inside the box. “Competitors” are organisms that are interacting, it is not a biotic interaction. So replace the names in such way that they reflect real interactions; “Competitors” should be “Competition”, “Predators” should be “Predation”, Mutualists should be mutualism. And think yourself how to rename “pathogens” and “engineers”.

AU RESPONSE 57: Thank you for identifying this inconsistency. We have corrected the terms as suggested, including “Pathogenicity” and “Facilitation”.

- “Engineer” is in my opinion not a biotic interaction. Facilitation is. Is that what you mean? If so, use facilitation. That is closer to your “community assembly theory”. Or do you mean that “ecosystem engineers” modify the landscape? Then it is in my opinion not a biotic interaction anymore.

AU RESPONSE 58: We replaced “Engineers” with “Facilitation” as above.

- Disturbance regime. If you define a disturbance as a sudden event that destroys biomass then flooding is NOT a disturbance regime (it is not sudden, and generally does not remove biomass), but a stress. I think it conceptually belongs to the environmental box, because flooding modifies resource availability by reducing oxygen and light, and increasing water availability and nutrient availability (through deposition). And please define in the legend “Mass movement” and “Igneous activity”. I did not have a clue what you were referring to.

AU RESPONSE 59: We draw an important distinction between successive flood (overbank flow) events and variations in stream flow, lake levels or currents (not overbank events). We interpret the former as a disturbance regime, but we agree that the latter are more logically placed in the Ambient Environment box. The concept of flood regimes (as sequences of successive inundation events that can be described in terms of frequency, depth, velocity, duration and extent) is well established in aquatic ecology. Flood regimes are consistent with all four components of our definition of disturbance regimes (see Glossary): 1) the inundation events are discrete in time, though not necessarily sudden (we avoid using that term in the definition of disturbance regime); 2) the events typically destroy some biomass, though this varies depending on the flood regime (especially velocity and duration); 3) flood events redistribute resources (water nutrients, and indirectly light through changes in ecosystem structure); and 4) floods trigger life history processes in some biota (e.g. seed germination, waterbird breeding, hatching of zooplankton diapause eggs). “Mass movement” and “Igneous activity” are defined in the Glossary in Appendix S4. We think they are widely understood terms so they could be excluded from the abridged glossary if space is an issue in the main text. We have added a cross reference to the Glossary in the caption for Fig. 1.

- Ecological traits. In the central circle make a distinction between what are your ecosystem traits and what are your species traits. For me as a functional trait ecologists it is VERY confusing that they are all listed as ecological traits, as the ecosystem traits are an emergent property of the species traits.

AU RESPONSE 60: See Response 62.

- Species pool. Move the word to a different place in your diagram. Now it erroneously suggest that it refers to the broken circle

AU RESPONSE 61: We deleted the broken-line ellipse (see Response 20), avoiding mis-association with the species pool.

3. CLARIFY CONCEPTUALLY ECOLOGICAL TRAITS AND ECOSYSTEM TRAITS. Ecological traits are the basis of your typology. Maybe I have missed it, but you nowhere define what a trait is. In traditional functional ecology a trait is a property of an individual, affecting its performance (growth, survival, reproduction, fitness) (Violle et al. 2014 PNAS). I can imagine that you assign a trait to a species. But then the question is: what is an “ecosystem level trait”? In my humble opinion, assembly theory is about the assembly of individuals and species into a community. A trait is therefore a property of an individual or a species. In my opinion, your ecosystem level traits (productivity, diversity, trophic structure and physiognomy) are the emergent consequences of the traits of the individuals in

combination with the environmental conditions. For example, primary productivity of a forest is determined by the number of individuals, and by their size, total leaf area and photosynthetic capacity (all properties of an individual), in combination with the environmental conditions (light, temperature) that determine photosynthetic rates). So using this analogy, I do not see why productivity would be an ecological “trait” of the ecosystem. It is simply the emergent consequence of species traits and environmental conditions. To circumvent this problem, you could define a “trait” simply as an attribute of an individual, species, or ecosystem. I am fine with that, but in your conceptual framework you suggest that ecosystems are filtered out by the environment based on their ecosystem traits, and that is of course not true. I think the environment does not filter out ecosystem A or B. It filters the individuals and species belonging to ecosystem A or B.

AU RESPONSE 62: Thank you for these thoughtful remarks. We see that our usage of the term “trait” was confusing in the previous version of our ms, especially in the absence of an explicit definition. An appropriate and concise collective term needs to fit within the format of the diagrammatic model. Thank you for a suggested definition of a “trait” to circumvent the problem. Rather than broaden an established definition of this term, we replaced “Ecological traits” with “Ecosystem properties”, which we defined as attributes of ecosystems and their component biota that result from assembly processes, and a collective term for ecosystem-level functions, ecological processes and species-level traits (see glossary for those terms).

The model and the explanatory text in the caption, main text and Appendix S2 explain that filters and evolutionary pressures operate on ecosystem properties. To ensure the text cannot be misinterpreted to suggest that ‘ecosystems are filtered out by the environment’, we revised the text to state even more clearly that ecosystem properties are outcomes of filtering, replacing text that previously introduced ecological traits on lines 119-125 as follows:

“The model (Fig. 1) frames working hypotheses about the processes (or ‘drivers’) that shape ecosystem properties and the interactions among drivers and properties. Ecosystem properties are attributes of ecosystems and their component biota that result from assembly processes. They include aggregate ecosystem functions (productivity, stocks and fluxes), ecological processes (e.g. trophic networks), structural features (e.g. 3-D spatial structure, diversity), and species-level traits of characteristic organisms (e.g. ecophysiology, life histories, morphology).”

We also added this definition to the Glossary in Appendix S4, along with definitions of species traits (based on Violle et al. 2014) and ecosystem functions (based on Pettorelli et al. 2018). We avoid reference to “emergent properties”, as some readers may interpret this to mean properties of groups that amount to more than the sum of their parts, which does not hold for ecosystem properties such as biomass or productivity.

4. CLARIFY HOW YOU CLASSIFIED YOUR ECOSYSTEMS BASED ON FUNCTIONAL TRAITS. Better explain and show how you used in practice these functional traits to classify the ecosystems. It seemed to me that you used a-priori defined ecosystems and described them afterwards with your functional typology, whereas it should of course have been done the other way around!

AU RESPONSE 63: We made substantial revisions and additions to the Methods section to clarify the sequence of development. The opening paragraph of the Methods now summarises this sequence as follows (lines 317-322):

“We developed the IUCN Global Ecosystem Typology in the following sequence of steps: design criteria; hierarchical structure and definition of levels; generic ecosystem assembly model; top-down classification of the upper hierarchical levels; iterative circumscription of the units and ecosystem-specific adaptations of the assembly models; full description of the units; and finally map compilation. Some iteration proved necessary as the description and review process sometimes revealed a need for circumscribing additional units.”

Removal of content related to the analysis (see Response 1) enabled us to restructure the Methods text to follow this sequence throughout and add detail to more fully describe the steps.

CONTENT-WISE COMMENTS MAIN TEXT

L139. You seem to have a very static view on ecosystems. Most are dynamic and resilient (to a certain extent)

AU RESPONSE 64: We emphasise ecosystem dynamics throughout the ms, including the revised opening of the second paragraph and the concluding sentence of the main text and several passages between them (e.g. lines 219-222). See Response 22 for more detail.

L140. ASSESSING THE RISK OF PRESSURES. Nice that you have a human pressure map. But who says that these are relevant to the functioning of ecosystems. So you assume a threat, but as you did not measure ecosystem response it is just a supposed threat. So how relevant is this exercise?

AU RESPONSE 65: We removed the spatial analyses and research findings from the manuscript and Appendix S6 as requested (see Response 1).

FIG 1. See my major comment 2 at the beginning of my review

AU RESPONSE 66: See Response 53.

FIG 2. I did not understand the figure, nor the legend

AU RESPONSE 67: Fig 2 has been replaced by a Table as suggested by the editor and Referee #1.

FIG 3. WHAT IS THE MEANING OF ABOVE OR BELOW AVERAGE. Everything is relative, so how important it is that an ecosystem faces above- or below pressures (your dotted lines). The question is how much a certain pressure results in an absolute response of the system (some systems are resistant or resilient, others are sensitive)

AU RESPONSE 68: We removed the spatial analyses and research findings from the manuscript and Appendix S6 as requested (see Response 1).

FIG3. WHAT IS THE MEANING THAT PROTECTION DECLINES WITH HIGH PRESSURE? It could be that people selected parks in remote areas as we do not need those (= probably areas with low pressure),

it could be that conservation results in less pressure. We simply can not tell. So what can we learn from this? In that case you should have done a temporal analysis (once it is a park, does the pressure decline).

AU RESPONSE 69: We removed the spatial analyses and research findings from the manuscript and Appendix S6 as requested (see Response 1).

FIG 4., Why a one-tailed test? I think it is more robust and appropriate if you would do two-tailed test. As an ecologists my prediction often was the opposite of what I found, so a one-tailed test makes little sense. And what is in the end the added value of having once conceptual framework if you still analyze terrestrial, freshwater, and marine systems separately?

AU RESPONSE 70: We removed the spatial analyses and research findings from the manuscript and Appendix S6 as requested (see Response 1).

MINOR COMMENTS ON INTRO MAIN TEXT:

DEFINE TERMS. If this approach is to be a vehicle for multidisciplinary or interdisciplinary collaboration, then all terms should be defined very clearly. It would be great if the authors would add in the main text a box with a glossary. Please define what you mean with “functionally similar responses” (l68), “functions” (l71), “Ecosystem Functional Groups” (l74), “resources” (l85), “ecological functions” (l102), “biota” (l102), “filters” (l115), “community assembly theory” (l116), “ecosystem traits” (l116) “disassembly and reassembly” (l125) “semi-deterministic resource appropriation” (l 125), “physical restructuring” (l124), “movement of biota” (l126), “convergent functional traits” (do you mean converging from an evolutionary perspective? (l128), “functionally based ecosystem typology” (l132), “generic indices” (l 148), “cryogenic ecosystems” (l167).

AU RESPONSE 71: Several of these terms were defined in the main text, while others were defined in the glossary. We placed the glossary in Appendix S4 (p24) in that appendix because much of its content is relevant to the descriptive profiles included there. However, we appreciate that this may make it less accessible to readers focussing on the main text. Rather than move or reproduce the entire glossary in the main text, we have prepared an abridged glossary (cross-referenced to the full version in Appendix S4) that could be included in the main article, methods, or as extended tables and figures (as preferred by the editor).

The abridged glossary includes definitions for terms used in the main text and Fig 1, including: functionally similar ecosystems, ecosystem function (also in main text), resources, filters, ecosystem attributes, ecosystem properties, species traits, disassembly, reassembly, resource appropriation, physical restructuring, convergent functional traits, and cryogenic. Ecosystem Functional Groups are defined and discussed in the main text, Methods and Appendix S3 in some detail, and described in Appendix S4. Other terms mentioned by Referee #3 above have been deleted or edited in response to comments from other referees.

CLARIFY TEXT. Large parts of the introductory text are very vague or totally unclear.

L66. What do you mean that “ecosystems contribute to biodiversity”?? I would say that ecosystems vary in their biodiversity

AU RESPONSE 72: We deleted “contributions to” as suggested.

L74-75. This sentence does not flow and it is unclear what <17 or <10 refers to, AU RESPONSE 73: Sentence deleted (see Response 1).

L75-678 This is a mixed bag of many things. Provide some conceptual structure. Now it feels like a random list of things.

L74-75. This sentence does not flow and it is unclear what <17 or <10 refers to, AU RESPONSE 74: Sentence deleted (see Response 1).

78-79: What do you mean with “globally comprehensive typology”, “policy transformation” and “ecosystem specific action.”? The links that you suggest are totally vague. L

AU RESPONSE 75: The Summary paragraph introduces these points briefly, and we explain them more fully in relevant sections of text. The typology is comprehensive because it encompasses the entire biosphere (design principle 3, lines 99-100), also mentioned in the preceding sentence (lines 97-98), “To support applications throughout Earth’s diverse ecosystems,...”. The typology can support policy transformation (i.e. major policy change) by establishing a consistent framework for ecosystem risk assessment, accounting and reporting on progress towards goals (lines 200-216 and Appendix S6). It can facilitate ecosystem-specific action including management and restoration strategies (lines 217-222) through knowledge transfer by grouping ecosystems with similar management needs (lines 224-226).

80: a typology is not an infrastructure, it is a tool.

AU RESPONSE 75: We see this global typology as more than a tool. It is a foundational framework for organising the evidence base needed to conserve biodiversity and sustain ecosystem services on a global scale. We clarified our use of this term by referring to “information infrastructure”.

L83-84: This is a complicated sentence.

AU RESPONSE 76: We rephrased it as follows (now on lines 81-83):

“Sustaining ecosystem functions and services (8) requires an understanding of ecological processes and mechanisms that drive ecosystem change, irrespective of specific biota within the ecosystems (5).”

L86: what is the difference between energy and biomass?

AU RESPONSE 77: Energy is widely understood as the capacity for doing ‘work’ (physics terminology), and may be expressed as light, thermal, chemical, kinetic, potential energy or various other forms. Biomass is the total mass of living tissue in a specified area or group of organisms (i.e. a form of matter). Einstein would have them as two sides of the same coin, but we think they are both essential to a clear definition of ecosystem function.

L87-88. What do you mean with “The identity of biota is also central to biodiversity concepts, conservation goals, and human values”?

AU RESPONSE 78: We clarified the sentence as follows (lines 87-89):

“Together with ecosystem function, the identity of constituent biota (individual species) is central to biodiversity concepts, conservation goals, and human values [13].”

L92. What do you mean with “design and resource on-ground solutions”?

AU RESPONSE 79: We added a phrase to clarify the sentence (lines 92-94):

“This limits our ability to diagnose trends and to design and resource on-ground management and policy solutions for slowing and reversing ongoing declines in biodiversity and ecosystem services.”

L83. Why is there a dual need for sustainability and conservation. Why do you need both? And do you suggest that there is a trade-off between the two?

AU RESPONSE 80: These dual needs are inherent in the UN Convention on Biological Diversity, its targets, and the UN Sustainable Development Goals. While there may be some trade-offs among them, that issue is beyond the scope of our paper. We made minor edits to clarify the text (lines 95-97):

“To serve dual needs of sustaining ecosystem services and conserving biodiversity, ecosystem assessments require a global typology to frame comparisons and standardise data aggregation for analysing ecosystem trends and diagnosis.”

L95-97. Why are especially these design principles crucial? Please justify

AU RESPONSE 81: The rationale underpinning each of the design principles is summarised in Table S1.1, with the justifications elaborated in text of Appendix S1. We have revised the commentary on lines 103-117 to clarify the justification of the design criteria. Key points in this text include: “the ability of ecosystem managers to learn from related ecosystems with similar operating mechanisms and drivers of change” (lines 105-106); “the ability to generalise about properties of ecosystems grouped together” (lines 107-108); “yield generalisations that are robust to new information because differences between groups are founded on causal relationships” (lines 109-110); and “a truly global approach” (line 115), i.e. consistency across terrestrial, freshwater and marine ecosystems.

L99. How different are these “biophysical attributes” from your stocks and fluxes?

AU RESPONSE 82: We added biophysical attributes to the Glossary as follows:

“Spatial representations of environmental variables (e.g. maps of climate variables, topography, bathymetry, substrate, etc.) that are assumed to represent the dimensions of species’ environmental niches. They are used to classify the earth’s surface into spatial units that represent the distributions of different assemblages of biota (see Table S1.1 for examples of biophysical classifications). Some biophysical attributes may be correlated with ecosystem stocks and fluxes.”

L99-100. Very good point!

Thank you

L102. What is the difference between an ecological function and an ecosystem function? Please use your terms consistently throughout. It seems that sometimes you use synonyms, which is confusing

AU RESPONSE 83: We rationalised and standardised this nomenclature throughout the main text and appendices, and included the terms in the glossary following definitions reviewed by Pettoirelli et al. (2018).

L105 Why would that “limit the ability to generalise about properties of ecosystems grouped together”?

AU RESPONSE 84: See Table S1.1 and text of Appendix S1. We elaborated and revised the main text on this point (lines 108-111):

“Ecological classifications based on tested and established theory are more likely to yield generalisations that are robust to new information because differences between groups are founded on causal relationships. In contrast, classifications based only on observed patterns and correlations may prove unstable when new information emerges.”

As an aside, we note that similar thinking was fundamental to the debate comparing phylogenetic taxonomic classifications to phenetic ones in the 1970s and 1980s, which ended with a shift to phylogenetic classifications underpinned by evolutionary theory.

L109. What do you mean with “to serve as a template”? What are your “units of classification”?

AU RESPONSE 85: As noted in the caption of Fig. 1, the generic model is a template that was adapted to help define and describe each of the 108 Ecosystem functional Groups in Level 3. The units of classification are the elements within each level of the classification (e.g. 5 realms, 25 biomes, 108 Ecosystem Functional Groups).

L111. What do you mean with “Interactions and dependencies amongst drivers and traits”. This is very vague

AU RESPONSE 86: We clarified by adding examples of interactions and feedbacks among drivers in the main text Lines 152-157 (see Response 20 for further discussion of interactions involving human activity). We also provide an extended discussion of other interactions in Appendix S2 (pp6-7).

L127. What is the difference between “dependency” and “inetractions”?

AU RESPONSE 87: We deleted ‘dependency’ to avoid repetition.

L131-134. I am totally lost

AU RESPONSE 88: We amended the sentence as follows and linked it to additional text on evolutionary convergence (see Response 19) to clarify our reasoning about robustness of generalisations (lines 164-172):

“Convergences in ecosystem properties are axiomatic to a functionally-based ecosystem typology because they underpin robust generalisations and predictions about ecosystem responses to environmental change and management. Convergences in species traits may arise from common evolutionary origins and niche conservatism [17] [22], but similarities in ecological drivers (selection pressures, assembly filters) may also produce functional convergences in independent lineages. These convergences are the enablers of a functional classification framework represented in the upper three levels of our typology. Functional constraints may be imposed by the species pool, which is a dynamic outcome of vicariance, dispersal and evolution, depending on ecosystem location and biogeographic history [19].”

A classification that explicitly represents differences and similarities in ecosystem function and biodiversity, should contain groupings that are more homogeneous with respect to those features than a classification based on other features. Homogeneous groups are less consistently correct predictors than more heterogeneous ones.

L134. To be adopted in 2020 or has it been adopted?

AU RESPONSE 89: Corrected – was adopted in 2020.

L136what you say about the upper and lower groupings is not understandable

AU RESPONSE 89: We edited the sentence as follows to clarify (lines 175-177):

“Three upper levels (Table S3.1) differentiate functional groupings and three lower levels (see Methods; Appendix S3, pp 19-20) accommodate differences in biotic composition among functionally convergent ecosystems.”

The differences between the upper and lower levels are explained in more detail in Methods (cross-ref on line 176) and Appendix S3.

MINOR COMMENTS ON APPENDIX S2

- P4 fifth par. Define assembly processes. This is crucial, as it is the cornerstone of your theory. Geomorphology and turbulence are not assembly processes. The assembly process is in my opinion dispersal, colonization, establishment, environmental filtering, etc.

AU RESPONSE 90: We added a following sentence to the end of the paragraph 3 on p5, and to the Glossary.

“The assembly processes include the combined action of the abiotic, biotic and dispersal filters described above (Fig. 1).”

Geomorphology and turbulence are part of environmental filtering. They are abiotic filters (i.e. ambient environment, Fig. 1) that influence assembly indirectly by moderating the availability of resources or the ability of organisms to acquire them (e.g. see Dodds’ et al. 2019 biome gradient concept in *Ecosphere*). Geomorphological features, for example, influence the flows and availability of nutrients in oceans and on land, while turbulence influences oxygen availability in aquatic ecosystems, as well as a number of other filtering processes. Dense communities of sessile predators and filter-feeders in submarine canyons that concentrate resources are examples of outcomes of assembly processes driven by geomorphology (see M3.2, Appendix S4). Filter feeders with traits allowing them to hold fast to substrates while capturing resources in swift flows and benthic biofilms are examples of assembly processes driven by turbulence in upland streams (see F1.1, Appendix S4).

- P2 last paragraph. Flooding is not a disturbance

AU RESPONSE 91: See Response 59.

- P4 first par. Define “Ecological traits” (I intuitively consider it a species attribute, rather than an ecosystem attribute.

AU RESPONSE 92: We revised terminology throughout the ms and appendices and define species traits in the Glossary (see Response 62).

- P4 third paragraph. Subterranean systems do have herbivores; nematodes and many other macrofauna browse on plant roots

AU RESPONSE 93: We amended the sentence as follows:

“In contrast, subterranean ecosystems have truncated trophic structure, generally lacking photoautotrophs, vertebrate herbivores and large predators.”

- P4 third par. Define “aphotic sensory mechanisms”. This is really jargon

AU RESPONSE 94: This is a direct quote from the cited reference. We added a phrase to explain:

“... aphotic sensory mechanisms, enabling foraging, reproduction and predator avoidance in darkness.”

COMMENTS ON TROPICAL ECOSYSTEM DESCRIPTIONS (S4)

T1.1 TROPICAL -SUBTROPICAL LOWLAND RAINFORESTS

Ecological traits: What do you mean with “Bottom-up regulatory processes are fuelled by large autochthonous energy sources”?

AU RESPONSE 95: Bottom-up resource supply processes determine productivity (cf. systems such as savannas where top-down herbivore activity, predation are relatively more important). The major source of energy (classified as autochthonous or allochthonous for consistency throughout the typology) is a standard component of all profiles, with terms defined in the Glossary. For all vegetated terrestrial systems autochthonous sources are dominant, whereas freshwater and marine and transitional ecosystems have varied levels of allochthonous input. We include it for completeness across all groups of the biosphere, not because it distinguishes tropical forests from most other terrestrial systems. The terminology is unfortunate technical jargon, but we opted to use it as accurate and standardised descriptors with definitions in the Glossary.

- It seems that you like to mention SLA but that you have not captured the concept, as every time you say that the SLA is high (or low) and it should be the other way around. SLA is the product of (1/leaf thickness) and (1 leaf density). Rainforest trees have thick and dense leaves and therefore a LOW SLA instead of high SLA, as they are adapted to low light or nutrient-poor soils. Low SLA comes along with long-lived persistent leaves. So they can retain the scarce carbon and nutrients (and rain can leach out nutrients less easily from the leaves) for a longer time.

AU RESPONSE 96: The descriptive profiles for all EFGs in the Global Ecosystem Typology are necessarily set in a global context (by definition) to enable meaningful comparisons. We think the reviewer’s remarks are framed on the tropics, rather than a global context. “High” and “low” descriptors of SLA (a species trait relevant to ecosystem function) are set in the context of the global leaf economic spectrum across all ecosystem types. Thus, although there is variation among plant species, the SLA of tropical lowland rainforest trees is accurately described as “high” because it is higher than in other tropical forests (dry, montane & heath forests in biome T1), most temperate and boreal forests (except deciduous forests in T2), shrublands (T3), most savannas (T4), deserts (T5) and cryogenic ecosystems (T6). As an aside, SLA is correlated with, but not calculated from leaf thickness. The correct calculation is given by Referee #4 under T1.3 (leaf mass/leaf area).

- Rainforest trees do not have rapid growth but slow growth (because they have low SLA and dense wood). A high growth potential does not make sense in a resource poor environment (low light, low nutrients). They grow on average 1mm in diameter per year. Only pioneer trees have fast growth.

AU RESPONSE 97: See Response 96. A similar context applies to the generalisation about growth rates and resource availability, which are correlated with SLA.

- Add palms to the life forms

AU RESPONSE 98: Added

Say “by tropical storms SUCH AS near coastal forests”. Large storms also occur in central amazonia where km² of trees might be blown down

AU RESPONSE 99: Added

- PLEASE remove “Many trees exhibit leaf form plasticity on a single individual” ALL plants over the world show a strong plastic response to light, this is not specific for tropical trees and it does not explain their success

AU RESPONSE 100: The full sentence reads,

“Many trees exhibit leaf plasticity enabling photosynthetic function and survival in deep shade, dappled light or full sun, even on a single individual.”

We reduced ‘leaf form plasticity’ to ‘leaf plasticity’ because plasticity extends to (non-visible) anatomical and physiological properties of leaves. However, we retained the sentence because a number of other ecosystem types do not exhibit such a high level of plasticity as tropical forest trees to *enable photosynthetic function and survival* in deep shade, including heath forests (T1.4), boreal/montane conifer forests, both groups of pyrogenic forests (T2.5, T2.6) and all shrublands (T3).

- Please remove “some species germinate on tree trunks” . This is a detail which is not typical for most rainforests and not an important functional response

AU RESPONSE 101: We agree, this is not a common feature in lowland tropical rainforests. However, it is much rarer in almost all other groups of ecosystems, and is exhibited by a major taxon (figs) and a few others. Therefore, we changed “Some” to “A few” and retained the sentence.

- Key ecological drivers. Add that soils range from very fertile, such as volcanic soils, to very infertile, such as on old weathered acidic soils. Because of high rainfall and strong weathering P often limits productivity

AU RESPONSE 102: We added an explanatory phrase as follows:

“Soils are moist but not regularly inundated or peaty (see FT1.3), and vary widely in nutrient status.”

- Conceptual diagram: what do you mean with “limited” competitive release? I guess you mean “strong winds” instead of “high winds”. Why do biotic interactions lead to “bottom-up regulation”. Add to resources “generally nutrient-limited”

AU RESPONSE 103: The referee’s interpretation of ‘limited competitive release’ is correct. Depending on the intensity of the storm, a varied proportion of living biomass and debris remain and this limits competitive release to varying degrees. Unfortunately, there is insufficient space to elaborate this explanation, but the referee’s remark suggests that the label is sufficient to convey the meaning. Bottom-up resource supply processes drive competition and determine productivity in tropical

lowland rainforests more than in systems such as savannas where top-down regulatory processes drive assembly through trophic interactions (herbivore activity, predation). Although nutrient status is varied among ecosystems within this group (Response 102), we do not think it is generally limited relative to most other Ecosystem Functional Groups (see Response 96).

T1.2 TROPICAL SUBTROPICAL DRY FORETS AND THICKETS

- Do you mean that tree and vertebrate diversity higher than most other TEMPERATE forest systems? If so, add temperate.

AU RESPONSE 104: No, we mean all forests, including tropical forest systems such as heath forests, and some montane forests.

- I was highly surprised that you said that trees have typically thin bark and low fire tolerance. Compared to what? Compared to savanna yes, compared to rainforest no. My experience is that trees can have thick barks (e.g., several Bombacaceae, Apocynaceae, Baobabs, especially the larger drought deciduous ones that store water in the stem. Next to that there are many thin barked ones, especially the shrubby plants that can resprout easily. So I would say that bark variation, and hence, fire tolerance varies a lot

AU RESPONSE 105: Fair point – we added “, *unlike many in savannas*” to make the context for comparison clear. This is one of the distinguishing features between tropical dry forests and fire-prone savannas and temperate forests. A number of Bombacaceae, Apocynaceae, Baobabs occur in both dry forest and savannas.

- I am highly surprised that you talk about gap-phase dynamics. Because dry forest have a seasonally deciduous and relatively open canopy and small leaflets, there is a lot of light in the understory. Many plants can recruit and hang on in the understory. Gap phase dynamics are generally NOT important, unless you talk about extremely light demanding species that may need large disturbances such as fires or hurricanes.

AU RESPONSE 106: The text does not say that gap dynamics is the only means of recruitment, but we think it is important relative to other mechanisms in these forests. Recruitment generally does not occur in dry season when the canopy is open, so gap phase dynamics are important, except in the most structurally open forms of dry forest where some recruitment may occur spontaneously. Disturbance-driven recruitment (as in fire-prone systems), is relatively unimportant in dry forests, except where fire incursions drive transition to savanna. We suggest that gap-phase is the most important means of tree recruitment, especially in the many dry forests where the tree canopy is dense when foliated.

- What do you mean with “These forests may be involved in fire-regulated stable state dynamics with savannas”? That is a totally vague and random remark

AU RESPONSE 107: There is quite a large and growing literature on this issue since a seminal paper by William Bond (2005 in J. Veg. Sci.). See several recent articles in <https://www.frontiersin.org/research-topics/6404/revisiting-the-biome-concept-with-a-functional->

lens#articles. The evidence on fire-mediated forest-savanna dynamics is now so abundant that we strengthened the qualification from “may be” to “are often”. We think the term stable-state dynamics is clear. Again, a large literature exists on alternative stable states across terrestrial, freshwater and marine environments (including some on forest-savanna dynamics in the tropics and subtropics), and we cite some key references in the main text and appendices (e.g. Scheffer et al. 2001).

- Maybe mention that many dry forests have a high abundance of nitrogen fixing species (e.g., M. Gei et al. 2018 Nature Ecology & Evolution)

AU RESPONSE 108: Added a phrase as follows:

“Fungi and other microbes are important decomposers of abundant leaf litter and N-fixing plants can be abundant.”

- In what areas are tropical storms important? In the Mexican Yucatan and Caribbean isles?? I can not think of any other dry forest area where this would be important, so I wonder whether this is the exception rather than the rule

AU RESPONSE 109: Yes tropical storms affect dry forests in meso-America, also in southeast Asia, southern Asia and some Indian Ocean islands.

- Conceptual diagram. I would not say that light is limiting. Maybe if you compare it to a savanna yes, but in general not. Say for warm temperatures that they have RELATIVELY low diurnal and seasonal variability. The variability is higher than in tropical rainforests, I think, due to less clouds, and cooling due to open night skies. Why do you say that there are canopy herbivores. In all forest systems there are canopy herbivores. Why do you say the canopy is dense? The canopy is in my opinion relatively open compared to rainforests, and temperate forests. The tree density is higher. I would remove shade tolerance and gap dynamics

AU RESPONSE 110: We amended the diagram to mention the variability in canopy cover. Canopy cover varies seasonally depending on the degree of deciduousness. It also varies spatially with water stress. Open forms of tropical dry forest are quite common in South America, as well as closed canopy forms. The open forms are less common in south and southeast Asia, Australia and southern Africa, because grasses more commonly are present where the canopy opens, and the system takes on characteristics of savanna. Hence, closed canopies (at least seasonally) are more common in those regions. In a global context of all terrestrial ecosystems (see Response 96), and given variation that we now acknowledge, we think shade tolerance and gap dynamics are still relevant. We added ‘relatively’ to qualify temperature variability (though variability is low compared to virtually all other forest groups except T1.1 and T1.4).

- Citations. Maybe cite Sanchez-Azofeifa et al. 2013 book: Tropical dry forests in the Americas: ecology, conservation, and management. I guess there is more ecosystem functioning there than in Toby Penningtons book (apologies!;)

AU RESPONSE 111: Thank you for the suggestion. We added it.

T1.3 TROPICAL-SUBTROPICAL MONTANE RAINFOREST

- It is wrong that montane rainforest have a high specific leaf area (SLA). SLA is leaf area divided by leaf mass. Montane species have thick, dense, leathery or coriaceous leaves to retain scarce nutrients and reduce damage by UV radiation. I am flabbergasted that the expert authors have overseen this

AU RESPONSE 112: As in Response 96, we think this comment is made in the context of other tropical forests, rather than a global context including all terrestrial ecosystems. We acknowledge that leaves of many trees in T1.3 are coriaceous, more so than in lowland rainforest and tropical dry forest, however SLA in T1.3 is higher than other biomes listed in Response 96. To account for the fact that a couple of terrestrial EFGs typically have higher SLA (T1.1, T1.2, T2.2), we qualified the descriptor, “*moderate-high SLA*”.

- Why is elfin woodland not a separate functional ecosystem?? They have a totally different structure and functioning compared to cloudforests!!! I would make it a separate category

AU RESPONSE 113: Yes, the working group considered this option carefully, recognising that a continuum exists from lowland rainforests to elfin forests. In an early draft of the typology, the circumscription of T1.3 was framed more narrowly around elfin forms (terminology after Whittaker 1975). However, further input from experts pointed to the need for a broader circumscription to encompass relevant variation in ecosystem properties related to diminishing temperatures, increasing cloud moisture and UV-B, relative to lowland rainforests (T1.1). Representation of ecological continua in classifications involves inevitable trade-offs and differences of opinion (see Appendix S3 for discussion of that issue). That said, with additional work we think there could be a case to subdivide tropical montane forests into elfin and non-elfin groups in a future version of the typology.

- What do you mean with “productivity is fueled by autochthonous energy??” That plants are autotrophic and photosynthesizing? If so say so, but this is an odd comment. An elevational transect through the Andes found that productivity is mainly limited by radiation (Fyllas et al. 2017 EcolLett Solar radiation and functional traits explain the decline of forest primary productivity along a tropical elevation gradient)

AU RESPONSE 114: The major source of energy (classified as autochthonous or allochthonous for consistency throughout the typology) is a standard component of all profiles (see response 96). Thank you for the reference. We re-ordered the relevant phrase as follows:

“Productivity... limited by high exposure to UV-B radiation, possibly also by cool temperatures, and sometimes by shallow soil and/or wind exposure.”

- Say that taxonomic TREE diversity is low, and add that there is a very high diversity of epiphytes (orchids, bromeliads, lichens, mosses, ferns)

AU RESPONSE 115: We altered the sentence as follows:

“Tree diversity is moderate to low, while epiphytes are diverse, but there is often high local endemism at higher altitudes in most groups, especially amphibians, birds, plants, and invertebrates.”

- Add that gap-phase dynamics are also driven by landslides (often driven by geologically young substrates, steep terrain, high rainfall and waterlogged soils)

AU RESPONSE 116: Thank you for the suggestion, we added landslides.

- Key ecological drivers. Say something that in general montane systems are especially limited by soil N? (Grubb 1977 Control of forest growth and distribution on wet tropical mountains: with special reference to mineral nutrition). Or has this proven to be wrong?

AU RESPONSE 117: Grubb identifies ‘air temperature and the radiation climate’ as main limiting factors for tropical montane forests (as noted in our text for T1.3), but infers there is ‘some evidence’ that plants in montane forests are adapted to low N and P because photosynthates are in short supply and large quantities of N and P are locked up in litter and humus. We acknowledge this secondary point and revised the text as follows:

“Moderate productivity fuelled by autochthonous energy is limited by high exposure to UV-B radiation, cool temperatures, and sometimes by shallow soil or wind exposure. Limited energy and sequestration in humic soils may limit N and P uptake.”

- Conceptual diagram. Why do you say that landslides lead to competitive release?? All trees are gone, so nobody is released. I can imagine that storms lead to competitive release. Plant competition is in nearly all closed vegetation cover systems of the world, so why do you mention it here? Because it is relatively shaded? Why do you say “bottom up regulation”? No clue what you are referring to. I disagree that diversity is moderate to low. It is moderate to low tree diversity but high epiphyte diversity. Add to “Abundant bryophytes” also “epiphytes and lichens”. Add also nutrients to your ambient environment, as it co-limits productivity?

AU RESPONSE 118: The recruits that emerge after landslides experience less competition from established neighbours relative to recruits in undisturbed forest. See Response 95 for bottom-up regulation and Response 96 for shade effects. The text states that endemism is high, but diversity is moderate to low (relative to other forests globally, but especially T1.1 tropical rainforests). This generalisation is robust for multiple taxonomic groups (plants, amphibians, invertebrates). Epiphytes are most notable for their abundance, but in some cases also their diversity. Unfortunately there was insufficient space on the diagram to include epiphytes and lichens as well as bryophytes – those are mentioned in the text. We added a resources box to the diagram to mention nutrient limitation.

T1.4 TROPICAL HEATH FOREST

- For a global classification I would NOT use the word “heath forest”. This comes traditionally from Asia (Kerangas forest) and is only used there. The term “Heidewald” was introduced by Winkler (1914). It gives to me associations with Ericaceae shrubland, which it is not. It is a forest on siliceous,

acid, nutrient poor soil, so I think “white sand forest” is more clear and appropriate in the spirit of your FUNCTIONAL classification, and it is also known as such in the Neotropics.

AU RESPONSE 119: Thank you for your thoughts on this. The working group considered the label for this group at some length, including the option of “white sand forest”. No term was considered ideal, and virtually all have regional usage (with “white sand forest” terminology originating in, and still mostly confined to South America). Although the best known expressions of these ecosystems are recorded on white sands, some reports mention other substrates (e.g. Adeney et al. 2016, Biotropica). Therefore we wanted to avoid implications that these forests occur only on white sands. More generally, we have avoided associations with substrate types in labels of Ecosystem Functional Groups. Although heath forests may originate from Bornean Kerangas, we reasoned that this term had been applied more widely (e.g. by Whittaker 1975), and that it conveyed an important functional feature – small leaf size related to nutritional poverty. Globally, “Heath” is applied more widely to sclerophyllous shrublands with small leaf sizes associated with low-nutrient soils (i.e. well beyond the strict taxonomic interpretation applied in a few regions). Although these ‘heath forests’ generally have larger leaves than heathland shrubs, we think the analogy drawn by Whittaker and others is reasonable.

- Ecological traits. I would add to the first sentence that it has a high density of thin stems. Also add that there are many plants and animals with special adaptations to low nutrients (pitcher plants), and that there are special mutualisms (abundant myrmecophytes). Richards mentions as feature a tendency towards dominance

AU RESPONSE 120: We added “*high density of thin stems*” to the first sentence (though a more detailed mention is made later in the text). We added “*ant mutualisms*” to the sentence on plant insectivory and N-fixing microbial associations.

- Drivers: Proctor suggests that aluminium toxicity plays a role. I do not know whether that has been accepted or rejected. Please check and add it to drivers and diagram if needed

AU RESPONSE 121: Aluminium toxicity is associated with acid soils in various ecosystems. We added it to the text and diagram.

- Conceptual diagram: sandy soils and shallow rooting lead not only to flooding but also to a water shortage in the dry season. So I would put an arrow from sandy substrate to resources, and not only put under resources water surplus but also shortage. In ecological traits you say “low diversity”. Low compared to what? Temperate forests? Tropical rainforest? I believe there are many species there. I would put high dominance as an ecological trait. Rather than microphyll I would say that leaves are leathery (that better reflects the nutrient limitation and functioning)

AU RESPONSE 122: We added seasonal surplus & deficit to the Water box and an arrow from sandy substrates to the Water box. Adeney et al. (2016) say, ‘[compared to surrounding forests]... white-sand ecosystems are species poor, dominated by a few woody species.’ Based on data and descriptions in cited references, we think ‘relatively low diversity, but high endemism’ also holds for comparisons with a number of temperate forest types and savannas. We mentioned high dominance and leathery leaves in the text, but there is insufficient space for them on the diagram.

- Distribution: P.W. Richards classifies Caatinga as heath forest. But you did not include it here on your map? Why not? You say that heath forest is not known from Africa, but Peace & MacDonald (Biotropica 1981) mention that it also occurs in Gabon (I do not know from where they have that wisdom)

AU RESPONSE 123: There is little white sand in the Caatinga. Subsequent to Richards' book, most authors classify most of the Caatinga as a type of tropical dry forest, with smaller areas of tropical heathland (T3.1). Thank you for the further information on Africa from Peace & MacDonald (1981). We changed the Distribution text as follows:

"Poorly known in Africa, but possibly in the Gabon region."

- References: they feel a bit detailed and random. Do you expect the readers to get an overview of the system based on these? Maybe better cite PW Richards 1996. And I must imagine that there is a better book or review article on these forests

AU RESPONSE 124: We agree there is a need for a modern global synthesis on heath forests. In its absence, we identified a selection of studies on different aspects from the two best known regions of occurrence. While Richards (1996) is a remarkable work, it is very general and now somewhat dated. We cite Adeney et al. (2016), who provide an overview of the Americas with some reference to Asia, as well as a more detailed contribution on southeast Asia.

T4.1 Trophic savannas.

- I am not a savanna ecologist, but have travelled through the systems.

- Ecological traits: I would add that the mammals can represent a large biomass, and that mammals show rotational grazing. And that in East Africa (and South Africa) mammals show strong migratory patterns aligned with the rainfall and nutrient needs when lactating. I am not sure whether this applies to West Africa. What do you mean with "sustaining the system through positive feedbacks and limiting fire fuels"? Why do you add "Nitrogen fixation, recycling, and deposition by animals exceeds volatilisation"? Why is nitrogen volatilisation a problem when there is little fire?

AU RESPONSE 125: We added *"These animals account for high biomass..."* and *"...kept short by vertebrate grazers that migrate in some regions."* Feedbacks operate as heavy grazing pressure promotes shortgrass over less nutritious tallgrass, promoting herbivore activity. The cited reference by Hempson et al. (2015) explain the feedbacks in detail. We do not identify N volatilisation as a 'problem', only that it is exceeded by N fixation and deposition in trophic savannas. The reverse is true in pyric savannas (see Archibald & Hempson 2016). We also note in the text that fire is part of trophic savanna ecosystems, but not the major driver.

- Key ecological drivers. You say that low intensity fires have return intervals of 5-50 years. I guess you refer here to natural fires? I think that in West Africa fire return interval is each year as people burn it regularly for hunting or to get rid of snakes etc., and to renew the grasses for their cattle

AU RESPONSE 126: Estimated return intervals are from Archibald & Hemson (2016). They base their inference on satellite data and life histories. Some sites may be burnt at annual intervals, but we think their data support a defensible generalisation about fire frequency.

- Distribution: In the map you classify the west African Sahel zone as trophic savanna. Maybe originally, but I guess now most animals are hunted and little is left. The same applies to the Indian subcontinent. At some places you can find still intact communities. Maybe the West African savannas are now de-facto pyrogenic savannas? In the description be explicit that “Asia” is the Indian subcontinent

AU RESPONSE 127: The map is based on Archibald & Hemson (2016), who modelled the distributions of both trophic and pyric savannas. Trophic savannas are not confined to India in Asia, they are also in Myanmar and possibly other countries.

- Conceptual diagram. Ecological traits: mention high mammal abundance. Biotic interactions: It is not clear what you mean with “engineers (+ve feedbacks)” and “strong and weak top down processes” Maybe use “Strong or weak herbivore control”?

AU RESPONSE 128: We added “Abundant megafauna” and replaced “Engineers” with “Facilitation” referring to positive feedbacks (see Response 125). Bottom-up and top-down regulation are widely used terms in ecology texts to describe the relative influence of primary producers and consumers. We added them to the Glossary.

T4.1 PYRIC TUSSOCK SAVANNAS

- Ecological traits: what do you mean with “grasses cure in winter” and “local endemism is low across all taxa”?

AU RESPONSE 129: Grass curing refers to the drying of grass and its increasing availability to burn due to its declining moisture content in the dry winter season. Drying is listed as a synonym of curing in Word thesaurus and most dictionaries, but we prefer curing in this context, consistent with usage by grassland and fire ecologists. Many savanna species have large distributions, hence there is low local endemism in the system, compared to systems where we described local endemism as high (e.g. tropical montane forests, heath forests).

- You highlight deciduousness, but some savannas are dominated by evergreen species (Australia) and others by deciduous species (South America).

AU RESPONSE 130: We qualified “Plant traits such as deciduous leaf phenology...”, adding “Common plant traits such as...”. Australian savannas in this group include a range of deciduous taxa, and some of the eucalypts are semi-deciduous, except in the wettest areas such as Bathurst Island.

- Shouldn't you mention the strong variation in tree cover (in the Brazilian cerrado you have at least campo sujo, cerrado, and cerradao) and the importance of gallery forests?

AU RESPONSE 131: We mentioned "*variable tree cover*" in the first sentence of the description. We now say "*highly variable...*". Gallery forests are rainforests (T1.1) or dry forests (T1.2).

- You say that plant defences against herbivores such as spinescence are less prominent. Then be explicit that you mention about MAMALLIAN herbivores, as insect herbivory can be high, and at least in the cerrado many plants defend themselves with high silicate concentrations.

AU RESPONSE 132: Added "*vertebrate herbivores*", although reference to mammals in preceding sentence and spinescence in this sentence should be clear.

- When you talk about detritivores, do you refer about termites? If so, say so.

AU RESPONSE 133: Termites are conspicuous but they are not the only invertebrate detritivores in the system, so we retained the more general term.

- In the cerrado the resouters and reseeders are important; there is a high diversity because of the fire regime and the reseeding plants. Maybe worth mentioning? I do not know whether that applies to other savannas.

AU RESPONSE 134: There is much variation between and within regions. Some of these savannas (e.g. southeast Africa, northern Australia) have very few reseeders. It may not be a strongly characteristic feature of this savanna type.

- Why do you say that plants have a high SLA? High for the deciduous species, but low for the evergreen species!

AU RESPONSE 135: We added a qualifier as follows: "*moderate-high SLA (low in some evergreens)...*"

- Maybe mention that belowground carbohydrate storage is important for resprouting success after fire, and many species have massive belowground storage organs?

AU RESPONSE 136: We amended the sentence to note storage, "*...fire-resistant storage organs (e.g. below-ground bud banks, thick bark).*"

Referee #5 (Remarks to the Author):

This paper represents a major step forward in a unified understanding and classification of the natural world in a way that can guide management and protection at very large scales. It directly

addresses a common criticism of environmental protection: regulation and management is mostly based upon conservation of particular species or specific human concerns such as pollutants entering water or air. The authors' approach provides an important step toward conservation or restoration of ecosystems using a more holistic approach. Global classification of biotic (biodiversity as well as ecosystem function) patterns is the most important issue with respect to the biodiversity crisis and maintaining ecosystem services vital to humanity.

A criticism was in the prior reviews that the classification was not perfect for several specific reasons. The fact is that humans like to classify things as a way to deal with them conceptually, and it is necessary to do things like write laws and treaties. This creates problems similar to those seen in ecology with biome classification. In the real world biomes grade into each other and there are not clean lines. For example, if you go to the Cerrado in Brazil, the uplands are savannah and if you go to the riparian zones, the vegetation is more like the Atlantic Rainforest to the east. As long as the authors explicitly note this reality in the caveats to the general framework, I do not think that criticisms of classification are too much of a worry. It is just a tool for humans to deal with the world, and we only need to be aware of the limitations of that tool.

AU RESPONSE 137: We thank the reviewer for these insightful comments. Our discussion under 'Discrete representation of continuous patterns in nature' at the conclusion of Appendix S3 very much aligned with these comments. To acknowledge this limitation more explicitly, we added the following paragraph to the main text before the concluding paragraph (lines 267-275):

"We acknowledge the limitations associated with discrete representation of continuous ecological patterns in nature (Appendix S3, p23). Even though our descriptive framework recognises core and transitional units, its discrete structure generates boundary uncertainties among ecosystems that are ultimately unavoidable, even with extensive splitting of classes [48]. However, this fallibility is outweighed by a classificatory approach founded in deep-seated cognitive processes that govern how humans understand and manage environmental, social economic and cultural dimensions of their conscious universe by dividing it into parts [49]. This will facilitate widespread uptake of the IUCN typology for effective storage, retrieval and transfer of ecosystem information."

The manuscript and methodology behind it have gone through extensive external review, and the authors have made a strong effort to respond to those reviews. Appendix s5 on development, review, and revision was a pillar of this work. I think that this is reflective of a developing system that will be continuously revised and refined in the future (sort of like the IUCN red list when new data, taxonomic information, or human caused changes become available or occur). Thus, I do not view this paper as a static result as much as a milestone report on an ongoing journey. I think a parallel is the publication of the human genome. It was not complete, and continues to be refined, but it was a large step forward.

AU RESPONSE 138: We emphasise the ongoing development of the typology into the future, with the addition of the following sentence (lines 225-227):

"We expect progressive improvements in future versions of the IUCN Global Ecosystem Typology as understanding develops. Several aspects of the typology warrant further development to address uncertainties."

Subsequent text identifies several specific aspects of the typology for further development. One of these is mapping - our framework provides a much needed target for mapping of ecosystems, which is where there will be considerable improvement in the future.

The functional diagram figure 1 should include seasonality? The methods do mention seasonality, but the key conceptual diagram only mentions it indirectly under climate change. Temporal factors are a missing major axis in classification that is ultimately included in the fine details, but not indicated at the top conceptual level.

AU RESPONSE 139: Specific models for several Ecosystem Functional Groups in the descriptive profiles (in Appendix S4) include seasonal variation, but Fig. 1 in the earlier version of our ms did not. We revised the figure to include 'Climate seasonality' in the Abiotic environment box in Fig. 1. Thank you for pointing out the omission.

The inclusion of subterranean habitats particularly impressed me. This highly underappreciated area needs protection, as well as representing an ecosystem that influences global biogeochemistry, linkages between land and aquatic habitats, and can be a hotspot of unique biodiversity. I am not aware of any efforts to make such classifications and determinations and applaud the authors for including this habitat.

Thank you

Some previous efforts have justified this manuscript. Whittaker (1970) stated biomes could be developed for aquatic systems based on physiognomy but noted that aquatic communities intergrade with each other in different ways and are less dependent on global climatic gradients than are terrestrial habitats (presumably because water is less limiting). My own efforts related to this paper might assist in the revision (Dodds et al. 2015, 2019). In these papers, we discuss the ideas of translating the terrestrial concept of biome to freshwaters. The Dodds et al. (2019) paper specifically discusses the fact that ecosystem characteristics, in addition to phylogenetic information, informs biomes. The author's manuscript acknowledges this view. Many of our predictions or reporting of patterns hinges upon the observation that the line between actual and potential evapotranspiration is a key factor in determination of fundamental characteristics of intermittency of freshwater habitats.

This is a key factor in all freshwater habitats (in addition to terrestrial biomes).

AU RESPONSE 140: Thank you for alerting us to these important references. They are highly relevant to several components of our approach, and we have added citations in the main text (lines 181190):

"Level 3 units of the typology (Ecosystem Functional Groups described in Appendix S4, pp36-168) are fundamental to generalisations and predictions about ecosystems with similar functional properties, and therefore have key roles in global synthesis and knowledge transfer for ecosystems. Their distribution across landscapes and seascapes (Fig. 2) is governed by expression of ecological drivers along temporally variable multidimensional gradients [23] [24]. Interactions between the drivers that operate at different spatial scales in this multidimensional space determine the dominant filters and evolutionary pressures that shape ecosystem properties in

different parts of the biosphere (see Methods on hierarchical levels and Appendix S3 for key drivers that differentiate Ecosystem Functional Groups along landscape and seascape gradients visualised in Fig. 2)."

We amended the Methods as follows:

Lines 386-389:

"Biome gradient concepts [45] highlight continuous variation in ecosystem properties, which is represented in the typology by transitional realms that mark the interfaces between the five core realms (e.g. floodplains (terrestrial/freshwater), estuaries (freshwater/marine))."

Lines 396-399:

"Our interpretation aligns broadly with 'functional biomes' described elsewhere [23] [52] [24], extended here to reflect dominant assembly filters and processes across all realms, rather than the more restricted basis of climate-vegetation relationships that traditionally underpin biome definition on land."

Lines 411-414:

"Resource gradients defined by flow regimes (hence catchment precipitation and evapotranspiration) and water chemistry, moderated by environmental gradients in temperature and geomorphology, differentiate functional groups of freshwater ecosystems [45]."

While Figures 1 and 2 help understand the classification approach, a conceptual figure on the approach for assigning pressures would be useful for the naive reader (myself included), as they appear to be a combination of two independent indices of pressure. Figure 2 is somewhat nebulous, but it is trying to encapsulate a very complex process, and I am not sure how to better do that. The earlier suggestion of a detailed supplementary table goes a good way toward that.

AU RESPONSE 141: We made several minor amendments to improve Fig. 1 and its caption in response to suggestions by other reviewers (see Responses 19, 20, 48, 53, 56, 90, 139). We replaced Fig. 2 with a new Table 1, which lists all units in the top three levels of the typology with brief descriptive details and shows the explicit hierarchical relationships among them. We also introduced a new Figure 2 showing idealised ecological relationships among major biomes in a stylised landscape/seascape. The schematic representation of the hierarchy is now in Appendix S3. Finally we deleted the analysis of pressures from the manuscript as recommended by the editor (see Response 1), hence the indices of pressure are no longer relevant.

Some minor comments:

This statement in the abstract

"Applying this typology, we find that the most degraded and least protected ecosystems on Earth include 18 of 85 terrestrial, freshwater and marine Ecosystem Functional Groups with >70% of extent exposed to high pressures and <17%, or <10% for marine, represented in protected areas."

Is a bit convoluted and could be made more strongly.

How about one fifth of Earths ecosystems are severely degraded and not protected with less than 70% of freshwater and marine areas protected

AU RESPONSE 142: Thank you for the suggestions, however, we removed the spatial analyses of pressures from the manuscript and Appendix S6 as requested by the editor (see Response 1).

If there is room maybe talk about some of the rarest or most endangered habitats that clearly need the most protection? Sort of like biodiversity hot-spots in need of conservation?

AU RESPONSE 143: We will pursue this in a separate publication analysing pressures on different ecosystem types (see Response 1), enabling us to elaborate on the rarest and most threatened ecosystem types in need of conservation.

Getting in the weeds a little, river regulation is a major impact, and I could not see any mention of dams or reservoirs.

AU RESPONSE 144: See Response 143

Line 70 define IUCN

AU RESPONSE 145: Now spelt out in the first paragraph:

“the International Union for the Conservation of Nature (IUCN)”

There is a link to the additional information on zenodo, and it needs updating. I got to the most recent version from there, but does not directly link.

AU RESPONSE 146: We have updated the link to the most recent version of the spatial data [<https://zenodo.org/record/4018314#.YOQRr-gzY2w>]. We update this archive as new data sets become available (see Response 138). All superseded versions in the archive are linked to the current version.

Dodds, W. K., L. Bruckerhoff, D. Batzer, A. Schechner, C. Pennock, E. Renner, F. Tromboni, K. Bigham, and S. Grieger. 2019. The freshwater biome gradient framework: predicting macroscale properties based on latitude, altitude, and precipitation. *Ecosphere* 10:e02786.

Dodds, W. K., K. Gido, M. R. Whiles, M. D. Daniels, and B. P. Grudzinski. 2015. The Stream Biome Gradient Concept: factors controlling lotic systems across broad biogeographic scales. *Freshwater Science* 34:1-19.

Whittaker, R. H. 1970. Communities and ecosystems.

Referee #6

The paper presents a new, hierarchical, and largely function-based typology of the world's ecosystems. It presents several analyses using this typology as an organizational framework as examples of globally important applications that can benefit from it. I was asked to comment specifically on the rigor of the human-modified component of the typology and the results presented. I will first present an overview of my assessment, and then expand on these points in more detail, as well as providing more specific assessments for the individual humandominated ecosystems.

I must say that this was probably the most challenging, and also the longest, review I have ever written. To really understand the things presented to me (which I hope I achieved), I ended up engaging quite intensively with all appendices as well as the original description of the IUCN ecosystem typology. I hope my comments are helpful to the authors (in particular).

Overview of my assessment

By and large, the typology is neat. I commend the authors on the massive work underpinning its development. There are, however, several significant flaws in the human-modified components that unfortunately prohibit me from given an overall positive assessment. Still, I generally consider its development a tremendous scientific achievement. For that reason, I would even potentially consider it worthy of publication in a high-impact multidisciplinary journal, although I do share an earlier reviewer's opinion that this journal is a somewhat odd choice for presenting it, given that other venues would offer so much more space to present this massive compilation of information and arguments.

There is a tremendous quality gap between the typology itself and the presented data and analyses. Unfortunately, a central argument of this paper is that this typology, specifically in combination with spatial data on ecosystem functional types (EFGs), can support detailed global analyses and therefore has great utility for tracking progress against global policy targets and other important applications. I am afraid that the flaws in the data and the presented applications undermine these claims. The paper provides evidence on the soundness of the typology itself, but it does not provide strong evidence for the conclusions from the exemplary analyses. I believe that the typology requires substantial alterations of, and expansions on, some of its human components. The spatial data of "indicative" global distributions of EFGs (an intermediate level in the presented hierarchy), which are presented as being associated with this typology, are in some cases very well chosen, but in many other cases these data are extremely crude, and the underlying data have in many cases very large and generally well-known uncertainties, that unfortunately are largely ignored here. In several cases, the chosen source data are conceptually inadequate for representing the respective EFG. The limited reliability of the presented indicative maps is verbally acknowledged through their presentation as "indicative", but there is generally no robust validation of even these indicative abilities, which is indeed questionable in some cases. Moreover, this verbal acknowledgement is undermined by using them for a set of spatially explicit global analyses for which they are not fit-for-purpose, which might even mislead other scientists or decision-makers about the quality of these data. By failing to account for these limitations, the presented analyses using this typology and associated data have severe methodological flaws, and the evidence provided is not strong enough to make the results trustworthy, which is the main reason why I do not provide an overall more positive assessment.

AU RESPONSE 147: We accept that a proportion of the maps are not suitable quality to support a spatial analysis and have deleted the analysis from the ms as suggested by the Editor. We now acknowledge this limitation more explicitly in the main text (Lines 240-253):

"Thirdly, the indicative global maps for ecosystem functional groups vary substantially in accuracy and precision. High resolution global maps of ecosystem functional groups are pivotal to

important applications of our typology (e.g. global synthesis for reporting on CBD targets). Many other uses, however, are national in scope, specific to particular ecosystem groups or biomes (e.g. forests, coral reefs, mangroves) or non-spatial (i.e. using the typology to frame context for knowledge transfer and generalisation), and thus do not require comprehensive and globally consistent maps. Maps that are most fit for purpose would be based on remote sensing and environmental predictors that align closely to the concept of their ecosystem functional group, incorporate spatially explicit ground observations and have low rates of omission and commission errors, high spatial resolution and time series of changes. While many current maps fall below one or more of these specifications, maps available for two-thirds of the 108 functional groups are of a high-intermediate quality suitable for global spatial analysis (Appendix S4, pp14-22). These represent an advance on global ecosystem distributions relative to proxies such as ecoregions, and new data sets are rapidly emerging.”

During this round of review for Nature, newly published studies and critical input enabled updates to maps for eight of the EFGs. More broadly, we draw attention to an important aspect of our approach (Lines 254-256):

“By decoupling the mapping process from prior development of the classification, our approach liberates the definition of ecosystem units from constraints imposed by current availability of spatial data and allows for progressive improvement (Appendix S4, p15).”

We removed all downstream analyses and focussed the revised manuscript on the typology itself (see Response 1). We note comments which are now not applicable in our response below.

Given the limitations in data that is available today, I am not convinced that this typology can indeed, at this point in time, support the claimed applications like global monitoring of conservation targets or national ecosystem accounting.

AU RESPONSE 148: These limitations apply to some of the map data, not the typology. The typology has already begun to support several applications as noted in Appendix S6, including a number of applications that do not depend on comprehensive coverage of high quality global maps. We discuss this in revised discussion on lines 241-246:

“High resolution global maps of ecosystem functional groups are pivotal to important applications of our typology (e.g. global synthesis for reporting on CBD targets). Many other uses, however, are national in scope, specific to particular ecosystem groups or biomes (e.g. forests, coral reefs, mangroves) or non-spatial (i.e. using the typology to frame context for knowledge transfer and generalisation), and thus do not require comprehensive and globally consistent maps.”

Following adoption by the IUCN in 2020, the typology was also adopted by the United Nations Statistics Commission in 2021 as the reference classification for ecosystem accounts in the UN System for Environmental Accounting – Ecosystem Accounts (UN SEEA-EA, noting that existing national maps can readily be related to the global typology for international reporting of national accounts (see Appendix S6). We recognise the need to improve globally consistent map data as soon as possible, and are progressively doing so – six maps were updated during the journal review process since the previous submission to Nature in October 2020.

I would recommend the authors to more carefully think through several of the descriptions of humandominated EFGs, and to only associate those EFGs with spatial data where the claim that

those indicate their global distributions has been confirmed through some kind of formal validation. In addition, I would recommend to do either of the following options:

- 1) sell the typology as a stand-alone scientific achievement without associated analyses,
- 2) restrict the analyses to validated data that are fit-for-purpose for these analyses, which would likely mean restricting them to only some EFGs and maybe only some regions of the world, or
- 3) invest in substantially more rigorous global analyses that appropriately consider the limitations in the used global data, and provide concrete evidence (sensitivity analyses, formal validation, or similar) that the presented results are robust to their limitations.

AU RESPONSE 149: Thank you for the suggested options, we revised the ms according to Option 1 (see Response 1). We address comments on anthropogenic ecosystems below.

Originality and significance

The typology is novel and generally well-conceived. I would consider the “natural” components of it, and at least some of the human-dominated components, both important and highly useful for ecologists, conservation biologists and some closely related fields. The paper will be mainly of interest to these audiences. I do not think the typology will be of substantial interest to neighboring where a conceptually sound and consistent classification of human-used systems is more crucial, such as land use science, agricultural economics, urban studies, forestry, agricultural science, or similar. The typology is strictly ecologically focused, which is fine for purely ecological applications but will make it less useful for interdisciplinary discourses in a broader sustainable development context, given that the boundary concept of “ecosystems” is used very differently across fields (such as setting of integrative targets for advancing SDGs, etc.). To be more usable to researchers accustomed to diverse other typologies, crosswalks like those in tables S3.3 and S3.4 would ideally be presented for all the typologies critiqued here.

AU RESPONSE 150: We agree that the IUCN Global Ecosystem Typology may not suit all purposes focussed on specific land uses, such as agricultural economics, urban studies, etc. These were not part of the design objectives, and would (as Referee #6 points out) require a different approach. The Introductory paragraphs of the main text and Appendix S1 endeavour to make it clear that the primary purpose of this typology is to support reporting, knowledge transfer and decision making for dual imperatives: 1) biodiversity conservation; and 2) sustainability of ecosystem services. These stem from the UN Convention on Biological Diversity, focussing more narrowly on biodiversity than the UN SDGs, which encompass environmental, cultural, social and economic dimensions.

The focus of our typology on ecological processes and ecosystem functions sets it apart from land use classifications (and also from biogeographic classifications such as ecoregions), which are designed to be fit for different purposes. We think it is appropriate that different tools are designed to serve contrasting purposes well, rather than a one-size-fits all classification that is suboptimal for some or most purposes because of trade-offs in the design objectives. We added the following discussion to Appendix S1 (p6) to elaborate on the scope of the ecosystem typology, as it relates to land use classification:

“We also draw a distinction between ecosystem typologies and land use classifications. Given fundamentally different purposes and trade-offs in design principles, the IUCN Global Ecosystem Typology will not be fit for all purposes that a land use classification can serve, and vice versa. Therefore we did not include land use classifications in our review of ecological typologies because they are classes of human economic, social and cultural activity that reflect the types and intensity of interactions between humans and their environment (Erb et al. 2017; Mayfroidt et al. 2018). Land use classifications typically group systems with low land use intensity into broad land cover categories based on plant life form, cover, height, and micropattern (Di Gregorio & Jansen 2000); attributes that are, at best, indirect proxies for ecosystem function. They classify higher intensity land uses on the basis of different criteria, for example, cultivated area are partitioned

by plant growth form, field size and spatial distribution, crop combination and cover-related cultural practices (Di Gregorio & Jansen 2000). Other land use classification and mapping approaches estimate the intensity of different land use types (e.g. Erb et al. 2007).

In a broad sense, the anthropogenic components of the IUCN Global Ecosystem Typology address similar themes to some land use classifications. For example, land use intensity mapping by Erb et al. (2007) corresponds broadly to five anthropogenic ecosystem functional groups in the terrestrial biome of the IUCN typology: cropping (T7.1 Annual croplands); grazing (T7.2 Sown pastures and fields, T7.5 Derived semi-natural pastures and old fields); forestry (T7.3 Perennial crops and plantations); and infrastructure (T7.4 Urban and industrial systems). However, anthropogenic ecosystem types (defined as those that are created and sustained by intensive human activities, see Glossary, Appendix S4) should not be confused with land use activities. For example, grazing and forestry activities occur at a wide range of intensities across natural systems, but in anthropogenic ecosystems (T7.2, T7.3), they occur at transformative intensities, associated with qualitatively different ecosystem properties and organisational processes that may not persist when the activity ceases.”

That said, we considered it crucial to encompass anthropogenic ecosystems (cf. land uses) in the typology, as these are now an important component of the biosphere, and relevant to the underlying goals of the CBD. We consulted with FAO land use classification specialists in the development of the descriptions of these units (UN SEEA Expert forum June 2020, Appendix S5, p2). Level 3 of the Global Ecosystem Typology includes 16 Ecosystem Functional Groups within six biomes (Level 2), a comparable but slightly greater level of detail to the global FAO Land Cover Classification System, which recognises eight units in the third level of its Dichotomous Phase, including four human-dominated units (Di Gregorio & Jansen 2000). Similarly, as noted in Appendix S1 text above, Erb et al. (2007) map the intensity of four land activities types that correspond broadly to five anthropogenic ecosystem types in the terrestrial realm.

Descriptions of Ecosystem Functional Groups recognise variability within them. For example, the description of T7.4 Urban systems (Appendix S4, p80) notes that this functional group includes a number of elements:

“These elements include: a) buildings; b) paved surfaces; c) transport infrastructure; d) treed areas; e) grassed areas; f) gardens; g) mines or quarries; h) bare ground; and i) refuse areas.”

Although it is in large parts beautifully conceived, I am not sure if the typology can be of immediate, practical relevance to many scholars even in ecology, because the majority of ecosystems are thus far not directly mapped, that is, as remote sensing based or other validated maps at resolutions where ecologists typically need to work, and given that for many of the EFGs, the discussed options for producing such maps are not overly convincing (see below). This fact alone makes me wonder how many scientists will actually adopt it. I do not consider this a limitation to its long-term utility, as it is still helpful to present a sound typology now that may guide future data collection. But this will likely limit its short-term impact. Except for few cases where temporal data are already available, it simply cannot be readily adopted for monitoring at this point (one of the main application fields claimed).

In this respect, I think there needs to be a more honest representation of the “spatially explicit” character of the typology. And even then, for all the anticipated temporal uses of the typology (ecosystem accounting, monitoring CBD targets), spatially explicit simply would not be enough. I am missing a nuanced statement in the discussion on what can currently, can maybe at some point, and maybe can never be mapped as a timeseries using deep time satellite images. The authors seem to

expect that most EFGs can be mapped globally and annually. Reading the sentence in the discussion on new AI technologies and deep time satellite archives, I expected several examples, but apparently, there are only two (tidal mudflats and glacial lakes; the habitat map does not look like one).

AU RESPONSE 151: The typology was developed to support a broad range of uses related to ecosystem management and research (see Appendix S6). Although maps are essential to some major applications (e.g. global CBD reporting), many uses are national in scope, specific to particular ecosystem groups or biomes (e.g. forests, coral reefs, mangroves) or non-spatial (i.e. using the typology to frame context for knowledge transfer and generalisation), and thus do not require a full set of globally consistent maps. High-quality, high-resolution global maps currently exist for a quarter of the (27 of 108) Ecosystem Functional Groups, maps of intermediate quality exist for a further 45% (48 of 108) EFGs, while the remaining 30% (33 of 108) of EFGs rely on low-quality indirect proxies (see Table S4.1 and preceding text in Appendix S4). Most maps in the first two categories are validated and peer-reviewed, and thus suitable for monitoring, and some are supporting applications which the typology can place in a global context of other ecosystems (e.g. Global Mangrove Watch).

Of those maps suitable for monitoring, a smaller number are advanced data cubes based on extended satellite time series. We give two contrasting examples of these in the main text (tidal mudflats and glacial lake), but others include several groups of lakes, sea ice, non-perennial rivers and streams, coral reefs, mangroves... (i.e. there are quite a few more than two, and the number is growing at a rate that surprises even us). Several new maps that have become available while our ms is in review have been incorporated into our archive (doi: 10.5281/zenodo.4018173). In further work, we are preparing a separate publication with a detailed review of map standards suitable for different applications.

Further, a number of 'bottom-up' subglobal maps (e.g. national vegetation maps) have already been linked to the global typology by local experts (see examples in Table S3.3 and S3.4; others include South Africa, Canada, USA, Finland, India, Australian states), enabling a range of fine-resolution spatial conservation planning applications in an international context. We elaborated the text on strengths and limitations of maps in the main text (Lines 241-267) – see Response 147.

As Reviewer #6 notes, the typology also establishes a vision and frames an agenda for future mapping efforts, highlighting groups of ecosystems, such as savannas and deserts, where global-scale mapping is currently poor. This agenda is rapidly advancing, with several relevant new data sets emerging in the past 1-2 years. We continually update our map repository in response to these advances, and note in the main text (lines 261-265):

“...recently-developed data cubes for a diverse range of species habitats [46] suggest that global high-resolution time series mapping should be possible for most ecosystem functional groups within the next decade. Future versions of the typology will progressively strengthen map standards and improve applications that depend on spatial analysis.”

In general, I would love to see some more careful discussion of the strengths and weaknesses of this typology.

AU RESPONSE 152: We substantially expanded discussion of strengths, weaknesses and aspects for future development of the typology in the main text, focussing on four main issues (see Response 4): assembly models for each Ecosystem Functional Group (lines 226-234, Response 9); modifications to

the classification of units within the hierarchical framework (lines 235-240); development of ecosystem maps (lines 241-267, see Response 151); and discrete representation of continuous patterns in nature (lines 267-276, Response 137).

Validity of the main claims with regard to the typology

Some of the claimed qualities of this typology seem overstated, at least for the human-modified components of it (see specific comments on the individual human-modified biomes and EFGs for details). Given the partial framing of this paper as contributing to the imperative to conserve all levels of biodiversity, I was a bit disappointed to see that cultivated biodiversity did not receive much attention. The distinctions within the human-dominated biomes are in parts not inclusive enough to really encompass all elements of cultivated biodiversity and to reliably distinguish more from less biodiverse human-dominated systems. For example, annual croplands are generally presented as low biodiversity systems, thus by definition precluding appreciation of higher over lower diversities of crop breeds.

I do not believe that the goal of grouping ecosystems with functionally similar responses is achieved for the terrestrial human dominated systems, as the presented EFGs refer to major land use classes and hide tremendous heterogeneity in land use intensity and land management practices. There is ample literature on the different responses of systems with different land management practices to stressors (think of differential resilience of more and less diverse cropping systems) and also on their different responses to ecological processes (e.g. Erb et al. 2017).

AU RESPONSE 153: We thank the reviewer for this thoughtful comment and the reference, which alerted us to other relevant literature. We agree that the functional groups may be quite heterogeneous in certain properties. In part, this relates to limitations associated with discrete representation of continuous properties in nature (see Response 137), which apply to anthropogenic ecosystems and processes as well as natural ones. In addition to relevant comments and edits in Response 150, we make three further points in response.

First, high levels of heterogeneity within groups can be addressed by segmenting continuous gradients of vegetation into a larger number of groups, with a trade-off through increased complexity of the typology. An important point is whether variation in properties is greater between groups than within them at the same level of classification. Greater heterogeneity within groups than among them would justify a trade-off to recognise more groups. We think the major functional distinctions among anthropogenic ecosystem functional groups are greater than variation within them. Conceptually, for example, they allow, market gardens to be distinguished from urban systems as belonging to different functional groups with major contrasts in ecosystem properties and drivers. Where fine-resolution spatial data allow them to be separately identified, market gardens can be mapped separately, even when they are positioned within city or village limits. Further, the level of detail at Levels 2 and 3 of the IUCN Global Ecosystem Typology is comparable to, or greater than detail in corresponding global classifications of land use (see Response 151 for two examples).

Second, the hierarchical structure of the typology allows heterogeneity within groups at one level to be characterised through the identification of multiple groups at a lower level. All Level 3 groups display substantial variation in properties – we are careful to define them as “*groups of related ecosystems within a biome that share common ecological drivers promoting convergence of ecosystem properties...*” (Table S3.1). This does not mean that the members of a group have the

same properties, but rather that the properties converge as a consequence of common causal factors (e.g. cultivation, species introductions and removals, etc.). While we agree with the reviewer, for example, that ‘appreciation of higher over lower diversities of crop breeds’ is very important for certain applications, we think this type of variation ought to be represented at Levels 5 or 6 of the typology, through recognition of multiple units that differ in these properties (within a single functional group). This approach is consistent with the definition of the hierarchical levels (see Response 150 regarding elements of urban ecosystems) and the treatment of non-anthropogenic ecosystems.

Third, where heterogeneity exists within a classification unit, the description of that unit should characterise the variation in its properties. We endeavoured to represent variability within EFGs within the descriptive profiles, so far as space constraints allow. We agree referee #6 that this variation could be better characterised in some anthropogenic functional groups. Accordingly, we amended relevant text to improve the characterisation of variability within groups (see responses to specific comments below).

A separate issue involves temporal variation within the units which, in anthropogenic ecosystems, is closely related to land use intensity and socio-economic processes. Although tracking change is largely now beyond the scope of this ms (see Response 1), we mention this as an example of feedbacks (lines 153-156):

“Similarly, feedbacks exist between ecosystem properties and the drivers. For example, changes in ecosystems initiated by human activity, such as land use intensification, influence human social structure, markets and consumption patterns, driving changes in resource appropriation and further change in ecosystem properties [16].”

The promised scalability seems to mostly apply to scaling across different levels of thematic detail. Scalability in the sense of applicability across spatial and temporal scales is more doubtful, given that such scales are rarely referred to even though several statements would strongly depend on the spatial scale considered. For example, the trait of “low diversity” of croplands may often be true at plot scales, but small farmlands may increase biodiversity at landscape scales. I think the attribute spatially explicit is woefully misleading (at least that term needs to be further qualified), as the provided indicative distribution maps in some cases are ill-conceived or too uncertain to really indicate much, beyond what is anyway trivial (for example, that a certain human-dominated EFG exists on most continents close to where there are humans).

AU RESPONSE 154: It is true that our emphasis is on thematic scalability, as this is most relevant to the design of the typology (reflected in Design principle 4, Table S1.1). We now justify this emphasis more clearly in the text, and link it to different thematic and spatial resolutions required for different applications (lines 177-181):

“The scalable hierarchical structure (principle 4, Table S1.1) and the explicit description of properties and drivers enables units at any thematic level to be mapped at different spatial scales. These may be tracked through different temporal scales, according to needs of specific applications and constraints on the resolution of available data.”

Our usage of ‘spatially explicit’ refers to the units being defined in a way that allows their distributions to be mapped. We demonstrate this by providing indicative maps, admittedly with variation in map quality, with improvements expected over time (see response 151). Conversely, a

classification with units that are poorly described is not spatially explicit if there is insufficient information to characterise their distributions. For example, a few of the classifications we reviewed in Appendix S1.1 only have text-string labels to describe the characteristics of their units.

Finally, although a detailed appraisal of spatial and temporal scaling is beyond the scope of our current paper (given that we excluded spatial analysis from the ms - see Response 1), we contrast the roles of ecological drivers in community assembly over contemporary and evolutionary time scales (lines 127-133) and note that different ecological drivers shape ecosystem assembly at different spatial scales (e.g. lines 141-142 and Fig. 1). We also address needs for spatial and temporal scaling to address related needs of the typology for ecosystem management applications at global, national and local scales. (Appendix S6).

I think the attribute of a sound theoretical basis is mainly achieved with regard to ecological functions. However, the definition of “human-made” systems cannot only rely on an ecological characterization. Unfortunately, the typology ignores existing theories and concepts from those sciences that focus on different “human-made” systems, even those that are also rooted in ecological theory (like the much clearer conceptualizations of land use intensity, for example, that exist in land use science).

AU RESPONSE 155: We now provide more theoretical context to support our rationale for constructing the anthropogenic component of the typology in the main text (lines 146-150):

“While our model portrays humans as integral drivers of ecosystem assembly, we separated human activity from other biotic interactions to highlight interactions and feedbacks between ecosystems and socio-economic systems [18], and the need to assess and mitigate the human impacts on biodiversity and ecosystem functioning.”

and in referring to feedbacks among assembly drivers and ecosystem properties in lines (153-156):

“Similarly, feedbacks exist between ecosystem properties and drivers. For example, human land use intensification initiates changes in ecosystems that, in turn, influence human social structure, markets and consumption patterns, driving changes in resource appropriation and further change in ecosystem properties [18].”

We elaborate in Appendices S2 (section on Human activity, pp3-4) and S3 (section on Dealing with anthropogenic influences on ecosystems, pp22-23). On the pervasiveness and variability of human influence, we note in Appendix S3 (p22):

“Human activity influences assembly of almost all ecosystems. Erb et al. (2017), for example, defined 10 different land management activities, quantified their variation in intensity, and estimated global-scale biophysical and biogeochemical effects on terrestrial ecosystems. The effects play out through complex interactions and feedbacks between ecosystems and socio-economic systems with varied settings for labour and capital inputs, technology, market dynamics, cultural beliefs, business or subsistence decision making and geopolitics (Meyfroidt et al. 2016). Erb et al. (2017) estimated that intensive land management activities occur on about 10% of Earth’s ice-free terrestrial surface based on subjectively thresholded metrics for each activity. Haberl et al. (2007) estimated that humans appropriate >40% of Net Primary Productivity over ~20% of Earth’s ice-free terrestrial surface.”

This typology also claims to be conceptually robust and to overcome other typologies weaknesses of not “fail[ing] to describe their units in sufficient detail for reliable identification or require[ing] diagnostic features that are hard to observe”. For the human-dominated systems, their descriptions are in parts reliant on vague, or insufficiently defined, boundary concepts (e.g. “intensive”, “natural”), or are incomprehensive and implicitly excluding regionally important examples of human-made systems (e.g. villages, non-woody permacultures). This makes a reliable identification of EFGs by strictly following this typology highly doubtful, especially if persons from diverse academic backgrounds are involved. The typology also used some diagnostic features that can be very difficult to observe (e.g. alien biota, selective breeding). For several of the ecological traits and the drivers listed in Fig. 1, data that would be needed for detailed mapping and monitoring are not available for many regions. Several human-made systems are defined either too vaguely or too restrictively, which leaves some systems unaccounted by any one group.

AU RESPONSE 156: The descriptions of Ecosystem Functional Groups in Appendix S4, including text on salient ecosystem properties and drivers, illustrative images, diagrammatic models of assembly processes, indicative global distribution maps and selected references, are more detailed, consistent and systematic than those provided for all other ecological typologies reviewed in Appendix S1. Some of the diagnostic features are recognisable in the field, others are detectable from remote sensing, while some require specialised measurements. As noted above, current map data varies in quality. The descriptive profiles have been subject to extensive peer review (Appendix S5). We are further developing interpretive resources and making them available to a wide audience through the IUCN Global Ecosystem Typology website <https://global-ecosystems.org/>. We acknowledge inherent challenges in simplifying complex continuous variation in nature within a discrete classificatory framework (main text lines 268-276; Appendix S3, p23). We also acknowledge the need to refine and update the descriptions themselves and thank the reviewer for detailed suggestions, which we address below.

Problems in the EFG datasets associated with the typology and validity of presented analyses I believe that the paper as a whole (that is, not just considering the typology per se, but also the presented analyses and data) has several severe flaws that undermine its scientific soundness. The authors frame several analyses around the high-level problem that “Decisions about effective action to conserve biodiversity and sustain ecosystem services [which] require evidence on which ecosystems are most exposed to impacts from particular pressures [...], which ecosystems are undergoing most rapid loss of biodiversity, and which ecosystems contribute most to particular human benefits”. The authors then claim that “[...] the IUCN Global Ecosystem Typology and associated spatial data [19] provide an ecologically robust and powerful framework for such synthesis”. As a central selling point to demonstrate this potential of the typology and associated data, the authors present concrete results on specific portions of the world’s ecosystems that are exposed to high pressures and/or are more poorly protected than others.

AU RESPONSE 157: Much of this criticism refers to our original spatial analyses which has now been deleted from the ms (see Response 1). The second phrase quoted applies only to some of the spatial data (see Response 151). We qualified the text to clarify as follows:

“... the IUCN Global Ecosystem Typology and a growing body of spatial data [23] will provide an ecologically robust and powerful framework for such synthesis”

(Lack of) treatment of uncertainties

There are tremendous uncertainties in several of the used data (see below for specific details on the data on human-dominated EFGs), and moreover severe scale mismatches between different datasets that are combined here. Both seriously hamper their fitness-for-use for assessing both which EFGs are how well protected, which ones are most pressured, and which ones experience specific pressures. As neither limitations are addressed in the presented analyses, nor is there any validation of results, the presented results cannot be considered trustworthy (see below for details). Some of the bigger picture applied conclusions, e.g., those stressing greater protection needs of specifically forests and grasslands compared to other terrestrial systems, are thus not sufficiently substantiated.

AU RESPONSE 158: We removed the spatial analyses and research findings from the manuscript and Appendix S6 as requested (see Response 1).

The same mentioned limitations of the typology-associated maps also make them inappropriate for any other claimed application of the typology that are sensitive to spatial misestimations, including as national ecosystem accounting and monitoring of CBD targets. In fact, the implicit message that a typology whose units can currently only be mapped indicatively could be a standard for these globally important applications seems dangerously misleading.

AU RESPONSE 159: In our ms, we noted that the United Nations had adopted our typology as the reference classification for ecosystem accounts but did not suggest that the indicative global maps in Appendix S4 should be the basis for national ecosystem accounting. On the contrary, the UN SEEA-EA (https://unstats.un.org/unsd/statcom/52nd-session/documents/BG-3f-SEEA-EA_Final_draft-E.pdf) has adopted the typology primarily to aid synthesis and upscaling of accounts from many semi-independent national sources. This allows use of the best available national maps for each country, with cross-walks developed to translate fine-resolution national ecosystem types to global Ecosystem Functional Groups as demonstrated for Chile and Myanmar in Tables S3.3 & S3.4, respectively. We now discuss how our approach decouples classification from mapping (lines 254256):

“By decoupling the mapping process from prior development of the classification, our approach liberates the definition of ecosystem units from constraints imposed by the current availability of spatial data and allows for progressive improvement (Appendix S4, p15).” And

and in the second paragraph of the Natural capital accounting section in Appendix S6, we discuss the use of maps for ecosystem accounting, as recommended for the UN SEEA-EA process:

“The IUCN Global Ecosystem Typology was adopted as the reference classification for implementing SEEA-EA (UNCEEA 2021). The core of the reference classification is Level 3, which will be used to summarise globally across national ecosystem accounts. Contributors to SEEA-EA are expected to use the best quality high-resolution classification available for their jurisdiction when developing their national accounts (i.e. Level 6 units), and assign the units of that classification to Ecosystem Functional Groups (Level 3) to enable consistent international reporting. Examples linking fine-grained national classifications to Level 3 of the IUCN typology are given in Tables S3.3 and S3.4. This flexible use of the IUCN typology will enable jurisdictions to

report detailed national accounts, while contributing data that can be scaled up and summarised to units of the international reference classification (UNSD 2019)."

I wonder how sensible it is in general to acknowledge the indicative nature of the maps, but then still present them as pixel-maps rather than in a format that is clearly distinguishable as "indicative", such as blob maps (compare the IUCN species maps). More so, I wonder why the earlier acknowledgement of limitations is later ignored presenting a pixel-based intersection with other maps. I have to give the authors credit in that they use a lot of leveraging language, like "indicative", "coarse-scale analysis", "Acknowledge uncertainties", etc. Yet, the uncertainties are merely acknowledged in writing, but are not reflected in the analyses, nor in the presentation of results. In fact, with the information provided, even the indicative validity of the EFG maps can only be ascertained by experts on the respective fields and data. I do feel that the burden to make uncertainties more clearly visible is on the authors, here, given that these indicative maps might be naively (ab)used for all kinds of applications.

AU RESPONSE 160: We removed the spatial analyses and research findings from the manuscript and Appendix S6 as requested (see Response 1). We think a global-scale pixel format with relevant caveats is appropriate to current uncertainties in distribution. We are familiar with the 'blob maps' stored in IUCN's Species Information Service. Most of those maps are compiled from hand-drawn polygons generated by species experts, usually with reference to occurrence records from a range of (unspecified) sources. The 'blobs' may include large areas of unoccupied habitat and the sources and development steps for individual maps are not well documented, yet they are used in a number of spatial analyses published in international journals. We believe the overall standard of EFG maps is at least as good as IUCN species maps (in many cases better), their derivation is more explicitly documented and they are updated regularly from published sources. While we agree that the lowest quality EFG maps preclude a comprehensive global spatial analysis, we believe all maps are sufficient quality for indicative thumbnail maps presented in Appendix S4 (see Response 1).

Lack of reproducibility:

The provided information on the data sources and the generic descriptions of mapping protocols are insufficient to make the indicative maps of EFGs reproducible, at least in several cases. I made the effort of reading through the IUCN ecosystem typology that is already published on the IUCN website (which extensively overlaps with information presented in the appendices for this paper). While there is substantially more information therein (e.g. Table 4), this is in many cases not sufficient for understanding specific data manipulations, let alone the underlying reasoning and the validity of the results. In some cases, using the specific referenced datasets as indications of the respective EFGs seems dubious at best (see comments on specific EFGs below). I have serious concerns about publishing any analysis built on a top of the EFG maps before publishing a detailed description of the specific mapping and validation protocol for each map. I also noted that the methodological description of degradation assessment relies on an "updated version" of a previously published terrestrial human footprint maps for other years. As these maps do not seem to be published, and it is unclear whether they were produced via the same protocol as the original maps, and how valid these new maps are, these analyses are not reproducible.

AU RESPONSE 161: We removed the spatial analyses and research findings from the manuscript and Appendix S6 as requested (see Response 1). We acknowledge that indicative maps for one-third of

the Ecosystem Functional Groups currently lack spatial data of suitable quality for quantitative analysis (lines 250-252).

Appropriate attribution of EFG map sources:

In this respect, I wonder in some cases which specific creative manipulations to the source maps were done to justify that they are here collectively referenced as “Keith et al. 2020”. With respect to spatial data, the main original achievement seems to have been to collate the pre-existing maps that were deemed fitting to the respective EFG, clip them to the broader regional boundaries of the higher ecosystem hierarchical levels, and making them available through a common portal. Where this is really all that happened, it would seem appropriate to also cite the original map sources in this paper.

AU RESPONSE 162: The reference cited by the referee refers to a compiled archive of maps. It provides the data agreed by specialist contributors for each description to be the best available representation of the concept for each Ecosystem Functional Group. The source data are in the public domain (Creative Commons) and fully cited in Table S4.1 and the archive, which includes Readme files and xml documents with map details and references for both the bundled and individual maps so that they are fully documented and attributed to sources. The 'creative manipulations' on the maps are different in each case with some EFGs almost identical to one original source map while others are combinations of several sources. The whole mapping process included reviewing and comparing alternative data sources, performing several spatial operations (transformation, reprojection, aggregating, resampling, cropping or clipping, intersections and unions, map algebra, etc), and finally harmonizing all of them to similar resolution and presentation. Even where the map of an EFG concept, such as the one for MFT1.1 Coastal deltas, is apparently identical to the source, it has undergone an evaluation process, has been selected instead of alternative maps, and has been harmonised for consistency with other maps in the archive.

Inconsistencies between EFG maps:

I am worried about inconsistencies among the datasets for the different EFGs. They originate from very different sources and were generated for different purposes, and range from hand-drawn blob maps, via dasymetric modelling outputs, to classified remote-sensing imagery. As far as I can tell, no efforts were made to adjust these layers to make them consistent. If the sole claimed purpose of these EFG maps was mapping their indicative global distributions, this would not be a problem. But this could affect several of the presented analyses. For example, the presented test for protection of different EFGs could be affected by pseudo-replication if small protected areas that really only encompass single EFGs were overlaid with crude distribution maps of multiple EFGs that are covering the same areas. This would also affect several of the other claimed application fields of the typology that would be sensitive to large misestimations of EFG areas resulting from such inconsistent mapping.

AU RESPONSE 163: We removed the spatial analyses and research findings from the manuscript and Appendix S6 as requested (see Response 1). The maps were harmonised to a consistent format, but represent a minimal necessary adjustment to the best available spatial data for each unit (see response 162). This allows users to make their own decisions about how to achieve a level of consistency between maps that suits their purpose.

Analysis of global protection status of ecosystems:

The authors aim to demonstrate “how the typology could support evaluation of Aichi target 11, ... [by analyzing whether] ... at least 17 per cent of terrestrial and inland water, and 10 per cent of coastal and marine areas ... [are] conserved through ecologically representative systems of protected areas ...”. The authors provide an analysis of EFG representation in global protected areas (PAs) based on spatial overlay of both datasets. However, like any globally “indicative” map, the EFG maps will in many cases misrepresent true areas where the EFGs exist at fine scales. The authors textually acknowledge this themselves, but the presented analyses fail to address the scale mismatch between the coarse-scale EFG maps and the PAs (which are mostly mapped at much finer scales). It has been repeatedly shown in the case of overlays with IUCN species maps that failing to account for such scale mismatches leads to biased estimations of species protection. For essentially the same reasons, a simple overlay of indicative EFG maps with PAs is unlikely to yield reliable results. Moreover, it is completely unclear just how certain or uncertain results for different EFGs might be. This makes the scientific value of this exercise highly questionable. Given the scale mismatches and unassessed uncertainties in the EFG maps, I am not convinced that the presented typology (with its currently associated data) can, indeed, at this point support such analyses for Aichi target 11, or similar global analyses.

AU RESPONSE 164: We removed the spatial analyses and research findings from the manuscript and Appendix S6 as requested (see Response 1).

Analysis of ecosystem degradation:

To assess which ecosystems face degradation pressures, the authors overlaid the indicative EFG maps with human-footprint maps for specific years. I have three main concerns about these analyses. Firstly, in addition to the same spatial-scale mismatches as in the PA-overlay analysis, this assessment is also subject to temporal mismatches. The human-footprint maps refer to the years 2000-2013, while the data underpinning the indicative EFG maps have many different focus years (and are in some cases not time-explicit). It seems likely that the areas of some EFGs have changed between the reference time of their respective original source data and the time period covered by the Human-influence data. Particularly in the parts of the world that have experienced very rapid land-use changes during short periods of time, it is highly possible that this temporal mismatch will bias the results. Secondly, the extent to which this assessment can provide evidence, rather than just indications, of “ecosystem degradation” seems to be slightly misrepresented. The authors acknowledge that their analyses are only based on information on presumed degradation drivers, rather than actual observations of degraded ecosystem states. But then they should arguably use more qualified language like “likely degraded ecosystems” or “ecosystems exposed to degradation pressure” throughout.

AU RESPONSE 165: We removed the spatial analyses and research findings from the manuscript and Appendix S6 as requested (see Response 1).

Finally, the discussion of these analyses is insufficiently referenced and/or too swiftly argued. For example, the authors contrast a need for protection of remnants for dry forests that recently saw accelerated degradation/conversion with the need to actively restore grasslands that have long been

exposed to pressures. I can only assume that the authors are hinting at the differential passive restoration potential of these systems, as natural species seedbanks would presumably still be largely intact in the former case but largely lost in the latter. In any case, it would be important to expand on and better support such statements, so the reader can understand the reasoning.

AU RESPONSE 166: We removed the spatial analyses and research findings from the manuscript and Appendix S6 as requested (see Response 1).

Analysis of ecosystem degradation:

The assessment of ecosystem portions that receive different combinations of degradation pressures (based on overlay with the Human-influence maps) and protection (based on the PA overlay) is particularly flawed. The human-influence maps are partly based on (and their patterns strongly driven by) maps of human population density (specifically, the Gridded-population-of-the-world dataset, GPW). However, the GPW dataset was generated by distributing statistical population data to pixels, and during this process, protected areas were masked out. There is thus circularity, and almost certainty bias, in the assessment of different ecosystems' combinations of degradation pressures and protection status (Figure S6.3). What is even more worrying is that the authors use this circular and likely biased assessment to derive concrete recommendations for nuanced management strategies for different ecosystems, depending on the specific combination.

AU RESPONSE 167: We removed the spatial analyses and research findings from the manuscript and Appendix S6 as requested (see Response 1).

Analysis of anthropogenic transformations:

The authors also overlaid the indicative maps with a global map product of anthropogenic biomes, which they used to “estimate the proportion of area exposed to high pressures that show evidence of transformation into artificial or human dominated ecosystems”. The used anthropogenic biomes product by Ellis et al. is based on a land use model with very well-known uncertainties (variously discussed by the authors themselves, and acknowledged in that paper). Yet, the transformations it shows seems to have been taken at face value here. In addition to these model uncertainties, I am also concerned about inconsistencies in the presented analysis, because the historical land-use model behind the anthropogenic biome product was informed by different land-cover maps than those underpinning the presented indicative ecosystem maps. None of these uncertainties is currently addressed. This makes it unclear which of the results shown in Fig. S6.6. indicate genuine ecosystem transformations and protection levels, and which ones merely reflect artefacts of overlaying uncertain and inconsistent products. Without some sort of sensitivity analyses it is unclear which patterns are interpretable and to what extent, and which ones are not.

I also wonder about the conceptual validity of the results shown in Fig. S6.6. It seems like this assessment assumes that EFG distributions remained static over the considered study period, and that they could only either remain intact, become degraded, or be destroyed, but never recover. We know for some regions that degraded lands were actively restored or abandoned and left to become wild again, and also that in some wilderness areas natural EFGs have advanced over others (like shrublands over arctic tundra in Canada). Both possibilities seem to have been discounted, which makes me wonder even more how reliable the relative thickness of bars shown in that figure may be.

AU RESPONSE 168: We removed the spatial analyses and research findings from the manuscript and Appendix S6 as requested (see Response 1).

General validity of typology

Generally, it is challenging to assess the specific validity of many of the statements placed across this massive paper, not least because many of them are not referenced. I am not sure if all those unreferenced statements are trivially clear or uncontentious, even among experts on the specific topic. For example, many statements in Table S3.2 arguably may fall into this category.

AU RESPONSE 169: We improved referencing and strengthened justifications for several aspects of the typology throughout the manuscript and appendices. We also revised the text to make it clear that the models compiled for each Ecosystem Functional Group (in Appendix S4) represent working hypotheses about drivers that shape observed ecosystem properties (see Responses 4 & 9). We added 26 references to the commentary on Table S3.2 (pp4-6, Realms section) to support the generalisations made there. In addition, the cited references in descriptions of each Ecosystem Functional Group in Appendix S4 are selected reviews and case studies with relevant evidence on ecosystem properties and postulated drivers. Table S3.2 is a summary of hypothesised relationships represented in the models for the Ecosystem Functional Groups across four realms. We revised the caption to make it clear that the relationships are postulated and that relevant sources of evidence are found in the text commentary the precedes it and the respective descriptive profiles within Appendix S4:

“Table S3.2. Synopsis of postulated assembly filters and ecological traits distinguishing ecosystems within the five realms of the biosphere (Fig. S3.1). Refer to Appendix S4 for glossary of selected terms and to text commentary (above) and respective descriptions of Ecosystem Functional Groups within each realm (Appendix S4).”

Conceptual choice of including community attributes as criteria

The typology is not only based on the functional attributes (which I think is very useful), but, for the defined ecosystems at levels 5 and 6 of the proposed hierarchy, also on their ecological communities.

I am not sure whether a typology that requires measurements of the integrity of a given community can be of much long-term stability, which should arguably be a requirement of any typology. Ecological communities have always changed, both over evolutionary and over ecological time scales. The biological lineages represented in communities may uniquely respond to different types of environmental changes (or “ecosystem drivers”, as referred to herein) and usually show only limited assemblage-wide synchrony in those responses. That many current communities will break apart and change into new ones should arguably be a baseline expectation of a typology that is designed to remain robust over multiple decades (i.e., policy and global monitoring time scales). In fact, all these criticisms have been phrased since several years ago in response to the very proposal of an IUCN Red List of Ecosystems (Boitani et al 2015). I would expect that a paper presenting an ecosystem ontology that defines ecosystems by community integrity would proactively address any such caveats and previous criticisms.

AU RESPONSE 170: We agree that long-term stability is an important quality for a workable typology. Two aspects of our approach to development of the typology contribute to this quality. Firstly, in lines 108-110 of the main text, we state:

“Ecological classifications based on tested and established theory are more likely to be robust to new information than classifications based only on observed patterns and correlations.”

Secondly, in Methods, we now comment further on how we constructed the typology to make it robust to changes in assemblages except where they involve substantial functional shifts (lines 351-361):

“While neither function nor composition were intended to take primacy within the typology, we reasoned that a hierarchy representing functional features in the upper levels is likely to support generalisations by leveraging evolutionary convergence. In contrast, a typology reflecting compositional similarities in its upper levels is less likely to be stable due to dynamism of species assemblages and evolving knowledge on species taxonomy and distributions. Furthermore, representation of compositional relationships at a global scale would require many more units in upper levels, and possibly more hierarchical levels. Therefore we concluded that a hierarchical structure recognising compositional variants at lower levels within broad functionally-based groupings at upper levels would be more parsimonious and robust (principle 6) than one representing composition at upper levels and functions at lower levels.”

For this and other reasons we did not consider ecosystem integrity in the design of the typology (none of the design criteria in Table S1.1 mention it). In our ms, we also carefully avoid confounding the concept of anthropogenic ecosystems and their definition with assessments of ecosystem integrity (We hope that excluding the Human Footprint analysis removes any confusion it may have generated on this point). Excluding integrity from definition of the classification units allows the typology to support applications by framing independent assessments of integrity for any type of ecosystem using appropriate methods and metrics.

As noted in the additional Methods text quoted above, the upper three levels of the typology are based on functional attributes, not biological lineages. The typology is therefore well buffered from instability when communities break apart. As a comparison, we point to species taxonomy which is in continual flux as species concepts change, new taxa are discovered and new data emerge on relationships of higher taxa, especially since the molecular revolution. This causes some inconvenience but does not preclude the usefulness of the classification system for a wide range of applications, including risk assessment (see Keith et al. 2015; <https://conbio.onlinelibrary.wiley.com/doi/full/10.1111/conl.12167> for elaboration on that issue). Similarly, nothing in our approach assumes that species components of ecosystems respond synchronously to environmental change. This should not be confused with our point that different ecosystem types included within the same Ecosystem Functional Group should be expected to exhibit similar responses to environmental change and management, not as a consequence of biological lineages, but because of functional convergence across different lineages (lines 164-166):

“Convergences in ecosystem properties are axiomatic to a functionally based ecosystem typology because they underpin robust generalisations and predictions about ecosystem responses to environmental change and management.”

The Boitani et al. (2015) paper raised some issues about an earlier paper presenting a risk assessment protocol for ecosystems (Keith et al. 2013; <https://journals.plos.org/plosone/article?id=10.1371/journal.pone.0062111>). While Boitani called

for development of an ecosystem typology to frame risk assessments (an identified need that we meet in this ms), most of the content of that paper concerned ecosystem risk assessment. We refer to our contribution in the same journal issue for discussion of ecosystem risk assessment and rebuttals of several claims made by Boitani et al (Keith et al. 2015; <https://conbio.onlinelibrary.wiley.com/doi/full/10.1111/conl.12167>).

It is unclear to me how the typology is expected to manage to deal with these community changes. What criteria will be applied to decide when a given ecosystem can no longer be considered that ecosystem as a result of its ecological communities being reshuffled? Will that create many novel ecosystems according to this typology, that are then still to be defined? Will a new, homogenous ecosystem that was created via a climate-driven reshuffling and homogenization of two neighboring, previously distinct communities (e.g., as may easily happen in some remote regions at high latitudes as previous climatic barriers are breaking down) be considered as a single new natural ecosystem, or as degraded version of the two ecosystems that were previously mapped there, and that the single new one replaced?

AU RESPONSE 171: Refer to Response 170. We also revised text in Appendix S6 (Natural capital accounting section) to address ecosystem shifts.

“When structured by Level 3 of the Global Ecosystem Typology, national and global ecosystem accounts should quantify major functional shifts and re-organisation of ecosystems. For example, transformation of the Aral Sea during 1980-2000 (Micklin 2006) from a large freshwater body (EFG F2.1) to ephemeral grasslands (degraded forms of T5.1) and hypersaline water bodies (F2.6), can be readily tracked at Level 3 of the typology. While we acknowledge uncertainties in ecosystem identification, such major shifts between groups should be relatively easy to detect and have major implications for human wellbeing and biodiversity. Other increasingly common types of ecosystem transformation are also readily detected and reportable through Level 3 of the typology. These include transitions driven by land use change (T1-T6 to T7, or T7.2 to T7.3), changes in fire regimes (e.g. T1 to T4), or climate-related shifts such as drying of freshwater aquatic systems (various transitions involving F1, F2 or FT1), advance of alpine treelines (e.g. T6.4 to T2.1) or repeated coral bleaching and ocean acidification (M1.3 to M1.6 or M1.7). Conversely, shifts that result from forest reconstruction (e.g. T7.2 to T2.2) or land use abandonment (e.g. T7.5 to T2.2) should also be detectable and reportable as shifts between different Ecosystem Functional Groups.

The upper levels of the typology should be robust to compositional changes that do not produce major functional shifts. However, more subtle shifts reshuffling of species composition, for example, in response to climate change or land degradation, could be represented at lower levels of the typology in national-level accounts (with consequent uncertainties in diagnosis); and upscaled to changes in global condition accounts.”

General comments on the human-modified aspects of the typology

After being extremely impressed by most of the natural components of the typology, I was quite disappointed with the human-modified components. I must say that I did not get the impression that either scientists specializing in these human altered systems (e.g. land-use scientists, agricultural scientists, foresters, urban scientists, aquaculture/fisheries scientists, etc.) nor specialists of human

natural systems extensively contributed to the definition of these systems. What matters here is of course not whether or not they were involved, but whether advances and contemporary definitions in those fields were sufficiently considered.

The typology heavily relies on specifically ecological assembly theory (although transferred from communities to entire ecosystems, including their abiotic components), and on using biological and physical system properties to distinguish “functionally different groups of ecosystems from one another by highlighting different ecological drivers that come to the fore in structuring their assembly”. For categorizing and tracking human-dominated systems, this may make this typology of limited use, as those human-dominated systems are characterized and driven by interacting social and biophysical processes and system properties. This may also make it difficult to reliably identify the specific distinctions between degraded “natural” and structurally and functionally similar but nevertheless “human-made” ecosystems. For example, a natural grassland with a long history of livestock grazing and an old seminatural pasture. Both may function very similarly now and may have similarly impoverished or neophyte-dominated grassland communities by now, and their only distinction knowable today might be their history of human use.

AU RESPONSE 172: This distinction is most likely to be problematic at fine spatial scales. We acknowledge uncertainties in representing continuous variation in a discrete classification (see Response 171). Also, we point to an important aspect of the definition of anthropogenic ecosystems (see Glossary),

“...For some of these systems, cessation of those [human] activities may lead to transformation into ecosystem types with different properties and organisational processes.”

Such processes are well-known in parts of Europe, North America and Australia where derived semi-natural grassland typically transit to secondary forests or woodlands, when they are abandoned or spelled from grazing. In contrast natural grasslands, should not undergo such transitions when grazing is removed, so long as the drivers that preclude tree establishment or growth remain functional.

Currently, all human system alterations are basically presented as an extrinsic “human activity” box. The specific ways in which human processes interact with the ecosystem drivers and traits that are used to distinguish EFGs are not always very clearly conceptualized, nor do they make clear connections to existing (ecologically inspired) conceptualizations of human-environment interactions. For example, the first group of abiotic drivers include the availability of five “fundamental resources essential to sustaining all life: water, nutrients, oxygen, carbon and energy”. I think this would have been an opportunity to characterize “intensive human systems” explicitly as those with a human appropriation of NPP (HANPP) above a certain threshold. HANPP is already one of system-level property of land-use systems that is commonly used by land use scientists to characterize different levels of land use intensity. It would seem to me that adding the proportion change in ecosystem productivity relative to a “natural” baseline that is due to harvesting by its top-predator/herbivore (humans), but also to fertilization, etc., as a system-level property and/or functional trait of the ecosystem might be helpful. Of course, this may in turn be influenced by human’s altering of its other traits and of the other drivers.

AU RESPONSE 173: We thank the reviewer for this suggestion, and agree that HANPP is an important signature for some anthropogenic systems, and that it could help to distinguish Ecosystem Functional Groups characterised by intensive land use those from natural and semi-natural systems

(i.e. with low intensity use). Such indicators are potentially useful in map development (we have used HANPP in revisions of several maps suggested by the referee #6), but using them to define the concepts of units within a classification raises several significant problems. Although thresholds in indices such as HANPP are attractive as explicit delimiters of classification units (most biophysical classifications reviewed in Table S1.2 use them for that reason), they tend to oversimplify complex multidimensional properties that distinguish groups of functionally similar ecosystems (see our definition of Ecosystem function in lines 81-82 – more than productivity alone). Use of aggregated univariate indices as thresholds to delimit ecosystem types confounds a clear perspective on some of these multiple properties and ignores others, ultimately producing a poor representation of variation in ecosystem types across the biosphere. Erb et al. (2013, p465) support this view, *‘Given the complexity of land-use intensity, providing a single, unambiguous and encompassing definition or indicator of land-use intensity does not appear to be an adequate target. A comprehensive analytical framework is required that considers the multidimensional nature of land-use intensity.’* Their proposed framework encompasses characterisation of land use inputs, outputs, their interactions and changes in system properties (for which HANPP is one of several example indices). We therefore avoided definition of EFGs by thresholding univariate aggregated indices. Another problematic issue is the estimation of HANPP, which relies on a number of (largely untestable) assumptions and subjective interpretations of ‘potential’ NPP, as well as sensitivities to data availability for actual NPP (discussed in some detail by Haberl et al. 2007, PNAS). Further, HANPP is relevant to production systems (and mainly on land), but may not be so useful for describing non-production anthropogenic systems, which account for many of the 16 anthropogenic Ecosystem Functional Groups in the typology.

Despite the limitations of quantitative application, we think HANPP (along with production inputs and outputs) has great value as one of the descriptors of production ecosystems, as well as a spatial predictor in mapping systems characterised by intensive production use (see Haberl et al. 2007, PNAS), and in the assessment of ecosystem status. Our approach to typology development, which decouples classification and mapping stages (see Response 151), enables a succession of mapping advances to be incorporated as data on potential and actual NPP improves. As noted above, we used HANPP and other spatial variables to revise maps for T7.2 and T7.5.

We incorporated discussion of these issues into several parts of the main text and appendices as follows:

Methods text (lines 477-484):

“Indices such as human appropriation of net primary productivity [58], combined with land-use maps [59], offer useful insights into the distribution of some anthropogenic ecosystems, but further development of indices is needed to adequately represent others, particularly in marine, and freshwater environments. Beyond land-use classification and mapping approaches (Appendix S1, p6), a more comprehensive elaboration of the intensity of human influence underpinning the diverse range of anthropogenic ecosystems requires a multidimensional framework incorporating land-use inputs, outputs, their interactions, legacies of earlier activity and changes in system properties [60].”

Appendix S2 (p4, 2nd para):

“The strength of these effects varies from negligible to transformative, resulting in changes between ecosystem types and variation within ecosystem types spatially and temporally. For example, global models of Human Appropriation of Net Primary Productivity (HANPP) estimate

variation from 0 to ~100%, as well as some areas where land use increases NPP above potential natural levels (Haberl et al. 2007). In the typology, we recognise major transformative outcomes of human activity by defining anthropogenic ecosystems as those created and sustained by human activity. We distinguish these types of ecosystems from lesser human influences where the underlying identity of a 'natural' system (i.e. its characteristic biota and ecological processes) is retained, albeit modified to some degree."

Appendix S2 (p4, 3rd para):

"Characterising the intensity of human activity requires analytical frameworks that consider the multidimensional nature of land-use, water-use or sea-use intensity (Erb et al. 2013). These frameworks are most advanced for production systems (agriculture, forestry, fisheries), with conceptual models encompassing land use inputs, outputs, their interactions, and changes in system properties (Erb et al. 2013). Spatial modelling of indices such as HANPP (Haberl et al. 2007), offer opportunities to map global distributions of certain anthropogenic ecosystem types, and to assess human impacts on natural and semi-natural types, particularly where there is continuous variation in the drivers of human activity."

and specific revisions to descriptive profiles for anthropogenic Ecosystem Functional Groups in Appendix S4 detailed below.

The third group of abiotic drivers includes disturbances, such as fire and flooding. Here, I am missing a clearer conceptual definition and quantification of "disturbance". Essentially every abiotic factor described in the first two groups might be a disturbance if it occurs more suddenly and/or more intensively than usual. When natural variation begins to be a disturbance is not defined. I would help if the framework made explicit reference to the spatial and temporal scales at which theEFG-defining criteria apply.

AU RESPONSE 174: Our definition of disturbance (see main text lines 136-139 and Glossary) is based on both the discrete nature of the recurring events and specific requirements on the ecosystem response (destroying biomass, resource redistribution, life history triggers). Together, these features distinguish disturbance regimes from other biotic filters. Our interpretation is consistent with a large literature on disturbance ecology across the different disturbance types that we mention. Fig 1 identifies disturbance as typically landscape-scale phenomena (i.e. regional-local), whereas resource availability and ambient environmental factors operate from global to local scales. We think there is considerable value in separating disturbance regimes of recurring events from other types of abiotic drivers, consistent with many other contributions on ecosystems, communities and populations. See also response 59.

I am not convinced that the anthropogenic assembly filters listed in table S3.2 are sufficient for distinguishing ecosystems under differing levels of anthropogenic influence. Humans, through different mechanisms, may alter almost any of the above-mentioned other filters and traits. I am probably missing something here, but none of these listed filters seems to capture humans' influence on ecosystems' energy flows through hunting, fishing, harvesting, etc. (e.g. only some of this human influence manifests via structural transformations).

AU RESPONSE: 175: We revised Fig. 1 to represent anthropogenic influences on all other drivers more transparently (see Response 19). Hunting, fishing, harvesting are different forms of resource

use by humans. We mention resource appropriation in the main text and resource use in Fig 1, but this was somehow omitted from Table S3.2. We added two rows to the table to address (abiotic) resource extraction and appropriation of biomass and productivity. Thank you for pointing out this omission.

I find the conceptualization of anthropogenic biomes as being “created by human activity, which continues to drive and maintain their assembly (Ellis et al 2010)” too simplistic. Few if any human dominated systems are entirely created by humans, and all still rely on at least some nonanthropogenic ecosystem processes. Few ecosystems, in turn, are completely free of human alterations. In most cases, humans have influenced ecosystems over centuries or even millennia, and in other cases, today’s ecosystems have been co-created by humans and other species, and/or the extent to which seemingly natural ecosystems are co-created is still debated. Many scholars in historical ecology and other fields have even come to question the very concept of “natural ecosystems” and instead conceptualize all systems as human-environmental systems (or social ecological systems, etc.). I think that it should be clearly defined what is meant by either, and any underlying value judgements should be reflected on and made transparent, and that less definite wording should maybe be used throughout, e.g. including qualifiers such as “strongly”, “mainly”, etc.

AU RESPONSE 176: We deleted the problematic phrase. Although a succinct and simple definition is needed (see Glossary), we discuss human activity and distinguish anthropogenic and non-anthropogenic ecosystems in terms of high and low intensity, referring to relevant literature, including that recommended by the reviewer. We acknowledge the profound and long effects of human activity at several points in our ms (lines 467-485, copied below) and in dedicated sections of Appendices S2 and S3:

Anthropogenic influences create challenges for ecosystem classification, as they may modify defining features of ecosystems to a degree that varies from negligible to major transformation across different locations and times. We addressed this problem by distinguishing transformative outcomes of human activity at Levels 2 and 3 of the typology from lesser human influences that may be represented either at Levels 5 and 6 or through measurements of ecosystem condition that reflect divergence from reference states arising from human activity (lines 472-477).

“Anthropogenic ecosystems grouped within Levels 2 and 3 were thus defined as those created and sustained by intensive human activities, or arising from extensive modification of natural ecosystems such that they function very differently. In many agricultural and aquacultural systems and some others, cessation of those activities may lead to transformation into ecosystem types with qualitatively different properties and organisational processes (see [56] and [57] for cropland and urban examples, respectively).”

Refer to Response 173 for further comment and text additions relevant to this point

This also applies to certain applied conservation statements, such as in line 282 of Appendix S6 (“Priority should be given to protecting the most intact areas that represent the range of ...”) – it would be appropriate to be transparent here, about the authors’ values on which these perceived priorities are based (I presume, a preference for wild nature), and whether decision-makers charged with representing different values (e.g. potentially preferring ecosystem services provision close to humans) would perceive the same priorities.

AU RESPONSE 177: The line referred to by the referee was an inference drawn from the results of the spatial analysis and has therefore been deleted from the ms (see Response 1). In Appendix S6 (section on Natural capital accounting) we explain how the typology can support decisions and actions founded on objectives to manage different values of ecosystems aligned with their dual roles in biodiversity conservation and human well-being. In the final paragraph of the section on natural capital accounting, for example, we state,

“one approach to sustaining ecosystem services is to identify, protect and manage the ecosystems that provide them (Bordt & Saner 2019). The Global Ecosystem Typology offers a consistent and comprehensive framework incorporating ecosystem functions as basis for attributing ecosystem services to ecosystem types. We suggest that such an approach is likely to be more informative for ecosystem management than land cover or land use categories (Di Gregorio & Jansen 2000).”

In this respect, I also find the characterization of several functional groups in the non-anthropogenic biomes “incomplete”, as they underrepresent the influence that human activities have long had in shaping how these systems look today. For example, there is ample evidence that pre-Columbian societies actively shaped (at least heavily contributed to shaping) today’s “pyric tussock savannas (T4.2)” in Eastern North America through human-made and/or human-altered fire regimes. Elsewhere, already prehistoric human societies have wiped out or compositionally altered megafaunal biota, etc., etc. Several EFGs, by their provided definitions, exclude several human-dominated systems that are not as intensive or as homogeneous as the definitions and examples imply and that, however, also do not fit into any of the other biomes (esp. the distinction of the T.7 biome into the five current EFGs). For example, “Annual croplands” by definition excludes permanent *Miscanthus giganteus* croplands, which are also excluded from “Plantations” due to their definition of being “woody”. Similarly, “Urban and industrial ecosystems” by definition exclude non-urban human settlements, and non-industrial infrastructure such military structures. Similarly, the distinctions provided do not adequately account for mixed systems such as agro-pastoral, silvo-pastoral systems (see specific issues related to this below). Since this typology is presented as being globally applicable, comprehensive, and systematic, and since it claims applicability in all kinds of fields, all human-made ecosystems should either be captured as individual classes, or the existing class definitions should be made more inclusive.

AU RESPONSE 178: We now explain in the ms (lines 485-491), that we focus on reference states of non-anthropogenic ecosystem types

“Where less intense human activities occur within non-anthropogenic ecosystem types, we focussed descriptions on low-impact reference states. Therefore, human activities are not shown as drivers in the assembly models for non-anthropogenic ecosystem groups, even though they may have important influences on the contemporary ecosystem distribution. This approach enables the degree and nature of human influence to be described and measured against these reference states using assessment methods such as the Red List of Ecosystems protocol [39], with appropriate data on ecosystem change.”

See further discussion in Appendix S3 (p22). Assessing the effects of human activity on ecosystem properties requires a wide range of appropriate methods (examples listed in Appendix S3). This is very large topic beyond the scope of our ms, which focusses on describing the typology, its rationale and process for development, and its current and potential applications. We address specific comments on T7.1 below.

Specific comments on the human-dominated biomes and EFGs

T7: Intensive land-use systems

There is no definition nor criteria for “intensive” land-use. I find this particularly problematic as “intensive land use” is a boundary concept that can have very different meanings to ecologists than it has to climate modelers or land-use scientists. Not all agricultural and settlement lands would usually be considered “intensive” by land-use scientists. Since nearly all lands are used to some extent, it makes sense to consider the intensity of use to distinguish these systems from the other EFGs. However, land-use intensity is a multi-dimensional concept, with land-use scientists distinguishing intensity in terms of (for example) different inputs, outputs, and system-level properties (see e.g. Erb et al. 2013). It is unfortunate that there is no reference to such conceptualizations of intensity by land-use scientists. Ideally, it would be reasoned, from an ecological point of view, which of these dimensions are most critical to consider in the definition and mapping of these ecosystems, and if reasoning for any thresholds on either of these dimensions were given. The first sentence under “T7: Intensive land-use systems” gives pastoralism as an example. However, pastoralism specifically tends to be one of the less intensive ways of producing livestock (e.g. compared to intensive pasture management, etc.), and many wilderness areas of the world are in fact subject to mild pastoral use (think reindeer herders in the Russian taiga). In many pastoral systems, human intervention is arguably not a “dominating influence”, nor would those systems necessarily change their key ecological characteristics if they were no longer “maintained” by continuing human intervention (e.g. as suggested by the third sentence, rather, low livestock densities would be replaced by similarly low densities of other herbivore species). Pastoral uses are not mentioned in the respective “natural” EFGs (e.g. grassland, savannah, taiga systems), but those also do not really seem fit in here. Similarly, T.7 includes “urbanization” but does not explicitly mention “any other type of permanent human settlement”, which according to common definitions would not be included in “urban”. At local scales, however, even small villages within otherwise fairly natural environments would meet many of the other criteria for T.7. Should I consider a settlement in Siberia as T.7 or as Taiga?

AU RESPONSE 179: In general, we do not think it is appropriate to give definitive or prescriptive criteria for identification of individual units in the typology (natural or anthropogenic). The variation of ecosystems is even more complex than variation in land-use intensity, which Erb et al. (2013) note, ‘Given the complexity of land-use intensity, providing a single, unambiguous and encompassing definition or indicator of land-use intensity does not appear to be an adequate target.’ Acknowledging this problem, we added the following guidance to Appendix S4 (T7 Intensive land use systems):

“The intensity of human influence on ecosystems forms a continuum that is best assessed by multidimensional analysis of inputs, outputs, their interactions and alterations to system properties. However, most intensive harvest-based land use systems exhibit a high (> c. 40%) Human Appropriation of Net Primary Productivity (HANPP), an aggregate measure of alteration to ecosystem properties.”

As well as this additional guidance, we also note an important distinguishing feature of some intensive land-use systems – that use maintains them in disequilibrium states that may undergo large transformative changes in properties and processes when intensive use ceases. This

distinguishes intensive systems from low intensity systems, such as taiga with reindeer herding activity.

We thank the reviewer for pointing to our loose usage of the term ‘pastoralism’ in the text on p78. We replaced it with the following to distinguish the activity from low-intensity rangeland grazing:

“high-density grazing of domesticated livestock”

We clarify interpretation of villages under T7.4 below.

Further, the statement in the fourth sentence suggests that those systems generally have low endemism and low functional and taxonomic diversity, but in some parts of the world (e.g. much of Europe), it is precisely the maintenance of certain forms of land-use systems that maintains much of those regions current biota. Also, the statement is generally only justifiable with explicit reference to spatial scale of analysis. Locally (at plot scale), intensive land-use systems may almost always suppress biodiversity relative to original vegetation, but beyond the local, diversity could even be increased as even intensive land-use systems can add to the overall environmental/habitat heterogeneity of a system. It might be that a high taxonomic diversity that is to a larger extent composed of generalist species coping well with human-altered systems is not what the authors have in mind, but instead only the “natural” biodiversity. If that is so, then the authors should very explicitly lay out their criteria and their implicit assumptions about what constitutes “natural”, and make their underlying values concerning different facets of biodiversity explicit.

AU RESPONSE 180: We think low endemism is accurate. There are very few known taxa that occur only within a small area of an intensive land-use system and nowhere else in the world – most biota are cosmopolitan. We also think low functional and taxonomic diversity is a robust generalisation, though we agree that statement requires a clearer context. We made the following revision to text:

“...typically low functional and taxonomic diversity relative to comparable systems under low-intensity use, although taxonomic diversity can be higher in some groups in some systems.”

Some statements are overly comprehensive. For example, the fifth sentence (“Target biota are ...”) includes a diversity of examples, many of which are not exclusive to intensive land-use systems. For example, selective breeding occurs among reindeer herders in the Siberian taiga, which typically shows up as a wilderness area and is presumably not included as intensive land-use systems here.

Other statements are not sufficiently comprehensive. For example, “The antecedent ecosystems that they replaced include forests, shrublands, grasslands and palustrine wetlands (biomes T1-T4 and FT1).” – e.g. cropland on land claimed from the ocean would be excluded by this statement. It seems appropriate that the authors commonly use examples and non-definite statements (e.g. “typically, but not exclusively”, “many intensive land-use systems are”, etc.), as they try to capture very heterogeneous systems by these few groups. However, this can also make the classification of systems as either belonging to T.7 or to a “natural” group somewhat subjective. I guess what I would ideally love to see is some kind of identification key, but I acknowledge that this is not easily possible.

AU RESPONSE 181: We think the reviewer interpreted the text on these issues more definitively than the phrasing actually states. For example, “Target biota are genetically manipulated...” does not imply that no genetic manipulation occurs in other systems. Genetic manipulation does occur in

other systems, but genetic manipulation is much more common and becoming more intensive in anthropogenic ecosystems (such as T7), than in other systems subject to less intensive human activity. Similarly, “...include forests, shrublands, grasslands and palustrine wetlands...” does not **exclude** other alternatives, but these are the most common systems applicable to the generalisation. We added transitional marine systems to the list, but these are much less commonly converted to intensive terrestrial uses on a global scale:

“...and more rarely transitional marine systems (biomes T1-T4, FT1, MT1, MFT1)”.

The idea of an identification key is something we have considered carefully. However, the development of such a key requires substantial additional analysis, design and extensive testing among user groups. We plan to embark on this work to develop a robust key and include it in web resources for the typology in the next stage of development.

What seems more doable would be clear distinctions between attributes that are “must haves” vs. “can haves”. If this is not possible, I feel that this caveat should be clearly acknowledged, and the typology should not be oversold as being systematic and rigorous. “On global and regional scales, intensive land-use systems are engaged in climate feedback processes via alterations to the water cycle and the release of greenhouse gases from vegetation, soils, livestock, and fossil fuels.” – something similar could also be said for some other ecosystems with strong dynamics between alternative states. E.g. the Savanna-Forest transition systems in the African savanna belt (Staver paper).

AU RESPONSE 182: We appreciate that users desire clarity, but we do not think a formal division of ecosystem properties into “must haves” vs. “can haves” reflects on systematic qualities or rigour. The design principles, conceptual framework, definitions of concepts and terms (Glossary), development process, and extensive review process contribute to systematic qualities or rigour of our typology. There are many other systematic and rigorous classifications of nature that are not based on artificially definitive membership rules, recognising inherent multidimensional variability and uncertainties in ecosystems. Indeed, some quantitative methods embrace and accommodate uncertainty and variability among classes by assessing overall evidence on group membership across multiple dimensions where quantitative data exist, e.g. fuzzy clustering. We draw attention to the introductory text to the descriptive profiles (Appendix S4, p13 2nd paragraph), which we have revised as follows to further address this issue:

“Inevitably, there are inherent uncertainties in assigning ecosystem types to unique EFGs because ecological classifications, in general, simplify complex multidimensional variation in nature by segmenting and categorising continuous gradients in multiple features (see Appendix S3; Regan et al. 2002). Thus, any given ecosystem type may possess a suite of features that are typical of different functional groups, and a single feature can rarely be definitive for ecosystem identification (e.g. Erb et al. 2013). For this reason we avoid prescriptive approaches to description of the units that seek to identify strictly exclusive or diagnostic ecosystem characteristics, and instead use appropriate qualifiers and caveats in descriptions where important exceptions apply to generalisations about ecosystem properties and postulated drivers. Users should assess and weigh evidence on all features to identify the most likely functional group and report the nature of uncertainties in group membership.”

T7.1 Annual croplands

Ecological traits:

The characterization of ecological traits starts with a qualifying phrase “Structurally simple, very low diversity, high-productivity annual croplands are ...” – this phrase obviously does not cover all croplands. I did not find another class in which the other croplands that are by definition excluded (at least if a land-use scientist reads these definitions), such as permanent croplands, low-productivity annual croplands, etc. The authors should either add additional classes to their ecosystem typology or use more accurate terms and adequately comprehensive definitions for the existing ones. Again, a distinction of defining characteristics into can-haves (e.g. “supplementation of nutrients”) and must-haves (e.g. “artificial disturbance regimes”) would seem in order.

Swidden agriculture does not seem to fit in to any described EFG. T7.1 might seem to include swidden agriculture according to the statement “... or subsistence production of food”. But then again, this land-use form would definitely be excluded by the statement “structurally simple”, “high productivity”, and “When actively managed systems are abandoned [...] these non-target biota [...] become dominant and may form a steady, self-maintaining state or a transitional phase to novel ecosystems.”

Some statements are so definitely phrased that they are unlikely to be true. And even if they were, I’m afraid the burden of proof whether or not they are always true is on the authors when making such definite statements. For example, consider the sentence “... these systems have very low functional, genetic, and taxonomic diversity and no local endemism”. How can the authors be so sure that annual croplands never, nowhere in the world, have even a single locally endemic host-specific crop pest species that coevolved with the crop? Another sentence that should be phrased as a “can-have” is “Target biota coexists with a cosmopolitan ruderal biota (e.g. weedy plants, mice, and starlings) that ...”. There are indeed many localized examples of species with restricted global distributions that primarily use natural ecosystems, but benefit from nearby intensive farming – that these species can effectively exploit those intensive cropland systems may depend just as much on the landscape composition as on the specific attributes of the croplands or the species per se. E.g., think of *Macaca nemestrina* opportunistically leaving rainforests fragments to hunt rats in SE-Asian palm-oil plantations – I do not think that these would classify as “cosmopolitan ruderal biota”. Even orangutans have been observed feeding in palm-oil plantations. I do not doubt that palm-oil plantations by and large harm the latter species, but the very restrictive definition provided is still not always true.

AU RESPONSE 183: Thank you for these suggestions. We made several revisions to T7.1 to address the issues raised. We deleted “Annual” and simplified the name of the group to “Croplands” and in the text replaced “annual” with “...typically dominated by one or few shallow-rooted *short-lived* plant species such as...”. We deleted “Structurally simple, very low diversity” from the opening sentence. We qualified herbicide and pesticide application, “...and *usually* by periodic application of herbicides and pesticides...”. We placed generalisations about diversity in context, “...compared to antecedent ‘natural’ systems, croplands are structurally simple, have low functional, genetic, and taxonomic diversity and *little or no* local endemism,” and contrasted tropical croplands with industrial croplands as follows, “*Subsistence croplands, including Swidden rotation systems, are typically more diverse than industrial croplands.*” After the sentence on ruderal biota we added, “*Native biota from adjoining non-anthropogenic systems may also interact with croplands.*”

T7.2 Sown pastures and fields

Ecological traits:

Again, a better distinction of “can-haves” and “must-haves” would be necessary to make some of these statements accurate. The first sentence, for example, connects several traits by “and” that in the real world do not always co-occur. Many of the comments already given for EFG “T7.1 Annual croplands” also apply here.

AU RESPONSE 184: We revised the name of this group to “*Intensive livestock pastures*”, recognising that sowing, while widespread, does not apply to all examples. We added qualifications and revised the text as follows. We deleted “*low diversity*” from first sentence. We revised and qualified the sentence on diversity and endemism as follows: “*Consequently, compared to antecedent rangeland systems and semi-natural pastures, these systems have low functional and taxonomic diversity and little or no local endemism.*” We added “*or maintenance*” to “*...harvested by humans continuously or periodically for consumption.*” We added “*Typically, at least 40% of net primary productivity is appropriated by humans.*” We deleted “*lawns and sporting fields*”, which were incorporated as elements within T7.4. We added a reference to mixed systems as follows “*Livestock pastures may be rotated inter-annually with non-woody crops (T7.1), or they may be managed as mixed silvo-pastoral systems (T7.3)*”. Finally, we note that “and” applies to collective characteristics of the group as a whole, not (necessarily) to co-occurrence in space.

Key ecological drivers:

Again, without a distinction of “can-haves” and “must-haves”, the described drivers do not allow for a reliable identification.

AU RESPONSE 185: We qualified “*is supplemented*” by “*is typically supplemented*” and added that fertilisers may be “*...applied at varied rates*”, noting that these may be inorganic chemical or organic forms of fertiliser.

Map:

The indicated data sources for this EFG were a set of livestock density maps (Gilbert et al.) and maps of areas that are equipped for irrigation (Siebert et al.). The latter appears to drive much of the patterns in this EFG map, as it was generated by overlaying irrigation-equipped areas with “major” (presumably thresholded) livestock densities. I believe this mapping protocol to be flawed on several conceptual grounds. Firstly, the livestock densities were not designed to depict any specific type of grazing. They were modelled across all land-cover types to account for the fact that in many parts of the world, livestock does not occur on lands dominated by forage herbs, but instead, for example, in silvo-pastoral systems, mixed cropping-livestock systems, land-less industrial farms, and even inner cities. Secondly, with a few exceptions, the areas that are globally equipped for irrigation are mainly used for agriculture, not for creating artificial pastures. High livestock densities intersecting irrigation equipped areas are certainly not globally indicative of specifically sown pastures. In many areas with extensive rice irrigation (especially in India, but also in SE-Asia), there are mainly mixed cropping livestock systems. In these systems, the irrigation is exclusively for crop production, whereas livestock mainly graze on patches of grassy vegetation in between the field (but those are certainly not sown) and on the field themselves (after harvest). In other areas, land-less livestock production systems co-occur with irrigation-cropping systems at the coarse grains where they were mapped. The illusion that the irrigation is in any way related to the livestock, there, is merely a result of the spatially coarse nature of these maps. Finally, it is not clear from the description what thresholds

were applied for livestock densities. Global thresholding is generally tricky here, as livestock densities supported by a grazing system depend as much on its natural vegetation productivity as on the artificial inputs it receives. This can certainly be done, but would definitely require some way of locally validating the resulting map. For the above reasons, I do not believe this EFG map to have high validity, not even as an indicative map.

AU RESPONSE 186: We agree with the referee that these proxies are not ideal predictors of the spatial distribution of sown pastures. We applied the irrigation layer mainly to exclude rangeland grazing areas from the map but, as the reviewer points out, there is considerable potential for false positives. We removed the spatial analyses and research findings dependent on maps from the manuscript, as recommended by the editor (see Response 1). We improved the indicative map by removing the irrigation layer and combining global maps estimating livestock density with the estimated human-appropriation of net primary productivity. We explored a number alternatives for combining these data and selected options that represented uncertainties through major and minor occurrences consistent with the concept of the EFG. We explain the data sources and derivation in Table S4.1:

“Mapping of intensive livestock pastures was based on fractional land use mapping (Ramankutty et al. 2008), dasymetric estimates of ruminant livestock density for cattle and sheep (Gilbert et al. 2010), and Human Appropriation of Net Primary Production (HANPP, Haberl et al. 2007). Fractional land use cover indicated firstly where pastures occur and secondly where they occupy a large portion of area relative to croplands. This helped to exclude intensive croplands that are also used to graze livestock, either through temporal rotation or on the margins of cropped paddocks (e.g. in south Asia). Livestock densities indicated where ruminants were important components of pasture systems, and helped exclude some rangelands with low livestock densities. Finally, HANPP helped exclude low productivity rangelands with high stocking rates and additional areas of cropland. We explored different combinations and thresholds for the input data layers, visually inspecting the output in South Asia, Australia, West Africa, and North and South America. We then mapped major occurrences where pasture area fraction greater than zero ($PAF > 0$) and greater than cropland area fraction ($PAF - CAF > 0$), densities of cattle or sheep were greater than 500 per cell, and $100 < HANPP < 700 \text{ gC/m}^2/\text{yr}$. We examined the sensitivity of mapped area to variation in these thresholds and found no appreciatble change in the global mapped area when livestock density was varied by $\pm 20\%$ and marginal change in mapped area with variation in the other thresholds by the same amount. To represent this uncertainty, we mapped minor occurrences as the additional area where $PAF > 0$, $PAF - CAP > -0.2$ and $80 < HANPP < 840 \text{ gC/m}^2/\text{yr}$.”

T7.3 Plantations

This EFG seems mostly well done.

Thanks.

T7.4 Urban and industrial ecosystems

Ecological traits:

This EFG seems mostly well done. However, by definition, this EFG excludes non-urban settlements (villages, etc.). These also fit in none of the other classes, but cover substantial surface area globally. I see this as a major gap in the presented typology.

AU RESPONSE 187: Thank you for this suggestion. We changed the label from “Urban and industrial ecosystems” to “Cities, villages and infrastructure” and replaced “urban systems” in the text with “urban/industrial/village systems”.

Map:

I wonder why no map of actual urban areas was used but instead an estimation from artificial nightlights. It is well known in the urban mapping scene that settlement indications based on artificial nightlights are regionally biased by wildfires and especially energy production platforms. The provided description in the IUCN Typology document does not suggest that this was accounted for.

AU RESPONSE 188: As noted above, we removed the spatial analyses and research findings dependent on maps from the manuscript, as requested by the editor (see Response 1). Although we believe night lights provided an adequate proxy for distribution of this group that excludes transient wildfires, we replaced it with a more recent land use data set. Both night lights and land use data may exclude weakly lit or unlit villages, but these are likely to account for a relatively small area. The source and derivation are described in Table S4.1:

“The distribution of urban and industrial infrastructure lands was taken from a global land use/land cover map (LULC class 7 ‘built areas’) for the year 2020 at 10 metre resolution (Karra et al. 2021). Class 7 includes major road and rail networks, large homogenous impervious surfaces including parking structures, office buildings and residential housing, dense. Sparse villages may not be represented. We calculated the proportion of built area per square kilometre and applied a threshold of 1 to 5 % for minor occurrences and >5% for major occurrences.”

T7.5 Derived semi-natural pastures and old fields

Ecological traits:

The description of the ecological traits of this EFG is more convincing than some of the others in this biome. However, some of these statements do not apply generally but are presented as if they did. For example, they are presented as generally being “structurally simpler than the systems from which they were derived”. This does not seem to be universally true, as in some cases, structurally highly homogeneous natural vegetation may be replaced with more complex mosaics of pasture and leftover original vegetation (small groups of trees, etc.). Adding a simple “typically” would help here. “Productivity [...] is generally [...] more stable than more intensive anthropogenic systems”. I would question the extent to which this claimed generality is really established.

AU RESPONSE 189: We think a sentence about structural complexity at site scales may have been misunderstood as a comment on landscape spatial complexity. We clarified the text as follows:

“Although structurally simpler at site scales than the systems from which they were derived, spatial complexity may be greater in fragmented landscapes and...”

We reasoned that productivity should generally be more stable in derived grasslands than in more intensive land-use systems because derived grasslands are generally less often and less intensively harvested, ploughed, fertilised, etc. than those other systems.

Map:

This EFG is not included in the Methods description in the IUCN Typology document, so I tried to speculate what this might have been, but frankly, I failed. It is puzzling to me how a meaningful map of this EFG should be derived from the indicated data sources. Those sources are a consensus landcover map distinguishing only coarse land-cover types (Tuanmu & Jetz) and a map of approximate areal boundaries of ecosystem extents (Dinerstein et al.), neither of which contains any information on the semi-natural character of any of the grasslands shown therein. Exploring the “indicative map” of this EFG and comparing it with the Tuanmu & Jetz data looks like this map was driven largely by the Tuanmu & Jetz class “Cultivated and Managed Vegetation” (e.g., close correspondence in patterns clearly visible for India, China, West Africa, Argentina, etc.). That class almost basically reflects indications of “cropland” in the different data sources that underpin the Tuanmu & Jetz consensus product. Without having access to any concrete description of the mapping protocols, I can only assume that this class might have been misinterpreted as showing “managed vegetation other than cropland”. Based on the information I have, I can only say that this map appears highly flawed. I do not believe that it has any validity even as an indicative map.

AU RESPONSE 190: Thank you for pointing out the omission from Table S4.1. We have now added the entry for T7.5 and used alternative sources data that more directly represent the suitability and use of these lands for grazing without intensive interventions that characterise T7.2. This group was especially challenging to map. For the indicative global map, we identified a plausible envelope of occurrence by identifying high suitability for grazing (based on Erb et al. 2007), excluding areas that Erb et al. (2007) estimated to have i) a higher proportional area of cropping or forestry; ii) a low proportion of grazing lands; and iii) a high HANPP. We reasoned that i) and iii) should exclude harvested crops (e.g. cereals), plantations and intensively managed sown pastures, while ii) should exclude wild low-productivity rangelands used for low intensity grazing. We are currently exploring other alternatives that sharpen the focus on the features that characterise this group of systems. We removed the spatial analyses and research findings dependent on maps from the manuscript, as recommended by the editor (see Response 1).

SF2. Anthropogenic subterranean freshwaters biome

I cannot comment on this biome, as this is outside my expertise.

F3. Artificial wetlands biome

From my assessment, there are fewer conceptual flaws in the definitions of EFGs in this biome than in the terrestrial biome. In particular, the descriptions of their traits and associated biota seem more nuanced than those for terrestrial human-modified EFGs. But my own expertise is also more in landuse systems than in water-use systems, so in case this is critical for the decision, I would suggest to consult an expert on artificial freshwater systems.

AU RESPONSE 191: We note that five freshwater specialists reviewed the profiles of Ecosystem Functional Groups within the Artificial freshwaters biome (see Appendix S5), and at least one of the eight reviewers for Nature was a freshwater specialist.

F3.1 Large reservoirs

Map: I believe the indicative map to be of little utility. There are much more complete datasets of dams and reservoirs available than the cited one: <http://globaldamwatch.org/>

AU RESPONSE 192: Thank you for drawing our attention to this update. We removed the spatial analyses and research findings dependent on maps from the manuscript, as recommended by the editor (see Response 1) but have included the revised map as a thumbnail in the profile for F3.1 in Appendix S4.

F3.3 Rice paddies

This EFG seems mostly well done.

Thanks

F3.2 Constructed lacustrine wetlands & F3.4 Freshwater aquafarms & F3.5 Canals, ditches and drains

Map:

The basis for mapping both types of EFGs has been the Freshwater Ecoregions of the World datasets by Abell et al. The criteria by which the maps freshwater ecosystems have been filtered to those “indicative” of either of these EFGs is unclear. I also think this has been with little grounding in actual evidence. Both EFGs certainly exist outside those indicated areas.

I do not understand why freshwater ecoregion maps were used for mapping these EFGs in the first place, as those ecoregion maps were not designed to capture signs of human alteration of “natural” systems. To quote from Abell et al.: "Ecoregions are intended to depict the estimated original extent of natural communities before major alterations caused by recent human activities". In any case, the coarse nature of these maps would arguably render their utility very low.

AU RESPONSE 193: We agree that freshwater ecoregions provide a poor template for mapping these Ecosystem Functional Groups. We removed the spatial analyses and research findings dependent on maps from the manuscript, as recommended by the editor (see Response 1) but we explored additional alternative options. All of these options have significant limitations, but are likely improvements on freshwater ecoregions. The global surface water database with small artificial water bodies intersected with grazing lands provide an indication for F3.2. F3.4 relies on incomplete and largely inaccessible industry information, and thus remains a very general envelope of potential occurrence. Maps of irrigation infrastructure and urban areas enabled an improved map for F3.5.

M4. Anthropogenic marine biome

I am missing an EFG for non-submerged marine artificial structures (e.g. artificial islands, oil platforms, etc.). I cannot comment in detail on the conceptual validity of these EFGs, as these systems are mostly outside of my expertise.

AU RESPONSE 194: Artificial islands and supratidal components of oil platforms are included within MT3.1 Artificial shorelines. The marine component of oil platforms is within M4.1.

M4.1 Submerged artificial structures

This EFG seems mostly well done, though I cannot confidently comment on the utility on the validity of the indicative map. It does seem empty in many parts of the world with substantial industrial activity in coastal areas, so it seems unlikely that this is very complete.

AU RESPONSE 195: We agree, this map is likely to be incomplete, as many wrecks and much marine infrastructure is not centrally documented or is undocumented. We removed the spatial analyses and research findings dependent on maps from the manuscript, as recommended by the editor (see Response 1) and noted the likely incomplete status of the map in the Distribution text,

"Map is incomplete but shows areas with many documented wrecks and marine infrastructure."

M4.2 Marine aquafarms

This EFG seems mostly well done.

Thanks.

MT3. Anthropogenic shorelines biome

I cannot comment on this biome, as this is outside my expertise.

References

Boitani, Luigi, Georgina M. Mace, and Carlo Rondinini. "Challenging the scientific foundations for an IUCN Red List of Ecosystems." *Conservation Letters* 8.2 (2015): 125-131.

Erb, Karl-Heinz, et al. "A conceptual framework for analysing and measuring land-use intensity." *Current opinion in environmental sustainability* 5.5 (2013): 464-470.

Erb, Karl-Heinz, et al. "Land management: data availability and process understanding for global change studies." *Global change biology* 23.2 (2017): 512-533.

Referee #7 (Remarks to the Author):

In this ambitious paper, the authors have attempted to distill a truly global classification scheme for ecosystems that can be used to guide international management, such as maintenance of biological diversity under the Aichi targets, by emphasizing and drawing on established ecological theory. The authors include a litany of supplemental material, lasting many of hundreds of pages, supporting their expert classifications that emerged from an international working group that initially convened in 2017 and considered two dozen existing typologies. They go on to show how their new classifications relate to protected status and human impacts, suggesting that fewer of these ecosystems are protected and more are vulnerable to human impacts than is needed to meet current sustainability goals.

In full disclosure, I did not review the first iteration of this manuscript but was brought on as additional referee with marine expertise. What the authors have produced is noteworthy based on sheer volume alone. But I honestly cannot say having read the paper and some of the appendices, I know exactly what the novel contribution is here, who is the intended audience, or how it is useful or improves on existing classifications. Some of this content is more clear in the response to the reviewers.

AU RESPONSE 196: We address these issues in the summary in the main text of the manuscript and elaborate in the appendices.

Novelty: This is the first time a consistent conceptual model has been applied to produce an ecosystem classification, integrating functional and compositional properties, across the entire biosphere (lines 70-77, 103-116 and 201-211 in the main text; Appendix S3 pp21-23). A significant advance and novelty of our study is that it places all ecosystems into a single theoretical context. We note several other novel aspects of our approach, including the integration of top-down and bottom-up hierarchical construction commended by Reviewer #2. This integration achieves global consistency in upper levels (1-4), while integrating established local classifications at lower levels (56) that benefit from considerable investment in data and expert knowledge (lines 439-442 in Methods text; Appendix S3 p21).

Audience: We summarise applications in the main text (lines 198-209) and elaborate in Appendix S6. We demonstrate already wide uptake of the typology with adoption by IUCN (1400 member organisations including 200 national governments), adoption as a reference classification by the UN Statistic Commission in the new system for ecosystem accounts (UN SEEA-EA) and as the methodological basis for headline indicators for ecosystems in the draft monitoring framework for the post-2020 CBD Global Biodiversity Framework. There are already examples of several applications at national levels. We expect many users who are active between in knowledge transfer between ecosystem scientists and managers (see detail in Appendix S6).

Usefulness: The typology was explicitly designed to support ecosystem management for dual goals of biodiversity conservation and sustainable development (lines 95-97 and 114-116, main text). We developed and applied six design principles to ensure the typology is fit for purpose. In lines 98-109 and Fig. 3 we summarise diverse applications of the typology and elaborate on its usefulness in Appendix S6.

A primary goal appears to be to produce a globally consistent and integrative classification scheme for ecosystems (Appendix S4), which is rooted entirely in ecological theory and therefore unbiased by existing perceptions or treatments of these biomes. They adopt an environmental filtering-limiting similarity approach, working from large earth systems (marine, terrestrial, atmospheric) down to individual biomes whose constituents—and therefore biodiversity—are determined by similar abiotic and biotic drivers. Putting aside for the moment that the notion of environmental

filtering and competitive exclusion can lead to both the same and wildly different assemblages (see transformative work by Jonathan Levine and Margaret Mayfield published in 2010 in Ecology), I am still struggling to understand how this new approach leads to categories that are fundamentally different than what we have always been managing for, e.g., coral reefs, temperate forests, etc. and therefore why this new typology is warranted. Or, rather, how including a comprehensive appendix crossing realms is any different than copy/pasting together 3 separate documents pertaining to terrestrial, marine, and freshwater systems. In other words, there is nothing synergistic about bringing these realms together under this new framework; it is merely the sum of its parts.

AU RESPONSE 197: Our approach reproduced some widely familiar ecosystem groups (coral reefs, several groups of temperate forests, etc.) and some ecosystem groups that were not widely recognised, but nonetheless with unique functional properties (e.g. subterranean ecosystems, as commended by Referee #5, subglacial lakes, abyssopelagic waters, etc. etc.). The typology would not be viewed as successful if it did not reproduce widely recognised ecosystems with large bodies of research and management activity such as coral reefs and various types of temperate forests. We used familiar vernacular names as labels for new functional descriptions, framed by the assembly model, to leverage disciplinary knowledge. A significant advance and novelty of our study is that it placing all of these ecosystems (familiar and unfamiliar) into a single theoretical context. The unifying conceptual framework advances previously disparate classifications for different segments of the biosphere, providing consistency to support practical applications by facilitating systematic syntheses, comparisons, and knowledge transfer. We adapted the community assembly model to focus thinking on convergence of functional properties of ecosystems and their components, rather than assemblages of species as prescribed in the original model. Our ms makes no claims about synergisms – like ecosystems themselves, a hierarchical typology is the sum of its parts and their interactions.

In fact, I can see one instance where the resulting categories have clearly suffered from pre-existing biases and do not necessarily follow the proposed scheme: seagrasses (M1.1) and kelps (M1.2). Both are marine primary producers inhabiting shallow coastal shelf regions, provide vertical structure that support diverse food webs, are limited by light, nutrients and oxygen, have similar dispersal strategies (forgiving reproductive differences, both use spores/seeds to disperse over large distances), and are subject to the same environmental (e.g., temperature, storms) and biological pressures (e.g., grazing), including human impacts (e.g., nutrient run-off, urbanization). They only differ in their geographic distribution (with seagrasses abundant in both temperate and tropical zones whereas kelps are largely temperate and restricted to a few areas in the tropics), but this is not a criterion in their Table S1.1. In fact, both kelp and seagrass ecosystems were recently grouped under “marine macrophytes” in a review for the Global Ocean Observing System (see Duffy et al. 2019 in *Frontiers in Marine Science*). I am wondering, given the description in Appendix S2, why these systems were kept separate? It seems that they would logically cluster together, as has been done by GOOS.

AU RESPONSE 198: Although we agree with referee #7 that there are some functional similarities between kelp forests and sea grass meadows, there are also important differences in ecosystem properties and drivers that justify recognition as separate functional groups, consistent with design principle 1 (Table S1.1). First, kelp forests are confined to cold waters and their declines with increasing sea temperatures suggest greater sensitivity than sea grass meadows, which are widely distributed across tropical and temperate waters. Second, kelp forests are almost exclusively associated with

hard substrates, whereas seagrass meadows occur on soft unconsolidated sediments. This has major implications for characteristic fauna (abundant sessile benthic organisms in kelp forests, abundant burrowing organisms in seagrass meadows) and response to disturbance. Third, there is a large literature on strong top-down trophic regulation of kelp forests, which is pivotal to ecosystem dynamics. Top-down regulation also occurs in sea grass meadows, but its influence on ecosystem dynamics is less profound. Fourth, upwelling of cold nutrient rich waters is a strong driver of kelp forest ecosystems (i.e. kelp dominance) and highly influential on their distribution and renowned high primary productivity. Sea grasses display different responses to nutrient availability and are not strongly associated with upwelling. Fifth, where sea grass meadows and kelp forests occur on the same coastlines, they are segregated parapatrically (kelp forests on higher energy subtidal rock platforms, sea grass meadows in lower-energy soft-sediment embayments), reflecting the influence of different assembly filters. Other functional differences include propensity for vegetation recovery, photosynthetic sensitivities to different wavelengths due to different chlorophylls, etc, etc. Finally, these functional distinctions are reflected in an abundant and strongly segregated literature. A quick WoS search (topic = kelp, topic = seagrass* or both) revealed 6184 papers on kelp, 12648 on seagrass, and only 252 (~1% total) on both. Many of the latter concern a passing comparison between the two contrasting system types, or resource subsidies between the ecosystems when juxtaposed. Given these functional distinctions, and that the two groups of ecosystems have distinguishable remote sensing signatures on a range of platforms, it is perplexing and disappointing to learn that GOOS lumped them. This type of misconception will not assist management of either ecosystem, and exemplifies a problem posing a barrier to management and monitoring that the IUCN Global Ecosystem Typology can help to resolve.

Nevertheless, such an overarching typology ignores the fact that most if not all of the management occurs not at the global (planetary) scale, but at the regional or even local scale, with rare exceptions (e.g., European Water Directive). I am not aware of management plans that consider globally broad categories and goals, but generally set targets based on regional historical baselines. Seagrasses in Florida are not managed the same as in Vancouver, or Norway, or Australia, even when attempting to meet international targets (such as Nationally Determined Contributions under the Paris Climate Agreement). As another example, pelagic fisheries stocks in the Atlantic are managed very differently than in the Pacific due to the much longer period of exploitation, even by the same country. I am not convinced by the current presentation of the benefits of plugging into this hierarchical topology (which again, leads to the same management entities, or EFGs, as previous schemes). Perhaps this is useful at the ultimate level, i.e., for international governance (IUCN), but the authors did not state who their audience or userbase is, so this is merely speculation on my part. (note: they elaborate more in the response to the reviewers, but this detail is not included in the manuscript, presumably due to space constraints).

AU RESPONSE 199: We substantially revised the summary and introductory paragraphs of the main text to more clearly articulate the need for the typology, and hence the audience:

Lines 67-74: "Ecosystems vary in their biota (4), service provision (5), and relative exposure to risks(6), yet there is no globally consistent classification of ecosystems that reflects functional responses to change and management. This hampers progress on developing conservation targets and sustainability goals. Here we present the International Union for the Conservation of Nature's (IUCN) global ecosystem typology, a new conceptually robust, scalable, spatially explicit approach for generalisations and predictions about functions, biota, risks and management remedies across the entire biosphere."

Lines 76-80: *“...a unifying theoretical context to guide transformation of ecosystem policy and management from global to local scales. This new information infrastructure will support knowledge transfer for ecosystem-specific management and restoration, globally standardised ecosystem risk assessments, natural capital accounting and progress on the post-2020 global biodiversity framework.”*

Lines 95-97: *“To serve dual needs for sustainability of ecosystem services and biodiversity conservation, ecosystem assessments require a global typology to frame comparisons and inform data aggregation for analysing ecosystem trends and diagnosing their causes.”*

We summarise applications in the main text (lines 186-221) and elaborate details on eight groups of applications (some already underway, others with potential for future implementation) in Appendix S6, each with their own user communities.

We specifically acknowledge that most ecosystem management and conservation action occurs at local levels (Methods text lines 422-424, Appendix S6 Ecosystem monitoring and management section).

The typology is designed support this local action in two ways. First, it provides a systematic framework for scaling up local data to assist countries in meeting their reporting obligations on CBD targets, SDGs, ecosystem accounts, world heritage outlook reports and other international commitments to which many countries are signatories (Appendix S6 pp2-3 for details). In Methods, we note (main text lines 422-424):

“... the lower levels are crucial for representing biodiversity in the typology, but also have important roles in scaling up information from established local-scale typologies (Appendix S3, pp19-20).”

Second, it facilitates knowledge transfer on management of functionally alike ecosystems. (Appendix S6, p5):

“Grouping ecosystems that share common mechanisms of response to environmental change, anthropogenic threats and remedial management actions (Fig. S6.2) establishes a powerful basis for adaptive management to reduce risks of ecosystem collapse (Keith et al. 2011; Williams 2011). Moreover, such a classificatory system provides a framework for information storage and retrieval and knowledge transfer by drawing attention to experience on nature-based solutions for managing functionally similar ecosystems...”

We outline a specific example for seagrass meadows (Appendix S6, p6), which share similar threats from intensifying levels of shipping and boating with associated turbulence, turbidity and uprooting of the structural dominants by anchors of recreational and commercial vessels. New designs of boat moorings that lift chains above the seafloor and citizen science initiatives to assist restoration of degraded meadows are developing with applications by local authorities in eastern Australia (see papers by Tan and colleagues in FMARS). The typology can support local seagrass managers in other regions by facilitating transfer of this knowledge and justify investment and implementation of effective strategies on sandy coasts and embayments that are exposed to similar threats. The typology would serve this purpose less effectively if functionally contrasting ecosystems were grouped together. Kelp forests, for example, are generally not exposed to such threats and require different local and regional management strategies to address a different suite of threats.

More generally, we draw attention to the knowledge transfer capabilities in the main text (lines 216224):

“Ecosystem groupings based on convergent drivers, properties and environmental relationships will reveal similarities in threats, mechanisms of degradation and, therefore, ecosystem-specific management strategies for recovery. Embracing the dynamic nature of ecosystems and its dependency on ecological processes is a key feature that differentiates the IUCN Global Ecosystem Typology from other ecological typologies (Table S1.2). This will enable policy and management actions to be targeted at causes of ecosystem degradation, with knowledge transfer and adaptive learning [40] about local ecosystems from functionally similar ecosystems elsewhere (Fig. S6.1).”

Indeed, the new scheme fails to yield new insights into the value of protection or response to human stressors. We already know that marine protected areas are generally concentrated on the coast, are therefore biased towards particular foundational species (e.g., corals), and are nominally effective depending on the size, isolation, enforcement, and age of the reserve (see review by Edgar et al. 2013 in Nature). We also know the unregulated areas that are under heavy exploitation (e.g., deep-sea mining) are also under greatest threat. There is not a conclusion here that is not echoed in other recent syntheses, or present novel statistics that integrate across realms in a way that is not possible by simply combining the results of these individual studies (again: this typology is just the sum of its parts).

AU RESPONSE 200: We removed the spatial analyses and research findings from the manuscript and Appendix S6, as requested by the editor (see Response 1).

I think the major issue is that, while 20 previous typologies did not consider ecological constraints EXPLICITLY, they do so implicitly. Terrestrial forests are different than kelp forests: we can see that based on where they are found and what they look like, captured by their evolutionary history. Clearly, an explicit consideration of ecological notions of filtering and competition has not led us to any different classifications or, if it has, the authors have not made that contrast clear. The paper would benefit immensely if the authors could contrast the novelty of their EFGs with those in other schema. I also find the suggestion that because this typology is already used by some entities and others have found it useful it should be published is rather weak: there are undoubtedly thousands of national and international schema that do not find their way into the pages of Nature because they are not stimulating groundbreaking science worth of the highest impact journal in the world, even if they are getting the job done.

AU RESPONSE 201: We think the reviewer posits an overly simplistic perspective on the purpose and value of our contribution by referring to glib differences between kelp forests and terrestrial forests. Response 193 summarises our purpose supported by multiple references to the main text and appendices that explain why the purpose is important.

Some of the problems in applying many of the other ecological typologies to ecosystem management relate to their implicit (cf. explicit) treatment of ecosystem functions. Implicit treatment of functional properties assumes that they will fall into line with other features (e.g. biogeography, biophysical patterns, etc.). Very few of the typologies that we reviewed justified that assumption. The exceptions are confined to one segment of the biosphere (e.g. Moncrieff et al. 2016 addresses only terrestrial areas, Appendix S1), are challenged by other constraints on their approach, and have failed to generate wide uptake. When representation of ecosystem function is included as a primary design principle, and interpretation of functional groupings is supported by a

conceptual model based on ecological theory, we argue that an ecosystem typology is much more likely to represent contrasts in salient functional features among its units than one that relies only on implicit connections with function (lines 108-110):

“Ecological classifications based on tested and established theory are more likely to be robust to new information than classifications based only on observed patterns and correlations.”

Beyond our systematic overview of existing typologies against design criteria to establish the need for a new approach (Appendix S1), we see little benefit in comparing the detail of EFGs with classification units of other ecological typologies (noting that only a small subset of them are ‘ecosystem’ typologies). Such an exercise is likely to reveal some similarities (some fundamental, some coincidental) and a number of differences, but equally unlikely to produce important insights. The only plausible conclusion from such a comparison is that differences reflect different purposes, inputs and developmental settings, some of which relate to disciplinary barriers. As a rare cross-disciplinary collaboration of specialists in different ecosystem types throughout the biosphere, the IUCN Global Ecosystem Typology transcends these barriers.

In sum, I wish I could be more positive about this paper. Much of the content is useful review: the original Reviewer 1 is correct, the massive appendices could be better served as a technical report (and I see will be published in a forthcoming IUCN report). But I also question the value of this new framework and whether it adds anything substantial to our understanding or our capacity to meet global sustainability goals based on the current presentation. The world is now facing a proliferation of schema, typologies, classifications, and all manner of guiding frameworks: while this manuscript is undoubtedly an achievement and represents considerable consensus, I was not convinced it is, as we are always searching for, a “better mousetrap.”

AU RESPONSE 202: We thank the reviewer for his/her consideration. We fully agree with the sentiments about, “proliferation of schema, typologies, classifications, and all manner of guiding frameworks”. Setting out on this endeavour, our intention was to adopt an existing typology that was most fit for purpose in relation to globally and locally established needs (see Response 193). Only after completing the review summarised in Appendix S1, we realised that no existing framework came close to fitness for purpose and that a new approach was needed. We hope our contribution is seen as a step change in capacity for ecosystem management, rather than simply adding to a proliferation of superficially similar classifications.

Reviewer Reports on the Second Revision:

Referees' comments:

Referee #1 (Remarks to the Author):

I have reviewed this ms since its first submission. I believe the authors have done a good job of addressing concerns and it is basically ready to publish.

My only suggestion is that the GIS files of the 108 regions be published with this manuscript even though the majority of "spatial analysis" (i.e. amount of each ecoregion that is protected and rate of human change) is regrettably but correctly postponed to another ms. I think the typology will be hard for anybody else to use without these GIS files (and this should become the anchor point for the use of the typology).

I understand concerns came up in review (I had one of them about being raster). But the utility so vastly outweighs the downsides of providing nothing that I think they should be published with appropriate caveats and limitations.

Otherwise, I think this is an exciting project.

Referee #2 (Remarks to the Author):

I thank the authors for their considerate replies to my comments. I am satisfied with the replies. I also took the time to read all 93 pages of the rebuttal document, and commend the authors for their care in addressing everybody else's comments, some of which I very much agree with in retrospect.

I think the paper now reads much better. Focusing on the typology and presenting it in more detail (and with more nuance) has in my view given the paper a more solid foundation. I appreciate the more expanded Methods section and the clearer figures. I am convinced this framework will become a landmark in the ecology and conservation literatures, and I thus recommend its publication.

Some comments and a few additional recommendations (optional):

In contrast with Figures 2 and 3, Figure 1 looks quite amateurish. Which is a shame, as this is arguably the most important figure, and one likely to be reproduced many times in the future. Any chance you could ask the people who did Fig 2 to help making this one a bit sleeker?

The new Figure 2 is a major improvement in relation to the previous version. It is very useful to see the full list of Realms and Biomes, and then examples of Ecosystem Functional Groups for most Biomes. I think this figure will be key to help the reader's mind navigate back and forth from conceptual to concrete. I predict it will be a key figure associated with this new typology in countless future documents.

Minor points/suggestions regarding Figure 2:

- 1) In some cases, the EFGs' rectangles (on the left side of the figure) appear merged (e.g., 8-9-10 in Lakes).
- 2) Good idea to represent three examples of multidimensional environmental gradients as arrows on the figure. To better drive home the message that these are independent axes, I would have drawn the top one (temperature) diagonally, rather than vertical as for light & nutrient availability.

Figure 3 is a nice addition too.

Regarding the very long Table 1, I think Referee 1 was right that this was missing from the previous version, but I find it much less crucial now that the new Figure 2 lists all Realms and Biomes, and illustrates some Ecosystem Functional Groups. As the authors point out, this table is really long. I would instead move it to Appendix S4, where I think it would make a useful summary (particularly if it included hyperlinks to each EFG).

Instead of Table 1, I would recommend adding an additional figure going into more detail for some of the EFGs from Fig 2, still in the spirit of helping the readers understand the concrete aspects of this conceptual framework. Specifically, I would recommend a figure with two (or three?) horizontal panels, each illustrating an EFG, and including: a brief summary of its typical key features, a (sleek) diagram of the drivers, and a little map with the distribution (and maybe the corresponding drawing from Fig 2, creating a link between the two figures). This would strengthen the value of the main text as a stand-alone piece, and encourage readers to go and see Appendix S4 for more detail.

Referee #3 (Remarks to the Author):

I read with much interest the revised version of this manuscript. I must first praise the authors for the work they have done during the revision of the manuscript (it is certainly not easy to deal with so many thorough reviews and comments!). I have carefully revised the response letter provided, and I feel they have effectively incorporated most of the criticisms I provided. Please note that I have not reviewed in-depth appendices other than S4, and within this one I focused on the typologies I revised in the previous section, so I assume that other reviewers will have specifically revised them.

Focusing on the novel typology itself has been a very good choice, as this makes the manuscript more focused and removes some major issues with the analyses included in the previous version. Overall, I think the ms has substantially improved and is easier to navigate, and I have not identified any major issue with this version of the text. Said so, I have some (mostly minor) comments, focused mainly on the structure/clarity of the text and its presentation, which I would advise the authors to consider:

- While the paper is now more focused, I found that the structure/order of the paragraphs makes the text not very attractive to read. Of course, I acknowledge that this is matter of style (and thus largely subjective) but I would encourage the authors to think about the structure of the text to

make it as attractive as possible. For example, some paragraphs would certainly benefit from an internal reordering (e.g., L103-116, see my suggestion below for this paragraph; it would be also good to add references when referring to existing classifications in this paragraph to provide a better context and to facilitate readers the comparison of the newly introduced typology with existing ones), others break the logical flow of the text. For instance, paragraph in lines 267-275 would be better place before the previous paragraph, so the final paragraphs of the ms can focus on the strengths (rather than on the limitations) of the proposed classification.

“We used these six design criteria to review a sample of 23 global-scale ecological typologies, finding none that explicitly represented both ecological functions and biota (Table S1.2). This limits the ability of ecosystem managers to learn from related ecosystems with similar operating mechanisms and drivers of change. Many of the existing typologies either failed to describe their units in sufficient detail for reliable identification, or required diagnostic features that are hard to observe. Others were based on biophysical attributes or biogeography, but approaches differed across terrestrial, freshwater and marine domains, precluding a truly global approach. Only three typologies encompassed the whole biosphere, but these lacked a clear theoretical basis, limiting their ability to generalise about properties of ecosystems grouped together. Ecological classifications based on tested and established theory are more likely to be robust to new information than classifications based only on observed patterns and correlations, which may prove unstable when new information emerges. In this study, we developed a global ecosystem typology that meets all six design criteria, thereby providing a stronger foundation for systematic ecosystem assessments, sustainable management, and biodiversity conservation”

- Title: The title could be more dynamic and appealing to readers, what about "A function-based typology for preserving and sustaining Earth's ecosystems"?

- L 74-75: I think “The outcome of a major cross-disciplinary collaboration” does not add much here, could be removed to save space

- L 82: I feel that “irrespective of specific biota within the ecosystems” is a little bit misleading and may confound readers as the processes and mechanisms that underpin ecosystem functioning rely on biota. Rewrite or delete this part of the sentence.

- L 78-79: perhaps this sentence could be better framed as “The classification introduced here can guide policy transformation for ecosystem-specific action, including ... (list some of the key actions this classification may be particularly useful for)”

- L 85: This sentence is confusing. To which functions do you refer (in the previous sentence you mention functions, biota...)?

- L 86: I found the definition of ecological processes given in the Glossary (“Activities that result from interactions among organisms, and between organisms and their environment (after Pettorelli et al. 2018)”) somewhat confusing. What do “activities” mean in this context? Please rewrite this definition for clarity or further explain it so everyone will understand the meaning of this important term (it is used multiple times throughout the text). Also, what is the difference between “ecological

processes” and “ecosystem processes” (the first term is defined, the second not but is embedded within other definitions, e.g. that of “ecosystem functions”).

- I know this does not have an easy solution, but Table 1 seems too large to be part of the printed main text. The editors will know better what would be a suitable alternative, but an interactive version online, which could also allow adding pictures for each EFG seems the best solution for this Table. Please also note that the * present together “Typical key features” is not defined, something that may confound readers (like me!).

- In Appendix S4 there is a mention to Appendix S7 (page 15) that does not exist. Should it be Appendix S6?

- In Appendix S4, I would also define “major occurrence”, as done with “minor occurrence” This will help readers to better interpret the maps provided. I would also include the justification you provided in the response letter “Omitting minor occurrences from maps, or merging minor and major occurrences (in which the EFG comprises the majority of the landscape matrix) would involve significant loss of information, notwithstanding acknowledged limitations on accuracy and precision” as part of Appendix S4 to further justify the approach followed.

- The new name of biome T4.5 is now more appropriate and will avoid many confusions, I think. I have not checked all the biome names (surely the authors will have done it), but please check them to minimize having names that would mislead readers (as it happened to me with the previous version of the T4.5 name).

Again, I applaud the authors for the effective revisions conducted and I hope that this new set of minor comments will be helpful to further polish and improve the presentation of this new typology, which undoubtedly will interest scientists, managers and policy makers alike around the world.

Referee #4 (Remarks to the Author):

The manuscript by Keith et al. describes the IUCN global typology of ecosystems, based on the functional traits of the species and ecosystems. In this era of global change and human impacts there is a dire need to make global comparisons of ecosystems, the impacts they receive, and their threats. As I said before, the manuscript has many strengths and will be widely used, and for these reasons it merits publication in Nature:

- 1) a unifying global typology that allows to classify bewildering different ecosystems that are often assessed separately (e.g., terrestrial, freshwater, marine),
- 2) a consistent classification based on community assembly and the functioning of organisms and ecosystems, which allows for a better mechanistic understanding, assessment and prediction of the consequences of environmental change,
- 3) a separate inclusion of humans as an environmental driver, which allows to assess the consequences of human activity for the biodiversity and functioning of the planet, and allows to design policies to change human activities or mitigate their effects,

4) the application of the typology by describing and mapping the 108 ecosystem types. The description in Appendix S1 with one page factsheets are a pleasure to read, as they are nice and concrete, succinct, well written, conceptually consistent by showing the same conceptual diagram with different drivers, and nicely illustrated with a clear beautiful photo conveying the message, and reference for further reading. This is a treasure for many biologists and interested readers to appreciate and understand the diversity and beauty of life, and will be a global reference for at least the coming 1.5 decade.

The authors have done a thorough job in replying to all comments, in a rebuttal letter of 93 (!) pages.

MAIN TEXT IS UP TO PUBLICATION STANDARDS. The main text has been substantially improved by taking out the threat analysis, and using the gained space to highlight the strengths, limitations, and potential of their classification. The main text is therefore now much more balanced and relevant because the cornerstone of the classification (the conceptual model) is better explained and now very well justified. It is now also crisp and clear because all jargon has been defined and also explained in a Table. The current version of the main manuscript is a pleasure to read and up to publication standards

MAJOR AND MINOR COMMENTS SOLVED. All my major issues have been solved, or in case the authors had a different opinion it is sufficiently well justified and I can live with. Specifically, I highly appreciate it that they have improved the conceptual diagram, cleaned up the terminology, and made it internally consistent. The difference between ecological traits and ecosystem traits is now also solved. Most of the minor comments have been solved as well.

A PHILOSOPHICAL QUESTION. I was specifically asked to comment on the description of tropical ecosystems. I am a functional ecologist and have worked and studied 30 years of my life in tropical forests over the world. Intriguingly, I was less convinced by the authors reply to my comments about their functional ecosystem descriptions; about 1/3 they implemented, about 1/3 they did not implement and I live with their counterarguments, and about 1/3 they did not implement and I was not convinced by their arguments and disagreed. Of course I acknowledge that it is difficult (and near to impossible) to accommodate and condense all people's priorities understandings and opinions about an ecosystem in one page. But the philosophical question to the authors is what this means?:

1) their functional, process and trait-based classification of ecosystems is less useful then they claim as even functional ecologists seem to disagree and have a different understanding of how these systems function?

2) Keith has worked mainly in tropical forests in Australia and I have worked mainly in West Africa and the Neotropics, we both have a pertinent understanding of our systems, but the relevant differences are only partly captured by the functional classification (tier 3), and more by biogeography and the peculiarities of the biota (tier 4-6)?, or

3) this functional classification is wonderful because by using the same functional ecosystems and jargon it facilitates comparison, highlights apparent discrepancies, and therefore facilitates discussion, spurs new research, and advances science?

I guess that all 3 reasons may partly explain what is going on.

KEEP THE MAPS. I concur with the plea of the authors to keep the maps when there is sufficient resolution for the scale at which they are shown (i.e., small global maps), as 1) it makes it more explicit to the reader what kind of ecosystems they refer to, 2) you can put it into an ecological context (i.e., whether it makes ecological sense given macroclimate, soils, biogeographic barriers), 3) you can see their convergent nature across the globe, and 4) it spurs interest to do comparative research. Without doubt these maps will be improved over time, but that is with all science we do. As long as the limitations are indicated it is totally fine.

GREAT PAPER AND PLEASE ACCEPT. I think the paper provides an important, timely, and solid contribution. I would therefore kindly suggest the editor and reviewers to accept the next revised version without sending it again out for review. We now have had 7 reviewers, 2 revision rounds, the manuscript has been expanded with 5 long encyclopedial appendices for further support, the authors have well justified their choices and indicated the limitations, and have written a careful rebuttal letter of 93 (!!) pages.

When in some cases issues have not been solved then I think it is fine, because it is difficult and near-to-impossible to accommodate everybody's ideas and concerns in 2500 words (or whatever the word limit is for this kind of Nature article). The main messages of the paper are solid, and over the lesser issues we should be able to agree to disagree as, after all, it is the authors paper and story and not the one of the reviewers. So I hope the editor and reviewers can be flexible, I congratulate the authors with this tremendous effort and result, and I am looking forward to see the paper in print.

With kind regards,
Prof. Lourens Poorter

Referee #5 (Remarks to the Author):

I have read the revised manuscript and the responses to reviewers and think that the authors have done an excellent job in taking into account the many points that were raised on this round of review. I hope that the editors realize the scope of this work and the fact that no system set up to classify all the global ecosystem types will satisfy everyone. Part of the issue is that many different sub-fields of biology and policy are represented in the materials the authors have produced, as well as the reviews this paper has received so far. There were some weaker points and the authors wisely agreed to remove them from this specific version of the paper. These can be addressed in their future work, and removing them does not weaken this specific paper.

The work reminds me of when the human genome was published. Many authors were involved, and the genome was incomplete, but the paper served as a milestone for a huge scientific advance. The paper was essentially descriptive, but that in no way diminished its importance. The approaches in that paper were further refined and the approach became even more useful as that happened.

Classification of functional types of ecosystems is an essential step in managing and conserving the Earth's ecosystems in a time of global human pressure (the Anthropocene). This is true both with respect to the preservation of biodiversity for its own sake as well as the maintenance of ecosystem services upon which humanity depends for its ultimate comfort and survival. A unified approach to such classification is an important step forward in our efforts to understand the natural world.

The authors have adequately responded to all of my specific points. I hope my input has strengthened the work and its utility. The authors should be congratulated on the breadth and the ambition of the work; this is a truly important contribution and represents a very significant scientific advance.

Referee #6 (Remarks to the Author):

Please see attached PDF.

Referee #7 (Remarks to the Author):

This is my second time reviewing this paper, and while I definitely appreciate the authors' hard work and thoughtful replies (I imagine it was quite difficult to generate the framework and consensus among so many experts and over a planetary scale, and 94 pages of reviewer replies is another manuscript in itself), I am beginning to appreciate that I am not the target audience for this article. While I have done a number of cross-system syntheses on ecosystem functioning and services, and currently work with local, regional, and international governments to meet biodiversity and climate targets, I just don't see the utility of the proposed typology for, or how it would significantly inform or improve the context of, my work. As the IUCN has already accepted this standard, I guess its not really my place to comment on the utility of the framework, since clearly it has already been vetted and accepted by a broad target audience of scientists, managers, and policymakers.

That said, as a scientific article, I still wonder how this will be used by the broader scientific community to shape the discourse around effective conservation and evaluation, which is a concern echoed by several other referees. I think what bothers me most about this article is that, on the surface, it is really attractive but ultimately rings hollow. As a classically-trained community ecologist, I love the idea of using, for example, niche partitioning or ecological interactions to define biomes. But as I dig deeper, the manuscript does not satisfyingly pay off on the promise of a truly integrative framework. Much of the framework is informed by expert opinion (lines 346-347) and the authors admit that truly empirical tests have yet to be conducted on assembly processes, linking multiple drivers, and so on (eg, lines 225-233, directly on lines 230-231, 250-252, 256-266). I agree it is valuable to draw attention to these deficiencies, but if there is a "gold star" example the authors could share, it would be a long way of convincing me that this framework can be fulfilled to the fullest extent rather than hypothetically promising.

I can also see biomes where this framework would fail to yield robust generalisations, as promised: "Ecosystem groupings based on convergent drivers, properties and environmental relationships will

reveal similarities in threats, mechanisms of degradation and, therefore, inform development of ecosystem-specific management strategies for recovery.” Take a widely-distributed marine habitat in seagrasses. Tropical seagrasses will probably thrive under climate change as their metabolisms ramp up and more CO₂ is available for photosynthesis (with some exceptions), while temperate species have already experienced significant decline and range retraction as a result of rising temperatures exceeding physiological thresholds. Even within a species, such as eelgrass, Atlantic populations are more threatened by climate change than Pacific ones, based on divergent evolutionary histories and consequent differences in biogeographic extent. The contrasting drivers within this system or even geographic region is summarized nicely in Table 1 of the recent “Out Of the Blue” UNEP report solely on seagrasses. I imagine that with any framework that is so broad, it would be possible to find exceptions to the rules (as pointed out by other referees). I am just concerned that the rules are so vaguely defined, seeing where they are usefully applied (that “gold star” example) would help negate the exceptions that immediately leap to mind.

The manuscript is also somewhat contradictory: in the very first sentence, the authors suggest the goal of “Sustaining ecosystem functions and services requires an understanding of ecological processes [irrespective] of specific biota.” They then immediately say: “[Ecosystem] functioning not only underpins biomass production, but also depends on [biota.]” (line 81). They then go on to say their 108 EFGs are specifically defined by “biotic composition” (line 176). So, sustaining functions is directly a consequence of invoking specific biota, as they do in their lengthy summary table. In the introduction, the authors also mix terminology, alternately referring to “ecosystem function,” “ecological processes,” and “ecosystem properties,” which include species traits (echoed by R4). The new glossary differentiates processes as those involving “interactions among organisms” although such processes, primarily consumption, have long been regarded as ecosystem functions (such as in reviews and meta-analyses by Hooper et al. 2005 and Cardinale et al. 2006, 2012). On a minor note, I would consider traits to be properties of the community, whereas ecosystem pertains to the general flow of energy among discrete compartments (eg, primary producers to consumers) (R4 also seems to agree on this point).

I could go on, but it would perhaps be a moot exercise as, based on other referees’ comments, I seem to be missing the utility of the contribution here. Without sufficiently compelling examples of applications (which the authors note in their reply to R7 that “There are already examples of several applications at national levels” but are seemingly mum on those), I can really only see myself or my colleagues citing this work to justify the discrete ecological units under investigation and how they may or may not be related (for example, through shared drivers or shared responses). Is that worthy of publication?

Referee #8 (Remarks to the Author):

I have been sent this manuscript to review from the perspective of an evolutionary ecologist. As will be seen below, I am comfortable with this manuscript from that perspective. I note that the manuscript has had MANY reviews already, and beyond that there has been extensive peer review of the whole typology, as is well documented by the authors. I do have some comments, which I place below.

Sincerely,
Kyle Dexter
University of Edinburgh

First and foremost, I think the hierarchical system they have developed makes good sense from an evolutionary ecological perspective. I agree with the authors' world view that biomes and ecosystems should be functional. I also agree that contingencies in the biota present in an ecosystem can modulate its function. Lastly, I agree that one key purpose of developing an ecosystem typology is to share lessons, for management or otherwise, in how ecosystems in different places may have similar behaviours. Thus, having ecosystems that span continents is almost a pre-requisite for the system, and allowing division of these core ecosystems (level 3) based on biogeographic region / biota (in level 4) present is appropriate. Yes, the biota can drive variation in how a level 3 ecosystem functions on different continents, but like the authors and most reviewers, I think the commonalities of these EFGs across continents generally supersedes among-EFG differences.

I think people who work in the field of eco-evolutionary feedbacks (and argue that many evolutionary processes operate on the same timescale as ecological processes) won't like the text in lines 130-131 about the longer timescales and lower relevance of evolutionary processes. But, I would tend to think that the evolutionary processes that play out on ecological timescales are not sufficiently powerful to create major shifts in ecosystem function to cause one EFG to shift to another (on ecological timescales). Thus, I am comfortable with what some might perceive as 'short shift' given to evolutionary processes in this context.

I have read through the main text, the response to previous reviews (including that by R2 who suggested an evolutionary ecologist) and read most of the appendices. I did not find anything in there that I find very problematic from an evolutionary ecological perspective.

I now shift to a few criticisms/comments, as I could hardly call myself an academic if I did not have some critical comments.

MAJOR COMMENTS

1) As has been commented by other reviewers, I do find this article to be a bit odd as an Article for Nature. For me it is more of a views and perspective type of piece than a research article per se. I have little doubt that the article will be well cited and also draw attention to the new typology, the latter being a good justification for publication in Nature.

My view that this is more of a perspective piece than a research article is down to the fact that ultimately I felt it was an expression of expert opinion. While the pool of experts consulted is large, it does not represent a formal, data-driven assessment of what functional biomes and ecosystems are. As the authors recognise, such comparable data across functional biomes and ecosystems are rare. Still, I think it could be useful for the authors to point a way forwards, by more clearly articulating

the critical need for comparable ecosystem function data across biomes and ecosystems (e.g. that generated by the GEM project for terrestrial forest systems, based at Oxford). As the authors point out, their functional biomes and ecosystems are hypotheses to be tested, and the typology will be revised. The authors could give readers some suggestions on how these hypotheses could/should be tested.

As an example, In AU153, they write "First, high levels of heterogeneity within groups can be addressed by segmenting continuous gradients of vegetation into a larger number of groups, with a trade-off through increased complexity of the typology. An important point is whether variation in properties is greater between groups than within them at the same level of classification. Greater heterogeneity within groups than among them would justify a trade-off to recognise more groups." This is the kind of analysis that is needed to test their hypothesised EFGs.

2) Another consequence of generating an expert-opinion based system is that the outcome depends on the experts involved. I don't want to nitpick at the typology as it has already received extensive scrutiny by people more qualified than myself. Rather, it is worth reflecting on the dual facts that Australia comes up as having multiple unique ecosystems (Temperate pyric humid forests, Hummock savanna, Sclerophyll hot deserts and semi-deserts) and that the lead authors and many of the authors are based in, or from, Australia (NB: I only looked at terrestrial EFGs). What would this typology have looked like if led by African authors, or South American authors? I am not suggesting changes to the system, but I wonder if a comment somewhere around the contingency in the outcome depending on experts involved might be useful.

3) I liked the mapping of major and minor occurrence as it helps communicate the fact that biomes/ecosystems can have spatially interdigitated distributions, that bits of one major EFG can be found within an expanse of another and that a given location in space can be in one of multiple alternate ecosystem states. While one would think such things are obvious to anyone who has been in mountain landscapes (or paid attention to lakes or forest-savanna transitions), this complexity seems to escape many users of biome/ecosystem maps. I think it could be useful for the authors to emphasise this issue more clearly in the main text of the manuscript somewhere. As they are no doubt aware, once they make raster layer(s) available for their EFGs, end-users will use the products to 'definitively' assign points in space to a given biome or EFG (e.g. when assigning a species collection record to occurring in a given biome/ecosystem). This is common practice with current biome data layers, at least in the terrestrial realm, and while it may lead to correct assignment much of the time, it does result in errors that have consequences for downstream interpretation and understanding. Again, even though some end-users will do this no matter what the authors write, I think the authors are in a good position to make a strong cautionary statement about assigning biome/ecosystem just based on lat/long in their paper.

4) Like R3, I felt the language seems unnecessarily complex in many places. R3 gives many examples, so I will only mention a few examples. The authors note in their response to R3 that curing is a synonym of drying, and that certain ecologists/managers use the term curing. If drying is a synonym of curing, why not use drying, which is a more easily understood term for the large majority of readers. Elsewhere, the authors refer to sessile photoautotrophs and note that they are vascular plants. Why not just use the term vascular plants? I think nature articles are meant to be accessible

to a broad audience, including physicists, mathematicians, social scientists, etc.

This language complexity also enters in the names of some of their EFGs. For example, I don't feel like the term pyric tussock savanna will be very accessible to non-savanna ecologists or other users. There is a classic division in Africa between moist/mesic and arid savannas (Huntley 1982). As their maps for 'pyric tussock savannas' and 'trophic savannas' largely match those previously conceived savanna types, why not use those terms that are more accessible and have precedence? If not the classic mesic versus arid, it could be 'fire-limited' versus 'herbivore-limited' savanna, which might also be more accessible to a broad audience. In general, it seems preferential to use accessible terms with precedence.

5) Like some of the reviewers, I am a bit uncomfortable with the assertion of the biotic processes that are important in each EFG, given the biases around what processes have been the focus of study in different EFGs and the expert opinion nature of the whole process. I think such statements are useful, but authors must be aware that this article will be used as cited evidence that such processes are important in X EFG, and that evidence will be accepted because the article is published in Nature. Thus, there is the real potential to create what some call 'zombie ideas' (c.f. Moles and Ollerton 2016 Biotropica and blog by Fox 2011 cited therein). Is there a way for the authors to show more caution around these assertions, in specific instances, or in general?

MINOR COMMENTS

For functional biomes, forests are separated around a temperature divide, but 'savannas and grasslands' and 'shrublands and woody shrublands' are not. Why is that?

Temperate pyric humid forests vs Temperate pyric sclerophyll forests and woodlands? Should these really be distinguished? [NB: This relates to the unique Australian biome comment above]

None of the EFGs have the word Mediterranean in their name. Many terrestrial researchers think of Med ecosystems as one of the main kinds of ecosystems in the world, and at least a couple of the EFGs seem to largely be found in Med areas. The absence of the term is a bit conspicuous.

Like R2, I don't like the term ecosystem collapse (line 162). I agree with R2 that this is actually just a form of ecosystem transition. Even if used by IUCN in a way to mean transition, most readers will not take it that way.

R2 did not like EFG. Why not use FEG? 'FEGs' rolls off the tongue a bit better than 'EFGs'.

I really like the conception of the term 'ecosystem properties'. I think it will be very useful.

Using ecoregions to split up Level 3 groups into Level 4 seems too fine-grained to me. Many ecoregions (e.g. in the Amazon) are not really that different.

Line 104: Do the authors mean ecosystem functions here rather than ecological functions? Elsewhere, the authors contrast ecological processes and ecosystem functions, so it is a bit confusing to then refer to ecological functions.

Line 148: I would use 'influence' or 'channel' rather than mitigate. The use of mitigate here could be interpreted in a normative fashion.

Line 433-434: A good example of this aggregation based on compositional resemblance can be found in Silva de Miranda et al. 2018. GEB. Apologies for self citation!

I review this paper for the second time. As requested before, I focus on the quality of the presented typology and paper with respect to its human-related components. In summary, I find that many elements have improved a lot since my last assessment, but I am still not happy with how the authors have addressed (or not) some of my earlier comments.

I focus my new comments (green below text) on whether or not concerns raised by me in the previous round (black text) have been adequately addressed by the authors' responses (blue text).

There is a tremendous quality gap between the typology itself and the presented data and analyses. Unfortunately, a central argument of this paper is that this typology, specifically in combination with spatial data on ecosystem functional types (EFGs), can support detailed global analyses and therefore has great utility for tracking progress against global policy targets and other important applications. I am afraid that the flaws in the data and the presented applications undermine these claims. The paper provides evidence on the soundness of the typology itself, but it does not provide strong evidence for the conclusions from the exemplary analyses. I believe that the typology requires substantial alterations of, and expansions on, some of its human components. The spatial data of "indicative" global distributions of EFGs (an intermediate level in the presented hierarchy), which are presented as being associated with this typology, are in some cases very well chosen, but in many other cases these data are extremely crude, and the underlying data have in many cases very large and generally well-known uncertainties, that unfortunately are largely ignored here. In several cases, the chosen source data are conceptually inadequate for representing the respective EFG. The limited reliability of the presented indicative maps is verbally acknowledged through their presentation as "indicative", but there is generally no robust validation of even these indicative abilities, which is indeed questionable in some cases. Moreover, this verbal acknowledgement is undermined by using them for a set of spatially explicit global analyses for which they are not fit-for-purpose, which might even mislead other scientists or decision-makers about the quality of these data. By failing to account for these limitations, the presented analyses using this typology and associated data have severe methodological flaws, and the evidence provided is not strong enough to make the results trustworthy, which is the main reason why I do not provide an overall more positive assessment.

AU RESPONSE 147: We accept that a proportion of the maps are not suitable quality to support a spatial analysis and have deleted the analysis from the ms as suggested by the Editor. We now acknowledge this limitation more explicitly in the main text (Lines 240-253):

"Thirdly, the indicative global maps for ecosystem functional groups vary substantially in accuracy and precision. High resolution global maps of ecosystem functional groups are pivotal to important applications of our typology (e.g. global synthesis for reporting on CBD targets). Many other uses, however, are national in scope, specific to particular ecosystem groups or biomes (e.g. forests, coral reefs, mangroves) or non-spatial (i.e. using the typology to frame context for knowledge transfer and generalisation), and thus do not require comprehensive and globally consistent maps. Maps that are most fit for purpose would be based on remote sensing and environmental predictors that align closely to the concept of their ecosystem functional group, incorporate spatially explicit ground observations and have low rates of omission and commission errors, high spatial resolution and time series of changes. While many current maps fall below one or more of these specifications, maps available for two-thirds of the 108 functional groups are of a high-intermediate quality suitable for global spatial analysis (Appendix S4, pp14-22). These represent an advance on global ecosystem distributions relative to proxies such as ecoregions, and new data sets are rapidly emerging."

During this round of review for Nature, newly published studies and critical input enabled updates to maps for eight of the EFGs. More broadly, we draw attention to an important aspect of our approach (Lines 254-256):

"By decoupling the mapping process from prior development of the classification, our approach liberates the definition of ecosystem units from constraints imposed by current availability of spatial data and allows for progressive improvement (Appendix S4, p15)."

We removed all downstream analyses and focussed the revised manuscript on the typology itself (see Response 1). We note comments which are now not applicable in our response below.

- The authors now claim (lines 250-252) that “While many current maps fall below one or more of these specifications, maps available for two-thirds of the 108 functional groups are of a high-intermediate quality suitable for global spatial analysis [25](Appendix S4, pp15-24)”.
- This statement is unsubstantiated, as neither the cited reference nor the information in Appendix S4 provides actual evidence for these claims. I personally firmly disagree with that statement. There is currently simply no assessment of the quality of nearly all of the presented ecosystem maps. Therefore, whether or not these maps might be of an adequate quality for supporting particular applications is speculative at best, and IMO highly doubtful considering the known quality problems in many of the gridded products that were used. Importantly, it is completely unclear what the authors even mean by either “global spatial analyses” or by “intermediate-high quality”. But regardless of what may be meant by the latter, I believe that without any formal validation of the presented maps, any suggestion of either the maps’ quality or applicability would be irresponsible. Also see my related responses to AU RESPONSE 151, AU RESPONSE 160 and AU RESPONSE 193.

Given the limitations in data that is available today, I am not convinced that this typology can indeed, at this point in time, support the claimed applications like global monitoring of conservation targets or national ecosystem accounting.

AU RESPONSE 148: These limitations apply to some of the map data, not the typology. The typology has already begun to support several applications as noted in Appendix S6, including a number of applications that do not depend on comprehensive coverage of high quality global maps. We discuss this in revised discussion on lines 241-246:

“High resolution global maps of ecosystem functional groups are pivotal to important applications of our typology (e.g. global synthesis for reporting on CBD targets). Many other uses, however, are national in scope, specific to particular ecosystem groups or biomes (e.g. forests, coral reefs, mangroves) or non-spatial (i.e. using the typology to frame context for knowledge transfer and generalisation), and thus do not require comprehensive and globally consistent maps.”

Following adoption by the IUCN in 2020, the typology was also adopted by the United Nations Statistics Commission in 2021 as the reference classification for ecosystem accounts in the UN System for Environmental Accounting – Ecosystem Accounts (UN SEEA-EA, noting that existing national maps can readily be related to the global typology for international reporting of national accounts (see Appendix S6). We recognise the need to improve globally consistent map data as soon as possible, and are progressively doing so – six maps were updated during the journal review process since the previous submission to Nature in October 2020.

- I appreciate that the authors added some more nuanced language and also the fact that there are ongoing updates.
- Still, the paper continues to claim that the new typology can currently support processes such as global ecosystem accounting under the SEEA or ecosystem monitoring under the Post-2020 framework. But the current text remains too vague on how exactly those processes would be enabled or supported by the presented typology.
- The presentation of the typology as “spatially explicit” and as associated with spatial information in the main text makes it sound as if anything spatial about the ecosystem would be supporting of these processes. However, as I stated in my earlier comments, maps alone with temporal change information (especially if those maps are merely indicative) are useless for either of those processes. As the authors themselves acknowledge, “High resolution global maps of ecosystem functional groups are pivotal to important applications of our typology (e.g. global synthesis for reporting on CBD targets)”. Together with the acknowledgement that such maps currently do not exist for most EFGs, I do wonder in how far and via which mechanisms the typology could support those processes at this point (i.e., rather than in some distant future when many more things might be mapped).

- Is the mechanism through which this typology would be useful then simply be that its EFGs can be used to summarize whatever types different countries are anyway, so that these can be formally synthesized in global reports? This may be useful, but is not the big leap forward, as previous IPBES assessments and the like already found ways of synthesizing globally heterogeneous information. Whatever the proclaimed mechanisms through which this typology may support those global accounting and monitoring processes are, this should be more clearly stated in the main text.
- The authors also claim (line 276) that “The hierarchical structure [...] enables global imperatives to be linked directly with on-ground nature-based solutions” – without concrete examples or the mechanisms for this being explained, this looks like an arbitrary claim. I could still see how the typology’s nested structure could enable local monitoring to be synthesized more easily into broader aggregate reports. But that global mandates formulated by, say, the CBD, could directly feed into local-scale solutions to real local problems, just because we have a hierarchical typology, seems like a stretch.

I would recommend the authors to more carefully think through several of the descriptions of humandominated EFGs, and to only associate those EFGs with spatial data where the claim that those indicate their global distributions has been confirmed through some kind of formal validation. In addition, I would recommend to do either of the following options:

- 1) sell the typology as a stand-alone scientific achievement without associated analyses,
- 2) restrict the analyses to validated data that are fit-for-purpose for these analyses, which would likely mean restricting them to only some EFGs and maybe only some regions of the world, or
- 3) invest in substantially more rigorous global analyses that appropriately consider the limitations in the used global data, and provide concrete evidence (sensitivity analyses, formal validation, or similar) that the presented results are robust to their limitations.

AU RESPONSE 149: Thank you for the suggested options, we revised the ms according to Option 1 (see Response 1). We address comments on anthropogenic ecosystems below.

- I find the authors’ choice of Option 1 very sensible, and I believe the quality of the presented paper improved tremendously as a result of this change.
- Still, I had also commented in my earlier comment that the authors should restrict the presentation of indicative maps to those EFGs for which the maps’ indicative character has indeed been confirmed via some type of formal validation, which the authors did not do. I continue to seriously doubt that some of the maps have even indicative value. As I also state elsewhere, maps should either be validated or not be presented in pixelated form at all.

Originality and significance

The typology is novel and generally well-conceived. I would consider the “natural” components of it, and at least some of the human-dominated components, both important and highly useful for ecologists, conservation biologists and some closely related fields. The paper will be mainly of interest to these audiences. I do not think the typology will be of substantial interest to neighboring where a conceptually sound and consistent classification of human-used systems is more crucial, such as land use science, agricultural economics, urban studies, forestry, agricultural science, or similar. The typology is strictly ecologically focused, which is fine for purely ecological applications but will make it less useful for interdisciplinary discourses in a broader sustainable development context, given that the boundary concept of “ecosystems” is used very differently across fields (such as setting of integrative targets for advancing SDGs, etc.). To be more usable to researchers accustomed to diverse other typologies, crosswalks like those in tables S3.3 and S3.4 would ideally be presented for all the typologies critiqued here.

AU RESPONSE 150: We agree that the IUCN Global Ecosystem Typology may not suit all purposes focussed on specific land uses, such as agricultural economics, urban studies, etc. These were not part of the design objectives, and would (as Referee #6 points out) require a different approach. The Introductory paragraphs of the main text and Appendix S1 endeavour to make it clear that the primary purpose of this typology is to support reporting, knowledge transfer and decision making for

dual imperatives: 1) biodiversity conservation; and 2) sustainability of ecosystem services. These stem from the UN Convention on Biological Diversity, focussing more narrowly on biodiversity than the UN SDGs, which encompass environmental, cultural, social and economic dimensions.

The focus of our typology on ecological processes and ecosystem functions sets it apart from land use classifications (and also from biogeographic classifications such as ecoregions), which are designed to be fit for different purposes. We think it is appropriate that different tools are designed to serve contrasting purposes well, rather than a one-size-fits all classification that is suboptimal for some or most purposes because of trade-offs in the design objectives. We added the following discussion to Appendix S1 (p6) to elaborate on the scope of the ecosystem typology, as it relates to land use classification:

“We also draw a distinction between ecosystem typologies and land use classifications. Given fundamentally different purposes and trade-offs in design principles, the IUCN Global Ecosystem Typology will not be fit for all purposes that a land use classification can serve, and vice versa. Therefore we did not include land use classifications in our review of ecological typologies because they are classes of human economic, social and cultural activity that reflect the types and intensity of interactions between humans and their environment (Erb et al. 2017; Mayfroidt et al. 2018). Land use classifications typically group systems with low land use intensity into broad land cover categories based on plant life form, cover, height, and micropattern (Di Gregorio & Jansen 2000); attributes that are, at best, indirect proxies for ecosystem function. They classify higher intensity land uses on the basis of different criteria, for example, cultivated area are partitioned by plant growth form, field size and spatial distribution, crop combination and cover-related cultural practices (Di Gregorio & Jansen 2000). Other land use classification and mapping approaches estimate the intensity of different land use types (e.g. Erb et al. 2007).

In a broad sense, the anthropogenic components of the IUCN Global Ecosystem Typology address similar themes to some land use classifications. For example, land use intensity mapping by Erb et al. (2007) corresponds broadly to five anthropogenic ecosystem functional groups in the terrestrial biome of the IUCN typology: cropping (T7.1 Annual croplands); grazing (T7.2 Sown pastures and fields, T7.5 Derived semi-natural pastures and old fields); forestry (T7.3 Perennial crops and plantations); and infrastructure (T7.4 Urban and industrial systems). However, anthropogenic ecosystem types (defined as those that are created and sustained by intensive human activities, see Glossary, Appendix S4) should not be confused with land use activities. For example, grazing and forestry activities occur at a wide range of intensities across natural systems, but in anthropogenic ecosystems (T7.2, T7.3), they occur at transformative intensities, associated with qualitatively different ecosystem properties and organisational processes that may not persist when the activity ceases.”

That said, we considered it crucial to encompass anthropogenic ecosystems (cf. land uses) in the typology, as these are now an important component of the biosphere, and relevant to the underlying goals of the CBD. We consulted with FAO land use classification specialists in the development of the descriptions of these units (UN SEEA Expert forum June 2020, Appendix S5, p2). Level 3 of the Global Ecosystem Typology includes 16 Ecosystem Functional Groups within six biomes (Level 2), a comparable but slightly greater level of detail to the global FAO Land Cover Classification System, which recognises eight units in the third level of its Dichotomous Phase, including four human-dominated units (Di Gregorio & Jansen 2000). Similarly, as noted in Appendix S1 text above, Erb et al. (2007) map the intensity of four land activities types that correspond broadly to five anthropogenic ecosystem types in the terrestrial realm.

Descriptions of Ecosystem Functional Groups recognise variability within them. For example, the description of T7.4 Urban systems (Appendix S4, p80) notes that this functional group includes a number of elements:

“These elements include: a) buildings; b) paved surfaces; c) transport infrastructure; d) treed areas; e) grassed areas; f) gardens; g) mines or quarries; h) bare ground; and i) refuse areas.”

- I agree with the authors' response that different application fields require different typologies, and I think it is indeed a fair choice to design this typology specifically for ecology- and biodiversity-focused applications.
- Yet, many prominent statements in the paper (incl. the title Abstract line 71, first paragraph of Appendix S1, last paragraph of section "Reporting on global goals and targets" in Appendix S6) are arguably misleading, then, as they suggest that this typology could also support "sustainability" applications, which, however, would require using concepts that can be clearly linked to those used in neighboring fields with which ecosystems interact (e.g., climate, water, food sciences, etc.). As this was not assured, I believe the term "sustainability" should be avoided, and related statements in above-mentioned sections should be toned down.

Although it is in large parts beautifully conceived, I am not sure if the typology can be of immediate, practical relevance to many scholars even in ecology, because the majority of ecosystems are thus far not directly mapped, that is, as remote sensing based or other validated maps at resolutions where ecologists typically need to work, and given that for many of the EFGs, the discussed options for producing such maps are not overly convincing (see below). This fact alone makes me wonder how many scientists will actually adopt it. I do not consider this a limitation to its long-term utility, as it is still helpful to present a sound typology now that may guide future data collection. But this will likely limit its short-term impact. Except for few cases where temporal data are already available, it simply cannot be readily adopted for monitoring at this point (one of the main application fields claimed).

In this respect, I think there needs to be a more honest representation of the "spatially explicit" character of the typology. And even then, for all the anticipated temporal uses of the typology (ecosystem accounting, monitoring CBD targets), spatially explicit simply would not be enough. I am missing a nuanced statement in the discussion on what can currently, can maybe at some point, and maybe can never be mapped as a timeseries using deep time satellite images. The authors seem to expect that most EFGs can be mapped globally and annually. Reading the sentence in the discussion on new AI technologies and deep time satellite archives, I expected several examples, but apparently, there are only two (tidal mudflats and glacial lakes; the habitat map does not look like one).

AU RESPONSE 151: The typology was developed to support a broad range of uses related to ecosystem management and research (see Appendix S6). Although maps are essential to some major applications (e.g. global CBD reporting), many uses are national in scope, specific to particular ecosystem groups or biomes (e.g. forests, coral reefs, mangroves) or non-spatial (i.e. using the typology to frame context for knowledge transfer and generalisation), and thus do not require a full set of globally consistent maps. High-quality, high-resolution global maps currently exist for a quarter of the (27 of 108) Ecosystem Functional Groups, maps of intermediate quality exist for a further 45% (48 of 108) EFGs, while the remaining 30% (33 of 108) of EFGs rely on low-quality indirect proxies (see Table S4.1 and preceding text in Appendix S4). Most maps in the first two categories are validated and peer-reviewed, and thus suitable for monitoring, and some are supporting applications which the typology can place in a global context of other ecosystems (e.g. Global Mangrove Watch).

- I do not find this response convincing. To cite from this paper's Appendix S4: "We found matching data sets for 27 EFGs comprising either polygons or rasters (e.g. MT1.2, T7.4, M1.3; Table S4.1) or point records (e.g. F3.1). For eight of those EFGs, we supplemented direct maps with biogeographic regions likely to contain minor occurrences (e.g. TF1.1)." – This is the only reference to a subset of 27 of the 108 EFGs I found, and frankly, this does not provide any proof of the "high quality" of those 27 maps. Similarly, the authors claim "intermediate quality" for EFG maps for which they write "For 34 EFGs that had no direct mapping, we assembled maps from simple combinations of remote sensing and/or environmental proxies, clipped by biogeographic regions where necessary". Given the well-known issue of oftentimes very low classification accuracies in global maps, I find those quality claims not only entirely

unsubstantiated but also dangerously misleading. The authors should IMO either present credible evidence of those maps' qualities (either assessed by themselves or quality information compiled from the original sources) or they should restrain from any claims of their quality.

Of those maps suitable for monitoring, a smaller number are advanced data cubes based on extended satellite time series. We give two contrasting examples of these in the main text (tidal mudflats and glacial lake), but others include several groups of lakes, sea ice, non-perennial rivers and streams, coral reefs, mangroves... (i.e. there are quite a few more than two, and the number is growing at a rate that surprises even us). Several new maps that have become available while our ms is in review have been incorporated into our archive (doi: 10.5281/zenodo.4018173). In further work, we are preparing a separate publication with a detailed review of map standards suitable for different applications.

- It is puzzling to me what the authors might mean by "maps suitable for monitoring"? As I also stated elsewhere, monitoring does not require maps, but time-series. Apart from what the authors describe here as "advanced data cubes", none of the other maps can support assessments of how EFGs change over time, so they are of not use for either ecosystem monitoring and annual ecosystem accounting.

Further, a number of 'bottom-up' subglobal maps (e.g. national vegetation maps) have already been linked to the global typology by local experts (see examples in Table S3.3 and S3.4; others include South Africa, Canada, USA, Finland, India, Australian states), enabling a range of fine-resolution spatial conservation planning applications in an international context. We elaborated the text on strengths and limitations of maps in the main text (Lines 241-267) – see Response 147.

As Reviewer #6 notes, the typology also establishes a vision and frames an agenda for future mapping efforts, highlighting groups of ecosystems, such as savannas and deserts, where global-scale mapping is currently poor. This agenda is rapidly advancing, with several relevant new data sets emerging in the past 1-2 years. We continually update our map repository in response to these advances, and note in the main text (lines 261-265):

"...recently-developed data cubes for a diverse range of species habitats [46] suggest that global high-resolution time series mapping should be possible for most ecosystem functional groups within the next decade. Future versions of the typology will progressively strengthen map standards and improve applications that depend on spatial analysis."

- As already stated in my previous assessment, there simply are no "recently-developed data cubes [*i.e.*, *time-series*] for a diverse range of species habitats [46]". The reference cited in that statement (now ref. #37) published a static map of different habitats for the year 2015. It promised that these maps may be extended into a time-series in the future, but this has not happened to date, so the associated sentence in lines 261-263 is misleading.

In general, I would love to see some more careful discussion of the strengths and weaknesses of this typology.

AU RESPONSE 152: We substantially expanded discussion of strengths, weaknesses and aspects for future development of the typology in the main text, focussing on four main issues (see Response 4): assembly models for each Ecosystem Functional Group (lines 226-234, Response 9); modifications to the classification of units within the hierarchical framework (lines 235-240); development of ecosystem maps (lines 241-267, see Response 151); and discrete representation of continuous patterns in nature (lines 267-276, Response 137).

- I applaud the authors for stating the remaining conceptual limitations so explicitly (lines 227 & following: "the models of assembly for each ecosystem functional group represent working hypotheses that require more evidence for critical evaluation. The models were developed by expert working groups and based on current understanding, but few formal research studies evaluate the relative influence of different ecosystem drivers").
- However, these statements really seem to imply that some unknown portion of the presented ecosystem concepts and their functional drivers may currently not stand on very strong

scientific foundations. This, in turn, seems to partly undermine claims elsewhere in the paper that the presented functions-based typology is conceptually sound, or rather, it seems like we cannot currently know for sure just how conceptually sound it is.

- It would indeed be good to see any form of comparative assessment of how strong the evidence underpinning each element of the typology is. Or, if this is not possible, a clearer explanation of which minimum quality criteria were applied. E.g., did every element included in the typology need to be supported by peer-reviewed literature, and if so, why do we not see references supporting each statement in the typology? Or was it sufficient if at least one expert said something to be true? I feel that much more transparency is needed here.

Validity of the main claims with regard to the typology

Some of the claimed qualities of this typology seem overstated, at least for the human-modified components of it (see specific comments on the individual human-modified biomes and EFGs for details). Given the partial framing of this paper as contributing to the imperative to conserve all levels of biodiversity, I was a bit disappointed to see that cultivated biodiversity did not receive much attention. The distinctions within the human-dominated biomes are in parts not inclusive enough to really encompass all elements of cultivated biodiversity and to reliably distinguish more from less biodiverse human-dominated systems. For example, annual croplands are generally presented as low biodiversity systems, thus by definition precluding appreciation of higher over lower diversities of crop breeds.

I do not believe that the goal of grouping ecosystems with functionally similar responses is achieved for the terrestrial human dominated systems, as the presented EFGs refer to major land use classes and hide tremendous heterogeneity in land use intensity and land management practices. There is ample literature on the different responses of systems with different land management practices to stressors (think of differential resilience of more and less diverse cropping systems) and also on their different responses to ecological processes (e.g. Erb et al. 2017).

AU RESPONSE 153: We thank the reviewer for this thoughtful comment and the reference, which alerted us to other relevant literature. We agree that the functional groups may be quite heterogeneous in certain properties. In part, this relates to limitations associated with discrete representation of continuous properties in nature (see Response 137), which apply to anthropogenic ecosystems and processes as well as natural ones. In addition to relevant comments and edits in Response 150, we make three further points in response.

First, high levels of heterogeneity within groups can be addressed by segmenting continuous gradients of vegetation into a larger number of groups, with a trade-off through increased complexity of the typology. An important point is whether variation in properties is greater between groups than within them at the same level of classification. Greater heterogeneity within groups than among them would justify a trade-off to recognise more groups. We think the major functional distinctions among anthropogenic ecosystem functional groups are greater than variation within them. Conceptually, for example, they allow, market gardens to be distinguished from urban systems as belonging to different functional groups with major contrasts in ecosystem properties and drivers. Where fine-resolution spatial data allow them to be separately identified, market gardens can be mapped separately, even when they are positioned within city or village limits. Further, the level of detail at Levels 2 and 3 of the IUCN Global Ecosystem Typology is comparable to, or greater than detail in corresponding global classifications of land use (see Response 151 for two examples).

Second, the hierarchical structure of the typology allows heterogeneity within groups at one level to be characterised through the identification of multiple groups at a lower level. All Level 3 groups display substantial variation in properties – we are careful to define them as “*groups of related ecosystems within a biome that share common ecological drivers promoting convergence of ecosystem properties...*” (Table S3.1). This does not mean that the members of a group have the same properties, but rather that the properties converge as a consequence of common causal factors (e.g. cultivation, species introductions and removals, etc.). While we agree with the reviewer,

for example, that ‘appreciation of higher over lower diversities of crop breeds’ is very important for certain applications, we think this type of variation ought to be represented at Levels 5 or 6 of the typology, through recognition of multiple units that differ in these properties (within a single functional group). This approach is consistent with the definition of the hierarchical levels (see Response 150 regarding elements of urban ecosystems) and the treatment of non-anthropogenic ecosystems.

Third, where heterogeneity exists within a classification unit, the description of that unit should characterise the variation in its properties. We endeavoured to represent variability within EFGs within the descriptive profiles, so far as space constraints allow. We agree referee #6 that this variation could be better characterised in some anthropogenic functional groups. Accordingly, we amended relevant text to improve the characterisation of variability within groups (see responses to specific comments below).

- I comment on those amendments at the end of these comments.

A separate issue involves temporal variation within the units which, in anthropogenic ecosystems, is closely related to land use intensity and socio-economic processes. Although tracking change is largely now beyond the scope of this ms (see Response 1), we mention this as an example of feedbacks (lines 153-156):

“Similarly, feedbacks exist between ecosystem properties and the drivers. For example, changes in ecosystems initiated by human activity, such as land use intensification, influence human social structure, markets and consumption patterns, driving changes in resource appropriation and further change in ecosystem properties [16].”

The promised scalability seems to mostly apply to scaling across different levels of thematic detail. Scalability in the sense of applicability across spatial and temporal scales is more doubtful, given that such scales are rarely referred to even though several statements would strongly depend on the spatial scale considered. For example, the trait of “low diversity” of croplands may often be true at plot scales, but small farmlands may increase biodiversity at landscape scales. I think the attribute spatially explicit is woefully misleading (at least that term needs to be further qualified), as the provided indicative distribution maps in some cases are ill-conceived or too uncertain to really indicate much, beyond what is anyway trivial (for example, that a certain human-dominated EFG exists on most continents close to where there are humans).

AU RESPONSE 154: It is true that our emphasis is on thematic scalability, as this is most relevant to the design of the typology (reflected in Design principle 4, Table S1.1). We now justify this emphasis more clearly in the text, and link it to different thematic and spatial resolutions required for different applications (lines 177-181):

“The scalable hierarchical structure (principle 4, Table S1.1) and the explicit description of properties and drivers enables units at any thematic level to be mapped at different spatial scales. These may be tracked through different temporal scales, according to needs of specific applications and constraints on the resolution of available data.”

Our usage of ‘spatially explicit’ refers to the units being defined in a way that allows their distributions to be mapped. We demonstrate this by providing indicative maps, admittedly with variation in map quality, with improvements expected over time (see response 151). Conversely, a classification with units that are poorly described is not spatially explicit if there is insufficient information to characterise their distributions. For example, a few of the classifications we reviewed in Appendix S1.1 only have text-string labels to describe the characteristics of their units. Finally, although a detailed appraisal of spatial and temporal scaling is beyond the scope of our current paper (given that we excluded spatial analysis from the ms - see Response 1), we contrast the roles of ecological drivers in community assembly over contemporary and evolutionary time scales (lines 127-133) and note that different ecological drivers shape ecosystem assembly at different spatial scales (e.g. lines 141-142 and Fig. 1). We also address needs for spatial and temporal scaling to address related needs of the typology for ecosystem management applications at global, national and local scales. (Appendix S6).

- This is actually the first time that I read in such explicit words that the term “spatially explicit” is merely meant to indicate that the units are described in a way that they can in principle be mapped. If this is really the intended meaning, this should be much more prominently clarified right where this “spatially explicit” characteristic of the typology is first claimed. Also, in the review of the other typologies, this should be reflected more fairly. I believe that nearly all existing ecosystem typologies refer to concepts that can in principle be mapped, so this should then arguably not be presented as a characteristic that makes specifically this typology stand out.
- Currently, there is clearly the claim in Table S1.2 that this typology is somehow associated with existing maps, and specifically with fine, spatially explicit units. This is simply untrue for a large portion of the ecosystem types, and for an additional even larger portion, it is entirely unclear how reliably existing maps can depict them.
- Also, there still remains an important difference between “can be mapped” and “can be mapped as a time-series”. E.g., looking at the sentence in line 191 & following: – As I have commented previously, having something as a map is simply not enough in the context of providing “evidence on which ecosystems are most exposed to collapse”, which requires information on temporal dynamics.

I think the attribute of a sound theoretical basis is mainly achieved with regard to ecological functions. However, the definition of “human-made” systems cannot only rely on an ecological characterization. Unfortunately, the typology ignores existing theories and concepts from those sciences that focus on different “human-made” systems, even those that are also rooted in ecological theory (like the much clearer conceptualizations of land use intensity, for example, that exist in land use science).

AU RESPONSE 155: We now provide more theoretical context to support our rationale for constructing the anthropogenic component of the typology in the main text (lines 146-150):

“While our model portrays humans as integral drivers of ecosystem assembly, we separated human activity from other biotic interactions to highlight interactions and feedbacks between ecosystems and socio-economic systems [18], and the need to assess and mitigate the human impacts on biodiversity and ecosystem functioning.”

and in referring to feedbacks among assembly drivers and ecosystem properties in lines (153-156):
“Similarly, feedbacks exist between ecosystem properties and drivers. For example, human land use intensification initiates changes in ecosystems that, in turn, influence human social structure, markets and consumption patterns, driving changes in resource appropriation and further change in ecosystem properties [18].”

We elaborate in Appendices S2 (section on Human activity, pp3-4) and S3 (section on Dealing with anthropogenic influences on ecosystems, pp22-23). On the pervasiveness and variability of human influence, we note in Appendix S3 (p22):

“Human activity influences assembly of almost all ecosystems. Erb et al. (2017), for example, defined 10 different land management activities, quantified their variation in intensity, and estimated global-scale biophysical and biogeochemical effects on terrestrial ecosystems. The effects play out through complex interactions and feedbacks between ecosystems and socio-economic systems with varied settings for labour and capital inputs, technology, market dynamics, cultural beliefs, business or subsistence decision making and geopolitics (Meyfroidt et al. 2016). Erb et al. (2017) estimated that intensive land management activities occur on about 10% of Earth’s ice-free terrestrial surface based on subjectively thresholded metrics for each activity. Haberl et al. (2007) estimated that humans appropriate >40% of Net Primary Productivity over ~20% of Earth’s ice-free terrestrial surface.”

- I appreciate that the authors make more extensive reference to land-use literature now. Nevertheless, this does not change my earlier assessment that the definition of anthropogenic ecosystems does not have the type of sound theoretical basis that may be true for the natural ecosystems.

- Assemblage theory may well be an adequate theoretical framework for defining the processes and filters that create and maintain natural ecosystem. But the fact that what the authors call anthropogenic ecosystems were co-created over millennia by human and natural processes means that their functioning and continued existence cannot be understood solely by those natural assemblage processes alone. The processes creating and maintaining human-made ecosystems additionally include economic processes, values, institutions, etc., etc. All this complexity is here largely condensed into a single “Humans” box, whereas the natural processes and assemblage filters at play in creating and maintaining natural ecosystems are distinguished in great detail.
- I am not suggesting that the same level of detail is even needed to describe how the natural and the human-made ecosystems systems are created and maintained. Given that the applications supported by this typology will anyway be primarily ecological, rather than social-ecological, this does not seem to be a great problem.
- Nevertheless, I was specifically asked to comment on the quality of the human components of this typology. And I consider the claim of a sound theoretical basis – however valid this may be for the natural components – an overstatement for the human-made components.

This typology also claims to be conceptually robust and to overcome other typologies weaknesses of not “fail[ing] to describe their units in sufficient detail for reliable identification or require[ing] diagnostic features that are hard to observe”. For the human-dominated systems, their descriptions are in parts reliant on vague, or insufficiently defined, boundary concepts (e.g. “intensive”, “natural”), or are incomprehensive and implicitly excluding regionally important examples of human-made systems (e.g. villages, non-woody permacultures). This makes a reliable identification of EFGs by strictly following this typology highly doubtful, especially if persons from diverse academic backgrounds are involved. The typology also used some diagnostic features that can be very difficult to observe (e.g. alien biota, selective breeding). For several of the ecological traits and the drivers listed in Fig. 1, data that would be needed for detailed mapping and monitoring are not available for many regions. Several human-made systems are defined either too vaguely or too restrictively, which leaves some systems unaccounted by any one group.

AU RESPONSE 156: The descriptions of Ecosystem Functional Groups in Appendix S4, including text on salient ecosystem properties and drivers, illustrative images, diagrammatic models of assembly processes, indicative global distribution maps and selected references, are more detailed, consistent and systematic than those provided for all other ecological typologies reviewed in Appendix S1. Some of the diagnostic features are recognisable in the field, others are detectable from remote sensing, while some require specialised measurements. As noted above, current map data varies in quality. The descriptive profiles have been subject to extensive peer review (Appendix S5). We are further developing interpretive resources and making them available to a wide audience through the IUCN Global Ecosystem Typology website <https://global-ecosystems.org/>. We acknowledge inherent challenges in simplifying complex continuous variation in nature within a discrete classificatory framework (main text lines 268-276; Appendix S3, p23). We also acknowledge the need to refine and update the descriptions themselves and thank the reviewer for detailed suggestions, which we address below.

- I agree with the text above. However, given that this typology too, extensively relies on diagnostic features that are difficult to observe (especially, but not exclusively, with respect to anthropogenic ecosystems), this should not be represented as an advantage of this over other typologies. Even statements that simply highlight that some other typologies have this “hard-to-observe diagnostic features” problem implicitly suggest that this typology was different. That seems unwarranted, and therefore, relevant text portions should IMO be changed to reflect that this is a general problem of this and the other typologies.
- E.g. L. 110-112: “Many typologies that we examined either failed to describe their units in sufficient detail for reliable identification, or required diagnostic features that are hard to

observe.” - This sentence is thus misleading as it implicitly suggests that the presented typology was superior.

Problems in the EFG datasets associated with the typology and validity of presented analyses I believe that the paper as a whole (that is, not just considering the typology per se, but also the presented analyses and data) has several severe flaws that undermine its scientific soundness. The authors frame several analyses around the high-level problem that “Decisions about effective action to conserve biodiversity and sustain ecosystem services [which] require evidence on which ecosystems are most exposed to impacts from particular pressures [...], which ecosystems are undergoing most rapid loss of biodiversity, and which ecosystems contribute most to particular human benefits”. The authors then claim that “[...] the IUCN Global Ecosystem Typology and associated spatial data [19] provide an ecologically robust and powerful framework for such synthesis”. As a central selling point to demonstrate this potential of the typology and associated data, the authors present concrete results on specific portions of the world’s ecosystems that are exposed to high pressures and/or are more poorly protected than others.

AU RESPONSE 157: Much of this criticism refers to our original spatial analyses which has now been deleted from the ms (see Response 1). The second phrase quoted applies only to some of the spatial data (see Response 151). We qualified the text to clarify as follows:

“... the IUCN Global Ecosystem Typology and a growing body of spatial data [23] will provide an ecologically robust and powerful framework for such synthesis”

(Lack of) treatment of uncertainties

There are tremendous uncertainties in several of the used data (see below for specific details on the data on human-dominated EFGs), and moreover severe scale mismatches between different datasets that are combined here. Both seriously hamper their fitness-for-use for assessing both which EFGs are how well protected, which ones are most pressured, and which ones experience specific pressures. As neither limitations are addressed in the presented analyses, nor is there any validation of results, the presented results cannot be considered trustworthy (see below for details). Some of the bigger picture applied conclusions, e.g., those stressing greater protection needs of specifically forests and grasslands compared to other terrestrial systems, are thus not sufficiently substantiated.

AU RESPONSE 158: We removed the spatial analyses and research findings from the manuscript and Appendix S6 as requested (see Response 1).

The same mentioned limitations of the typology-associated maps also make them inappropriate for any other claimed application of the typology that are sensitive to spatial misestimations, including as national ecosystem accounting and monitoring of CBD targets. In fact, the implicit message that a typology whose units can currently only be mapped indicatively could be a standard for these globally important applications seems dangerously misleading.

AU RESPONSE 159: In our ms, we noted that the United Nations had adopted our typology as the reference classification for ecosystem accounts but did not suggest that the indicative global maps in Appendix S4 should be the basis for national ecosystem accounting. On the contrary, the UN SEEA-EA (https://unstats.un.org/unsd/statcom/52nd-session/documents/BG-3f-SEEA-EA_Final_draft-E.pdf) has adopted the typology primarily to aid synthesis and upscaling of accounts from many semi-independent national sources. This allows use of the best available national maps for each country, with cross-walks developed to translate fine-resolution national ecosystem types to global Ecosystem Functional Groups as demonstrated for Chile and Myanmar in Tables S3.3 & S3.4, respectively. We now discuss how our approach decouples classification from mapping (lines 254-256):

“By decoupling the mapping process from prior development of the classification, our approach liberates the definition of ecosystem units from constraints imposed by the current availability of spatial data and allows for progressive improvement (Appendix S4, p15).” And

and in the second paragraph of the Natural capital accounting section in Appendix S6, we discuss the use of maps for ecosystem accounting, as recommended for the UN SEEA-EA process:

“The IUCN Global Ecosystem Typology was adopted as the reference classification for implementing SEEA-EA (UNCEEA 2021). The core of the reference classification is Level 3, which will be used to summarise globally across national ecosystem accounts. Contributors to SEEA-EA are expected to use the best quality high-resolution classification available for their jurisdiction when developing their national accounts (i.e. Level 6 units), and assign the units of that classification to Ecosystem Functional Groups (Level 3) to enable consistent international reporting. Examples linking fine-grained national classifications to Level 3 of the IUCN typology are given in Tables S3.3 and S3.4. This flexible use of the IUCN typology will enable jurisdictions to report detailed national accounts, while contributing data that can be scaled up and summarised to units of the international reference classification (UNSD 2019).”

I wonder how sensible it is in general to acknowledge the indicative nature of the maps, but then still present them as pixel-maps rather than in a format that is clearly distinguishable as “indicative”, such as blob maps (compare the IUCN species maps). More so, I wonder why the earlier acknowledgement of limitations is later ignored presenting a pixel-based intersection with other maps. I have to give the authors credit in that they use a lot of leveraging language, like “indicative”, “coarse-scale analysis”, “Acknowledge uncertainties”, etc. Yet, the uncertainties are merely acknowledged in writing, but are not reflected in the analyses, nor in the presentation of results. In fact, with the information provided, even the indicative validity of the EFG maps can only be ascertained by experts on the respective fields and data. I do feel that the burden to make uncertainties more clearly visible is on the authors, here, given that these indicative maps might be naively (ab)used for all kinds of applications.

AU RESPONSE 160: We removed the spatial analyses and research findings from the manuscript and Appendix S6 as requested (see Response 1). We think a global-scale pixel format with relevant caveats is appropriate to current uncertainties in distribution. We are familiar with the ‘blob maps’ stored in IUCN’s Species Information Service. Most of those maps are compiled from hand-drawn polygons generated by species experts, usually with reference to occurrence records from a range of (unspecified) sources. The ‘blobs’ may include large areas of unoccupied habitat and the sources and development steps for individual maps are not well documented, yet they are used in a number of spatial analyses published in international journals. We believe the overall standard of EFG maps is at least as good as IUCN species maps (in many cases better), their derivation is more explicitly documented and they are updated regularly from published sources. While we agree that the lowest quality EFG maps preclude a comprehensive global spatial analysis, we believe all maps are sufficient quality for indicative thumbnail maps presented in Appendix S4 (see Response 1).

- I disagree with the authors’ response that “... a global-scale pixel format with relevant caveats is appropriate to current uncertainties in distribution”.
- The point of vectorizing uncertain map information into coarse blobs would be precisely to make it clear that they should not be used for a pixel-level analysis. Referring to my earlier example: by distributing the species maps as blob vectors, the IUCN is actually being honest about the non-precise nature of knowledge of where the species occur. If they instead distributed the maps as 1-km pixelated raster maps, that would be dishonest. This principle could be translated to ecosystems, too. Distributing them as generalized blobs would make their merely indicative character obvious to users and high-resolution pixel-level applications would automatically be discouraged.
- I did not suggest that the IUCN species maps would represent any acceptable standard for the EFG indicative maps. To the contrary, the IUCN species maps are woefully non-transparent and non-reproducible. But the IUCN species maps are also not to be published in a highly reputable scientific journal, where I believe different mapping standards apply. Disseminating rasterized maps at a resolution where the mapped results were not quality-assessed indeed

violates widely followed scientific mapping standards in the geographical, biogeographical, and environmental sciences.

- As a side remark, I am aware that IUCN species blob maps include areas not occupied by those species, but this is precisely the point I am trying to make. Those maps were not designed to show which areas are occupied, but only to indicate the approximate boundaries of the species ranges. Representing this as a crude blob is appropriate, and this only then become a problem if those blobs are incorrectly interpreted by ill-informed users as being pixel-level indications of where the species exists.

Lack of reproducibility:

The provided information on the data sources and the generic descriptions of mapping protocols are insufficient to make the indicative maps of EFGs reproducible, at least in several cases. I made the effort of reading through the IUCN ecosystem typology that is already published on the IUCN website (which extensively overlaps with information presented in the appendices for this paper). While there is substantially more information therein (e.g. Table 4), this is in many cases not sufficient for understanding specific data manipulations, let alone the underlying reasoning and the validity of the results. In some cases, using the specific referenced datasets as indications of the respective EFGs seems dubious at best (see comments on specific EFGs below). I have serious concerns about publishing any analysis built on a top of the EFG maps before publishing a detailed description of the specific mapping and validation protocol for each map. I also noted that the methodological description of degradation assessment relies on an “updated version” of a previously published terrestrial human footprint maps for other years. As these maps do not seem to be published, and it is unclear whether were produced via the same protocol as the original maps, and how valid these new maps are, these analyses are not reproducible.

AU RESPONSE 161: We removed the spatial analyses and research findings from the manuscript and Appendix S6 as requested (see Response 1). We acknowledge that indicative maps for one-third of the Ecosystem Functional Groups currently lack spatial data of suitable quality for quantitative analysis (lines 250-252).

Appropriate attribution of EFG map sources:

In this respect, I wonder in some cases which specific creative manipulations to the source maps were done to justify that they are here collectively referenced as “Keith et al. 2020”. With respect to spatial data, the main original achievement seems to have been to collate the pre-existing maps that were deemed fitting to the respective EFG, clip them to the broader regional boundaries of the higher ecosystem hierarchical levels, and making them available through a common portal. Where this is really all that happened, it would seem appropriate to also cite the original map sources in this paper.

AU RESPONSE 162: The reference cited by the referee refers to a compiled archive of maps. It provides the data agreed by specialist contributors for each description to be the best available representation of the concept for each Ecosystem Functional Group. The source data are in the public domain (Creative Commons) and fully cited in Table S4.1 and the archive, which includes Readme files and xml documents with map details and references for both the bundled and individual maps so that they are fully documented and attributed to sources. The 'creative manipulations' on the maps are different in each case with some EFGs almost identical to one original source map while others are combinations of several sources. The whole mapping process included reviewing and comparing alternative data sources, performing several spatial operations (transformation, reprojection, aggregating, resampling, cropping or clipping, intersections and unions, map algebra, etc), and finally harmonizing all of them to similar resolution and presentation. Even where the map of an EFG concept, such as the one for MFT1.1 Coastal deltas, is apparently identical to the source, it has undergone an evaluation process, has been selected instead of alternative maps, and has been harmonised for consistency with other maps in the archive.

- I did not intent to suggest that no creative work went into the compilation referred to here as Keith et al 2020, and I indeed very much acknowledge that this is scientific work in and of

itself. I am specifically worried about those cases where a given pre-existing map that precisely defined a given EFG was directly adopted as its indicative map. Where that was the case, the mere inclusion of such map in a larger compilation (along with trivial technicalities such as reprojections, etc.) arguably does not constitute enough of an intellectual input to justify that only the larger compilation is henceforth cited, i.e., without citing the original references of the adopted components alongside. Anyway, opinions on this might differ.

Inconsistencies between EFG maps:

I am worried about inconsistencies among the datasets for the different EFGs. They originate from very different sources and were generated for different purposes, and range from hand-drawn blob maps, via dasymetric modelling outputs, to classified remote-sensing imagery. As far as I can tell, no efforts were made to adjust these layers to make them consistent. If the sole claimed purpose of these EFG maps was mapping their indicative global distributions, this would not be a problem. But this could affect several of the presented analyses. For example, the presented test for protection of different EFGs could be affected by pseudo-replication if small protected areas that really only encompass single EFGs were overlaid with crude distribution maps of multiple EFGs that are covering the same areas. This would also affect several of the other claimed application fields of the typology that would be sensitive to large misestimations of EFG areas resulting from such inconsistent mapping.

AU RESPONSE 163: We removed the spatial analyses and research findings from the manuscript and Appendix S6 as requested (see Response 1). The maps were harmonised to a consistent format, but represent a minimal necessary adjustment to the best available spatial data for each unit (see response 162). This allows users to make their own decisions about how to achieve a level of consistency between maps that suits their purpose.

Analysis of global protection status of ecosystems:

The authors aim to demonstrate “how the typology could support evaluation of Aichi target 11, ... [by analyzing whether] ... at least 17 per cent of terrestrial and inland water, and 10 per cent of coastal and marine areas ... [are] conserved through ecologically representative systems of protected areas ...”. The authors provide an analysis of EFG representation in global protected areas (PAs) based on spatial overlay of both datasets. However, like any globally “indicative” map, the EFG maps will in many cases misrepresent true areas where the EFGs exist at fine scales. The authors textually acknowledge this themselves, but the presented analyses fail to address the scale mismatch between the coarse-scale EFG maps and the PAs (which are mostly mapped at much finer scales). It has been repeatedly shown in the case of overlays with IUCN species maps that failing to account for such scale mismatches leads to biased estimations of species protection. For essentially the same reasons, a simple overlay of indicative EFG maps with PAs is unlikely to yield reliable results. Moreover, it is completely unclear just how certain or uncertain results for different EFGs might be. This makes the scientific value of this exercise highly questionable. Given the scale mismatches and unassessed uncertainties in the EFG maps, I am not convinced that the presented typology (with its currently associated data) can, indeed, at this point support such analyses for Aichi target 11, or similar global analyses.

AU RESPONSE 164: We removed the spatial analyses and research findings from the manuscript and Appendix S6 as requested (see Response 1).

Analysis of ecosystem degradation:

To assess which ecosystems face degradation pressures, the authors overlaid the indicative EFG maps with human-footprint maps for specific years. I have three main concerns about these analyses. Firstly, in addition to the same spatial-scale mismatches as in the PA-overlay analysis, this assessment is also subject to temporal mismatches. The human-footprint maps refer to the years 2000-2013, while the data underpinning the indicative EFG maps have many different focus years (and are in some cases not time-explicit). It seems likely that the areas of some EFGs have changed between the reference time of their respective original source data and the time period covered by

the Human-influence data. Particularly in the parts of the world that have experienced very rapid land-use changes during short periods of time, it is highly possible that this temporal mismatch will bias the results. Secondly, the extent to which this assessment can provide evidence, rather than just indications, of “ecosystem degradation” seems to be slightly misrepresented. The authors acknowledge that their analyses are only based on information on presumed degradation drivers, rather than actual observations of degraded ecosystem states. But then they should arguably use more qualified language like “likely degraded ecosystems” or “ecosystems exposed to degradation pressure” throughout.

AU RESPONSE 165: We removed the spatial analyses and research findings from the manuscript and Appendix S6 as requested (see Response 1).

Finally, the discussion of these analyses is insufficiently referenced and/or too swiftly argued. For example, the authors contrast a need for protection of remnants for dry forests that recently saw accelerated degradation/conversion with the need to actively restore grasslands that have long been exposed to pressures. I can only assume that the authors are hinting at the differential passive restoration potential of these systems, as natural species seedbanks would presumably still be largely intact in the former case but largely lost in the latter. In any case, it would be important to expand on and better support such statements, so the reader can understand the reasoning.

AU RESPONSE 166: We removed the spatial analyses and research findings from the manuscript and Appendix S6 as requested (see Response 1).

Analysis of ecosystem degradation:

The assessment of ecosystem portions that receive different combinations of degradation pressures (based on overlay with the Human-influence maps) and protection (based on the PA overlay) is particularly flawed. The human-influence maps are partly based on (and their patterns strongly driven by) maps of human population density (specifically, the Gridded-population-of-the-world dataset, GPW). However, the GPW dataset was generated by distributing statistical population data to pixels, and during this process, protected areas were masked out. There is thus circularity, and almost certainty bias, in the assessment of different ecosystems’ combinations of degradation pressures and protection status (Figure S6.3). What is even more worrying is that the authors use this circular and likely biased assessment to derive concrete recommendations for nuanced management strategies for different ecosystems, depending on the specific combination.

AU RESPONSE 167: We removed the spatial analyses and research findings from the manuscript and Appendix S6 as requested (see Response 1).

Analysis of anthropogenic transformations:

The authors also overlaid the indicative maps with a global map product of anthropogenic biomes, which they used to “estimate the proportion of area exposed to high pressures that show evidence of transformation into artificial or human dominated ecosystems”. The used anthropogenic biomes product by Ellis et al. is based on a land use model with very well-known uncertainties (variously discussed by the authors themselves, and acknowledged in that paper). Yet, the transformations it shows seems to have been taken at face value here. In addition to these model uncertainties, I am also concerned about inconsistencies in the presented analysis, because the historical land-use model behind the anthropogenic biome product was informed by different land-cover maps than those underpinning the presented indicative ecosystem maps. None of these uncertainties is currently addressed. This makes it unclear which of the results shown in Fig. S6.6. indicate genuine ecosystem transformations and protection levels, and which ones merely reflect artefacts of overlaying uncertain and inconsistent products. Without some sort of sensitivity analyses it is unclear which patterns are interpretable and to what extent, and which ones are not.

I also wonder about the conceptual validity of the results shown in Fig. S6.6. It seems like this assessment assumes that EFG distributions remained static over the considered study period, and that they could only either remain intact, become degraded, or be destroyed, but never recover. We know for some regions that degraded lands were actively restored or abandoned and left to become wild again, and also that in some wilderness areas natural EFGs have advanced over others (like

shrublands over arctic tundra in Canada). Both possibilities seem to have been discounted, which makes me wonder even more how reliable the relative thickness of bars shown in that figure may be.

AU RESPONSE 168: We removed the spatial analyses and research findings from the manuscript and Appendix S6 as requested (see Response 1).

General validity of typology

Generally, it is challenging to assess the specific validity of many of the statements placed across this massive paper, not least because many of them are not referenced. I am not sure if all those unreferenced statements are trivially clear or uncontentious, even among experts on the specific topic. For example, many statements in Table S3.2 arguably may fall into this category.

AU RESPONSE 169: We improved referencing and strengthened justifications for several aspects of the typology throughout the manuscript and appendices. We also revised the text to make it clear that the models compiled for each Ecosystem Functional Group (in Appendix S4) represent working hypotheses about drivers that shape observed ecosystem properties (see Responses 4 & 9). We added 26 references to the commentary on Table S3.2 (pp4-6, Realms section) to support the generalisations made there. In addition, the cited references in descriptions of each Ecosystem Functional Group in Appendix S4 are selected reviews and case studies with relevant evidence on ecosystem properties and postulated drivers. Table S3.2 is a summary of hypothesised relationships represented in the models for the Ecosystem Functional Groups across four realms. We revised the caption to make it clear that the relationships are postulated and that relevant sources of evidence are found in the text commentary the precedes it and the respective descriptive profiles within Appendix S4:

“Table S3.2. Synopsis of postulated assembly filters and ecological traits distinguishing ecosystems within the five realms of the biosphere (Fig. S3.1). Refer to Appendix S4 for glossary of selected terms and to text commentary (above) and respective descriptions of Ecosystem Functional Groups within each realm (Appendix S4).”

- The hypothesized assembly filters still seem under-referenced (at least, I could not easily find justification/explanation in all cases that were not immediately clear to me). I am not sure what the standards would be, here, for a typology. In any case, I acknowledge that this is a massive piece of work and that referencing every component of this work would be a massive piece of work in itself. I do appreciate the more careful framing as “working hypotheses”, “postulated assembly drivers” etc. This is very helpful!

Conceptual choice of including community attributes as criteria

The typology is not only based on the functional attributes (which I think is very useful), but, for the defined ecosystems at levels 5 and 6 of the proposed hierarchy, also on their ecological communities.

I am not sure whether a typology that requires measurements of the integrity of a given community can be of much long-term stability, which should arguably be a requirement of any typology.

Ecological communities have always changed, both over evolutionary and over ecological time scales. The biological lineages represented in communities may uniquely respond to different types of environmental changes (or “ecosystem drivers”, as referred to herein) and usually show only limited assemblage-wide synchrony in those responses. That many current communities will break apart and change into new ones should arguably be a baseline expectation of a typology that is designed to remain robust over multiple decades (i.e., policy and global monitoring time scales). In fact, all these criticisms have been phrased since several years ago in response to the very proposal of an IUCN Red List of Ecosystems (Boitani et al 2015). I would expect that a paper presenting an ecosystem ontology that defines ecosystems by community integrity would proactively address any such caveats and previous criticisms.

AU RESPONSE 170: We agree that long-term stability is an important quality for a workable typology. Two aspects of our approach to development of the typology contribute to this quality. Firstly, in lines 108-110 of the main text, we state:

“Ecological classifications based on tested and established theory are more likely to be robust to new information than classifications based only on observed patterns and correlations.”

- I believe the claim of the enabled long-term stability of this presented typology should generally be toned down.
- In lines 108 & following, the authors state “Ecological classifications based on tested and established theory are more likely to be robust to new information than classifications based only on observed patterns and correlations, which may prove unstable when new information emerges.”
- I fully agree, however, yet the human components of this typology, at least, are not based on much theory either, and their stability is thus generally more doubtful. We simply cannot foresee how stable the types of anthropogenic systems that exist today will be, as, e.g. advances in technology, changing cultural preferences, and societal adaptations to environmental changes all come together. Who knows, we might rely on deep-sea or high-atmospheric food production systems 100 years from now.
- Also, this claimed stability is countered by the (rightful) acknowledgement further (lines 235 & following): “By highlighting poorly known systems in the atmosphere, deep sea floors, subterranean freshwaters, lithosphere and beneath ice, and by prompting researchers and other users to ask where particular ecosystems belong in the scheme, we foresee the typology promoting research to fill significant knowledge gaps that will inform future amendments of the typology structure, as well as descriptions of the units.”
-

Secondly, in Methods, we now comment further on how we constructed the typology to make it robust to changes in assemblages except where they involve substantial functional shifts (lines 351-361):

“While neither function nor composition were intended to take primacy within the typology, we reasoned that a hierarchy representing functional features in the upper levels is likely to support generalisations by leveraging evolutionary convergence. In contrast, a typology reflecting compositional similarities in its upper levels is less likely to be stable due to dynamism of species assemblages and evolving knowledge on species taxonomy and distributions. Furthermore, representation of compositional relationships at a global scale would require many more units in upper levels, and possibly more hierarchical levels. Therefore we concluded that a hierarchical structure recognising compositional variants at lower levels within broad functionally-based groupings at upper levels would be more parsimonious and robust (principle 6) than one representing composition at upper levels and functions at lower levels.”

- I am happy with these text additions.

For this and other reasons we did not consider ecosystem integrity in the design of the typology (none of the design criteria in Table S1.1 mention it). In our ms, we also carefully avoid confounding the concept of anthropogenic ecosystems and their definition with assessments of ecosystem integrity (We hope that excluding the Human Footprint analysis removes any confusion it may have generated on this point). Excluding integrity from definition of the classification units allows the typology to support applications by framing independent assessments of integrity for any type of ecosystem using appropriate methods and metrics.

- I think it is a smart (indeed, a critical) choice to not confound ecosystem identity with ecosystem integrity.

As noted in the additional Methods text quoted above, the upper three levels of the typology are based on functional attributes, not biological lineages. The typology is therefore well buffered from instability when communities break apart. As a comparison, we point to species taxonomy which is in continual flux as species concepts change, new taxa are discovered and new data emerge on

relationships of higher taxa, especially since the molecular revolution. This causes some inconvenience but does not preclude the usefulness of the classification system for a wide range of applications, including risk assessment (see Keith et al. 2015; <https://conbio.onlinelibrary.wiley.com/doi/full/10.1111/conl.12167> for elaboration on that issue). Similarly, nothing in our approach assumes that species components of ecosystems respond synchronously to environmental change. This should not be confused with our point that different ecosystem types included within the same Ecosystem Functional Group should be expected to exhibit similar responses to environmental change and management, not as a consequence of biological lineages, but because of functional convergence across different lineages (lines 164-166): *“Convergences in ecosystem properties are axiomatic to a functionally based ecosystem typology because they underpin robust generalisations and predictions about ecosystem responses to environmental change and management.”*

The Boitani et al. (2015) paper raised some issues about an earlier paper presenting a risk assessment protocol for ecosystems (Keith et al. 2013; <https://journals.plos.org/plosone/article?id=10.1371/journal.pone.0062111>). While Boitani called for development of an ecosystem typology to frame risk assessments (an identified need that we meet in this ms), most of the content of that paper concerned ecosystem risk assessment. We refer to our contribution in the same journal issue for discussion of ecosystem risk assessment and rebuttals of several claims made by Boitani et al (Keith et al. 2015; <https://conbio.onlinelibrary.wiley.com/doi/full/10.1111/conl.12167>).

It is unclear to me how the typology is expected to manage to deal with these community changes. What criteria will be applied to decide when a given ecosystem can no longer be considered that ecosystem as a result of its ecological communities being reshuffled? Will that create many novel ecosystems according to this typology, that are then still to be defined? Will a new, homogenous ecosystem that was created via a climate-driven reshuffling and homogenization of two neighboring, previously distinct communities (e.g., as may easily happen in some remote regions at high latitudes as previous climatic barriers are breaking down) be considered as a single new natural ecosystem, or as degraded version of the two ecosystems that were previously mapped there, and that the single new one replaced?

AU RESPONSE 171: Refer to Response 170. We also revised text in Appendix S6 (Natural capital accounting section) to address ecosystem shifts.

“When structured by Level 3 of the Global Ecosystem Typology, national and global ecosystem accounts should quantify major functional shifts and re-organisation of ecosystems. For example, transformation of the Aral Sea during 1980-2000 (Micklin 2006) from a large freshwater body (EFG F2.1) to ephemeral grasslands (degraded forms of T5.1) and hypersaline water bodies (F2.6), can be readily tracked at Level 3 of the typology. While we acknowledge uncertainties in ecosystem identification, such major shifts between groups should be relatively easy to detect and have major implications for human wellbeing and biodiversity. Other increasingly common types of ecosystem transformation are also readily detected and reportable through Level 3 of the typology. These include transitions driven by land use change (T1-T6 to T7, or T7.2 to T7.3), changes in fire regimes (e.g. T1 to T4), or climate-related shifts such as drying of freshwater aquatic systems (various transitions involving F1, F2 or FT1), advance of alpine treelines (e.g. T6.4 to T2.1) or repeated coral bleaching and ocean acidification (M1.3 to M1.6 or M1.7). Conversely, shifts that result from forest reconstruction (e.g. T7.2 to T2.2) or land use abandonment (e.g. T7.5 to T2.2) should also be detectable and reportable as shifts between different Ecosystem Functional Groups.

The upper levels of the typology should be robust to compositional changes that do not produce major functional shifts. However, more subtle shifts reshuffling of species composition, for example, in response to climate change or land degradation, could be represented at lower levels of the typology in national-level accounts (with consequent uncertainties in diagnosis); and upscaled to changes in global condition accounts.”

- It is still not clear to me how compositional characteristics and changes would be reflected in practice, even in the lower levels of this typology, but this is not a critical point for me.

General comments on the human-modified aspects of the typology

After being extremely impressed by most of the natural components of the typology, I was quite disappointed with the human-modified components. I must say that I did not get the impression that either scientists specializing in these human altered systems (e.g. land-use scientists, agricultural scientists, foresters, urban scientists, aquaculture/fisheries scientists, etc.) nor specialists of human natural systems extensively contributed to the definition of these systems. What matters here is of course not whether or not they were involved, but whether advances and contemporary definitions in those fields were sufficiently considered.

The typology heavily relies on specifically ecological assembly theory (although transferred from communities to entire ecosystems, including their abiotic components), and on using biological and physical system properties to distinguish “functionally different groups of ecosystems from one another by highlighting different ecological drivers that come to the fore in structuring their assembly”. For categorizing and tracking human-dominated systems, this may make this typology of limited use, as those human-dominated systems are characterized and driven by interacting social and biophysical processes and system properties. This may also make it difficult to reliably identify the specific distinctions between degraded “natural” and structurally and functionally similar but nevertheless “human-made” ecosystems. For example, a natural grassland with a long history of livestock grazing and an old seminatural pasture. Both may function very similarly now and may have similarly impoverished or neophyte-dominated grassland communities by now, and their only distinction knowable today might be their history of human use.

AU RESPONSE 172: This distinction is most likely to be problematic at fine spatial scales. We acknowledge uncertainties in representing continuous variation in a discrete classification (see Response 171). Also, we point to an important aspect of the definition of anthropogenic ecosystems (see Glossary),

“...For some of these systems, cessation of those [human] activities may lead to transformation into ecosystem types with different properties and organisational processes.”

Such processes are well-known in parts of Europe, North America and Australia where derived semi-natural grassland typically transit to secondary forests or woodlands, when they are abandoned or spelled from grazing. In contrast natural grasslands, should not undergo such transitions when grazing is removed, so long as the drivers that preclude tree establishment or growth remain functional.

- I must say I am still not convinced that the distinction between anthropogenic and natural systems can always be as straightforward as suggested here, and I doubt that the provided descriptions and the glossary suffice for a reliable diagnosis.
- In general, I think this typology could be much stronger on the anthropogenic ecosystems if it more extensively referred to historical human activities as part of the diagnosis. Sure, these are often difficult to know, but this may at last be increasingly possible with pollen reconstruction, archaeological evidence, etc. By contrast, I cannot foresee that that it will ever be possible to credibly establish (at scale) what would happen to a given system if the human activities in place were to cease, given that people actually live in these systems and any changes of that fact would face major societal hurdles.
- It is also unclear how the current definition of anthropogenic ecosystems deals with the opposite case, i.e., landscapes with a long history of human activities where it is nowadays precisely the continued human management interventions on one or more ecosystem drivers that keep the remaining natural systems intact and/or helps restore them. If, say, a wetland is lost due to climate warning and humans restored a different site to a similar wetland state, but without continued interventions on the water regime, that wetland would return into a non-wetland state, would this now be a natural or an anthropogenic ecosystem?

Currently, all human system alterations are basically presented as an extrinsic “human activity” box. The specific ways in which human processes interact with the ecosystem drivers and traits that are used to distinguish EFGs are not always very clearly conceptualized, nor do they make clear connections to existing (ecologically inspired) conceptualizations of human-environment interactions. For example, the first group of abiotic drivers include the availability of five “fundamental resources essential to sustaining all life: water, nutrients, oxygen, carbon and energy”. I think this would have been an opportunity to characterize “intensive human systems” explicitly as those with a human appropriation of NPP (HANPP) above a certain threshold. HANPP is already one of system-level property of land-use systems that is commonly used by land use scientists to characterize different levels of land use intensity. It would seem to me that adding the proportion change in ecosystem productivity relative to a “natural” baseline that is due to harvesting by its top-predator/herbivore (humans), but also to fertilization, etc., as a system-level property and/or functional trait of the ecosystem might be helpful. Of course, this may in turn be influenced by human’s altering of its other traits and of the other drivers.

AU RESPONSE 173: We thank the reviewer for this suggestion, and agree that HANPP is an important signature for some anthropogenic systems, and that it could help to distinguish Ecosystem Functional Groups characterised by intensive land use those from natural and semi-natural systems (i.e. with low intensity use). Such indicators are potentially useful in map development (we have used HANPP in revisions of several maps suggested by the referee #6), but using them to define the concepts of units within a classification raises several significant problems. Although thresholds in indices such as HANPP are attractive as explicit delimiters of classification units (most biophysical classifications reviewed in Table S1.2 use them for that reason), they tend to oversimplify complex multidimensional properties that distinguish groups of functionally similar ecosystems (see our definition of Ecosystem function in lines 81-82 – more than productivity alone). Use of aggregated univariate indices as thresholds to delimit ecosystem types confounds a clear perspective on some of these multiple properties and ignores others, ultimately producing a poor representation of variation in ecosystem types across the biosphere. Erb et al. (2013, p465) support this view, ‘*Given the complexity of land-use intensity, providing a single, unambiguous and encompassing definition or indicator of land-use intensity does not appear to be an adequate target. A comprehensive analytical framework is required that considers the multidimensional nature of land-use intensity.*’ Their proposed framework encompasses characterisation of land use inputs, outputs, their interactions and changes in system properties (for which HANPP is one of several example indices). We therefore avoided definition of EFGs by thresholding univariate aggregated indices. Another problematic issue is the estimation of HANPP, which relies on a number of (largely untestable) assumptions and subjective interpretations of ‘potential’ NPP, as well as sensitivities to data availability for actual NPP (discussed in some detail by Haberl et al. 2007, PNAS). Further, HANPP is relevant to production systems (and mainly on land), but may not be so useful for describing non-production anthropogenic systems, which account for many of the 16 anthropogenic Ecosystem Functional Groups in the typology.

Despite the limitations of quantitative application, we think HANPP (along with production inputs and outputs) has great value as one of the descriptors of production ecosystems, as well as a spatial predictor in mapping systems characterised by intensive production use (see Haberl et al. 2007, PNAS), and in the assessment of ecosystem status. Our approach to typology development, which decouples classification and mapping stages (see Response 151), enables a succession of mapping advances to be incorporated as data on potential and actual NPP improves. As noted above, we used HANPP and other spatial variables to revise maps for T7.2 and T7.5.

We incorporated discussion of these issues into several parts of the main text and appendices as follows:

Methods text (lines 477-484):

“Indices such as human appropriation of net primary productivity [58], combined with land-use maps [59], offer useful insights into the distribution of some anthropogenic ecosystems, but further

development of indices is needed to adequately represent others, particularly in marine, and freshwater environments. Beyond land-use classification and mapping approaches (Appendix S1, p6), a more comprehensive elaboration of the intensity of human influence underpinning the diverse range of anthropogenic ecosystems requires a multidimensional framework incorporating land-use inputs, outputs, their interactions, legacies of earlier activity and changes in system properties [60]." Appendix S2 (p4, 2nd para):

"The strength of these effects varies from negligible to transformative, resulting in changes between ecosystems types and variation within ecosystem types spatially and temporally. For example, global models of Human Appropriation of Net Primary Productivity (HANPP) estimate variation from 0 to ~100%, as well as some areas where land use increases NPP above potential natural levels (Haberl et al. 2007). In the typology, we recognise major transformative outcomes of human activity by defining anthropogenic ecosystems as those created and sustained by human activity. We distinguish these types of ecosystems from lesser human influences where the underlying identity of a 'natural' system (i.e. its characteristic biota and ecological processes) is retained, albeit modified to some degree." Appendix S2 (p4, 3rd para):

"Characterising the intensity of human activity requires analytical frameworks that consider the multidimensional nature of land-use, water-use or sea-use intensity (Erb et al. 2013). These frameworks are most advanced for production systems (agriculture, forestry, fisheries), with conceptual models encompassing land use inputs, outputs, their interactions, and changes in system properties (Erb et al. 2013). Spatial modelling of indices such as HANPP (Haberl et al. 2007), offer opportunities to map global distributions of certain anthropogenic ecosystem types, and to assess human impacts on natural and semi-natural types, particularly where there is continuous variation in the drivers of human activity."

and specific revisions to descriptive profiles for anthropogenic Ecosystem Functional Groups in Appendix S4 detailed below.

The third group of abiotic drivers includes disturbances, such as fire and flooding. Here, I am missing a clearer conceptual definition and quantification of "disturbance". Essentially every abiotic factor described in the first two groups might be a disturbance if it occurs more suddenly and/or more intensively than usual. When natural variation begins to be a disturbance is not defined. I would help if the framework made explicit reference to the spatial and temporal scales at which the EFG-defining criteria apply.

AU RESPONSE 174: Our definition of disturbance (see main text lines 136-139 and Glossary) is based on both the discrete nature of the recurring events and specific requirements on the ecosystem response (destroying biomass, resource redistribution, life history triggers). Together, these features distinguish disturbance regimes from other biotic filters. Our interpretation is consistent with a large literature on disturbance ecology across the different disturbance types that we mention. Fig 1 identifies disturbance as typically landscape-scale phenomena (i.e. regional-local), whereas resource availability and ambient environmental factors operate from global to local scales. We think there is considerable value in separating disturbance regimes of recurring events from other types of abiotic drivers, consistent with many other contributions on ecosystems, communities and populations. See also response 59.

- I am happy with the changes.

I am not convinced that the anthropogenic assembly filters listed in table S3.2 are sufficient for distinguishing ecosystems under differing levels of anthropogenic influence. Humans, through different mechanisms, may alter almost any of the above-mentioned other filters and traits. I am probably missing something here, but none of these listed filters seems to capture humans' influence on ecosystems' energy flows through hunting, fishing, harvesting, etc. (e.g. only some of this human influence manifests via structural transformations).

AU RESPONSE: 175: We revised Fig. 1 to represent anthropogenic influences on all other drivers more transparently (see Response 19). Hunting, fishing, harvesting are different forms of resource use by humans. We mention resource appropriation in the main text and resource use in Fig 1,

but this was somehow omitted from Table S3.2. We added two rows to the table to address (abiotic) resource extraction and appropriation of biomass and productivity. Thank you for pointing out this omission.

- I am happy with the changes.

I find the conceptualization of anthropogenic biomes as being “created by human activity, which continues to drive and maintain their assembly (Ellis et al 2010)” too simplistic. Few if any human dominated systems are entirely created by humans, and all still rely on at least some nonanthropogenic ecosystem processes. Few ecosystems, in turn, are completely free of human alterations. In most cases, humans have influenced ecosystems over centuries or even millennia, and in other cases, today’s ecosystems have been co-created by humans and other species, and/or the extent to which seemingly natural ecosystems are co-created is still debated. Many scholars in historical ecology and other fields have even come to question the very concept of “natural ecosystems” and instead conceptualize all systems as human-environmental systems (or social ecological systems, etc.). I think that it should be clearly defined what is meant by either, and any underlying value judgements should be reflected on and made transparent, and that less definite wording should maybe be used throughout, e.g. including qualifiers such as “strongly”, “mainly”, etc.

AU RESPONSE 176: We deleted the problematic phrase. Although a succinct and simple definition is needed (see Glossary), we discuss human activity and distinguish anthropogenic and non-anthropogenic ecosystems in terms of high and low intensity, referring to relevant literature, including that recommended by the reviewer. We acknowledge the profound and long effects of human activity at several points in our ms (lines 467-485, copied below) and in dedicated sections of Appendices S2 and S3:

- I think the updated glossary helps a lot. However, I’m afraid it is still not sufficiently comprehensive and clear. For instance, one poorly conceptualized distinction (“human-created” vs. “natural”) cannot be resolved by simply adding another one to define it (“high intensity” vs. “low intensity”). I am sorry to belabor this so much. But even among ecologists, there will be too many different interpretations of all of these terms. And if you add interpretations in neighboring fields, you have exactly the type of ontological mess that typologies ought to help avoid.

Anthropogenic influences create challenges for ecosystem classification, as they may modify defining features of ecosystems to a degree that varies from negligible to major transformation across different locations and times. We addressed this problem by distinguishing transformative outcomes of human activity at Levels 2 and 3 of the typology from lesser human influences that may be represented either at Levels 5 and 6 or through measurements of ecosystem condition that reflect divergence from reference states arising from human activity (lines 472-477).

“Anthropogenic ecosystems grouped within Levels 2 and 3 were thus defined as those created and sustained by intensive human activities, or arising from extensive modification of natural ecosystems such that they function very differently. In many agricultural and aquacultural systems and some others, cessation of those activities may lead to transformation into ecosystem types with qualitatively different properties and organisational processes (see [56] and [57] for cropland and urban examples, respectively).”

- I appreciate the response. Still, there are some ecosystems that would be characterized as belonging to certain “natural” EFGs according to this typology, but for which the natural vs. anthropogenic origin of even their fundamental identities and functional properties is hotly debated (e.g., think of the possibly anthropogenic origins of certain savannas in North America or South-Eastern Brazil, to give just two examples). I think this should be acknowledged somewhere in the text.

Refer to Response 173 for further comment and text additions relevant to this point

This also applies to certain applied conservation statements, such as in line 282 of Appendix S6 (“Priority should be given to protecting the most intact areas that represent the range of ...”) – it would be appropriate to be transparent here, about the authors’ values on which these perceived

priorities are based (I presume, a preference for wild nature), and whether decision-makers charged with representing different values (e.g. potentially preferring ecosystem services provision close to humans) would perceive the same priorities.

AU RESPONSE 177: The line referred to by the referee was an inference drawn from the results of the spatial analysis and has therefore been deleted from the ms (see Response 1). In Appendix S6 (section on Natural capital accounting) we explain how the typology can support decisions and actions founded on objectives to manage different values of ecosystems aligned with their dual roles in biodiversity conservation and human well-being. In the final paragraph of the section on natural capital accounting, for example, we state,

“one approach to sustaining ecosystem services is to identify, protect and manage the ecosystems that provide them (Bordt & Saner 2019). The Global Ecosystem Typology offers a consistent and comprehensive framework incorporating ecosystem functions as basis for attributing ecosystem services to ecosystem types. We suggest that such an approach is likely to be more informative for ecosystem management than land cover or land use categories (Di Gregorio & Jansen 2000).”

In this respect, I also find the characterization of several functional groups in the non-anthropogenic biomes “incomplete”, as they underrepresent the influence that human activities have long had in shaping how these systems look today. For example, there is ample evidence that pre-Columbian societies actively shaped (at least heavily contributed to shaping) today’s “pyric tussock savannas (T4.2)” in Eastern North America through human-made and/or human-altered fire regimes.

Elsewhere, already prehistoric human societies have wiped out or compositionally altered megafaunal biota, etc., etc. Several EFGs, by their provided definitions, exclude several human-dominated systems that are not as intensive or as homogeneous as the definitions and examples imply and that, however, also do not fit into any of the other biomes (esp. the distinction of the T.7 biome into the five current EFGs). For example, “Annual croplands” by definition excludes permanent *Miscanthus giganteus* croplands, which are also excluded from “Plantations” due to their definition of being “woody”. Similarly, “Urban and industrial ecosystems” by definition exclude non-urban human settlements, and non-industrial infrastructure such military structures. Similarly, the distinctions provided do not adequately account for mixed systems such as agro-pastoral, silvo-pastoral systems (see specific issues related to this below). Since this typology is presented as being globally applicable, comprehensive, and systematic, and since it claims applicability in all kinds of fields, all human-made ecosystems should either be captured as individual classes, or the existing class definitions should be made more inclusive.

AU RESPONSE 178: We now explain in the ms (lines 485-491), that we focus on reference states of non-anthropogenic ecosystem types

“Where less intense human activities occur within non-anthropogenic ecosystem types, we focussed descriptions on low-impact reference states. Therefore, human activities are not shown as drivers in the assembly models for non-anthropogenic ecosystem groups, even though they may have important influences on the contemporary ecosystem distribution. This approach enables the degree and nature of human influence to be described and measured against these reference states using assessment methods such as the Red List of Ecosystems protocol [39], with appropriate data on ecosystem change.”

- I appreciate that the diagnostic EFG descriptions are focused on low-impact reference states, and this is good to have written down explicitly. Yet, by extension, this also means that the EFG descriptions may have a much more limited diagnostic value in cases where natural EFGs were already severely altered by human activities (but still not sufficiently so to qualify for inclusion in the human-created EFGs). This should be clearly acknowledged in the text, along with recommendations on how to use this typology for EFG diagnosis in such cases.

See further discussion in Appendix S3 (p22). Assessing the effects of human activity on ecosystem properties requires a wide range of appropriate methods (examples listed in Appendix S3). This is very large topic beyond the scope of our ms, which focusses on describing the typology, its rationale

and process for development, and its current and potential applications. We address specific comments on T7.1 below.

Specific comments on the human-dominated biomes and EFGs

T7: Intensive land-use systems

There is no definition nor criteria for “intensive” land-use. I find this particularly problematic as “intensive land use” is a boundary concept that can have very different meanings to ecologists than it has to climate modelers or land-use scientists. Not all agricultural and settlement lands would usually be considered “intensive” by land-use scientists. Since nearly all lands are used to some extent, it makes sense to consider the intensity of use to distinguish these systems from the other EFGs. However, land-use intensity is a multi-dimensional concept, with land-use scientists distinguishing intensity in terms of (for example) different inputs, outputs, and system-level properties (see e.g. Erb et al. 2013). It is unfortunate that there is no reference to such conceptualizations of intensity by land-use scientists. Ideally, it would be reasoned, from an ecological point of view, which of these dimensions are most critical to consider in the definition and mapping of these ecosystems, and if reasoning for any thresholds on either of these dimensions were given. The first sentence under “T7: Intensive land-use systems” gives pastoralism as an example. However, pastoralism specifically tends to be one of the less intensive ways of producing livestock (e.g. compared to intensive pasture management, etc.), and many wilderness areas of the world are in fact subject to mild pastoral use (think reindeer herders in the Russian taiga). In many pastoral systems, human intervention is arguably not a “dominating influence”, nor would those systems necessarily change their key ecological characteristics if they were no longer “maintained” by continuing human intervention (e.g. as suggested by the third sentence, rather, low livestock densities would be replaced by similarly low densities of other herbivore species). Pastoral uses are not mentioned in the respective “natural” EFGs (e.g. grassland, savannah, taiga systems), but those also do not really seem fit in here. Similarly, T.7 includes “urbanization” but does not explicitly mention “any other type of permanent human settlement”, which according to common definitions would not be included in “urban”. At local scales, however, even small villages within otherwise fairly natural environments would meet many of the other criteria for T.7. Should I consider a settlement in Siberia as T.7 or as Taiga?

AU RESPONSE 179: In general, we do not think it is appropriate to give definitive or prescriptive criteria for identification of individual units in the typology (natural or anthropogenic). The variation of ecosystems is even more complex than variation in land-use intensity, which Erb et al. (2013) note, ‘Given the complexity of land-use intensity, providing a single, unambiguous and encompassing definition or indicator of land-use intensity does not appear to be an adequate target.’

Acknowledging this problem, we added the following guidance to Appendix S4 (T7 Intensive land use systems):

“The intensity of human influence on ecosystems forms a continuum that is best assessed by multidimensional analysis of inputs, outputs, their interactions and alterations to system properties. However, most intensive harvest-based land use systems exhibit a high (> c. 40%) Human Appropriation of Net Primary Productivity (HANPP), an aggregate measure of alteration to ecosystem properties.”

As well as this additional guidance, we also note an important distinguishing feature of some intensive land-use systems – that use maintains them in disequilibrium states that may undergo large transformative changes in properties and processes when intensive use ceases. This distinguishes intensive systems from low intensity systems, such as taiga with reindeer herding activity.

We thank the reviewer for pointing to our loose usage of the term ‘pastoralism’ in the text on p78.

We replaced it with the following to distinguish the activity from low-intensity rangeland grazing:

“high-density grazing of domesticated livestock”

We clarify interpretation of villages under T7.4 below.

Further, the statement in the fourth sentence suggests that those systems generally have low endemism and low functional and taxonomic diversity, but in some parts of the world (e.g. much of Europe), it is precisely the maintenance of certain forms of land-use systems that maintains much of those regions current biota. Also, the statement is generally only justifiable with explicit reference to spatial scale of analysis. Locally (at plot scale), intensive land-use systems may almost always suppress biodiversity relative to original vegetation, but beyond the local, diversity could even be increased as even intensive land-use systems can add to the overall environmental/habitat heterogeneity of a system. It might be that a high taxonomic diversity that is to a larger extent composed of generalist species coping well with human-altered systems is not what the authors have in mind, but instead only the “natural” biodiversity. If that is so, then the authors should very explicitly lay out their criteria and their implicit assumptions about what constitutes “natural”, and make their underlying values concerning different facets of biodiversity explicit.

AU RESPONSE 180: We think low endemism is accurate. There are very few known taxa that occur only within a small area of an intensive land-use system and nowhere else in the world – most biota are cosmopolitan. We also think low functional and taxonomic diversity is a robust generalisation, though we agree that statement requires a clearer context. We made the following revision to text: “...typically *low functional and taxonomic diversity relative to comparable systems under low-intensity use, although taxonomic diversity can be higher in some groups in some systems.*”

Some statements are overly comprehensive. For example, the fifth sentence (“Target biota are ...”) includes a diversity of examples, many of which are not exclusive to intensive land-use systems. For example, selective breeding occurs among reindeer herders in the Siberian taiga, which typically shows up as a wilderness area and is presumably not included as intensive land-use systems here. Other statements are not sufficiently comprehensive. For example, “The antecedent ecosystems that they replaced include forests, shrublands, grasslands and palustrine wetlands (biomes T1-T4 and FT1).” – e.g. cropland on land claimed from the ocean would be excluded by this statement. It seems appropriate that the authors commonly use examples and non-definite statements (e.g. “typically, but not exclusively”, “many intensive land-use systems are”, etc.), as they try to capture very heterogeneous systems by these few groups. However, this can also make the classification of systems as either belonging to T.7 or to a “natural” group somewhat subjective. I guess what I would ideally love to see is some kind of identification key, but I acknowledge that this is not easily possible.

AU RESPONSE 181: We think the reviewer interpreted the text on these issues more definitively than the phrasing actually states. For example, “*Target biota are genetically manipulated...*” does not imply that no genetic manipulation occurs in other systems. Genetic manipulation does occur in other systems, but genetic manipulation is much more common and becoming more intensive in anthropogenic ecosystems (such as T7), than in other systems subject to less intensive human activity. Similarly, “...*include forests, shrublands, grasslands and palustrine wetlands...*” does not **exclude** other alternatives, but these are the most common systems applicable to the generalisation. We added transitional marine systems to the list, but these are much less commonly converted to intensive terrestrial uses on a global scale:

“...*and more rarely transitional marine systems (biomes T1-T4, FT1, MT1, MFT1)*”.

The idea of an identification key is something we have considered carefully. However, the development of such a key requires substantial additional analysis, design and extensive testing among user groups. We plan to embark on this work to develop a robust key and include it in web resources for the typology in the next stage of development.

What seems more doable would be clear distinctions between attributes that are “must haves” vs. “can haves”. If this is not possible, I feel that this caveat should be clearly acknowledged, and the typology should not be oversold as being systematic and rigorous. “On global and regional scales, intensive land-use systems are engaged in climate feedback processes via alterations to the water cycle and the release of greenhouse gases from vegetation, soils, livestock, and fossil fuels.” – something similar could also be said for some other ecosystems with strong dynamics between

alternative states. E.g. the Savanna-Forest transition systems in the African savanna belt (Staver paper).

AU RESPONSE 182: We appreciate that users desire clarity, but we do not think a formal division of ecosystem properties into “must haves” vs. “can haves” reflects on systematic qualities or rigour. The design principles, conceptual framework, definitions of concepts and terms (Glossary), development process, and extensive review process contribute to systematic qualities or rigour of our typology. There are many other systematic and rigorous classifications of nature that are not based on artificially definitive membership rules, recognising inherent multidimensional variability and uncertainties in ecosystems. Indeed, some quantitative methods embrace and accommodate uncertainty and variability among classes by assessing overall evidence on group membership across multiple dimensions where quantitative data exist, e.g. fuzzy clustering. We draw attention to the introductory text to the descriptive profiles (Appendix S4, p13 2nd paragraph), which we have revised as follows to further address this issue:

“Inevitably, there are inherent uncertainties in assigning ecosystem types to unique EFGs because ecological classifications, in general, simplify complex multidimensional variation in nature by segmenting and categorising continuous gradients in multiple features (see Appendix S3; Regan et al. 2002). Thus, any given ecosystem type may possess a suite of features that are typical of different functional groups, and a single feature can rarely be definitive for ecosystem identification (e.g. Erb et al. 2013). For this reason we avoid prescriptive approaches to description of the units that seek to identify strictly exclusive or diagnostic ecosystem characteristics, and instead use appropriate qualifiers and caveats in descriptions where important exceptions apply to generalisations about ecosystem properties and postulated drivers. Users should assess and weigh evidence on all features to identify the most likely functional group and report the nature of uncertainties in group membership.”

- I like this revised text and agree with the statements therein. However, these statements acknowledging the inherent difficulty of diagnosing ecosystems do contrast with the much bolder (and/or less differentiated) claims elsewhere in the paper that this typology does have a high diagnostic utility (e.g., lines 214 & following, or in sections contrasting this with the lower diagnostic power of other typologies, e.g., lines 110-112). I think that this acknowledgement should be reflected in more careful wording throughout the main manuscript Appendix S1.
- My other comments (specifically that some statements seem too exclusive and some seem too definitive), as well as my thoughts on a possible distinction of “can haves” and “must haves”, were not to imply that the authors need to do this (specifically), but simply to indicate that all ecosystem descriptive profiles should arguably reflect this multidimensionality of fuzziness. An oversimplified statement “X is caused by Y” can too easily be misunderstood by readers as “Y is generally the exclusive cause of X”. To avoid misinterpretations, I suggest that the authors should include qualifiers such as “typically”, “often”, or “rarely” rather generously.

T7.1 Annual croplands

Ecological traits:

The characterization of ecological traits starts with a qualifying phrase “Structurally simple, very low diversity, high-productivity annual croplands are ...” – this phrase obviously does not cover all croplands. I did not find another class in which the other croplands that are by definition excluded (at least if a land-use scientist reads these definitions), such as permanent croplands, low-productivity annual croplands, etc. The authors should either add additional classes to their ecosystem typology or use more accurate terms and adequately comprehensive definitions for the existing ones. Again, a distinction of defining characteristics into can-haves (e.g. “supplementation of nutrients”) and must haves (e.g. “artificial disturbance regimes”) would seem in order.

Swidden agriculture does not seem to fit in to any described EFG. T7.1 might seem to include swidden agriculture according to the statement "... or subsistence production of food". But then again, this land-use form would definitely be excluded by the statement "structurally simple", "high productivity", and "When actively managed systems are abandoned [...] these non-target biota [...] become dominant and may form a steady, self-maintaining state or a transitional phase to novel ecosystems."

Some statements are so definitely phrased that they are unlikely to be true. And even if they were, I'm afraid the burden of proof whether or not they are always true is on the authors when making such definite statements. For example, consider the sentence "... these systems have very low functional, genetic, and taxonomic diversity and no local endemism". How can the authors be so sure that annual croplands never, nowhere in the world, have even a single locally endemic host-specific crop pest species that coevolved with the crop? Another sentence that should be phrased as a "can-have" is "Target biota coexists with a cosmopolitan ruderal biota (e.g. weedy plants, mice, and starlings) that ...". There are indeed many localized examples of species with restricted global distributions that primarily use natural ecosystems, but benefit from nearby intensive farming – that these species can effectively exploit those intensive cropland systems may depend just as much on the landscape composition as on the specific attributes of the croplands or the species per se. E.g., think of *Macaca nemestrina* opportunistically leaving rainforests fragments to hunt rats in SE-Asian palm-oil plantations – I do not think that these would classify as "cosmopolitan ruderal biota". Even orangutans have been observed feeding in palm-oil plantations. I do not doubt that palm-oil plantations by and large harm the latter species, but the very restrictive definition provided is still not always true.

AU RESPONSE 183: Thank you for these suggestions. We made several revisions to T7.1 to address the issues raised. We deleted "Annual" and simplified the name of the group to "Croplands" and in the text replaced "annual" with "...typically dominated by one or few shallow-rooted short-lived plant species such as...". We deleted "Structurally simple, very low diversity" from the opening sentence. We qualified herbicide and pesticide application, "...and usually by periodic application of herbicides and pesticides...". We placed generalisations about diversity in context, "...compared to antecedent 'natural' systems, croplands are structurally simple, have low functional, genetic, and taxonomic diversity and little or no local endemism," and contrasted tropical croplands with industrial croplands as follows, "Subsistence croplands, including Swidden rotation systems, are typically more diverse than industrial croplands." After the sentence on ruderal biota we added, "Native biota from adjoining non-anthropogenic systems may also interact with croplands."

- I think the changes made have much improved this EFG description. In the first sentence under "Ecological drivers", I suggest adding the word "often" before "...supplemented by human inputs ...", as there are many entirely rainfed and unfertilized croplands in the world.

T7.2 Sown pastures and fields

Ecological traits:

Again, a better distinction of "can-haves" and "must-haves" would be necessary to make some of these statements accurate. The first sentence, for example, connects several traits by "and" that in the real world do not always co-occur. Many of the comments already given for EFG "T7.1 Annual croplands" also apply here.

AU RESPONSE 184: We revised the name of this group to "Intensive livestock pastures", recognising that sowing, while widespread, does not apply to all examples. We added qualifications and revised the text as follows. We deleted "low diversity" from first sentence. We revised and qualified the sentence on diversity and endemism as follows: "Consequently, compared to antecedent rangeland systems and semi-natural pastures, these systems have low functional and taxonomic diversity and little or no local endemism." We added "or maintenance" to "...harvested by humans continuously or periodically for consumption." We added "Typically, at least 40% of net primary productivity is

appropriated by humans.” We deleted “lawns and sporting fields”, which were incorporated as elements within T7.4. We added a reference to mixed systems as follows “Livestock pastures may be rotated inter-annually with non-woody crops (T7.1), or they may be managed as mixed silvo-pastoral systems (T7.3)”. Finally, we note that “and” applies to collective characteristics of the group as a whole, not (necessarily) to co-occurrence in space.

Key ecological drivers:

Again, without a distinction of “can-haves” and “must-haves”, the described drivers do not allow for a reliable identification.

AU RESPONSE 185: We qualified “is supplemented” by “is typically supplemented” and added that fertilisers may be “...applied at varied rates”, noting that these may be inorganic chemical or organic forms of fertiliser.

- The description is improved. However, I wonder where the 40% HANPP threshold comes from? Is this the authors’ assumption or is there any basis for this?
- I would additionally add a statement “Native biota from adjoining non-anthropogenic systems may also interact with ...” similar to what was added to the T7.1 description.

Map:

The indicated data sources for this EFG were a set of livestock density maps (Gilbert et al.) and maps of areas that are equipped for irrigation (Siebert et al.). The latter appears to drive much of the patterns in this EFG map, as it was generated by overlaying irrigation-equipped areas with “major” (presumably thresholded) livestock densities. I believe this mapping protocol to be flawed on several conceptual grounds. Firstly, the livestock densities were not designed to depict any specific type of grazing. They were modelled across all land-cover types to account for the fact that in many parts of the world, livestock does not occur on lands dominated by forage herbs, but instead, for example, in silvo-pastoral systems, mixed cropping-livestock systems, land-less industrial farms, and even inner cities. Secondly, with a few exceptions, the areas that are globally equipped for irrigation are mainly used for agriculture, not for creating artificial pastures. High livestock densities intersecting irrigation equipped areas are certainly not globally indicative of specifically sown pastures. In many areas with extensive rice irrigation (especially in India, but also in SE-Asia), there are mainly mixed cropping livestock systems. In these systems, the irrigation is exclusively for crop production, whereas livestock mainly graze on patches of grassy vegetation in between the field (but those are certainly not sown) and on the field themselves (after harvest). In other areas, land-less livestock production systems co-occur with irrigation-cropping systems at the coarse grains where they were mapped. The illusion that the irrigation is in any way related to the livestock, there, is merely a result of the spatially coarse nature of these maps. Finally, it is not clear from the description what thresholds were applied for livestock densities. Global thresholding is generally tricky here, as livestock densities supported by a grazing system depend as much on its natural vegetation productivity as on the artificial inputs it receives. This can certainly be done, but would definitely require some way of locally validating the resulting map. For the above reasons, I do not believe this EFG map to have high validity, not even as an indicative map.

AU RESPONSE 186: We agree with the referee that these proxies are not ideal predictors of the spatial distribution of sown pastures. We applied the irrigation layer mainly to exclude rangeland grazing areas from the map but, as the reviewer points out, there is considerable potential for false positives. We removed the spatial analyses and research findings dependent on maps from the manuscript, as recommended by the editor (see Response 1). We improved the indicative map by removing the irrigation layer and combining global maps estimating livestock density with the estimated human-appropriation of net primary productivity. We explored a number alternatives for combining these data and selected options that represented uncertainties through major and minor occurrences consistent with the concept of the EFG. We explain the data sources and derivation in Table S4.1:

“Mapping of intensive livestock pastures was based on fractional land use mapping (Ramankutty et al. 2008), dasymetric estimates of ruminant livestock density for cattle and sheep (Gilbert et al.

2010), and Human Appropriation of Net Primary Production (HANPP, Haberl et al. 2007). Fractional land use cover indicated firstly where pastures occur and secondly where they occupy a large portion of area relative to croplands. This helped to exclude intensive croplands that are also used to graze livestock, either through temporal rotation or on the margins of cropped paddocks (e.g. in south Asia). Livestock densities indicated where ruminants were important components of pasture systems, and helped exclude some rangelands with low livestock densities. Finally, HANPP helped exclude low productivity rangelands with high stocking rates and additional areas of cropland. We explored different combinations and thresholds for the input data layers, visually inspecting the output in South Asia, Australia, West Africa, and North and South America. We then mapped major occurrences where pasture area fraction greater than zero ($PAF > 0$) and greater than cropland area fraction ($PAF - CAF > 0$), densities of cattle or sheep were greater than 500 per cell, and $100 < HANPP < 700 \text{ gC/m}^2/\text{yr}$. We examined the sensitivity of mapped area to variation in these thresholds and found no appreciable change in the global mapped area when livestock density was varied by $\pm 20\%$ and marginal change in mapped area with variation in the other thresholds by the same amount. To represent this uncertainty, we mapped minor occurrences as the additional area where $PAF > 0$, $PAF - CAP > -0.2$ and $80 < HANPP < 840 \text{ gC/m}^2/\text{yr}$.”

- This is an example where I firmly context that the presented map has even indicative value for this EFG. Many larger areas mapped in red in South America and Africa (in particular), certainly do not meet any of the management criteria described for this EFG, but instead correspond to semi-natural pastures and rangelands (e.g., to be included in biomes T4 and T5).
- Note that the fractional land-use maps of Ramankutty et al. 2008 cannot reliably distinguish intensive livestock pastures as described here from semi-natural pastures and rangelands in biomes T4 and T5, due to inconsistent definitions in the land-use data underpinning this map. This limitation is acknowledged by Ramankutty and colleagues and generally appreciated in the land-use science community.
- Using a livestock density as a standalone indicator to distinguish intensive livestock pastures as described from semi-natural pastures and rangelands is also problematic, as there can be low-livestock-density artificial pastures just as there can be high-livestock-density natural pastures and rangelands. Supported livestock densities on certain rangeland systems (i.e. in natural grasslands or savannas) with a high natural vegetation productivity can be very high, even without any of the specific management measures listed as qualifiers for this EFG.

T7.3 Plantations

This EFG seems mostly well done.

Thanks.

T7.4 Urban and industrial ecosystems

Ecological traits:

This EFG seems mostly well done. However, by definition, this EFG excludes non-urban settlements (villages, etc.). These also fit in none of the other classes, but cover substantial surface area globally. I see this as a major gap in the presented typology.

AU RESPONSE 187: Thank you for this suggestion. We changed the label from “Urban and industrial ecosystems” to “Cities, villages and infrastructure” and replaced “urban systems” in the text with “urban/industrial/village systems”.

- I am happy with these changes.

Map:

I wonder why no map of actual urban areas was used but instead an estimation from artificial nightlights. It is well known in the urban mapping scene that settlement indications based on artificial nightlights are regionally biased by wildfires and especially energy production platforms. The provided description in the IUCN Typology document does not suggest that this was accounted for.

AU RESPONSE 188: As noted above, we removed the spatial analyses and research findings dependent on maps from the manuscript, as requested by the editor (see Response 1). Although we believe night lights provided an adequate proxy for distribution of this group that excludes transient wildfires, we replaced it with a more recent land use data set. Both night lights and land use data may exclude weakly lit or unlit villages, but these are likely to account for a relatively small area. The source and derivation are described in Table S4.1:

“The distribution of urban and industrial infrastructure lands was taken from a global land use/land cover map (LULC class 7 ‘built areas’) for the year 2020 at 10 metre resolution (Karra et al. 2021). Class 7 includes major road and rail networks, large homogenous impervious surfaces including parking structures, office buildings and residential housing, dense. Sparse villages may not be represented. We calculated the proportion of built area per square kilometre and applied a threshold of 1 to 5 % for minor occurrences and >5% for major occurrences.”

- I am happy with these changes.

T7.5 Derived semi-natural pastures and old fields

Ecological traits:

The description of the ecological traits of this EFG is more convincing than some of the others in this biome. However, some of these statements do not apply generally but are presented as if they did. For example, they are presented as generally being “structurally simpler than the systems from which they were derived”. This does not seem to be universally true, as in some cases, structurally highly homogeneous natural vegetation may be replaced with more complex mosaics of pasture and leftover original vegetation (small groups of trees, etc.). Adding a simple “typically” would help here. “Productivity [...] is generally [...] more stable than more intensive anthropogenic systems”. I would question the extent to which this claimed generality is really established.

AU RESPONSE 189: We think a sentence about structural complexity at site scales may have been misunderstood as a comment on landscape spatial complexity. We clarified the text as follows:

“Although structurally simpler at site scales than the systems from which they were derived, spatial complexity may be greater in fragmented landscapes and...”

We reasoned that productivity should generally be more stable in derived grasslands than in more intensive land-use systems because derived grasslands are generally less often and less intensively harvested, ploughed, fertilised, etc. than those other systems.

- I am happy with the updated description of this EFG.

Map:

This EFG is not included in the Methods description in the IUCN Typology document, so I tried to speculate what this might have been, but frankly, I failed. It is puzzling to me how a meaningful map of this EFG should be derived from the indicated data sources. Those sources are a consensus landcover map distinguishing only coarse land-cover types (Tuanmu & Jetz) and a map of approximate areal boundaries of ecosystem extents (Dinerstein et al.), neither of which contains any information on the semi-natural character of any of the grasslands shown therein. Exploring the “indicative map” of this EFG and comparing it with the Tuanmu & Jetz data looks like this map was driven largely by the Tuanmu & Jetz class “Cultivated and Managed Vegetation” (e.g., close correspondence in patterns clearly visible for India, China, West Africa, Argentina, etc.). That class almost basically reflects indications of “cropland” in the different data sources that underpin the Tuanmu & Jetz consensus product. Without having access to any concrete description of the mapping protocols, I can only assume that this class might have been misinterpreted as showing “managed vegetation other than cropland”. Based on the information I have, I can only say that this map appears highly flawed. I do not believe that it has any validity even as an indicative map.

AU RESPONSE 190: Thank you for pointing out the omission from Table S4.1. We have now added the entry for T7.5 and used alternative sources data that more directly represent the suitability and use of these lands for grazing without intensive interventions that characterise T7.2. This group was especially challenging to map. For the indicative global map, we identified a plausible envelope of occurrence by identifying high suitability for grazing (based on Erb et al. 2007), excluding areas that

Erb et al. (2007) estimated to have i) a higher proportional area of cropping or forestry; ii) a low proportion of grazing lands; and iii) a high HANPP. We reasoned that i) and iii) should exclude harvested crops (e.g. cereals), plantations and intensively managed sown pastures, while ii) should exclude wild low-productivity rangelands used for low intensity grazing. We are currently exploring other alternatives that sharpen the focus on the features that characterise this group of systems. We removed the spatial analyses and research findings dependent on maps from the manuscript, as recommended by the editor (see Response 1).

- I see a number of issues in this mapping methodology:
- At the mapping resolution of the Erb et al. product, the exclusion of areas with a higher proportional area of cropping or forestry / a lower proportion of grazing lands, as described, makes no sense in parts of the world where semi-natural pastures are small (e.g. much of Central Europe).
- I do not see how excluding areas with a low proportion of grazing land would help separate derived semi-natural pastures/old fields from livestock-grazed rangelands. Both systems occupy large and small areas.
- High HANPP is also not a sensible discriminatory factor here. In many drylands (especially in overgrazed ones), the rangelands are naturally fairly unproductive and thus only support very low livestock stocking densities. Yet, those few livestock individuals make up for a very high HANPP on a proportional scale (i.e., the little productivity that there is largely appropriated)
- Also, as noted further up, linking the concept of “rangeland” to that of “low productivity” is generally problematic. Rangelands exist that have rather high livestock densities, but are still distinct from the systems describes in T7.2 and T7.5.
- Honestly, the challenges of mapping different livestock systems are so widely acknowledged in the land-use mapping community, that I would find it extremely problematic to present any maps at all (however they were derived) as indicative of the distinguished livestock-related EFGs if those were not made subject to any formal validation.

SF2. Anthropogenic subterranean freshwaters biome

I cannot comment on this biome, as this is outside my expertise.

F3. Artificial wetlands biome

From my assessment, there are fewer conceptual flaws in the definitions of EFGs in this biome than in the terrestrial biome. In particular, the descriptions of their traits and associated biota seem more nuanced than those for terrestrial human-modified EFGs. But my own expertise is also more in landuse systems than in water-use systems, so in case this is critical for the decision, I would suggest to consult an expert on artificial freshwater systems.

AU RESPONSE 191: We note that five freshwater specialists reviewed the profiles of Ecosystem Functional Groups within the Artificial freshwaters biome (see Appendix S5), and at least one of the eight reviewers for Nature was a freshwater specialist.

F3.1 Large reservoirs

Map: I believe the indicative map to be of little utility. There are much more complete datasets of dams and reservoirs available than the cited one: <http://globaldamwatch.org/>

AU RESPONSE 192: Thank you for drawing our attention to this update. We removed the spatial analyses and research findings dependent on maps from the manuscript, as recommended by the editor (see Response 1) but have included the revised map as a thumbnail in the profile for F3.1 in Appendix S4.

- As the different databases of dams and reservoirs are all based on ongoing inventory work that so far is spatially highly incomplete (i.e., missing entire countries due to lack of data), I suggest that the map is not represented as “indicative” but instead as “indicative but regionally incomplete map”, or similar.

F3.3 Rice paddies

This EFG seems mostly well done.

Thanks

F3.2 Constructed lacustrine wetlands & F3.4 Freshwater aquafarms & F3.5 Canals, ditches and drains
Map:

The basis for mapping both types of EFGs has been the Freshwater Ecoregions of the World datasets by Abell et al. The criteria by which the maps freshwater ecosystems have been filtered to those “indicative” of either of these EFGs is unclear. I also think this has been with little grounding in actual evidence. Both EFGs certainly exist outside those indicated areas.

I do not understand why freshwater ecoregion maps were used for mapping these EFGs in the first place, as those ecoregion maps were not designed to capture signs of human alteration of “natural” systems. To quote from Abell et al.: "Ecoregions are intended to depict the estimated original extent of natural communities before major alterations caused by recent human activities". In any case, the coarse nature of these maps would arguably render their utility very low.

AU RESPONSE 193: We agree that freshwater ecoregions provide a poor template for mapping these Ecosystem Functional Groups. We removed the spatial analyses and research findings dependent on maps from the manuscript, as recommended by the editor (see Response 1) but we explored additional alternative options. All of these options have significant limitations, but are likely improvements on freshwater ecoregions. The global surface water database with small artificial water bodies intersected with grazing lands provide an indication for F3.2. F3.4 relies on incomplete and largely inaccessible industry information, and thus remains a very general envelope of potential occurrence. Maps of irrigation infrastructure and urban areas enabled an improved map for F3.5.

- I do not think that much worth lies in a claimed but unquantified “improvement” in mapping a specific system relative to a map that explicitly did not want to map that system. Since the authors so clearly acknowledge that some of the EFGs cannot currently be mapped with any reliability, I do seriously wonder why they insist on mapping them at all. This certainly does not meet standards for map-making in geography nor biogeography.
- Therefore, I strongly believe that the value of this typology for the community would be greater if for some of the EFGs it was simply acknowledged that even indicative global mapping is currently not possible based on available information. That would at least be honest and not overrepresent the state of knowledge.

M4. Anthropogenic marine biome

I am missing an EFG for non-submerged marine artificial structures (e.g. artificial islands, oil platforms, etc.). I cannot comment in detail on the conceptual validity of these EFGs, as these systems are mostly outside of my expertise.

AU RESPONSE 194: Artificial islands and supratidal components of oil platforms are included within MT3.1 Artificial shorelines. The marine component of oil platforms is within M4.1.

M4.1 Submerged artificial structures

This EFG seems mostly well done, though I cannot confidently comment on the utility on the validity of the indicative map. It does seem empty in many parts of the world with substantial industrial activity in coastal areas, so it seems unlikely that this is very complete.

AU RESPONSE 195: We agree, this map is likely to be incomplete, as many wrecks and much marine infrastructure is not centrally documented or is undocumented. We removed the spatial analyses and research findings dependent on maps from the manuscript, as recommended by the editor (see Response 1) and noted the likely incomplete status of the map in the Distribution text,

“Map is incomplete but shows areas with many documented wrecks and marine infrastructure.”

- I like the inclusion of this statement. Similar statements should be included in all maps of EFGs that are based on globally incomplete inventories (e.g., dams, caves, etc.)

M4.2 Marine aquafarms

This EFG seems mostly well done.

Thanks.

MT3. Anthropogenic shorelines biome

I cannot comment on this biome, as this is outside my expertise.

References

Boitani, Luigi, Georgina M. Mace, and Carlo Rondinini. "Challenging the scientific foundations for an IUCN Red List of Ecosystems." *Conservation Letters* 8.2 (2015): 125-131.

Erb, Karl-Heinz, et al. "A conceptual framework for analysing and measuring land-use intensity." *Current opinion in environmental sustainability* 5.5 (2013): 464-470.

Erb, Karl-Heinz, et al. "Land management: data availability and process understanding for global change studies." *Global change biology* 23.2 (2017): 512-533.

Author Rebuttals to Second Revision:

Referees' comments:

Referee #1 (Remarks to the author)"

I have reviewed this ms since its first submission. I believe the authors have done a good job of addressing concerns and it is basically ready to publish.

My only suggestion is that the GIS files of the 108 regions be published with this manuscript even though the majority of "spatial analysis" (i.e. amount of each ecoregion that is protected and rate of human change) is regrettably but correctly postponed to another ms. I think the typology will be hard for anybody else to use without these GIS files (and this should become the anchor point for the use of the typology).

I understand concerns came up in review (I had one of them about being raster). **But the utility so vastly outweighs the downsides of providing nothing that I think they should be published with appropriate caveats and limitations.**

Otherwise, I think this is an exciting project.

AU RESPONSE 1.1: Thank you for the positive remarks. We are comfortable with a compromise to cite the spatial data in a separate data publication (ref 15 in main text) and to present the maps only as indicative thumbnails in Appendix S4 of this paper.

Referee #2 (Remarks to the author):

I thank the authors for their considerate replies to my comments. I am satisfied with the replies. I also took the time to read all 93 pages of the rebuttal document, and commend the authors for their care in addressing everybody else's comments, some of which I very much agree with in retrospect.

I think the paper now reads much better. Focusing on the typology and presenting it in more detail (and with more nuance) has in my view given the paper a more solid foundation. I appreciate the more expanded Methods section and the clearer figures. I am convinced **this framework will become a landmark in the ecology and conservation literatures, and I thus recommend its publication.**

AU RESPONSE: Thank you for the positive remarks.

Some comments and a few additional recommendations (optional):

In contrast with Figures 2 and 3, **Figure 1 looks quite amateurish.** Which is a shame, as this is arguably the most important figure, and one likely to be reproduced many times in the future. Any chance you could ask the people who did Fig 2 to help making this one a bit sleeker?

AU RESPONSE 2.1: Thank you for the suggestion. We have made some amendments to Figure 1 to improve its appearance. We would be happy to work with production staff to improve it further. An important consideration is to ensure that Figure 1 remains in a comparable form to those in the descriptive profiles in Appendix S2 because it is the template on which the assembly models for all ecosystem functional groups are based.

The new Figure 2 is a major improvement in relation to the previous version. It is very useful to see the full list of Realms and Biomes, and then examples of Ecosystem Functional Groups for most Biomes. I think this figure will be key to help the reader's mind navigate back and forth from

conceptual to concrete. I predict it will be a key figure associated with this new typology in countless future documents.

AU RESPONSE: Thank you for recognising the significance of Figure 2.

Minor points/suggestions regarding Figure 2:

1) In some cases, the EFGs' rectangles (on the left side of the figure) appear merged (e.g., 8-9-10 in Lakes).

AU RESPONSE 2.2: We think this is an artefact of importing the figure into Word and then converting it to pdf. The original figure does not have this problem.

2) Good idea to represent three examples of multidimensional environmental gradients as arrows on the figure. To better drive home the message that these are independent axes, I would have drawn the top one (temperature) diagonally, rather than vertical as for light & nutrient availability.

AU RESPONSE 2.2: Thank you for the suggestion, we wanted to portray the temperature gradient vertically because of its correlation with altitude.

Figure 3 is a nice addition too.

Regarding the very long Table 1, I think Referee 1 was right that this was missing from the previous version, but I find it much less crucial now that the new Figure 2 lists all Realms and Biomes, and illustrates some Ecosystem Functional Groups. As the authors point out, this table is really long. I would instead move it to Appendix S4, where I think it would make a useful summary (particularly if it included hyperlinks to each EFG).

AU RESPONSE 2.3: Thank you for this suggestion. We agree that Table 1 is too long for the main text. However, we do not think it sits readily in Appendix S4. Instead, we propose to present it as Extended Data where it will be more accessible as further detail to Figure 2.

Instead of Table 1, I would recommend adding an additional figure going into more detail for some of the EFGs from Fig 2, still in the spirit of helping the readers understand the concrete aspects of this conceptual framework. Specifically, I would recommend a figure with two (or three?) horizontal panels, each illustrating an EFG, and including: a brief summary of its typical key features, a (sleek) diagram of the drivers, and a little map with the distribution (and maybe the corresponding drawing from Fig 2, creating a link between the two figures). This would strengthen the value of the main text as a stand-alone piece, and encourage readers to go and see Appendix S4 for more detail.

AU RESPONSE 2.4: Thank you, we agree that it would be helpful to include an additional figure in lieu of Table 1 to consolidate the conceptual framework. Rather than a graphical summary that samples content already presented in Appendix S4, we designed a new figure that presents hypothesised relationships among ecosystem functional groups and their associations with gradients representing major assembly filters. Figure 3 illustrates these relationships in three panels with examples from terrestrial, freshwater and marine realms. By representing ecosystem functional groups with 'soft' boundaries, the new figure also portrays the units of the typology in the context of continuous variation in nature (main text lines 288-296).

Referee #3 (Remarks to the Author):

I read with much interest the revised version of this manuscript. I must first praise the authors for the work they have done during the revision of the manuscript (it is certainly not easy to deal with so many thorough reviews and comments!). I have carefully revised the response letter provided, and I feel they have effectively incorporated most of the criticisms I provided. Please note that I have not reviewed in-depth appendices other than S4, and within this one I focused on the typologies I revised in the previous section, so I assume that other reviewers will have specifically revised them.

Focusing on the novel typology itself has been a very good choice, as this makes the manuscript more focused and removes some major issues with the analyses included in the previous version. Overall, I think the ms has substantially improved and is easier to navigate, and I have not identified any major issue with this version of the text.

AU RESPONSE 3.1: Thank you for the positive comments on our manuscript.

Said so, I have some (mostly minor) comments, focused mainly on the structure/clarity of the text and its presentation, which I would advise the authors to consider:

- While the paper is now more focused, I found that the structure/order of the paragraphs makes the text not very attractive to read. Of course, I acknowledge that this is matter of style (and thus largely subjective) but I would encourage the authors to think about the structure of the text to make it as attractive as possible. For example, some paragraphs would certainly benefit from an internal reordering (e.g., L103-116, see my suggestion below for this paragraph; it would be also good to add references when referring to existing classifications in this paragraph to provide a better context and to facilitate readers the comparison of the newly introduced typology with existing ones), others break the logical flow of the text. For instance, paragraph in lines 267-275 would be better place before the previous paragraph, so the final paragraphs of the ms can focus on the strengths (rather than on the limitations) of the proposed classification.

“We used these six design criteria to review a sample of 23 global-scale ecological typologies, finding none that explicitly represented both ecological functions and biota (Table S1.2). This limits the ability of ecosystem managers to learn from related ecosystems with similar operating mechanisms and drivers of change. Many of the existing typologies either failed to describe their units in sufficient detail for reliable identification, or required diagnostic features that are hard to observe. Others were based on biophysical attributes or biogeography, but approaches differed across terrestrial, freshwater and marine domains, precluding a truly global approach. Only three typologies encompassed the whole biosphere, but these lacked a clear theoretical basis, limiting their ability to generalise about properties of ecosystems grouped together. Ecological classifications based on tested and established theory are more likely to be robust to new information than classifications based only on observed patterns and correlations, which may prove unstable when new information emerges. In this study, we developed a global ecosystem typology that meets all six design criteria, thereby providing a stronger foundation for systematic ecosystem assessments, sustainable management, and biodiversity conservation”

AU RESPONSE 3.2: The original order of text followed the order of the design criteria Table S1.1. We agree that this is not essential and have re-arranged this paragraph as suggested.

- Title: The title could be more dynamic and appealing to readers, what about "A function-based typology for preserving and sustaining Earth's ecosystems"?

AU RESPONSE 3.3: Thank you for the suggestion. On balance, we think a title that begins with 'Earth's ecosystems' is more likely to attract attention than one that begins with reference to the typology.

- L 74-75: I think "The outcome of a major cross-disciplinary collaboration" does not add much here, could be removed to save space

AU RESPONSE 3.4: We deleted this phrase.

- L 82: I feel that "irrespective of specific biota within the ecosystems" is a little bit misleading and may confound readers as the processes and mechanisms that underpin ecosystem functioning rely on biota. Rewrite or delete this part of the sentence.

AU RESPONSE 3.5: We deleted this phrase and added a sentence on lines 87-88 to make the point more clearly:

"Ecosystems with different species composition may show functional convergence, if their biota share similar traits and contribute to similar ecological processes (e.g. [8])."

- L 78-79: perhaps this sentence could be better framed as "The classification introduced here can guide policy transformation for ecosystem-specific action, including ... (list some of the key actions this classification may be particularly useful for)"

AU RESPONSE 3.6: Thank you for the suggestion, but we think this would change the meaning of the sentence, which identifies both general and ecosystem-specific applications.

- L 85: This sentence is confusing. To which functions do you refer (in the previous sentence you mention functions, biota...)?

AU RESPONSE 3.7: We edited the preceding sentence (lines 82-84) to make it clear that the functions refer to biomass production and stocks and fluxes of resources, energy and biota:

"Ecosystem functioning not only underpins biomass production, but also depends on, and regulates the stocks and fluxes of resources, energy, and biota [7]."

- L 86: I found the definition of ecological processes given in the Glossary ("Activities that result from interactions among organisms, and between organisms and their environment (after Pettorelli et al. 2018)") somewhat confusing. What do "activities" mean in this context? Please rewrite this definition for clarity or further explain it so everyone will understand the meaning of this important term (it is used multiple times throughout the text). Also, what is the difference between "ecological processes" and "ecosystem processes" (the first term is defined, the second not but is embedded within other definitions, e.g. that of "ecosystem functions").

AU RESPONSE 3.8: We are reluctant to change this definition, which comes from Pettorelli et al. 2018, who selected it from several alternative definitions that they reviewed. The common language meaning of activity applies (condition in which things are happening), hence a definition in our paper should not be required. Pettorelli et al. (2018) distinguish ecological processes () from ecosystem

processes.

- I know this does not have an easy solution, but Table 1 seems too large to be part of the printed main text. The editors will know better what would be a suitable alternative, but an interactive version online, which could also allow adding pictures for each EFG seems the best solution for this Table. Please also note that the * present together “Typical key features” is not defined, something that may confound readers (like me!).

AU RESPONSE 3.9: We agree and have moved Table 1 to Extended Data.

- In Appendix S4 there is a mention to Appendix S7 (page 15) that does not exist. Should it be Appendix S6?

AU RESPONSE 3.10: Corrected to S6 – thank you.

- In Appendix S4, I would also define “major occurrence”, as done with “minor occurrence” This will help readers to better interpret the maps provided. I would also include the justification you provided in the response letter “Omitting minor occurrences from maps, or merging minor and major occurrences (in which the EFG comprises the majority of the landscape matrix) would involve significant loss of information, notwithstanding acknowledged limitations on accuracy and precision” as part of Appendix S4 to further justify the approach followed.

AU RESPONSE 3.11: We expanded the relevant text in Appendix S4 (in the Indicative distribution maps section) so that it now reads as follows:

“The maps show areas of the world containing major (in red) or minor occurrences (in yellow) of each EFG. Major occurrences indicate areas where an EFG occupies a large portion (generally >20%) of a landscape or seascape. Minor occurrences are areas where an EFG is scattered in patches within matrices of other EFGs or where they occur in large patches but only within a segment of a larger region. Distributions that are uncertain are mapped as minor occurrences across large geographic envelopes. Small but important occurrences are identified with black ellipses. In landscapes or seascapes occupied by mosaics of ecosystems, EFGs comprising the matrix of the mosaic are mapped as major occurrences, and those distributed in patches are mapped as minor occurrences.”

- The new name of biome T4.5 is now more appropriate and will avoid many confusions, I think. I have not checked all the biome names (surely the authors will have done it), but please check them to minimize having names that would mislead readers (as it happened to me with the previous version of the T4.5 name).

AU RESPONSE 3.12: We checked the names with reference to previous interactions with co-authors and reviewers. We follow the nomenclature principles outlined on p13 of Appendix S4.

Again, I applaud the authors for the effective revisions conducted and I hope that this new set of minor comments will be helpful to further polish and improve the presentation of this new typology, which undoubtedly will interest scientists, managers and policy makers alike around the world. **AU**

RESPONSE 3.13: Thanks for these positive remarks.

Referee #4 (Remarks to the Author):

The manuscript by Keith et al. describes the IUCN global typology of ecosystems, based on the

functional traits of the species and ecosystems. In this era of global change and human impacts there is a dire need to make global comparisons of ecosystems, the impacts they receive, and their threats. As I said before, the manuscript has many strengths and will be widely used, and for these reasons it merits publication in Nature:

- 1) a unifying global typology that allows to classify bewildering different ecosystems that are often assessed separately (e.g., terrestrial, freshwater, marine),
- 2) a consistent classification based on community assembly and the functioning of organisms and ecosystems, which allows for a better mechanistic understanding, assessment and prediction of the consequences of environmental change,
- 3) a separate inclusion of humans as an environmental driver, which allows to assess the consequences of human activity for the biodiversity and functioning of the planet, and allows to design policies to change human activities or mitigate their effects,
- 4) the application of the typology by describing and mapping the 108 ecosystem types. The description in Appendix S1 with one page factsheets are a pleasure to read, as they are nice and concrete, succinct, well written, conceptually consistent by showing the same conceptual diagram with different drivers, and nicely illustrated with a clear beautiful photo conveying the message, and reference for further reading. This is a treasure for many biologists and interested readers to appreciate and understand the diversity and beauty of life, and will be a global reference for at least the coming 1.5 decade.

The authors have done a thorough job in replying to all comments, in a rebuttal letter of 93 (!) pages.

AU RESPONSE 4.1: Thank you for these very positive remarks.

MAIN TEXT IS UP TO PUBLICATION STANDARDS. The main text has been substantially been improved by taking out the threat analysis, and using the gained space to highlight the strengths, limitations, and potential of their classification. The main text is therefore now much more balanced and relevant because the cornerstone of the classification (the conceptual model) is better explained and now very well justified. It is now also crisp and clear because all jargon has been defined and also explained in a Table. The current version of the main manuscript is a pleasure to read and up to publication standards

MAJOR AND MINOR COMMENTS SOLVED. All my major issues have been solved, or in case the authors had a different opinion it is sufficiently well justified and I can live with. Specifically, I highly appreciate it that they have improved the conceptual diagram, cleaned up the terminology, and made it internally consistent. The difference between ecological traits and ecosystem traits is now also solved. Most of the minor comments have been solved as well.

AU RESPONSE 4.2: We are pleased that our revisions have resolved these points.

A PHILOSOPHICAL QUESTION. I was specifically asked to comment on the description of tropical ecosystems. I am a functional ecologists and have worked and studied 30 years of my life in tropical forests over the world. Intriguingly, I was less convinced by the authors reply to my comments about their functional ecosystem descriptions; about 1/3 they implemented, about 1/3 they did not implement and I live with their counterarguments, and about 1/3 they did not implement and I was not convinced by their arguments and disagreed. Of course I acknowledge that it is difficult (and near to impossible) to accommodate and condense all people's priorities understandings and

opinions about an ecosystem in one page. But the philosophical question to the authors is what this means?:

- 1) their functional, process and trait-based classification of ecosystems is less useful than they claim as even functional ecologists seem to disagree and have a different understanding of how these systems function?
- 2) Keith has worked mainly in tropical forests in Australia and I have worked mainly in West Africa and the Neotropics, we both have a pertinent understanding of our systems, but the relevant differences are only partly captured by the functional classification (tier 3), and more by biogeography and the peculiarities of the biota (tier 4-6)?, or
- 3) this functional classification is wonderful because by using the same functional ecosystems and jargon it facilitates comparison, highlights apparent discrepancies, and therefore facilitates discussion, spurs new research, and advances science?

I guess that all 3 reasons may partly explain what is going on.

AU RESPONSE 4.3: We thank the reviewer for these thoughtful remarks. The current groupings for tropical forests at Level 3 are based on convergence of some ecosystem properties across lineages that evolved relatively independently on different land masses, but we certainly agree that substantial variation in functional properties represented within those groups, and that they could be subdivided into more functionally homogeneous groupings that could warrant recognition at Level 3 of the typology. Some of this variability is expressed within landscapes (e.g. productivity and associated leaf traits across gradients in nutrient availability), while some is expressed at larger biogeographic scales (e.g. between land masses). In the latter case, legacies of evolutionarily independent lineages have led to functional differences (e.g. in seed dispersal syndromes, functionally distinctive animal assemblages such as primates and parrots, etc.), which are recognised by default at Level 4 of the typology because functional differences align with major biogeographic and compositional patterns. Thus it is inevitable that some differences in functional properties are represented at tiers 3-6 because function and composition are inseparable, but groupings in tiers 1-3 are founded on functional similarities, irrespective of whether they share compositional similarities (see lines 87-88, main text). We believe this aligns with the reviewer's think in point 2 above. Further, as the reviewer suggests in point 3 above, publication of the framework "facilitates discussion, spurs new research, and advances science" providing the knowledge base for refinement of future versions of the typology (see lines 228-229 and Appendix S3).

KEEP THE MAPS. I concur with the plea of the authors to keep the maps when there is sufficient resolution for the scale at which they are shown (i.e., small global maps), as 1) it makes it more explicit to the reader what kind of ecosystems they refer to, 2) you can put it into an ecological context (i.e., whether it makes ecologically sense given macroclimate, soils, biogeographic barriers), 3) you can see their convergent nature across the globe, and 4) it spurs interest to do comparative research. Without doubt these maps will be improved over time, but that is with all science we do. As long as the limitations are indicated it is totally fine.

AU RESPONSE 4.4: Thank you, we agree with this reasoning.

GREAT PAPER AND PLEASE ACCEPT. I think the paper provides an important, timely, and solid contribution. I would therefore kindly suggest the editor and reviewers to accept the next revised version without sending it again out for review. We now have had 7 reviewers, 2 revision rounds, the manuscript has been expanded with 5 long encyclopedial appendices for further support, the authors have well justified their choices and indicated the limitations, and have written a careful

rebuttal letter of 93 (!!) pages.

When in some cases issues have not been solved then I think it is fine, because it is difficult and near-to-impossible to accommodate everybody's ideas and concerns in 2500 words (or whatever the word limit is for this kind of Nature article). The main messages of the paper are solid, and over the lesser issues we should be able to agree to disagree as, after all, it is the authors paper and story and not the one of the reviewers. So I hope the editor and reviewers can be flexible, I congratulate the authors with this tremendous effort and result, and I am looking forward to see the paper in print.

AU RESPONSE 4.5: Again, many thanks for a thoughtful and constructive review

With kind regards,
Prof. Lourens Poorter

Referee #5 (Remarks to the Author):

I have read the revised manuscript and the responses to reviewers and think that the authors have done an excellent job in taking into account the many points that were raised on this round of review. I hope that the editors realize the scope of this work and the fact that no system set up to classify all the global ecosystem types will satisfy everyone. Part of the issue is that many different sub-fields of biology and policy are represented in the materials the authors have produced, as well as the reviews this paper has received so far. There were some weaker points and the authors wisely agreed to remove them from this specific version of the paper. These can be addressed in their future work, and removing them does not weaken this specific paper.

The work reminds me of when the human genome was published. Many authors were involved, and the genome was incomplete, but the paper served as a milestone for a huge scientific advance. The paper was essentially descriptive, but that in no way diminished its importance. The approaches in that paper were further refined and the approach became even more useful as that happened.

Classification of functional types of ecosystems is an essential step in managing and conserving the Earth's ecosystems in a time of global human pressure (the Anthropocene). This is true both with respect to the preservation of biodiversity for its own sake as well as the maintenance of ecosystem services upon which humanity depends for its ultimate comfort and survival. A unified approach to such classification is an important step forward in our efforts to understand the natural world.

The authors have adequately responded to all of my specific points. I hope my input has strengthened the work and its utility. The authors should be congratulated on the breadth and the ambition of the work; this is a truly important contribution and represents a very significant scientific advance.

AU RESPONSE 5.1: We thank the reviewer for these positive comments.

Referee #6

Referee # 6 acknowledges substantial improvement in our revision, “many elements have improved a lot ... the paper provides evidence on the soundness of the typology itself”, but raises some residual concerns, primary centred on the spatial data, which is not part of this paper (separate data publication cited as ref 25 in main text). The key issues raised in this round of review are:

AU RESPONSE 6.1: Context of the (spatial) data in this paper.

Several of the reviewer’s comments reflect a need to comprehend the relevance of the map data to: a) the **needs** for a global ecosystem typology; b) the **capability** of our typological framework to meet them; and c) its **readiness** for specific applications (only some of which require global maps).

a) Our introductory text (lines 62-102) addresses the **needs** for a global ecosystem typology (including needs for spatially explicit units, now defined in the Glossary). These needs are pertinent and important, regardless of whether currently available data are fit and ready for specific applications.

b) The main text on lines 175-227 addresses the **capability** of the typological framework to serve multiple needs. While much of this text focusses on the conceptual qualities of the framework, it notes (lines 179-181),

“The scalable hierarchical structure (principle 4, Table S1.1) and the explicit description of properties and drivers enables units at any thematic level to be mapped at different spatial scales.”

The word “enables” refers to capability, rather than readiness of current data, a point that we emphasize in the following sentence in the text, “...according to needs of specific applications and availability of data at resolution suitable to those needs”. Our point here (and below in (c)) is that, contrary to common belief that higher spatial resolution is always better, usefulness is not contingent on some absolute resolution, but that a useful spatial resolution depends on the question, the application, or the needs.

We further clarify the distinction between capability and readiness with edits to existing text in lines 198-199,

“...a rapidly growing body of spatial data [25] has established an ecologically robust and powerful capability and growing readiness for such syntheses.”

c) Lines 240-247 address limitations in the spatial data, their **readiness**, and future prospects for specific applications, elaborated in the Limitations section of the Methods (lines 546-564). We distinguish applications that do not require high-resolution global maps (lines 242-245) from global syntheses that do require such maps (lines 245-247). In Methods (lines 547-551) we identify qualities of maps that make them most fit for global synthesis and summarise the additional information (lines 551-556) from in a revised Table S4 (see AU response 6.6i). On this basis, we suggest that spatial data for 60-80 of EFGs (c. two-thirds of the total) are potentially suitable for global analysis, while we identify the remainder as priorities for additional work to bring them to a similar standard (lines 556-564).

While several passages of our main text describe the growing readiness of map data for different types of applications (lines 199-200, 214, 257-259), it does not state that map data can support global assessments “currently” or “at this time” – those are the reviewer’s phrases.

Overall, by i) distinguishing needs, capability and readiness for application, ii) identifying the limitations on the readiness of existing spatial data for some applications, and iii) providing additional information on the conceptual alignment of maps to EFGs, validation and spatial resolution of source data (Table S4.1), we think our paper gives a balanced and defensible account of the data used to construct thumbnail maps presented in Appendix S4, noting that the spatial data itself is not part of this paper (see further comments on this issue in AU response 6.5 – 6.8).

AU RESPONSE 6.2: Use of the term ‘sustainability’.

We note the reviewer’s concerns that a hierarchical typology is [not] all that is needed to address sustainability goals or ecosystem accounts. We checked our paper thoroughly and nothing in its text suggests an ecological typology is the only requisite for measuring sustainability or for supporting sustainability decisions. We agree with the reviewer that socio-economic frameworks are also needed to assess sustainability, and acknowledge the importance of socio-economic drivers in revised text in lines 471-473, but those and other requisites are not the subject of our paper. Our key point on this subject is that neither conservation nor sustainability (the dual goals implicit in the CBD and SDGs) can be done well without an ecological typology that represents function. Our text makes this clear on lines 70-71:

“This [i.e. the current lack of a function-based ecosystem typology] hampers progress on developing conservation targets and sustainability goals,”

and lines 96-98:

“To serve dual needs of sustaining ecosystem services and conserving biodiversity, ecosystem assessments require a global typology to frame comparisons and standardise data aggregation for analysing ecosystem trends and diagnosis.”

Our statements in the text that the typology can “support” sustainability, as well as conservation, are appropriate measured because “support” does not imply exclusivity. The two design principles for our typology that enable it to represent both ecosystem functions and biodiversity, distinguish it from other ecological classifications (Appendix S1). We reason that any future spatial analyses of ecosystem sustainability will be better served by a typological framework with spatial units that represent ecological functions, than frameworks that do not represent functions.

AU RESPONSE 6.3: Referee #6 queried the nature of evidence “that the typology can currently support processes such as global ecosystem accounting under the SEEA or ecosystem monitoring under the Post-2020 framework”.

Given the short time since its development, the following points are relevant to evidence of the typology’s role in these processes:

- i) Policy adoption in both the UN CBD post-2020 framework and the UN SEEA-EA standard is significant evidence of utility and uptake because it signals a clear intent for implementation. Both are major international agreements negotiated during extensive global consultation processes. Adoption would not have occurred if representatives of member nations had doubts about map suitability and its rate of development.

- ii) After policy adoption, there is early evidence of mobilisation. UN staff have initiated a collaboration with relevant coauthors of our paper to develop refined maps of EFGs suitable for ecosystem accounting needs, further showing their intent to advance readiness of global maps. Ecosystem indicators developed by our research group (see Nicholson et al. 2021, Nature Sustainability) have been adopted into the Biodiversity Indicators Partnership. This demonstrates an intent in the developing post-2020 CBD framework to measure progress on targets for ecosystem extent, integrity and risk, which derive from the IUCN Red List of Ecosystems and Global Ecosystem Typology. Since last review of our ms, a new publication (Vanderrabano et al. 2021) identified several roles for the typology in applying ecosystem risk assessment science to ecosystem restoration in the UN Decade of Ecosystem Restoration. We have added commentary on this to Appendix S6 (p7):

“More broadly, at the outset of the UN Decade of Ecosystem Restoration, new approaches applying risk assessment science to ecosystem restoration are under development (Valderrábano et al. 2021). The Global Ecosystem Typology is recognised as a useful tool for framing consistent descriptions of ecosystems when planning restoration priorities and strategies for risk reduction (Nicholson 2021; Etter et al. 2021), and for knowledge transfer on ecosystems with similar restoration requirements around the world, building capacity and increasing a shared global understanding (Nelson 2021).”

AU RESPONSE 6.4: Reviewer #6 seeks examples to illustrate how our Global Ecosystem Typology supports translation of global policy goals to on-ground nature-based solutions, with reference to main text lines (270-271), *“The hierarchical structure [...] enables global imperatives to be linked directly with on-ground nature-based solutions...”*

- i) This quote comes from the concluding paragraph of the main text. Our intention for this final paragraph of the ms is to flag emerging applications and look forward to potential applications of the typology (elaborated in Appendix S6). Examples of nature-based solutions are only beginning to emerge (see below), given the recent development, adoption and release of the typology. To clarify this context, we modified the phrasing, *“...should enable...”* in the passage quoted above.
- ii) The typology should enable these linkages in three main ways. Firstly, when practitioners reference their ecosystems to the Global Ecosystem Typology, it will provide a framework to build a knowledge base that aggregates learnings from a multiplicity of on-ground management and restoration actions in functionally alike ecosystems. Secondly, it illuminates the global significance of local ecosystems, which should be considered in local decision-making and for their management. The typology will help apply this principle more equitably to poorly understood and hitherto poorly defined ecosystems such as heath forests and rhodolith beds, counterbalancing existing biases in philanthropic, government and business investment decisions that favour action to conserve widely familiar ecosystem types such as tropical rainforests and coral reefs. Thirdly, the typology facilitates knowledge transfer by framing a common language for ecosystem description, reducing confusion that emerges from use of inconsistent concepts of ecosystem

types (see example of differing concepts of ‘savanna’ in our previous response to reviews).

- iii) We discuss emerging examples of these applications in Appendix S6 (Ecosystem monitoring). We are aware of other examples that are not yet sufficiently advanced to cite outcomes, such as the use of the typology to frame ecosystem management strategies that will be shared across public, private and indigenous land tenures.

AU RESPONSE 6.5: Role of high-quality global maps in ecosystem accounting.

A full set of global high-resolution maps is not essential for global ecosystem accounting according to the UN SEEA-EA Standard as assumed by Referee #6. Sections 3.54 – 3.56 of the adopted UN Standard for Ecosystem Accounting (cited in our paper, UNCEEA 2021) describe how our Global Ecosystem Typology will be used as the reference classification. The Standard recommends “Crossreferencing of [national] spatial units to the SEEA EA reference classification, the IUCN Global Ecosystem Typology (IUCN GET), will enable national level accounts to be scaled up and compared between countries.”

This approach allows use of the best high-resolution spatial data available in each country, with our typology enabling global synthesis of otherwise disparate data sources at a broader level classification. Hence, high-resolution global maps are not essential for global synthesis of ecosystem accounts or compliance with the UN SEEA-EA standard. In Appendix S6 we explain this role of the Global Ecosystem Typology (under Natural Capital Accounting, 2nd paragraph as follows:

“Contributors to SEEA-EA are expected to use the best quality classification and maps available for their jurisdiction when developing their national accounts (i.e. Level 6 units), and assign the units of that classification to Ecosystem Functional Groups (Level 3) to enable consistent international reporting.”

In Appendix 3 (Tables S3.3, S3.4), we provide two examples where detailed national ecological classifications have been cross-reference to Level 3 of Global Ecosystem Typology (EFGs), as recommended in the UN SEEA-EA Standard.

AU RESPONSE 6.6: Limitations on the spatial data.

Referee #6 posits that there is “currently simply no assessment of the quality of nearly all of the presented ecosystem maps”. We respond with the following points:

- i) The reviewer’s assertion is incorrect, however, we agree that more information should be provided to assist readers in understanding the efficacy of source maps. Accordingly, we reformatted and extended Table S4.1 to include additional information on the spatial data used to construct the indicative maps for each Ecosystem Functional Group, including: the alignment of the map data with the concept of their corresponding Ecosystem Functional Group; the type of map evaluation undertaken by the authors of the source data; spatial resolution of the map data; and methods of assembling the relevant spatial layers. We added a brief summary of this tabulated information to the Methods text to support our inference about the suitability of maps for global spatial analysis (lines 551-558):

“Sixty of the maps currently in our archive [25] aligned directly or mostly with the concept of their corresponding ecosystem functional group, while the remainder were based on indirect spatial proxies, and most were derived from polygon data or rasters of 30 arc-seconds or finer (Table S4.1). Maps for 81 functional groups were based either on known records, or on spatial data validated by quantitative assessments of accuracy or efficacy. Therefore, we suggest that maps currently available for 60-80 of the 110 functional groups are potentially suitable for global spatial analysis of ecosystem distributions..”

- ii) Regarding concerns about data quality, almost all spatial data sets used in the global maps of EFGs are published in peer-reviewed scientific journals, with map evaluations that range from quantitative accuracy assessments to expert review (details in revised Table S4.1).
- iii) Our paper contains no spatial analyses that depend on the map data. The thumbnail maps included in Appendix S4 cannot be used for spatial analysis and are simplified for global-scale visualisation of ecosystem distribution patterns. Several reviewers (1, 2, 4) noted that presentation of maps as thumbnails, helps to describe the current concept of each EFG relative to others in the typology at a global level, a point also made in our previous response. In turn, this helps ongoing efforts, both to improve the concepts of the EFGs, and to set the stage for globally consistent refinements in the maps themselves.
- iv) As noted in the text (lines 246-260), decoupling of the typology from the maps allows the typology to benefit from rapid developments in mapping technology and spatial data availability. We continue to actively update the archive of maps for EFGs (ref 25) to accommodate newly available data sets and improvements in mapping. Since the last revision of our ms, we have updated six of the maps.
- v) We agree with the reviewer that maps for pastures T7.2 and T7.5 are among the weakest across the typology. As the reviewer notes, the challenges of mapping different livestock systems are “widely acknowledged in the land-use mapping community.” We have further strengthened the caveats on both maps accordingly (Table S4.1, and cross-referenced in respective profiles):

“We acknowledge significant untested assumptions and limitations on spatial predictors that challenge the global-scale delineation of pasture ecosystems with varied levels of human influence. Therefore, we advise appropriate caution in use of the spatial data for quantitative analysis.”

We think there is value in the inclusion of thumbnail maps for these functional groups because they represent our best (albeit imperfect) attempt at delineation based on key variables (livestock density and productivity), notwithstanding the current paucity of suitable spatial data for other relevant variables (e.g. retention of native biota, nutrient addition, tillage). As noted by Referee #1, the utility of including such maps with strong caveats “vastly outweighs the downsides of providing nothing” and helps to highlight the need for further development of the spatial data, with progressive updating of the maps as improvements emerge.

AU RESPONSE 6.7: Spatial resolution and time series of maps.

Referee #6 queried whether fine-scale spatial data are available for the Global Ecosystem Typology (with reference to Table S1.2) and whether time series of spatial data are available. We make the following points in response:

- i) Fine-scale spatial data are available for the Global Ecosystem Typology in a data archive that is not part of this publication (ref 25 cited in main text). The resolution of maps in the archive varies from grid cells of 30 arc-seconds up to 1 degree. Relative to other global data sets, the resolution of most maps is relatively fine, and in most cases has been aggregated from even finer resolution source data (details in revised Table S4.1).
- ii) The typologies reviewed in Table S2.1 include some with raster- and some with polygon-format spatial data so their spatial resolution cannot be compared quantitatively. Nonetheless, the existence of digital maps of global extent is sufficient to assess principle 5 (whether units are spatially explicit, i.e. mapped). Although most of the typologies had maps, the resolution was very coarse in four typologies and their maps are currently unavailable in digital format. These four typologies are identified in Table S1.2 as ‘coarse resolution’ under principle 5. The remainder of spatially explicit typologies, including ours, had global-scale polygon data (e.g. Dinerstein et al. 2017) or raster data with grid dimensions that range mostly between 0.5 degree and <30 arc-seconds, and were accordingly ranked as ‘fine resolution’. We added this explanation to the commentary in Appendix S1 under Principle 5 ‘Spatially explicit units’.
- iii) Presentation of time series data is not within the scope of our paper. We draw attention to the **need** for time series data to support (spatial) ecosystem monitoring, as well as the **capability** to develop required data sets and **readiness** of currently available data (see response 1). We believe citation of examples for specific Ecosystem Functional Groups (e.g. Murray et al. 2018; Fetterer et al. 2017) and established generic data cubes that cover large portions of terrestrial and freshwater realms (e.g. Hansen et al. 2013; Pekel et al. 2016; GLIMS and NSIDC 2005-2018) are sufficient to support our inferences about capability and readiness.

RESPONSE 6.8: Appropriate display of maps in this paper.

We thank the reviewer for further commentary on “blob” vs grid map format. We make the following points in response:

- i) Comments on the format of the spatial data relate to a separate data publication that is not part of our *Nature* submission. That archive (cited as reference 25 in main text) was published in early 2020 with subsequent updates.
- ii) The only maps presented in our *Nature* submission are thumbnail images included in Appendix S4. These are simplified global-scale visualisations of ecosystem distribution patterns (using data from ref 25) that cannot be used for spatial analysis. The format of the underlying spatial data (raster or polygon) is not discernible on the thumbnails, except for a few based on the coarsest resolution grids. See also Response 6.6iii).
- iii) Quantitative spatial analyses can be, and are carried out on coarse-resolution spatial data, irrespective of its format as polygons or rasters, i.e. conversion of rasters to polygons is not a deterrent to spatial analysis of coarse resolution or low-quality map data. We note that there are many examples where the expert-based ‘blob’ maps downloaded from the IUCN

Red List of Threatened Species (RLTS) website have been used in coarse-grain global spatial analysis in well-cited papers published in high-impact journals, including:

- Strassburg, B.B.N., Iribarrem, A., Beyer, H.L. *et al.* Global priority areas for ecosystem restoration. *Nature* **586**, 724–729 (2020). <https://doi.org/10.1038/s41586-020-2784-9>
- Leclère, D., Obersteiner, M., Barrett, M. *et al.* Bending the curve of terrestrial biodiversity needs an integrated strategy. *Nature* **585**, 551–556 (2020). <https://doi.org/10.1038/s41586-020-2705-y>
- Venter O, Fuller RA, Segan DB, Carwardine J, Brooks T, Butchart SHM, et al. (2014) Targeting Global Protected Area Expansion for Imperiled Biodiversity. *PLoS Biol* **12**(6): e1001891. <https://doi.org/10.1371/journal.pbio.1001891>
- Mair, L., Bennun, L.A., Brooks, T.M. *et al.* A metric for spatially explicit contributions to science-based species targets. *Nat Ecol Evol* **5**, 836–844 (2021). <https://doi.org/10.1038/s41559-021-01432-0>

There are many older examples. While the efficacy of spatial data should always be evaluated against trade-offs, these studies provide context for the quality of our indicative EFG maps. Many of the RLTS range maps of species used in these studies are coarse resolution, based on intuitive judgements and not independently validated except by museum records used to inform the participating experts who created the maps (see AU response 160 of our previous response to reviews) cf Table S4.1. The last paper listed above uses the RLTS “blob” maps to calculate a metric for assessing species-based targets proposed for the post-2020 CBD framework. By that standard, our appraisal of the readiness of ecosystem maps for such spatial analysis is conservative.

- iv) Contrary to the reviewer’s suggestion, the foundational spatial science literature recommends raster (grid) formats, in preference to polygon (blob) formats, for representation of imprecise boundaries or spatial uncertainty, and cautions that vector polygons can give a sense of false precision.

Goodchild (1987) notes that vector maps can imply false precision more than coarse raster maps: “A vector system may be able to represent the apparent precision of pen-drawn maps, but its precision is often meaningless in relation to the data from which the map was derived and is therefore of no benefit.” Rasters are commonly used for global scale land cover map because the resolution of those rasters is related to the scale (resolution) and precision of the underlying data (Congalton et al 2014). Rocchini et al. (2013) also note that coarse rasters can usefully represent low spatial precision of boundaries, and recommend against generalization of vectors as a visual representation of uncertainty. Consistent with these established principles and practices in spatial science, we use coarser rasters to represent distributions with greater spatial uncertainty or boundary imprecision in our map archive (ref 25) (Goodchild 1987; Rocchini et al. 2013; Congalton et al. 2014).

Congalton, R.G., Gu, J., Yadav, K., Thenkabail, P. & Ozdogan, M. (2014) Global land cover mapping: A review and uncertainty analysis. *Remote Sensing*, **6**, 12070-12093.

Goodchild, M.F. (1987) A spatial analytical perspective on geographical information systems, *International Journal of Geographical Information System*, **1:4**, 327-334, DOI: 10.1080/02693798708927820

Rocchini, D., Foody, G.M., Nagendra, H., Ricotta, C., Anand, M., He, K.S., Amici, V., Kleinschmit, B., Förster, M. & Schmidlein, S. (2013) Uncertainty in ecosystem mapping by remote sensing. *Computers & Geosciences*, 50, 128-135.

AU RESPONSE 6.9: Anthropogenic ecosystems.

We appreciate the reviewer's interests and emphasis on anthropogenic ecosystems and acknowledgement of improvements in this aspect of our paper. We fully agree with the reviewer on the importance of socio-economic and cultural-spiritual processes that underpin human activities that, in turn, shape ecosystems. We added a sentence to the relevant section of Methods to acknowledge this point (lines 471-473):

"These activities ultimately are driven by socio-economic and cultural-spiritual processes that operate from local to global scales of human organisation."

However, our focus in this paper is on the proximal drivers of ecosystem assembly, i.e. the human activities themselves, rather than the distal, though important, factors that determine the nature and variability of those assembly filters. Similarly, in our assembly model, we focus on the availability of resources and nature of disturbance regimes as filters, rather than the geophysical processes that ultimately control them. In this context, we believe the level of detail on human activity and 'natural' drivers of assembly is appropriately balanced. We also note that very few of the global typologies that we reviewed in Appendix S1 incorporate anthropogenic ecosystems, and when they do, it is largely as an appendage to a list of natural ecosystems. We added a sentence on this point (lines 461-462):

"Very few ecological typologies reviewed in Appendix S1 incorporate anthropogenic ecosystems in their classificatory frameworks."

The Global Ecosystem Typology identifies 6 of 25 biomes and 16 of 110 Ecosystem Functional Groups as anthropogenic, acknowledging that almost all the world's ecosystems are influenced by human activities to differing degrees (lines 463-464). In that context, we believe our typology is the first systematic global integration of natural and anthropogenic ecosystems and a significant advance on previous efforts.

Referee #6 also comments on the difficulty of distinguishing anthropogenic and non-anthropogenic ecosystems. We agree this can be challenging, but our typology supports users by distinguishing anthropogenic from non-anthropogenic ecosystems at three different levels. Firstly, we provide a broad conceptual definition of anthropogenic ecosystems in the Glossary. Secondly, we describe diagnostic characteristics for each anthropogenic Ecosystem Functional Groups in Appendix S4. Thirdly, we operationalise the concept and descriptions into indicative global maps and assembly rules for mapping the distribution of each EFG (as applied in ref 25). Each of these levels provides a different trade-off between generality and specificity of the distinction between anthropogenic and non-anthropogenic ecosystems. If taken in isolation, each level of distinction may be inadequate for general or particular purposes. In combination, however, all three levels provide a workable means of differentiating anthropogenic ecosystems, given the inherent degree of uncertainty that stems from continuous variation (acknowledged in lines 261-269 of our main text) and the current state of knowledge (including available spatial data) on particular ecosystem types (acknowledged in lines 230-239 of the main text and Limitations section of the Methods text).

AU RESPONSE 6.10: Diagnostic utility.

Referee #6 suggests a need to tone down inferences about the diagnostic utility of our typology in lines 110-112, 214 and onwards and Appendix S1 (comments on AU response #182 in the previous revision). This comment relates to our evaluation of Design principle 6 (Appendix S1), and therefore we interpret 'diagnostic utility' as the adequacy of information provided to users for diagnosing different units within the typology. In response, we make the following points.

- i) Design principle 6 states that descriptions of classification units should be sufficiently detailed to enable users to readily interpret and diagnose the features of their units in the field (Appendix S1).
- ii) To evaluate existing typologies against this principle, we reviewed the level of detail in the descriptions of their units, including the range of features described. We found that descriptive detail was poor in many of existing typologies (e.g. some had only a legend label and map).
- iii) We inferred that the IUCN Global Ecosystem Typology is well supported by diagnostic and descriptive materials because the descriptions of ecosystem functional groups address a greater range of ecosystem features in more detail than most other global ecological typologies. Table S1.2 states that our Global Ecosystem Typology includes "*detailed text descriptions, assembly models, illustrations and global maps available for all units in top 3 levels*". No other global typology in Table S1.2 has all of these descriptive components.

We checked the phrasing and context of the text in lines 110-112 and Appendix S1 and think this comparative context is clear and that the inferences are justified by the review of descriptive detail summarised in Table S1.2 (note that text from line 214 is about conceptual models and ecosystem dynamics, rather than descriptive material).

AU RESPONSE 6.11: Minor issues addressed:

- i) We replaced Jung et al with Hansen et al. 2013 and Pekel et al. 2016 in lines 266-267
- ii) We added 'spatially explicit' to the Glossary
- iii) We have amended text as suggested for profiles of T7.1, T7.2 and F3.1.

Referee #7 (Remarks to the author):

Referee #7: This is my second time reviewing this paper, and while I definitely appreciate the authors' hard work and thoughtful replies (I imagine it was quite difficult to generate the framework and consensus among so many experts and over a planetary scale, and 94 pages of reviewer replies is another manuscript in itself), I am beginning to appreciate that I am not the target audience for this article.

While I have done a number of cross-system syntheses on ecosystem functioning and services, and currently work with local, regional, and international governments to meet biodiversity and climate targets, I just don't see the utility of the proposed typology for, or how it would significantly inform or improve the context of, my work. As the IUCN has already accepted this standard, I guess its not really my place to comment on the utility of the framework, since clearly it has already been vetted

and accepted by a broad target audience of scientists, managers, and policymakers. That said, as a scientific article, I still wonder how this will be used by the broader scientific community to shape the discourse around effective conservation and evaluation, which is a concern echoed by several other referees. I think what bothers me most about this article is that, on the surface, it is really attractive but ultimately rings hollow. As a classically-trained community ecologist, I love the idea of using, for example, niche partitioning or ecological interactions to define biomes. But as I dig deeper, the manuscript does not satisfyingly pay off on the promise of a truly integrative framework.

Much of the framework is informed by expert opinion (lines 346-347) and the authors admit that truly empirical tests have yet to be conducted on assembly processes, linking multiple drivers, and so on (eg, lines 225-233, directly on lines 230-231, 250-252, 256-266). I agree it is valuable to draw attention to these deficiencies, but if there is a “gold star” example the authors could share, it would be a long way of convincing me that this framework can be fulfilled to the fullest extent rather than hypothetically promising.

AU RESPONSE 7.1: We greatly appreciate the Reviewer’s comments about the challenges of quantifying the concepts associated with community assembly. To consolidate our conceptual contribution, we offer the following responses:

- i) While it is beyond our scope and available space to include a full critical evaluation of a ‘gold-star’ example of how out typology can “shape the discourse around effective conservation and evaluation”, we expanded the Methods text to draw attention to a number of ecosystem types with a strong empirical knowledge base. Tropical savannas provide a good example, where previous definitions, classifications and maps of those ecosystems focus unduly on tree cover and fail to address the importance of top-down assembly processes, including herbivore activity and fire regimes. We cite a recent review (Bond 2020), which summarises a large empirical and modelling literature on the topic, and a recent policy perspective that identifies the impact of ‘biome awareness disparity’ on conservation action and research. We thus point to the need for similar studies to test our assembly hypotheses in a large number of other ecosystems. The revised text is in the Limitations section within Methods on lines 522-545:

“Uneven knowledge of Earth’s biosphere has constrained the delimitation and description of units within the typology. There is a considerable research bias across the full range of Earth’s ecosystems, with few formal research studies evaluating the relative influence of different ecosystem drivers in many of the functional groups, and abiotic assembly filters generally receiving more attention than biotic and dispersal filters. This poses challenges for developing standardised models of assembly for each ecosystem functional group. The models therefore represent working hypotheses, for which available evidence varies from large bodies of published empirical evidence to informal knowledge of ecosystem experts and their extensive research networks. Large numbers of empirical studies for some forest functional groups, savannas, temperate heathlands in Mediterranean-type climates, coral reefs, rocky shores, kelp forests, trophic webs in pelagic waters, small permanent freshwater lakes, etc. (see references in respective profiles, Appendix S4). For example, Bond [32] recently reviewed empirical and modelling evidence on the assembly and function of tropical savannas that make up three ecosystem functional groups, showing that they have a large global biophysical envelope that overlaps with tropical dry forests, and that their distribution

and dynamics within that envelope is strongly influenced by top-down regulation via biotic filters (large herbivores and their predators) and recurrent disturbance regimes (fires). Despite the development of this critical knowledge base, savannas suffer from an awareness disparity that hinders effective conservation and management [59]. In other ecosystems, our assembly models rely more heavily on inferences and generalisations of experts drawn from related ecosystems, are more sensitive to interpretations of participating experts, and await empirical testing and adjustment as understanding improves. Empirical tests could examine hypothesised variation in ecosystem properties along gradients within and between ecosystem functional groups and should return incremental improvements on group delineation and description of assembly processes.”

- ii) We believe the most important contribution of our paper are the advances in bringing concepts of ecosystem function and dynamics into ecological classification, which has previously been the domain of more static approaches focussed on contemporary pattern, rather than processes that sustain them or drive change. Our approach to synthesis needed to be flexible to accommodate qualitatively different cross-disciplinary inputs of ideas on assembly models, ecosystem dynamics, trophic relationships, interactions between humans and nature, spatial modelling, remote sensing and conservation science and policy across the full range of ecosystems on Earth. Our approach also required pragmatism to deliver a workable product that is needed now to establish targets and indicators for the CBD post-2020 framework, the new global system for ecosystem accounting, the UN decade on ecosystem restoration and other policy initiatives that require a common descriptive framework for ecosystems that represents both their function and biodiversity.
- iii) The two contextual factors mentioned above in i) and ii) shaped the balance of expert input and more formal analysis of empirical evidence in our synthesis. At one extreme, reliance on rigid meta-analysis of empirical evidence, with due appraisal of experimental design and other limitations, would likely have constrained the conceptual development of the framework by excluding emergent sources of knowledge (from both the literature and field) and could not have delivered a product capable of implementation in policy-relevant time frames. At the other extreme, failure to address published evidence from well-designed experiments would undermine the efficacy of the conceptual framework. Therefore, we used expert input to identify strategic examples of evidence from the literature, supplemented by inferences and ideas drawn from collective experience of collaborators and their respective research networks.

Referee #7: I can also see biomes where this framework would fail to yield robust generalisations, as promised: “Ecosystem groupings based on convergent drivers, properties and environmental relationships will reveal similarities in threats, mechanisms of degradation and, therefore, inform development of ecosystem-specific management strategies for recovery.” Take a widely-distributed marine habitat in seagrasses. Tropical seagrasses will probably thrive under climate change as their metabolisms ramp up and more CO₂ is available for photosynthesis (with some exceptions), while temperate species have already experienced significant decline and range retraction as a result of rising temperatures exceeding physiological thresholds. Even within a species, such as eelgrass, Atlantic populations are more threatened by climate change than Pacific ones, based on divergent evolutionary histories and consequent differences in biogeographic extent. The contrasting drivers within this system or even geographic region is summarized nicely in Table 1 of the recent “Out of

the Blue” UNEP report solely on seagrasses. I imagine that with any framework that is so broad, it would be possible to find exceptions to the rules (as pointed out by other referees). I am just concerned that the rules are so vaguely defined, seeing where they are usefully applied (that “gold star” example) would help negate the exceptions that immediately leap to mind.

AU RESPONSE 7.2: These comments highlight important variability within Level 3 units of our typology (ecosystem functional groups). Using the reviewer’s example, variation within the seagrass functional group could be recognised by defining separate tropical and temperate groupings. Uncertainty related to groupings stems in part from discrete representation of continuous ecological patterns in nature (discussed in lines 261-269, main text). However, splitting seagrass meadows at Level 3, for example, would come at the cost of more units and greater complexity at that level of the classification, and can instead be accommodated by definition and description of finer groupings at Level 4 and below. We think this is appropriate for different kinds of seagrass ecosystems (tropical vs temperate, Pacific vs Atlantic), as these share fundamental features that govern their function (dominance by submerged vascular macrophytes rooted in soft substrates at shallow depths, supporting epibiota and large marine herbivores, and sensitive to turbidity and substrate disturbance etc.). Within that broad specification, variation among seagrass ecosystems is readily recognised as biogeographic variations in biotic composition and diversity that closely associated with biogeography, and therefore appropriately recognised at Level 4. To pilot development of Level 4 units, we are currently collaborating with mangrove specialists to define level 4 biogeographic units within the mangrove ecosystem functional group (MFT1.2).

Referee #7: The manuscript is also somewhat contradictory: in the very first sentence, the authors suggest the goal of “Sustaining ecosystem functions and services requires an understanding of ecological processes [irrespective] of specific biota.” They then immediately say: “[Ecosystem] functioning not only underpins biomass production, but also depends on [biota.]” (line 81). They then go on to say their 108 EFGs are specifically defined by “biotic composition” (line 176). So, sustaining functions is directly a consequence of invoking specific biota, as they do in their lengthy summary table. In the introduction, the authors also mix terminology, alternately referring to “ecosystem function,” “ecological processes,” and “ecosystem properties,” which include species traits (echoed by R4). The new glossary differentiates processes as those involving “interactions among organisms” although such processes, primarily consumption, have long been regarded as ecosystem functions (such as in reviews and meta-analyses by Hooper et al. 2005 and Cardinale et al. 2006, 2012).

AU RESPONSE 7.3: Thank you for pointing out this problem, which results from earlier revisions. Our point is that organisms that belong to different species, genera and families may contribute to very similar processes and functions in different ecosystems – Primack and Corlett (2005), for example, point to tropical forests in Africa, Madagascar, Asia, Oceania and South America that share broadly similar ecological features derived from evolutionary lineages that have been independent for tens of millions of years. We have revised and re-arranged the text to clarify (now on lines 87-88):

“Ecosystems with different species composition may show functional convergence if their biota share similar traits and contribute to similar ecological processes (e.g. [8]).”

Referee #7: On a minor note, I would consider traits to be properties of the community, whereas ecosystem pertains to the general flow of energy among discrete compartments (eg, primary producers to consumers) (R4 also seems to agree on this point).

AU RESPONSE 7.4: Referee #8 identified our conception of ecosystem properties as a strength (see Response 8.14). Both species and communities are ecosystem components, reflecting their hierarchical relationships. Therefore, we think it is reasonable to describe them collectively, along with physical properties, ecological processes and fluxes, as ecosystem properties.

Referee #7: I could go on, but it would perhaps be a moot exercise as, based on other referees' comments, I seem to be missing the utility of the contribution here. Without sufficiently compelling examples of applications (which the authors note in their reply to R7 that "There are already examples of several applications at national levels" but are seemingly mum on those), I can really only see myself or my colleagues citing this work to justify the discrete ecological units under investigation and how they may or may not be related (for example, through shared drivers or shared responses). Is that worthy of publication?

AU RESPONSE 7.5: As noted in responses to Referee #6, adoption into major global policies is strong evidence of widespread intent for application, noting that few applications can be expected to have been completed and published at this early stage after development and adoption. We do cite some early published examples of applications in Appendix S6, including Murray et al. (2020) who used the global typology to define a national typology and identify Key Biodiversity Areas, and Etter et al. (2021), who identified priorities for ecosystem restoration using a risk assessment based on a national typology that has since been linked by the authors to the global typology in the manner shown in Appendix S3. We believe these are forerunners of many other applications that should emerge in future as a result of adoption in policy. We also note that other international frameworks driven by policy imperatives begin with national implementation and are later scaled up to global application (e.g. <https://www.fao.org/in-action/forest-landscape-restoration-mechanism/resources/detail/es/c/1315004/>).

Referee #8 (Remarks to the author):

I have been sent this manuscript to review from the perspective of an evolutionary ecologist. As will be seen below, I am comfortable with this manuscript from that perspective. I note that the manuscript has had MANY reviews already, and beyond that there has been extensive peer review of the whole typology, as is well documented by the authors. I do have some comments, which I place below.

Sincerely,
Kyle Dexter
University of Edinburgh

Referee #8: First and foremost, I think the hierarchical system they have developed makes good sense from an evolutionary ecological perspective. I agree with the authors' world view that biomes and ecosystems should be functional. I also agree that contingencies in the biota present in an ecosystem can modulate its function. Lastly, I agree that one key purpose of developing an

ecosystem typology is to share lessons, for management or otherwise, in how ecosystems in different places may have similar behaviours. Thus, having ecosystems that span continents is almost a pre-requisite for the system, and allowing division of these core ecosystems (level 3) based on biogeographic region / biota (in level 4) present is appropriate. Yes, the biota can drive variation in how a level 3 ecosystem functions on different continents, but like the authors and most reviewers, I think the commonalities of these EFGs across continents generally supersedes among-EFG differences.

AU RESPONSE 8.1: We thank the reviewer for these positive remarks. It is very pleasing that a fresh reviewer has so readily understood our motivations, approach and the key concepts underpinning the development of the typology, as well as its structure.

REFEREE #8: I think people who work in the field of eco-evolutionary feedbacks (and argue that many evolutionary processes operate on the same timescale as ecological processes) won't like the text in lines 130-131 about the longer timescales and lower relevance of evolutionary processes. But, I would tend to think that the evolutionary processes that play out on ecological timescales are not sufficiently powerful to create major shifts in ecosystem function to cause one EFG to shift to another (on ecological timescales). Thus, I am comfortable with what some might perceive as 'short shift' given to evolutionary processes in this context.

AU RESPONSE 8.2: Again, we thank the reviewer for this insight. We fully agree that evolutionary processes can play out over relatively short time scales, depending on the generation lengths of the organisms involved, but in general, those evolutionary processes that influence ecosystem function operate over longer time scales than ecological processes, especially given the context of evidence on biome conservatism of lineages illuminated by Crisp et al. (2011) and subsequent authors.

Referee #8: I have read through the main text, the response to previous reviews (including that by R2 who suggested an evolutionary ecologist) and read most of the appendices. I did not find anything in there that I find very problematic from an evolutionary ecological perspective.

AU RESPONSE: Thank you

Referee #8: I now shift to a few criticisms/comments, as I could hardly call myself an academic if I did not have some critical comments.

MAJOR COMMENTS

1) As has been commented by other reviewers, I do find this article to be a bit odd as an Article for Nature. For me it is more of a views and perspective type of piece than a research article per se. I have little doubt that the article will be well cited and also draw attention to the new typology, the latter being a good justification for publication in Nature.

My view that this is more of a perspective piece than a research article is down to the fact that ultimately I felt it was an expression of expert opinion. While the pool of experts consulted is large, it does not represent a formal, data-driven assessment of what functional biomes and ecosystems are. As the authors recognise, such comparable data across functional biomes and ecosystems are rare. Still, I think it could be useful for the authors to point a way forwards, by more clearly articulating the critical need for comparable ecosystem function data across biomes and ecosystems (e.g. that

generated by the GEM project for terrestrial forest systems, based at Oxford). As the authors point out, their functional biomes and ecosystems are hypotheses to be tested, and the typology will be revised. The authors could give readers some suggestions on how these hypotheses could/should be tested.

As an example, In AU153, they write "First, high levels of heterogeneity within groups can be addressed by segmenting continuous gradients of vegetation into a larger number of groups, with a trade-off through increased complexity of the typology. An important point is whether variation in properties is greater between groups than within them at the same level of classification. Greater heterogeneity within groups than among them would justify a trade-off to recognise more groups." This is the kind of analysis that is needed to test their hypothesised EFGs.

AU RESPONSE 8.3: We thank the reviewer for this input and address this point briefly with revised text on lines 230-234:

"models of assembly for each ecosystem functional group represent working hypotheses, for which available empirical evidence varies greatly (see Methods – limitations). Redressing research biases across different ecosystem types and among different assembly filters will help improve not only the assembly models, but the distinctions between ecosystem functional groups and units within other levels of the typology."

and elaborate in the Methods (lines 522-545, see AU Response 7.1i) above for the text).

Given knowledge biases and awareness disparities, we think it is important that expert input is viewed as a mechanism for incorporating published and other sources of evidence, as well as informed hypotheses into the typology. Hence our revised text above refers to examples for which a substantial body of empirical and modelling evidence exists on assembly mechanisms, as well as noting the many functional groups with limited evidence on the hypothesised drivers and mechanisms.

We understand from previous correspondence that a formal meta-analysis would not be a requirement for a research letter (that approach would have greatly constrained the input of relevant information and knowledge), but are happy to take advice from the editor on the suitability of alternative article types. The final sentences of revised text above suggest an approach to hypothesis testing, along the lines suggested by the reviewer.

Referee #8: 2) Another consequence of generating an expert-opinion based system is that the outcome depends on the experts involved. I don't want to nitpick at the typology as it has already received extensive scrutiny by people more qualified than myself. Rather, it is worth reflecting on the dual facts that Australia comes up as having multiple unique ecosystems (Temperate pyric humid forests, Hummock savanna, Sclerophyll hot deserts and semi-deserts) and that the lead authors and many of the authors are based in, or from, Australia (NB: I only looked at terrestrial EFGs). What would this typology have looked like if led by African authors, or South American authors? I am not suggesting changes to the system, but I wonder if a comment somewhere around the contingency in the outcome depending on experts involved might be useful.

AU RESPONSE 8.4: We added lines 540-543 to acknowledge this point to text on Limitations in Methods:

“In other ecosystems, our assembly models rely more heavily on inferences and generalisations of experts drawn from related ecosystems, are more sensitive to interpretations of participating experts, and await empirical testing and adjustment as understanding improves.”

As an aside, three of the 110 EFGs and none of the biomes are unique to Australasia. Many non-Australian authors have noted the global uniqueness of certain Australian ecosystems dominated by eucalypt or hummock grass life forms and life histories, including Bond (2020), Orians & Milewski (2007), Pianka (1981, 1989) and others. In the context of several EFGs unique to other regions of the world, we think the delineation of these units at level 3 is well-justified and not heavily influenced by Australian representation among experts.

Referee #8: 3) I liked the mapping of major and minor occurrence as it helps communicate the fact that biomes/ecosystems can have spatially interdigitated distributions, that bits of one major EFG can be found within an expanse of another and that a given location in space can be in one of multiple alternate ecosystem states. While one would think such things are obvious to anyone who has been in mountain landscapes (or paid attention to lakes or forest-savanna transitions), this complexity seems to escape many users of biome/ecosystem maps. I think it could be useful for the authors to emphasise this issue more clearly in the main text of the manuscript somewhere. As they are no doubt aware, once they make raster layer(s) available for their EFGs, end-users will use the products to 'definitively' assign points in space to a given biome or EFG (e.g. when assigning a species collection record to occurring in a given biome/ecosystem). This is common practice with current biome data layers, at least in the terrestrial realm, and while it may lead to correct assignment much of the time, it does result in errors that have consequences for downstream interpretation and understanding. Again, even though some end-users will do this no matter what the authors write, I think the authors are in a good position to make a strong cautionary statement about assigning biome/ecosystem just based on lat/long in their paper.

AU RESPONSE 8.5: Thank you for the suggestion. We added the following text to Methods (lines 508-513):

“We mapped ecosystem functional groups as major occurrences where they dominated a landscape or seascape matrix and minor occurrences where they were present, but not dominant in landscape/seascape mosaics, or where dominance was uncertain. Although these two categories in combination communicate more information about ecosystem distribution than binary maps, simple spatial overlays using minor occurrences are likely to inflate spatial statistics.”

Referee #8: 4) Like R3, I felt the language seems unnecessarily complex in many places. R3 gives many examples, so I will only mention a few examples. The authors note in their response to R3 that curing is a synonym of drying, and that certain ecologists/managers use the term curing. If drying is a synonym of curing, why not use drying, which is a more easily understood term for the large majority of readers. Elsewhere, the authors refer to sessile photoautotrophs and note that they are vascular plants. Why not just use the term vascular plants? I think nature articles are meant to be accessible to a broad audience, including physicists, mathematicians, social scientists, etc.

AU RESPONSE 8.6: We have tried to simplify text wherever possible and have included a Glossary for essential terms. As suggested by both reviewers, we have now replaced 'curing' with 'drying' on p75 and p76 in Appendix S4 and used 'vascular plants' with reference to the mobility and trophic status in parentheses on p5 of Appendix S2.

Referee #8: This language complexity also enters in the names of some of their EFGs. For example, I don't feel like the term pyric tussock savanna will be very accessible to non-savanna ecologists or other users. There is a classic division in Africa between moist/mesic and arid savannas (Huntley 1982). As their maps for 'pyric tussock savannas' and 'trophic savannas' largely match those previously conceived savanna types, why not use those terms that are more accessible and have precedence? If not the classic mesic versus arid, it could be 'fire-limited' versus 'herbivore-limited' savanna, which might also be more accessible to a broad audience. In general, it seems preferential to use accessible terms with precedence.

AU RESPONSE 8.7: We workshopped names of ecosystem functional groups extensively and made further amendments in response to the reviews of descriptive profiles described in Appendix S5. In general, we used familiar terms (see Nomenclature text in Appendix S4), but used less familiar terms where it was important to emphasise departure from traditionally recognised units. Tropical savannas are one such case where we align with recent literature in recognising savannas shaped primarily by herbivores and their predators (T4.1 Trophic savannas), those shaped primarily by fire and dominated by tussock grasses (T4.2 Pyric tussock savannas), and oligotrophic savannas dominated by hummock grasses, but also shaped by fire (T4.3 Hummock savannas). The relationships between these ecosystem types and older concepts of wet and dry savannas is not straightforward (see papers by Archibald, Lehmann, Bond and others), hence we avoid those terms.

Referee #8: 5) Like some of the reviewers, I am a bit uncomfortable with the assertion of the biotic processes that are important in each EFG, given the biases around what processes have been the focus of study in different EFGs and the expert opinion nature of the whole process. I think such statements are useful, but authors must be aware that this article will be used as cited evidence that such processes are important in X EFG, and that evidence will be accepted because the article is published in Nature. Thus, there is the real potential to create what some call 'zombie ideas' (c.f. Moles and Ollerton 2016 *Biotropica* and blog by Fox 2011 cited therein). Is there a way for the authors to show more caution around these assertions, in specific instances, or in general?

AU RESPONSE 8.8: We understand this concern. We have tried to balance a realistic representation of evidence on biotic assembly processes and disturbance filters with risks of omission of these potentially important assembly processes. These risks are well noted by authors such as Bond (2019) (cited as ref 48)., Estes et al. 2016 (in PNAS) and others. We have tried to be careful to avoid propagating 'zombie ideas', first by including paragraph in the main text (lines 230-251) explaining our sources and the fact that inferred relationships are hypotheses supported by varying levels of evidence. Second, we clearly label diagrammatically represented relationships in Figs 1-3 as hypotheses. Third, throughout Appendix S4, we use qualifiers such as 'may', 'typically', 'posited' or 'suggested' where appropriate and cite review papers that provide a gateway to relevant published evidence. Finally, we re-read our text to ensure that appropriate caveats were used and are confident that expression of ideas is appropriate to the evidence.

MINOR COMMENTS

Referee #8: For functional biomes, forests are separated around a temperature divide, but 'savannas and grasslands' and 'shrublands and woody shrublands' are not. Why is that?

AU RESPONSE 8.9: Shrublands and grasslands in low temperature environments are represented in the Polar/alpine (cryogenic) biome in (T6.3-T6.5). Hence there T1/T2, T3/T6 and T4/T6 all represent expression of temperature gradients across functional biomes in properties of tree-, shrub- and grass-dominated systems, respectively.

Referee #8: Temperate pyric humid forests vs Temperate pyric sclerophyll forests and woodlands? Should these really be distinguished? [NB: This relates to the unique Australian biome comment above]

AU RESPONSE 8.10: There are multiple properties that distinguish these two groups, which include the world's most fire-prone forests (some Boreal and temperate high montane forests and woodlands are also fire-prone). The humid forests have greater stature and biomass, faster decomposition rates, a more diverse arboreal fauna including vertebrates dependent on tree cavities, a more mesic understorey dominated by ferns, forbs and non-sclerophyll shrubs (cf. sclerophyll shrubs), and occupies a moister, more fertile biophysical niche than the sclerophyll forests. The pyric humid forests are restricted to Australasia and are always dominated by eucalypts, whereas the pyric sclerophyll forests have a more extensive distribution on three continents and are not always dominated by eucalypts.

Referee #8: None of the EFGs have the word Mediterranean in their name. Many terrestrial researchers think of Med ecosystems as one of the main kinds of ecosystems in the world, and at least a couple of the EFGs seem to largely be found in Med areas. The absence of the term is a bit conspicuous.

AU RESPONSE 8.11: Seasonally dry temperate heath and shrublands (T3.2) includes Mediterranean-type shrublands, but very similar shrub-dominated ecosystems occur outside the five regions with strictly Mediterranean-type climates with winter maximum rainfall (see profile in Appendix S4 and Keeley et al. (2012) cited therein). These systems outside the five archetypal Mediterranean climate regions generally have aseasonal rainfall patterns, but still a substantial summer water deficit due to high evapotranspiration rates. Similarly, Mediterranean forest ecosystems are within T2.6 along with a number of other fire-prone tree-dominated ecosystems with seasonal patterns of flammability.

Referee #8: Like R2, I don't like the term ecosystem collapse (line 162). I agree with R2 that this is actually just a form of ecosystem transition. Even if used by IUCN in a way to mean transition, most readers will not take it that way.

AU RESPONSE 8.12: We think 'Ecosystem transition' is unsuitable for our purpose because it is a more generic term with no formal definition that could involve large or small changes in ecosystem properties in any direction (e.g. ecosystem restoration is a form of transition). We strongly prefer to use the term with an explicit definition that is agreed in an international standard (the IUCN Red List of Ecosystems), viz. a major transformation in ecosystem properties, involving loss or biodiversity or function that may be sudden or gradual.

Referee #8: R2 did not like EFG. Why not use FEG? 'FEGs' rolls off the tongue a bit better than 'EFGs'.

AU RESPONSE 8.13: Thank you for the suggestion, but we do not believe this rearrangement would make a substantive improvement to reader comprehension.

Referee #8: I really like the conception of the term 'ecosystem properties'. I think it will be very useful.

AU RESPONSE 8.14: thank you

Referee #8: Using ecoregions to split up Level 3 groups into Level 4 seems too fine-grained to me. Many ecoregions (e.g. in the Amazon) are not really that different.

AU RESPONSE 8.15: We tried to provide a minimal level of explanation to assist readers to understand the role of the three lower levels of typology, rather than a full explanation of how they are developed, which is out of scope for this paper. The same set of ecoregional boundaries may not be appropriate for all EFGs (e.g. tropical lowland rainforests and tropical savannas). Development of Level 4 units will only begin with a simple intersection of Level 3 with a suitable ecoregionalisation. This will undergo a review and adjustment process by ecosystem specialists, informed by analyses where suitable data on biotic composition exist (Silva de Miranda et al. 2018, mentioned below, is a good example of such analysis for Tropical dry forests). Adjustments may split ecoregions, merge them or shift boundaries to accommodate biogeographic patterns in ecosystem biota. We are currently trialling this approach with mangrove specialists and species distributions of key taxa (trees, crustacea, gastropods) to delineate Level 4 units within MFT1.2 Intertidal forests. Thus, if two or more adjacent ecoregions in the Amazon contain similar tropical lowland rainforests, they will likely be included within the same Level 4 unit.

Referee #8: Line 104: Do the authors mean ecosystem functions here rather than ecological functions? Elsewhere, the authors contrast ecological processes and ecosystem functions, so it is a bit confusing to then refer to ecological functions.

AU RESPONSE 8.16: corrected.

Referee #8: *Line 148: I would use 'influence' or 'channel' rather than mitigate. The use of mitigate here could be interpreted in a normative fashion.*

AU RESPONSE 8.17: We mean to refer to active response on impact reduction, not just influence, so we think impact mitigation is an appropriate and widely understood phrase.

Referee #8: *Line 433-434: A good example of this aggregation based on compositional resemblance can be found in Silva de Miranda et al. 2018. GEB. Apologies for self citation!*

AU RESPONSE 8.18: Thanks, we agree this is a good example of bottom-up construction of a classification, and have added the citation.

Reviewer Reports on the Third Revision:

Referees' comments:

Referee #6 (Remarks to the Author):

Dear authors,

I restrict my comments to the issue of presentation of data uncertainties, as this was indicated to me as the key issue for determining the publication of the paper in Nature by the responsible editor. First of all, let me thank you for revising your paper once again and for responding to my earlier comments in detail.

The editor asked me whether publishing the EFG maps would be acceptable if they included an uncertainty disclaimer at the top of each page/map. The short answer is Yes.

To be clear, I do agree with the authors and other reviewers that including purely indicative maps is more useful than problematic. I believe we were mainly in disagreement regarding the level of detail that should be depicted in those maps and possibly on the claims that should be made regarding the maps' utility.

Personally, I find it not ultimately relevant whether or not the maps themselves are published as part of this paper or have instead been published previously (not least because the publication of the archive referenced as [26] on Zenodo did not formally require peer-review of the quality of the final, published maps). If this paper is to be published in Nature, then any statement in it must in my humble opinion be subject to the very highest standards. And in terms of the spatial data in reference [26], it is definitely so that this paper variously discusses these data and uses them as part of arguments, so these statements have to be correct.

As I understand it, the purpose of the authors' maps provided alongside the typology (the thumbnails) is to "indicate" approximate EFG distributions, and from their responses to earlier comments, I also gather that the authors are quite happy to openly convey to any potential users that specific spatial patterns of most EFGs are indeed not known. In this spirit, I urge them to do their best in this paper to help avoid any potential misuses of the maps published in reference [26]. Specifically, any suggestions that the quality of those specific maps was formally assessed should be deleted, for all cases where such a formal assessment was only made for some "base data", but not for specific EFG concepts depicted in the final maps. Moreover, the authors should delete/adjust any remaining suggestions that EFG maps that have not undergone such formal validation of the actual EFG concepts could be fit for purposes of spatial analyses. In this respect, I plea that the authors help promote best practices, rather than settling for lower standards that might be exemplified in prior literature (irrespective of the journals in which that was published).

The authors state that my proposal to generalize maps of uncertain quality to spatial blobs would contradict recommendations in foundational spatial science literature. I welcome their reference to this litReferees' comments:

Referee #6 (Remarks to the Author):

Dear authors,

I restrict my comments to the issue of presentation of data uncertainties, as this was indicated to me as the key issue for determining the publication of the paper in Nature by the responsible editor. First of all, let me thank you for revising your paper once again and for responding to my earlier comments in detail.

The editor asked me whether publishing the EFG maps would be acceptable if they included an uncertainty disclaimer at the top of each page/map. The short answer is Yes.

To be clear, I do agree with the authors and other reviewers that including purely indicative maps is more useful than problematic. I believe we were mainly in disagreement regarding the level of detail that should be depicted in those maps and possibly on the claims that should be made regarding the maps' utility.

Personally, I find it not ultimately relevant whether or not the maps themselves are published as part of this paper or have instead been published previously (not least because the publication of the archive referenced as [26] on Zenodo did not formally require peer-review of the quality of the final, published maps). If this paper is to be published in Nature, then any statement in it must in my humble opinion be subject to the very highest standards. And in terms of the spatial data in reference [26], it is definitely so that this paper variously discusses these data and uses them as part of arguments, so these statements have to be correct.

As I understand it, the purpose of the authors' maps provided alongside the typology (the thumbnails) is to "indicate" approximate EFG distributions, and from their responses to earlier comments, I also gather that the authors are quite happy to openly convey to any potential users that specific spatial patterns of most EFGs are indeed not known. In this spirit, I urge them to do their best in this paper to help avoid any potential misuses of the maps published in reference [26]. Specifically, any suggestions that the quality of those specific maps was formally assessed should be deleted, for all cases where such a formal assessment was only made for some "base data", but not for specific EFG concepts depicted in the final maps. Moreover, the authors should delete/adjust any remaining suggestions that EFG maps that have not undergone such formal validation of the actual EFG concepts could be fit for purposes of spatial analyses. In this respect, I plea that the authors help promote best practices, rather than settling for lower standards that might be exemplified in prior literature (irrespective of the journals in which that was published).

The authors state that my proposal to generalize maps of uncertain quality to spatial blobs would contradict recommendations in foundational spatial science literature. I welcome their reference to this literature. I had based my suggestion for blobs on guidelines by Weibel (1996)* that recommend that maps should generalize uncertain information to reduce error and ensure robustness by simplifying the representation of detail as appropriate to the scale and/or the purpose of the map, which may involve lowering the spatial resolution, and/or simplifying shapes, and/or other

measures. But my intention was certainly not to promote blob maps over coarse gridded maps, but simply to argue that no fine resolution maps that did not undergo formal accuracy assessment and uncertainty quantification should be distributed, nor promoted as fit for purpose of any spatial analyses. This follows the basic principle that one should only map detail that is supported by evidence, which I am sure the authors and also the foundational papers cited by them agree with. I will be very happy if the authors accompany the ecosystem typology with thumbnail maps with a coarsened distribution, as suggested by the authors.

*Weibel, Robert. "Generalization of spatial data: Principles and selected algorithms." Advanced School on the Algorithmic Foundations of Geographic Information Systems. Springer, Berlin, Heidelberg, 1996.

The authors further argue that my earlier comment 6.6 (i.e., that most depicted EFGs underwent no validation) is incorrect, referring to information they provide on the source maps. But I never intended to argue that the quality of most SOURCE maps is unknown - in fact, it is in many cases known to be very poor. Instead, I argued (I continue to argue) that the quality of the PROVIDED ECOSYSTEM MAPS is unknown in most cases. Just as we cannot infer the quality of a meal from the quality of just some of its ingredients, it is simply not sufficient to know that some portion of the different input datasets used to infer the EFG indirectly has been validated. A validated map of tree cover can reliably inform on tree cover, nothing more and nothing less. But if that layer is used to map something more specific than tree cover, such as a dry forest EFG, and if the additional thematic information that allowed calling the tree cover "dry forest" comes from an expert-based blob map that was only ever intended to map the broad extends of a region in which dry forest would be naturally dominant, then the accuracy of the final EFG map is still entirely unknown. I do not believe this is a question of debate. This is simply deductive logic. The only broadly accepted way of knowing the new map's accuracy would be via a formal accuracy assessment, i.e., one carried out at the thematic detail of the EFG concept in question. Certainly, a more accurate tree cover layer should be expected to contribute to a higher overall accuracy of the derived dry forest EFG map, but we cannot know how large that contribution is, so any suggestion that the accuracy of the EFG map is also high, or that the EFG map was validated, is simply misleading. The statement might well be correct, but it should not be made because we do not know this.

The authors suggest that 60-80 EFG types may be fit for purpose of spatial analysis, which they seem to have (illogically) deduced from the fact that such a number of maps EITHER directly/closely corresponded to the EFGs (60) OR were based on validated spatial data (81). However, as the authors will surely agree, reliable spatial analyses require reliable spatial data (i.e., data of high average pixel-level accuracy) on the specific concepts that are the subject of analysis, not merely on some more or less closely related concept. Fitness for purpose of spatial analysis should thus only be attributed to the subset of the 38 EFGs with "spatial data sets that directly matched the concepts ..." that additionally fulfil the criterion that those maps underwent a formal quantitative accuracy assessment (i.e., one that have yielded reasonably high accuracies), or additionally, to any further EFG maps that were derived indirectly but that still underwent formal validation. I very much welcome the new Table S4.1. However, for the reasons given above, the column on quantitative accuracy assessments of "base data" in that table is largely irrelevant information to users of the final EFG maps, who instead need to know whether the EFG maps themselves were validated. The latter should be clearly indicated in this table. This much higher bar for fitness for purpose of spatial

analyses will certainly be met by at least some of the EFGs, so I cannot imagine any reason why the authors might wish to set a lower bar, here. Any statements on map utility in the main text or appendices should be rephrased accordingly, as appropriate. For example, the sentences in lines 554-558 might instead read as "Typical spatial analyses of EFG patterns will require maps where the EFG occurrences were quantitatively validated and shown to be sufficiently accurate (e.g., typically >80-90%, depending on the specific demands of the application). This is currently the case for XYZ# of EFGs (indicated in Table S4.1). For other EFG maps, currently available spatial information [26] may only serve to indicate broad areas in which finding the respective EFG may be most likely" - or something along those lines. I don't wish to be prescriptive or hair-splitting and I am sure the authors understand my point.

My long answer to the editor's question is thus 'Yes, under the conditions that such disclaimer clearly indicates that the maps are for indicative purposes only and do not support spatial analyses, or, if the authors wish to make this distinction among EFGs, only assigns fitness for purpose of spatial analyses to those EFGs whose maps (not just the base data) were formally validated.

Finally, I wish to stress that I continue to consider this typology a very important contribution and an impressive achievement. I do support its publication in Nature. I also do acknowledge and appreciate the effort the authors have made to respond to my own and the other reviewers' comments already, and I look forward to seeing the final version of the paper published soon.

Minor comments:

In the sentence starting in l. 530 of the main text, the verb seems to be missing.

Referee #7 (Remarks to the Author):

I have been asked to review the two additions to the marine and coastal ecosystem functional groups: M1.10 Rhodolith/Maërl beds and MT2.2 Large seabird and pinniped colonies. I believe the first is a legitimate classification as CCA and/or suspension feeders are a legitimate benthic type, and indeed we are seeing shifts to CCA as the dominant feature of the sea floor in iconic systems such as overgrazed Alaskan Kelp forests (providing nominal shelter and food for urchins and other benthic organisms).

I do have a small reservation on the naming of the second category, which is the only categorization that explicitly cites higher trophic level organisms as the defining "function." I understand why-- because the presence of large colonies can lead to excessive nitrogen input through guano. I suppose the closest analogue is "Trophic savanna" which recognizes similar contributions by large herbivores and their predators, and perhaps "Cities, villages, and infrastructure" which can be characterized by high nutrient concentrations (eg, sewage outflow). As the colonies are a unique feature of the landscape and will likely have to be managed/monitored differently than other functional categories, I am willing to concede on the naming here, but it does stick out a bit relative to the other classifications.

Referee #8 (Remarks to the Author):

I am happy with this revision. I think the authors have addressed most of my concerns to satisfaction, and the remaining differences could be put down to opinion and style. I had based my suggestion for blobs on guidelines by Weibel (1996)* that recommend that maps should generalize uncertain information to reduce error and ensure robustness by simplifying the representation of detail as appropriate to the scale and/or the purpose of the map, which may involve lowering the spatial resolution, and/or simplifying shapes, and/or other measures. But my intention was certainly not to promote blob maps over coarse gridded maps, but simply to argue that no fine resolution maps that did not undergo formal accuracy assessment and uncertainty quantification should be distributed, nor promoted as fit for purpose of any spatial analyses. This follows the basic principle that one should only map detail that is supported by evidence, which I am sure the authors and also the foundational papers cited by them agree with. I will be very happy if the authors accompany the ecosystem typology with thumbnail maps with a coarsened distribution, as suggested by the authors.

*Weibel, Robert. "Generalization of spatial data: Principles and selected algorithms." Advanced School on the Algorithmic Foundations of Geographic Information Systems. Springer, Berlin, Heidelberg, 1996.

The authors further argue that my earlier comment 6.6 (i.e., that most depicted EFGs underwent no validation) is incorrect, referring to information they provide on the source maps. But I never intended to argue that the quality of most SOURCE maps is unknown - in fact, it is in many cases known to be very poor. Instead, I argued (I continue to argue) that the quality of the PROVIDED ECOSYSTEM MAPS is unknown in most cases. Just as we cannot infer the quality of a meal from the quality of just some of its ingredients, it is simply not sufficient to know that some portion of the different input datasets used to infer the EFG indirectly has been validated. A validated map of tree cover can reliably inform on tree cover, nothing more and nothing less. But if that layer is used to map something more specific than tree cover, such as a dry forest EFG, and if the additional thematic information that allowed calling the tree cover "dry forest" comes from an expert-based blob map that was only ever intended to map the broad extends of a region in which dry forest would be naturally dominant, then the accuracy of the final EFG map is still entirely unknown. I do not believe this is a question of debate. This is simply deductive logic. The only broadly accepted way of knowing the new map's accuracy would be via a formal accuracy assessment, i.e., one carried out at the thematic detail of the EFG concept in question. Certainly, a more accurate tree cover layer should be expected to contribute to a higher overall accuracy of the derived dry forest EFG map, but we cannot know how large that contribution is, so any suggestion that the accuracy of the EFG map is also high, or that the EFG map was validated, is simply misleading. The statement might well be correct, but it should not be made because we do not know this.

The authors suggest that 60-80 EFG types may be fit for purpose of spatial analysis, which they seem to have (illogically) deduced from the fact that such a number of maps EITHER directly/closely corresponded to the EFGs (60) OR were based on validated spatial data (81). However, as the authors will surely agree, reliable spatial analyses require reliable spatial data (i.e., data of high average pixel-level accuracy) on the specific concepts that are the subject of analysis, not merely on some more or less closely related concept. Fitness for purpose of spatial analysis should thus only be

attributed to the subset of the 38 EFGs with "spatial data sets that directly matched the concepts ..." that additionally fulfil the criterion that those maps underwent a formal quantitative accuracy assessment (i.e., one that have yielded reasonably high accuracies), or additionally, to any further EFG maps that were derived indirectly but that still underwent formal validation. I very much welcome the new Table S4.1. However, for the reasons given above, the column on quantitative accuracy assessments of "base data" in that table is largely irrelevant information to users of the final EFG maps, who instead need to know whether the EFG maps themselves were validated. The latter should be clearly indicated in this table. This much higher bar for fitness for purpose of spatial analyses will certainly be met by at least some of the EFGs, so I cannot imagine any reason why the authors might wish to set a lower bar, here. Any statements on map utility in the main text or appendices should be rephrased accordingly, as appropriate. For example, the sentences in lines 554-558 might instead read as "Typical spatial analyses of EFG patterns will require maps where the EFG occurrences were quantitatively validated and shown to be sufficiently accurate (e.g., typically >80-90%, depending on the specific demands of the application). This is currently the case for XYZ# of EFGs (indicated in Table S4.1). For other EFG maps, currently available spatial information [26] may only serve to indicate broad areas in which finding the respective EFG may be most likely" - or something along those lines. I don't wish to be prescriptive or hair-splitting and I am sure the authors understand my point.

My long answer to the editor's question is thus 'Yes, under the conditions that such disclaimer clearly indicates that the maps are for indicative purposes only and do not support spatial analyses, or, if the authors wish to make this distinction among EFGs, only assigns fitness for purpose of spatial analyses to those EFGs whose maps (not just the base data) were formally validated.

Finally, I wish to stress that I continue to consider this typology a very important contribution and an impressive achievement. I do support its publication in Nature. I also do acknowledge and appreciate the effort the authors have made to respond to my own and the other reviewers' comments already, and I look forward to seeing the final version of the paper published soon.

Minor comments:

In the sentence starting in l. 530 of the main text, the verb seems to be missing.

Referee #7 (Remarks to the Author):

I have been asked to review the two additions to the marine and coastal ecosystem functional groups: M1.10 Rhodolith/Maërl beds and MT2.2 Large seabird and pinniped colonies. I believe the first is a legitimate classification as CCA and/or suspension feeders are a legitimate benthic type, and indeed we are seeing shifts to CCA as the dominant feature of the sea floor in iconic systems such as overgrazed Alaskan Kelp forests (providing nominal shelter and food for urchins and other benthic organisms).

I do have a small reservation on the naming of the second category, which is the only categorization that explicitly cites higher trophic level organisms as the defining "function." I understand why-- because the presence of large colonies can lead to excessive nitrogen input through guano. I

suppose the closest analogue is "Trophic savanna" which recognizes similar contributions by large herbivores and their predators, and perhaps "Cities, villages, and infrastructure" which can be characterized by high nutrient concentrations (eg, sewage outflow). As the colonies are a unique feature of the landscape and will likely have to be managed/monitored differently than other functional categories, I am willing to concede on the naming here, but it does stick out a bit relative to the other classifications.

Referee #8 (Remarks to the Author):

I am happy with this revision. I think the authors have addressed most of my concerns to satisfaction, and the remaining differences could be put down to opinion and style.

Author Rebuttals to Third Revision:

Referees' comments:

Referee #6 (Remarks to the Author):

Dear authors,

I restrict my comments to the issue of presentation of data uncertainties, as this was indicated to me as the key issue for determining the publication of the paper in Nature by the responsible editor. First of all, let me thank you for revising your paper once again and for responding to my earlier comments in detail.

The editor asked me whether publishing the EFG maps would be acceptable if they included an uncertainty disclaimer at the top of each page/map. The short answer is Yes.

To be clear, I do agree with the authors and other reviewers that including purely indicative maps is more useful than problematic. I believe we were mainly in disagreement regarding the level of detail that should be depicted in those maps and possibly on the claims that should be made regarding the maps' utility.

Personally, I find it not ultimately relevant whether or not the maps themselves are published as part of this paper or have instead been published previously (not least because the publication of the archive referenced as [26] on Zenodo did not formally require peer-review of the quality of the final, published maps). If this paper is to be published in Nature, then any statement in it must in my humble opinion be subject to the very highest standards. And in terms of the spatial data in reference [26], it is definitely so that this paper variously discusses these data and uses them as part of arguments, so these statements have to be correct.

As I understand it, the purpose of the authors' maps provided alongside the typology "indicate" approximate EFG distributions, and from their responses (the thumbnails) is to

to earlier comments, I also gather that the authors are quite happy to openly convey to any potential users that specific spatial patterns of most EFGs are indeed not known. In this spirit, I urge them to do their best in this paper to help avoid any potential misuses of the maps published in reference [26]. Specifically, any suggestions that the quality of those specific maps was formally assessed should be deleted, for all cases where such a formal assessment was only made for some "base data", but not for specific EFG concepts depicted in the final maps. Moreover, the authors should delete/adjust any remaining suggestions that EFG maps that have not undergone such formal validation of the actual EFG concepts could be fit for purposes of spatial analyses. In this respect, I plea that the authors help promote best practices, rather than settling for lower standards that might be exemplified in prior literature (irrespective of the journals in which that was published).

The authors state that my proposal to generalize maps of uncertain quality to spatial blobs would contradict recommendations in foundational spatial science literature. I welcome their reference to this literature. I had based my suggestion for blobs on guidelines by Weibel (1996)* that recommend that maps should generalize uncertain information to reduce error and ensure robustness by simplifying the representation of detail as appropriate to the scale and/or the purpose of the map, which may involve lowering the spatial resolution, and/or simplifying shapes, and/or other measures. But my intention was certainly not to promote blob maps over coarse gridded maps, but simply to argue that no fine resolution maps that did not undergo formal accuracy assessment and uncertainty quantification should be distributed, nor promoted as fit for purpose of any spatial analyses. This follows the basic principle that one should only map detail that is supported by evidence, which I am sure the authors and also the foundational papers cited by them agree with. I will be very happy if

the authors accompany the ecosystem typology with thumbnail maps with a coarsened distribution, as suggested by the authors.

*Weibel, Robert. "Generalization of spatial data: Principles and selected algorithms." Advanced School on the Algorithmic Foundations of Geographic Information Systems. Springer, Berlin, Heidelberg, 1996.

The authors further argue that my earlier comment 6.6 (i.e., that most depicted EFGs underwent no validation) is incorrect, referring to information they provide on the source maps. But I never intended to argue that the quality of most SOURCE maps is unknown - in fact, it is in many cases known to be very poor. Instead, I argued (I continue to argue) that the quality of the PROVIDED ECOSYSTEM MAPS is unknown in most cases. Just as we cannot infer the quality of a meal from the quality of just some of its ingredients, it is simply not sufficient to know that some portion of the different input datasets used to infer the EFG indirectly has been validated. A validated map of tree cover can reliably inform on tree cover, nothing more and nothing less. But if that layer is used to map something more specific than tree cover, such as a dry forest EFG, and if the additional thematic information that allowed calling the tree cover "dry forest" comes from an expert-based blob map that was

only ever intended to map the broad extends of a region in which dry forest would be naturally dominant, then the accuracy of the final EFG map is still entirely unknown. I do not believe this is a question of debate. This is simply deductive logic. The only broadly accepted way of knowing the new map's accuracy would be via a formal accuracy assessment, i.e., one carried out at the thematic detail of the EFG concept in question. Certainly, a more accurate tree cover layer should be expected to contribute to a higher overall accuracy of the derived dry forest EFG map, but we cannot know how large that contribution is, so any suggestion that the accuracy of the EFG map is also high, or that the EFG map was validated, is simply misleading. The statement might well be correct, but it should not be made because we do not know this.

The authors suggest that 60-80 EFG types may be fit for purpose of spatial analysis, which they seem to have (illogically) deduced from the fact that such a number of maps EITHER directly/closely corresponded to the EFGs (60) OR were based on validated spatial data (81). However, as the authors will surely agree, reliable spatial analyses require reliable spatial data (i.e., data of high average pixel-level accuracy) on the specific concepts that are the subject of analysis, not merely on some more or less closely related concept. Fitness for purpose of spatial analysis should thus only be attributed to the subset of the 38 EFGs with "spatial data sets that directly matched the concepts ..." that additionally fulfil the criterion that those maps underwent a formal quantitative accuracy assessment (i.e., one that have yielded reasonably high accuracies), or additionally, to any further EFG maps that were derived indirectly but that still underwent formal validation. I very much

welcome the new Table S4.1. However, for the reasons given above, the column on

quantitative accuracy assessments of "base data" in that table is largely irrelevant information to users of the final EFG maps, who instead need to know whether the EFG maps themselves were validated. The latter should be clearly indicated in this table. This much higher bar for fitness for purpose of spatial analyses will certainly be met by at least some of the EFGs, so I cannot imagine any reason why the authors might wish to set a lower bar, here. Any statements on map utility in the main text or appendices should be rephrased accordingly, as appropriate. For example, the sentences in lines 554-558 might instead read as "Typical spatial analyses of EFG patterns will require maps where the EFG occurrences were quantitatively validated and shown to be sufficiently accurate (e.g., typically >80-90%, depending on the specific demands of the application).

This is currently the case for XYZ# of EFGs

(indicated in Table S4.1). For other EFG maps, currently available spatial information [26] may only serve to indicate broad areas in which finding the respective EFG may be most likely" - or something along those lines. I don't wish to be prescriptive or hair-splitting and I am sure the authors understand my point.

My long answer to the editor's question is thus 'Yes, under the conditions that such disclaimer clearly indicates that the maps are for indicative purposes only and do not support spatial analyses, or, if the authors wish to make this distinction among EFGs, only assigns fitness for purpose of spatial analyses to those EFGs whose maps (not just the base data) were formally validated.

Finally, I wish to stress that I continue to consider this typology a very important contribution and an impressive achievement. I do support its publication in Nature. I also do acknowledge and appreciate the effort the authors have made to respond to my own and the other reviewers' comments already, and I look forward to seeing the final version of the paper published soon.

Minor comments:

In the sentence starting in l. 530 of the main text, the verb seems to be missing.

AU RESPONSE: We agree with the reviewer's general sentiments on map accuracy and are keen to report transparently the strengths and weaknesses in map data. In this revision of our ms, we further encourage responsible use of spatial data by adding a caveat to every profile in Appendix S4 where thumbnail maps are displayed. We also refer to these caveats in the main text (line 214) and include a further statement on map usage in the Methods section as follows,

"The purpose of maps for our study was to visualise global distributions. Prior to applications of map data to spatial analysis, we recommend critical review of methods and validation outcomes reported in each data source to ensure fitness for purpose (Appendix S4)."

Ideally, maps of ecosystems and other environmental features would only be used where they meet the highest standards with systematic field validation. The reality, in our experience, however, is that the decision-makers and other map users are driven by imperatives that do not wait for scientists to meet data quality standards. This is a vexed trade-off because bad data can result in bad inferences and decisions, yet decisions will be made, regardless of whether scientists are entirely satisfied with data quality. Our challenge, is to provide: i) the best available data to support particular decisions at the time they must be made; ii) transparency about methods, strengths and weaknesses; iii) a pathway for updating and progressively improving data quality; and iii) demonstrate how users can accommodate uncertainties that are inevitably inherent in any spatial data sets, including those with extensive validation. Fitness for purpose of any data set, by definition, will depend on the application, acceptable margins of error and other specific considerations (i.e. some maps will be suitable for some analyses, but not others). Therefore, general and prescriptive caveats about whether particular data sets should or should not be used for any "typical" analysis (such as that suggested for lines 554-558) are not that helpful to users. We also note problems interpreting "typical" analysis when applications are so diverse. Some users will be prepared to tolerate larger errors than others to realise benefits of spatial analysis for their particular applications. In our preceding response to reviews, we point to multiple existing publications in Nature where authors have made such judgements. Our job is to equip them with information to support those judgements. Overall, by presenting the best available maps with appraisals, sources and caveats, we think we have struck a good balance between utility (the maps are widely useful, at the very least for global display and in many cases much more), transparency about limitations, recommendations for responsible use and a pathway for data improvement.

Thank you for picking up the omission on l.530. We added "Large numbers of empirical studies 'exist' for..."

Referee #7 (Remarks to the Author):

I have been asked to review the two additions to the marine and coastal ecosystem functional groups: M1.10 Rhodolith/Maërl beds and MT2.2 Large seabird and pinniped colonies. I believe the first is a legitimate classification as CCA and/or suspension feeders are a legitimate benthic type, and indeed we are seeing shifts to CCA as the dominant feature of the sea floor in iconic systems such as overgrazed Alaskan Kelp forests (providing nominal shelter and food for urchins and other benthic organisms).

I do have a small reservation on the naming of the second category, which is the only categorization that explicitly cites higher trophic level organisms as the defining "function." I understand why--because the presence of large colonies can lead to excessive nitrogen input through guano. I suppose the closest analogue is "Trophic

savanna" which recognizes similar contributions by large herbivores and their predators, and perhaps "Cities, villages, and infrastructure" which can be characterized by high nutrient concentrations (eg, sewage outflow). As the colonies are a unique feature of the landscape and will likely have to be managed/monitored differently than other functional categories, I am willing to concede on the naming here, but it does stick out a bit relative to the other classifications.

AU RESPONSE: Thank you for these thoughts on the name of MT2.2. The titles of Ecosystem Functional Groups involve some difficult trade-offs – we aim to convey information about ecosystem structure and functions, but also use terms that are widely familiar to potential users and not too technical. We considered alternative titles, such as 'Eutrophic coasts' to represent nutrient influx, but after consultation with experts who work with such systems, decided on the more pragmatic option that identifies the vectors of nutrient influx, primarily because that is recognisable to a wide audience.

Referee #8 (Remarks to the Author):

I am happy with this revision. I think the authors have addressed most of my concerns to satisfaction, and the remaining differences could be put down to opinion and style.

AU RESPONSE: Thank you